# Microsaccadic sampling of moving image information provides *Drosophila* hyperacute vision

Mikko Juusola[1,2†*], An Dau[2†], Zhuoyi Song[2†], Narendra Solanki[2], Diana Rien[1,2], David Jaciuch[2], Sidhartha Anil Dongre[2], Florence Blanchard[2], Gonzalo G de Polavieja[3], Roger C Hardie[4], Jouni Takalo[2]

[1]National Key Laboratory of Cognitive Neuroscience and Learning, Beijing Normal University, Beijing, China; [2]Department of Biomedical Science, University of Sheffield, Sheffield, United Kingdom; [3]Champalimaud Neuroscience Programme, Champalimaud Center for the Unknown, Lisbon, Portugal; [4]Department of Physiology Development and Neuroscience, Cambridge University, Cambridge, United Kingdom

**Abstract** Small fly eyes should not see fine image details. Because flies exhibit saccadic visual behaviors and their compound eyes have relatively few ommatidia (sampling points), their photoreceptors would be expected to generate blurry and coarse retinal images of the world. Here we demonstrate that *Drosophila* see the world far better than predicted from the classic theories. By using electrophysiological, optical and behavioral assays, we found that R1-R6 photoreceptors' encoding capacity *in time* is maximized to fast high-contrast bursts, which resemble their light input during saccadic behaviors. Whilst *over space*, R1-R6s resolve moving objects at saccadic speeds beyond the predicted motion-blur-limit. Our results show how refractory phototransduction and rapid photomechanical photoreceptor contractions jointly sharpen retinal images of moving objects *in space-time*, enabling hyperacute vision, and explain how such microsaccadic information sampling exceeds the compound eyes' optical limits. These discoveries elucidate how acuity depends upon photoreceptor function and eye movements.
DOI: https://doi.org/10.7554/eLife.26117.001

**\*For correspondence:**
m.juusola@sheffield.ac.uk

†These authors contributed equally to this work

**Competing interests:** The authors declare that no competing interests exist.

## Introduction

The acuity of an eye is limited by its photoreceptor spacing, which provides the grain of the retinal image. To resolve two stationary objects, at least three photoreceptors are needed for detecting the intensity difference in between. To resolve two moving objects is harder, as vision becomes further limited by each photoreceptor's finite integration time and receptive field size (*Srinivasan and Bernard, 1975*; *Juusola and French, 1997*; *Land, 1997*).

Nevertheless, animals - from insects to man - view the world by using saccades, fast movements, which direct their eyes to the surroundings, and fixation intervals between the saccades, during which gaze is held near stationary (*Land, 1999*). Because of photoreceptors' slow integration-time, saccades should blur image details and these are thought to be sampled when gaze is stabilized. Thus, information would be captured during fixations whilst during saccades animals would be effectively blind. This viewpoint, however, ignores fast photoreceptor adaptation, which causes perceptual fading during fixation (*Ditchburn and Ginsborg, 1952*; *Riggs and Ratliff, 1952*), reducing visual information and possibly rendering perception to mean light only. Therefore, to maximize information and acuity, it is plausible that evolution has optimized photoreceptor function in respect to visual behaviors and needs.

**eLife digest** Fruit flies have five eyes: two large compound eyes which support vision, plus three smaller single lens eyes which are used for navigation. Each compound eye monitors 180° of space and consists of roughly 750 units, each containing eight light-sensitive cells called photoreceptors. This relatively wide spacing of photoreceptors is thought to limit the sharpness, or acuity, of vision in fruit flies. The area of the human retina (the light-sensitive surface at back of our eyes) that generates our sharpest vision contains photoreceptors that are 500 times more densely packed.

Despite their differing designs, human and fruit fly eyes work via the same general principles. If we, or a fruit fly, were to hold our gaze completely steady, the world would gradually fade from view as the eye adapted to the unchanging visual stimulus. To ensure this does not happen, animals continuously make rapid, automatic eye movements called microsaccades. These refresh the image on the retina and prevent it from fading. Yet it is not known why do they not also cause blurred vision.

Standard accounts of vision assume that the retina and the brain perform most of the information processing required, with photoreceptors simply detecting how much light enters the eye. However, Juusola, Dau, Song et al. now challenge this idea by showing that photoreceptors are specially adapted to detect the fluctuating patterns of light that enter the eye as a result of microsaccades. Moreover, fruit fly eyes resolve small moving objects far better than would be predicted based on the spacing of their photoreceptors.

The discovery that photoreceptors are well adapted to deal with eye movements changes our understanding of insect vision. The findings also disprove the 100-year-old dogma that the spacing of photoreceptors limits the sharpness of vision in compound eyes. Further studies are required to determine whether photoreceptors in the retinas of other animals, including humans, have similar properties.

DOI: https://doi.org/10.7554/eLife.26117.002

We have now devised a suite of new experimental and theoretical methods to study this question both in time and over space in *Drosophila* R1-R6 photoreceptors. The *Drosophila* compound eyes are composed of ~750 seemingly regular lens-capped modules called ommatidia, which should provide the fly a panoramic visual field of low optical resolution (*Barlow, 1952*; *Land, 1997*). Each ommatidium contains eight photoreceptor cells (R1-R8), pointing to seven different directions. The ultraviolet and blue-green-sensitive outer photoreceptors, R1-R6, initiate the motion vision pathway, whilst the central R7 and R8, which lie on top of each other, detect different colors from one direction (*Wardill et al., 2012*). Owing to the eye's neural superposition principle, R1, R2, R3, R4, R5 and R6, each from a separate neighboring ommatidium, also point to the same direction. By pooling their output for synaptic transmission, the photoreceptor spacing (spatial resolution) effectively matches the ommatidium spacing (average interommatidial angle, $\Delta\varphi$ = 4.5° (*Götz, 1964*; *Land, 1997*; *Gonzalez-Bellido et al., 2011*) but the signal-to-noise ratio of the transmitted image could improve by $\sqrt{6}$ (*de Ruyter van Steveninck and Laughlin, 1996*; *Zheng et al., 2006*).

Here we show how evolution has improved *Drosophila* vision beyond these classic ideas, suggesting that light information sampling in R1-R6 photoreceptors is tuned to saccadic behavior.

Our intracellular recordings reveal that R1-R6s capture 2-to-4-times more information in time than previous maximum estimates (*Juusola and Hardie, 2001a*; *Song et al., 2012*; *Song and Juusola, 2014*) when responding to high-contrast bursts (periods of rapid light changes followed by quiescent periods) that resemble light input from natural scenes generated by saccadic viewing. Biophysically-realistic model simulations suggest that this improvement largely results from interspersed 'fixation' intervals, which allow photoreceptors to sample more information from phasic light changes by relieving them from refractoriness (*Song et al., 2012*; *Song and Juusola, 2014*; *Juusola et al., 2015*).

Remarkably, over space, our intracellular recordings, high-speed microscopy and modeling further reveal how photomechanical photoreceptor contractions (*Hardie and Franze, 2012*) work together with refractory sampling to improve spatial acuity. We discover that by actively modulating light input and photoreceptor output, these processes reduce motion blur during saccades and

adaptation during gaze fixation, which otherwise could fade vision (*Ditchburn and Ginsborg, 1952*; *Riggs and Ratliff, 1952*; *Land, 1997*). The resulting phasic responses sharpen retinal images by highlighting the times when visual objects cross a photoreceptor's receptive field, thereby encoding space in time (see also: *Ahissar and Arieli, 2001*; *Donner and Hemilä, 2007*; *Rucci et al., 2007*; *Kuang et al., 2012a*; *Kuang et al., 2012*; *Franceschini et al., 2014*; *Viollet, 2014*). Thus, neither saccades nor fixations blind the flies, but together improve vision.

Incorporation of this novel opto-mechano-electric mechanism into our 'microsaccadic sampling'-model predicts that *Drosophila* can see >4 fold finer details than their eyes' spatial sampling limit – a prediction directly confirmed by optomotor behavior experiments. By demonstrating how fly photoreceptors' fast microsaccadic information sampling provides hyperacute vision of moving images, these results change our understanding of insect vision, whilst showing an important relationship between eye movements and visual acuity.

## Results

These results establish that *Drosophila* exploit image motion (through eye movements) to see spatial details, down to hyperacute resolution. A fly's visual acuity is limited by how well its photoreceptors resolve different photon rate changes, and their receptive field sizes. However, because each photoreceptor's signal-to-noise ratio and receptive field size adapt dynamically to light changes, acuity also depends upon the eye movements that cause them. To make these relationships clear, the results are presented in the following order:

1. We show that photoreceptors capture most visual information from high-contrast bursts, and reveal how this is achieved by refractory photon sampling and connectivity (Figures 1–5).
2. We show that saccades and gaze fixations in natural environment result in such high-contrast bursts, implying that eye movements work with refractory sampling to improve vision (Figure 6).
3. We demonstrate that photoreceptors contract to light *in vivo* and explain how these microsaccades move and narrow their receptive fields (Figures 7–8) to sharpen light input and photoreceptor output in time.
4. Collectively, these dynamics predict that *Drosophila* see finer spatial details than their compound eyes' optical resolution over a broad range of image velocities (Figure 9), and we verify this by optomotor behavior (Figure 10).

Videos 1-4 and Appendixes 1–10 explain in detail the new ideas, methods, experiments and theory behind these results.

### Breaking the code by coupling experiments with theory

To work out how well a *Drosophila* R1-R6 photoreceptor can see the world, we compared intracellular recordings with realistic theoretical predictions from extensive quantal light information sampling simulations (Appendixes 1–3), having the following physical limits and properties (*Song et al., 2012*; *Song and Juusola, 2014*; *Juusola et al., 2015*; *Song et al., 2016*):

- A photoreceptor counts photons and integrates these samples to an estimate, a macroscopic voltage response, of light changes within its receptive field.
- This estimate is counted by 30,000 microvilli, which form its light-sensor, the rhabdomere. Each microvillus is a photon sampling unit, capable of transducing a photon's energy to a unitary response (quantum bump or sample) (*Henderson et al., 2000*; *Juusola and Hardie, 2001a*; *Song et al., 2012*; *Song and Juusola, 2014*).
- Following each bump, the light-activated microvillus becomes *refractory* (*Song et al., 2012*; *Song and Juusola, 2014*; *Juusola et al., 2015*) for 50–300 ms. Therefore, with brightening light, a photoreceptor's sample rate gradually saturates, as fewer microvilli are available to generate bumps.
- Although refractory sampling makes photoreceptors imperfect photon counters, it benefits vision by representing a fast automatic adaptation mechanism, reducing sensitivity in proportion to background intensity (*Song et al., 2012*; *Song and Juusola, 2014*), whilst accentuating responses to contrast changes (*Song and Juusola, 2014*).

As previously described for a variety of other stimuli (*Song et al., 2012*; *Song and Juusola, 2014*; *Juusola et al., 2015*), we found a close correspondence between the recordings and

simulations (waveforms, noise, adaptation dynamics and information transfer) for all the tested stimuli, establishing how refractory quantal sampling is tuned by light changes. Conversely, control models without refractoriness or based on the Volterra black-box method (*Juusola and French, 1997*) failed to predict R1-R6s' information sampling and adaptation dynamics. Nevertheless, these limitations and differences gave us vital clues into the hidden/combined mechanisms that underpin photoreceptor function (Appendixes 2–9). We now analyze and explain the key results step-by-step.

## High-contrast 'saccadic' bursts maximize encoding

A well-known trade-off of fast adaptation is that it causes perceptual fading during fixation (*Ditchburn and Ginsborg, 1952*; *Riggs and Ratliff, 1952*), and to see the world requires motion or self-motion: body, head and eye movements (*Hengstenberg, 1971*; *Land, 1973*; *Franceschini and Chagneux, 1997*; *Schilstra and van Hateren, 1998*; *Blaj and van Hateren, 2004*; *Martinez-Conde et al., 2013*), which remove adaptation. However, it remains unclear whether or how the fly photoreceptors' information sampling dynamics is tuned to visual behaviors to see the world better. To start unravelling these questions, we first surveyed what kind of stimuli drove their information transfer maximally (*Figure 1*), ranging from high-contrast bursts, in which transient intensity fluctuations were briefer than *Drosophila*'s normal head/body-saccades (*Fry et al., 2003*; *Geurten et al., 2014*), to Gaussian white-noise (GWN). These stimuli tested systematically different contrast and bandwidth patterns over R1-R6s' diurnal encoding gamut.

Intracellular recordings (*Figure 1A*) revealed that photoreceptors responded most vigorously to high-contrast bursts, which contained fast transient events with darker intervals. *Figure 1B* shows the averages (signals; thick) and individual responses (thin) of a typical R1-R6, grouped by the stimulus bandwidth and mean contrast. For all the bandwidths (columns), the responses increased with contrast, while for all the contrasts (rows), the responses decreased with the increasing bandwidth (left). Therefore, the slowest high-contrast bursts (red; top-left) with the longest darker intervals, which theoretically (*Song et al., 2012*; *Song and Juusola, 2014*; *Juusola et al., 2015*) should relieve most refractory microvilli (Appendixes 1–3), evoked the largest peak-to-peak responses (43.4 ± 5.6 mV; mean ± SD, n = 16 cells; *Figure 1—figure supplement 1*). Whereas the fastest low-contrast GWN (blue; bottom-right), which would keep more microvilli refractory, evoked the smallest responses (3.7 ± 1.1 mV; n = 4).

Notably, whilst all the stimuli were very bright, the largest responses (to bursts) were induced at the dimmest background (BG0, darkness) and the smallest responses (to GWN) at the brightest background (BG1.5) (*Figure 1B*). Thus, the mean emitted photon rate and light information at the source was lower for the bursts and higher for the GWNs (the signal-to-noise ratio of the observable world increases with brightening illumination; e.g. *Appendix 2—figures 5D and H*). However, in very bright stimulation, the global mean light intensity (over the experiment) becomes less critical for good vision as the eye self-regulates its own input more. Photons galore are lost to intracellular pupil (*Howard et al., 1987*; *Song and Juusola, 2014*) and refractory microvilli (*Song et al., 2012*), which reduce quantum efficiency. Although a R1-R6's receptive field could be bombarded by $10^6$–$10^9$ photons/s (in daylight), due to the dramatic drop in quantum efficiency, the photoreceptor could only count up ~80,000–800,000 quantum bumps/s (Appendix 2). Therefore, the stimulus contrast and bandwidth, which drive the dynamic quantum bump rate changes, summing up the photoreceptor output, are confounded with changes in mean intensity. And, as such, this stimulus design, by containing four different BGs, makes it difficult to see the exact contributions of contrast, bandwidth and mean in controlling the responses.

Information theoretical analysis (*Figure 2* and *Figure 2—figure supplement 1*) indicated that the response differences largely reflected differences in their quantum bump counts. The maximum signal power spectra to bursty stimuli could be up to ~6,000 times larger than that of the noise, which was effectively stimulus-invariant (*Figure 2—figure supplement 2A*). Because the noise power spectrum largely represents the average quantum bump's frequency composition (*Wong et al., 1982*; *Juusola and Hardie, 2001a*; *Song and Juusola, 2014*), the bumps adapted to a similar size. Here, given the brightness of the stimuli, the bumps had light-adapted close to their minimum (*Juusola and Hardie, 2001a*). Thereby, the larger responses simply comprised more bumps. Moreover, with Poisson light statistics, the response precision - how well it estimated photon flux changes - should increase with the square root of bump count until saturation; when more microvilli remained refractory (*Song and Juusola, 2014*). Accordingly, signaling performance (*Figure 2A,C*) increased

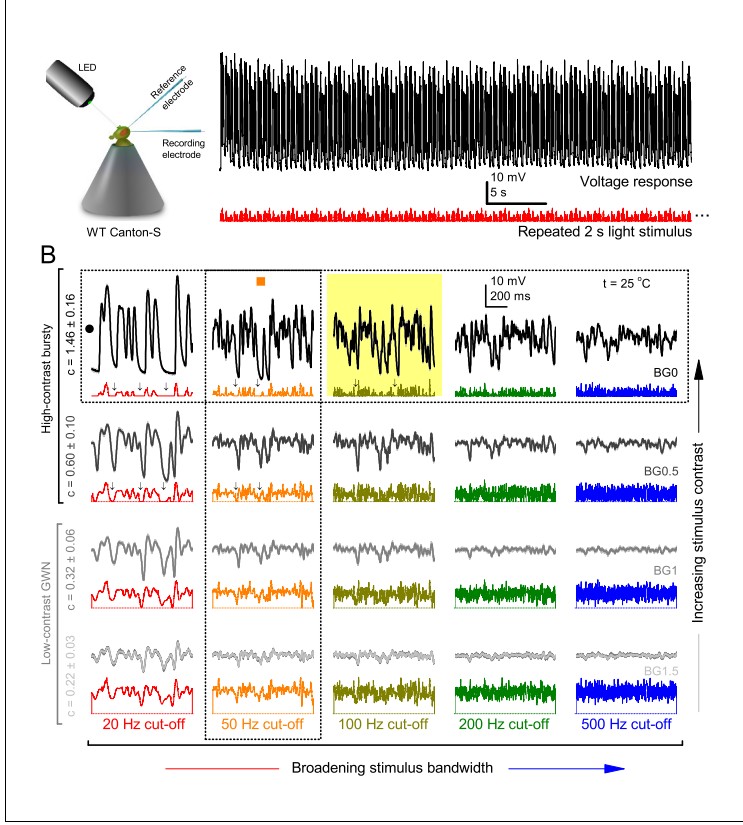

**Figure 1.** Photoreceptors respond best to high-contrast bursts. (**A**) Schematic of intracellular recordings to repeated bursty light intensity time series (20 Hz bandwidth). Responses changed little (minimal adaptation) during bursts. (**B**) Testing a R1-R6 photoreceptor's diurnal encoding gamut. Means (thick traces) and 20 individual responses (thin; near-perfectly overlapped) to 20 different stimuli; each with a specific bandwidth (columns: from 20 Hz, red, to 500 Hz, blue) and mean contrast (rows). Reducing the background (BG) of Gaussian white-noise stimuli (GWN; 2-unit peak-to-peak modulation) from bright (1.5-unit, bottom) to dark (0-unit, top) halved their modulation, generating bursts of increasing contrast: the lower the BG, the higher the contrast. Left-top: responses from (**A**). Yellow box: maximum information responses. Arrows: dark intervals. Because of half-Gaussian waveforms, light bursts carried fewer photons (see *Figure 2—figure supplement 3*). Yet their larger responses comply with the stochastic adaptive visual information sampling theory (*Song et al., 2012*; *Song and Juusola, 2014*; *Juusola et al., 2015*) (**Appendixes 1–3**), whereby dark intervals rescue refractory microvilli for transducing high-frequency (1–20 ms) saccadic photon surges (of high contrast) into quantum bumps efficiently. Thus, larger responses would incorporate more bumps. Recordings are from the same photoreceptor. Vertical dotted rectangle (orange square) and horizontal rectangle (black circle): responses for contrast and bandwidth analyses in *Figure 2A*. Similar R1-R6 population data is in *Figure 1—figure supplement 1*.

DOI: https://doi.org/10.7554/eLife.26117.003

The following source data and figure supplement are available for figure 1:

**Source data 1.** Intracellular voltage responses of the same R1-R6 photoreceptor to very bright 20 Hz, 50 Hz, 100 Hz, 200 Hz and 500 Hz bursty light stimuli at BG0 (darkness).

DOI: https://doi.org/10.7554/eLife.26117.005

**Source data 2.** Intracellular voltage responses of the same R1-R6 photoreceptor to very bright 20 Hz, 50 Hz, 100 Hz, 200 Hz and 500 Hz bursty light stimuli at BG0.5.

DOI: https://doi.org/10.7554/eLife.26117.006

**Source data 3.** Intracellular voltage responses of the same R1-R6 photoreceptor to very bright 20 Hz, 50 Hz, 100 Hz, 200 Hz and 500 Hz bursty light stimuli at BG1.

DOI: https://doi.org/10.7554/eLife.26117.007

**Source data 4.** Intracellular voltage responses of the same R1-R6 photoreceptor to very bright 20 Hz, 50 Hz, 100 Hz, 200 Hz and 500 Hz bursty light stimuli at BG1.5.

DOI: https://doi.org/10.7554/eLife.26117.008

*Figure 1 continued on next page*

*Figure 1 continued*

**Figure supplement 1.** R1-R6 output varies more cell-to-cell than trial-to-trial (*cf.Figure 1*) but show consistent stimulus-dependent dynamics over the whole encoding range.

DOI: https://doi.org/10.7554/eLife.26117.004

both with the stimulus bandwidth (left) and contrast (right), until these became too fast to follow. Information transfer peaked at 100 Hz bursts, which allocated the R1-R6's limited bandwidth and amplitude range near-optimally, generating the broadest frequency (*Figure 2A* and *Figure 2—figure supplement 1A*) and (Gaussian) voltage distributions (*Figure 2B* and *Figure 2—figure supplement 1B*).

Thus, with the right mixture of bright 'saccadic' bursts (to maximally activate microvilli) and darker 'fixation' intervals (to recover from refractoriness) forming the high-contrast input, a photoreceptor's information transfer approached the capacity (*Shannon, 1948*), the theoretical maximum, where every symbol (voltage value) of a message (macroscopic voltage response) is transmitted equally often (*Figure 2C* and *Figure 2—figure supplement 1C*). Remarkably, this performance (610–850 bits/s) was 2-to-4-times of that for GWN (200–350 bits/s), which has often been used for characterizing maximal encoding (*Juusola and Hardie, 2001a*), and twice of that for rich naturalistic stimuli (380–510 bits/s) (*Song and Juusola, 2014*) (*Figure 2—figure supplement 3*). GWN, especially, lacks longer darker events, which should make microvilli refractory (*Song and Juusola, 2014*) with fewer sampled photons limiting information transfer (Appendixes 2–3).

There are two reasons why these information rate estimates, which were calculated from equal-sized data chunks by the Shannon formula (*Equation 1*, Material and methods), should be robust and largely bias-free. First, apart from the responses to 20 Hz high-contrast bursts (*Figure 2B*, red trace), the responses to all the other stimuli had broadly Gaussian signal and noise distributions, obeying the Shannon formula's major assumptions (*Shannon, 1948*). Second, our previous tests in comparing the Shannon formula to triple extrapolation method (*Juusola and de Polavieja, 2003*), which is directly derived from Shannon's information theory, have shown that for sufficiently large sets of data both these methods provide similar estimates even for this type of highly non-Gaussian responses (~5–20% maximal differences) (*Song and Juusola, 2014*; *Dau et al., 2016*). And, indeed, new tests using additional recordings to longer stimulus repetitions (*Figure 2—figure supplement 4*) indicated the same. Thus here, the Shannon formula should provide a sufficiently accurate information estimate also for the 20 Hz high-contrast burst responses, making this evaluation fair (see Appendix 2).

## Simulations reveal network contribution

These findings were largely replicated by stochastic simulations (*Figures 3–4*). A biophysically realistic photoreceptor model, which contains 30,000 microvilli (*Song et al., 2012*), sampled light information much like a real R1-R6, generating authentic responses to all the test stimuli (*Figure 3A–B*). Yet, markedly, the model lacked the intracellular pupil (or any structural adaptation), which protects microvilli from saturation (*Howard et al., 1987*; *Song and Juusola, 2014*), and network connections (*Zheng et al., 2006*; *Rivera-Alba et al., 2011*; *Wardill et al., 2012*). In real photoreceptors, the pupil screens off excess light to maximize information transfer (*Howard et al., 1987*; *Song and Juusola, 2014*). Similarly, in the simulations, the mean light intensity of each stimulus was optimized (Appendix 2) for maximum information (*Figure 4A–C*), establishing the photon absorption rate for a R1-R6 photoreceptor's best signaling performance (bits/s).

At its peak, the model transferred 633 ± 20 bits/s (mean ± SD; *Figure 4C*) for 100 Hz bursts of 8 × 10⁵ photons/s, with further brightening reducing information as more microvilli became refractory. This performance matches that of many real R1-R6s (*Figure 2—figure supplement 1C*), but is ~200 bits/s less than in some recordings (*Figure 2C*). The real R1-R6s, on balance, receive extra information from their neighbors (*Rivera-Alba et al., 2011*; *Wardill et al., 2012*), which through superposition (*Zheng et al., 2006*) sample information from overlapping receptive fields. In other words, since our stimuli (from a white LED) were spatially homogenous, these synaptic feedbacks should be able to enhance the system's signal-to-noise by averaging the photoreceptors' independent photon count estimates from the same visual area, reducing noise (*Zheng et al., 2006*; *Juusola and Song, 2017*).

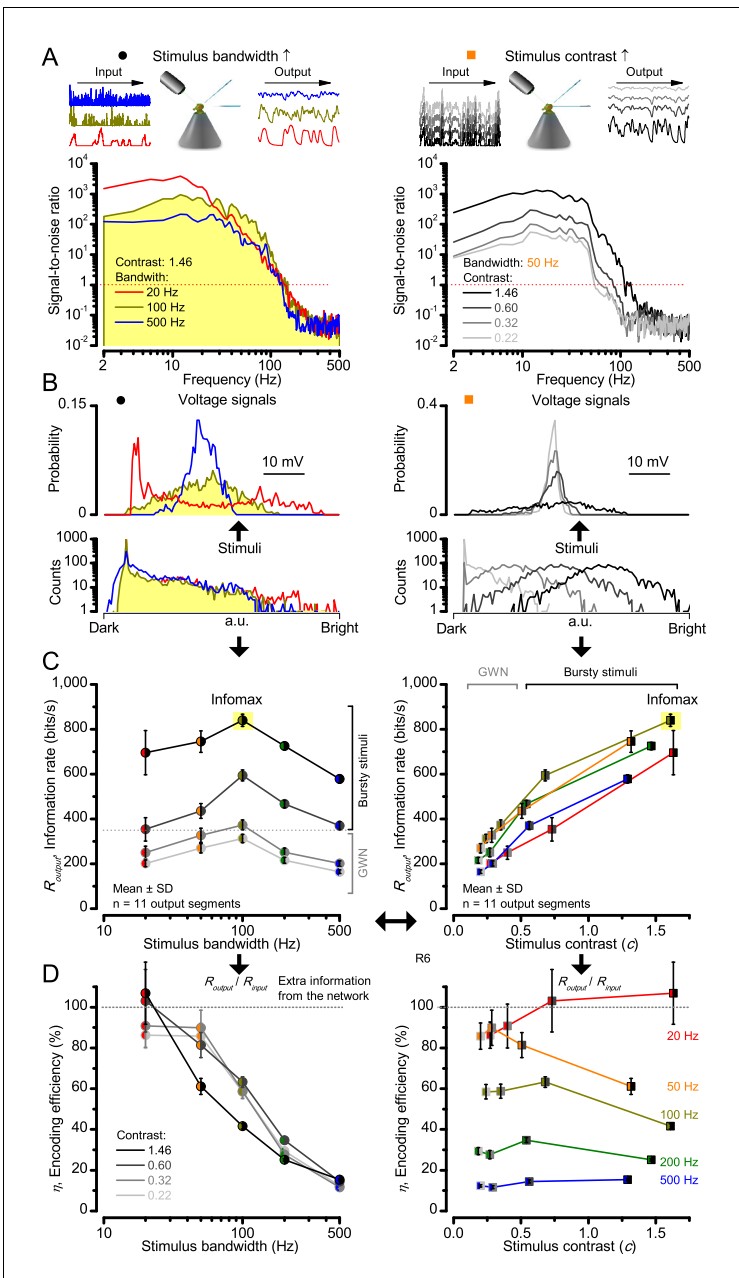

**Figure 2.** High-contrast bursts drive maximal encoding. A R1-R6's information transfer to high-frequency 100 Hz bursts exceeded 2-to-4-times the previous estimates. (**A**) Response signal-to-noise ratio (SNR, left) to 20 (red), 100 (yellow) and 500 Hz (blue) bursts, and to 50 Hz bandwidth stimuli of different contrasts (right); data from *Figure 1*. SNR increased with contrast (right), reaching the maximum (~6,000) for 20 Hz bursts (left, red) and the broadest frequency range for 100 Hz bursts (yellow). (**B**) Skewed bursts drove largely Gaussian responses (exception: 20 Hz bursts, red), with 100 Hz bursts evoking the broadest amplitude range (yellow). (**C**) Information transfer peaked for 100 Hz stimuli, irrespective of contrast (or BG; left), having the global maximum of ~850 bits/s (capacity, infomax) for the high-frequency high-contrast bursts. (**D**) Encoding efficiency, the ratio between input and output information ($R_{output}/R_{input}$), was > 100% for 20 Hz bursts. Extra information came from the neighboring cells. $R_{input}$ at each BG was determined for the optimal mean light intensity, which maximized a biophysically realistic photoreceptor model's information transfer (Appendix 2). Encoding efficiency fell with stimulus bandwidth but remained more constant with contrast. Population dynamics are in *Figure 2—figure supplement 1*.
DOI: https://doi.org/10.7554/eLife.26117.009

The following figure supplements are available for figure 2:

**Figure supplement 1.** Signaling performance vary cell-to-cell but adapts similarly to given stimulus statistics.
*Figure 2 continued on next page*

**eLIFE** Research article
Computational and Systems Biology | Neuroscience

*Figure 2 continued*

DOI: https://doi.org/10.7554/eLife.26117.010

**Figure supplement 2.** Light-adapted R1-R6 noise is similar for all the test stimuli, with its high-frequencies reflecting the mean quantum bump shape and its low-frequencies the rhabdomere jitter.

DOI: https://doi.org/10.7554/eLife.26117.011

**Figure supplement 3.** Strong responses to naturalistic stimulation (NS) carry only about half the information of the strongest responses to bursts.

DOI: https://doi.org/10.7554/eLife.26117.012

**Figure supplement 4.** *Drosophila* R1-R6 photoreceptor output information transfer rate estimates to bursty stimuli are consistent.

DOI: https://doi.org/10.7554/eLife.26117.013

Moreover, as their rhabdomere sizes (*Figure 5A–B*) and connectivity vary systematically (*Rivera-Alba et al., 2011*), each R1-R6 receives different amounts of information (*Figure 5C–D*) (see also: *Wardill et al., 2012*). Here, R6s, with large rhabdomeres (*Figure 5B*) and gap-junctions to R8 (*Figure 5C*), should receive the most (*Wardill et al., 2012*), suggesting that the best performing cells (*e.g. Figures 1* and *2*) might be of the R6-type (*Figure 5E*). And yet whilst R7s also share gap-junctions with R6s (*Shaw et al., 1989*), our stimuli contained little UV component to drive them.

Encoding efficiency for the different stimuli (*Figure 2D* and *Figure 2—figure supplement 1D*) was determined as the ratio between the related photoreceptor and light information rates ($R_{output}$/$R_{input}$); with $R_{input}$ estimated from the simulated Poisson stimulus repeats, which maximized information in R1-R6 model output (*Figures 3B* and *4C*; Appendix 2). Thus, as $R_{input}$ included the photon loss by the intracellular pupil and other structural adaptations (*Howard et al., 1987*; *Song and Juusola, 2014*), it was less than at the light source. Moreover, *in vivo*, the combined stimulus information captured simultaneously by other photoreceptors in the retina network must be more than that by a single R1-R6 (*Zheng et al., 2006*). *E.g.* as summation reduces noise, the signal-to-noise of a postsynaptic interneuron, LMC, which receives similar inputs from six R1-R6s, can be $\sqrt{6}$-times higher than that of a R1-R6 (*de Ruyter van Steveninck and Laughlin, 1996*; *Zheng et al., 2006*), but lower than what is broadcasted from the source (*Song and Juusola, 2014*). Thus, information is lost during sampling and processing, with the analysis obeying data processing theorem (*Shannon, 1948*; *Cover and Thomas, 1991*). Finally, as the LED light source's photon emission statistics were untested (if sub-Poisson, $R_{input}$ would be higher), the efficiency estimates represented the theoretical upper bounds.

We found that encoding efficiency for both the recordings (*Figure 2D* and *Figure 2—figure supplement 1D*) and simulations (*Figure 4D*) weakened with the increasing bandwidth (left) but less so with contrast (right). This was because $R_{input}$ estimates (Appendix 2) increased monotonically with bandwidth (*Song and Juusola, 2014*) and contrast, while $R_{output}$ for bandwidth did not (*Figure 2C*). However, as predicted, some recordings showed >100% efficiency for 20 Hz bursts, presumably due to their extra network information (*Figure 5* and *Figure 5—figure supplement 1*) (*Zheng et al., 2006*; *Wardill et al., 2012*; *Dau et al., 2016*).

A locomoting *Drosophila* generates ~1–5 head/body-saccades/s, which direct its gaze in high velocities to the surroundings (*Fry et al., 2003*; *Geurten et al., 2014*). Here, our recordings and simulations suggested that the refractoriness in R1-R6s' phototransduction, together with network inputs, might be tuned for capturing information during such fast light changes in time.

## Saccades and fixations increase information capture from natural scenes

To test this idea more directly, we used published body yaw velocities (*Geurten et al., 2014*) of a walking *Drosophila* (*Figure 6A*) to sample light intensity information from natural images (of characteristic $1/f$-statistics [*van Hateren, 1997a*]) (*Figure 6B*). This resulted in time series of contrasts (*Figure 6C*, blue) that (i) mimicked light input to a R1-R6 photoreceptor during normal visual behavior, containing fixations, translational movements and saccadic turns. As controls, we further used light inputs resulting from corresponding (ii) linear median (red) and (iii) shuffled (gray) velocity walks across the same images (*Video 1*). These stimuli were then played back to R1-R6s in intracellular experiments and stochastic refractory model simulations.

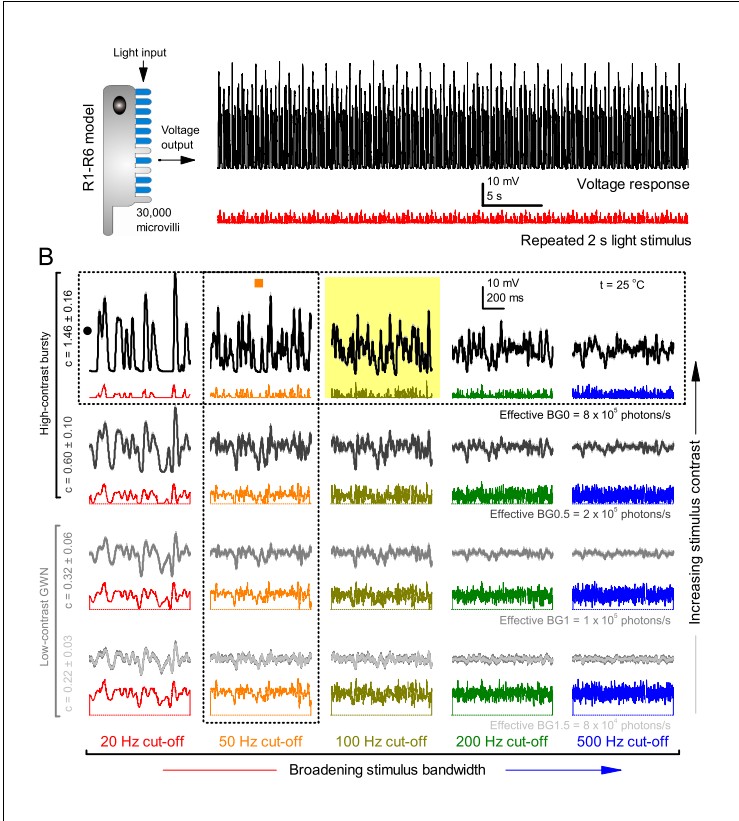

**Figure 3.** Model output is realistic over the whole encoding range. (**A–B**) Simulated responses of a stochastic *Drosophila* R1-R6 model to the tested light stimuli show similar response dynamics to the corresponding real recordings (*cf. Figure 1* and *Figure 1—figure supplement 1*). The model has 30,000 microvilli (sampling units) that convert absorbed photons to quantum bumps (samples). Simulations at each background (BG) were set for the mean light level (effective or absorbed photons/s) that generated responses with the maximum information transfer (Appendix 2). These effective light levels should correspond to the optimal photomechanical screening throughput (by intracellular pupil mechanism and photomechanical rhabdomere contractions, Appendix 7), which minimize saturation effects (refractory microvilli) on a *Drosophila* photoreceptor; see *Figure 4*. Notice that the model had no free parameters - it was the same in all simulations and had not been fitted to data. Thus, these macroscopic voltage responses emerged naturally as a by-product of refractory information sampling by 30,000 microvilli. Yellow box: maximum information responses. Vertical dotted rectangle (orange square) and horizontal rectangle (black circle): responses for contrast and bandwidth analyses in *Figure 4*.
DOI: https://doi.org/10.7554/eLife.26117.014

The following source data is available for figure 3:

**Source data 1.** Simulated voltage responses of a biophysically realistic R1-R6 photoreceptor model to very bright 20 Hz, 50 Hz, 100 Hz, 200 Hz and 500 Hz bursty light stimuli at BG0 (darkness).
DOI: https://doi.org/10.7554/eLife.26117.015

**Source data 2.** Simulated voltage responses of a biophysically realistic R1-R6 photoreceptor model to very bright 20 Hz, 50 Hz, 100 Hz, 200 Hz and 500 Hz bursty light stimuli at BG0.5.
DOI: https://doi.org/10.7554/eLife.26117.016

**Source data 3.** Simulated voltage responses of a biophysically realistic R1-R6 photoreceptor model to very bright 20 Hz, 50 Hz, 100 Hz, 200 Hz and 500 Hz GWN light stimuli at BG1.
DOI: https://doi.org/10.7554/eLife.26117.017

**Source data 4.** Simulated voltage responses of a biophysically realistic R1-R6 photoreceptor model to very bright 20 Hz, 50 Hz, 100 Hz, 200 Hz and 500 Hz GWN light stimuli at BG1.5.
DOI: https://doi.org/10.7554/eLife.26117.018

We found that saccadic viewing of natural images (*Figure 6C,i*), even without visual selection (*i.e.* without the fly choosing what it gazes), transformed the resulting light input to resemble the bursty

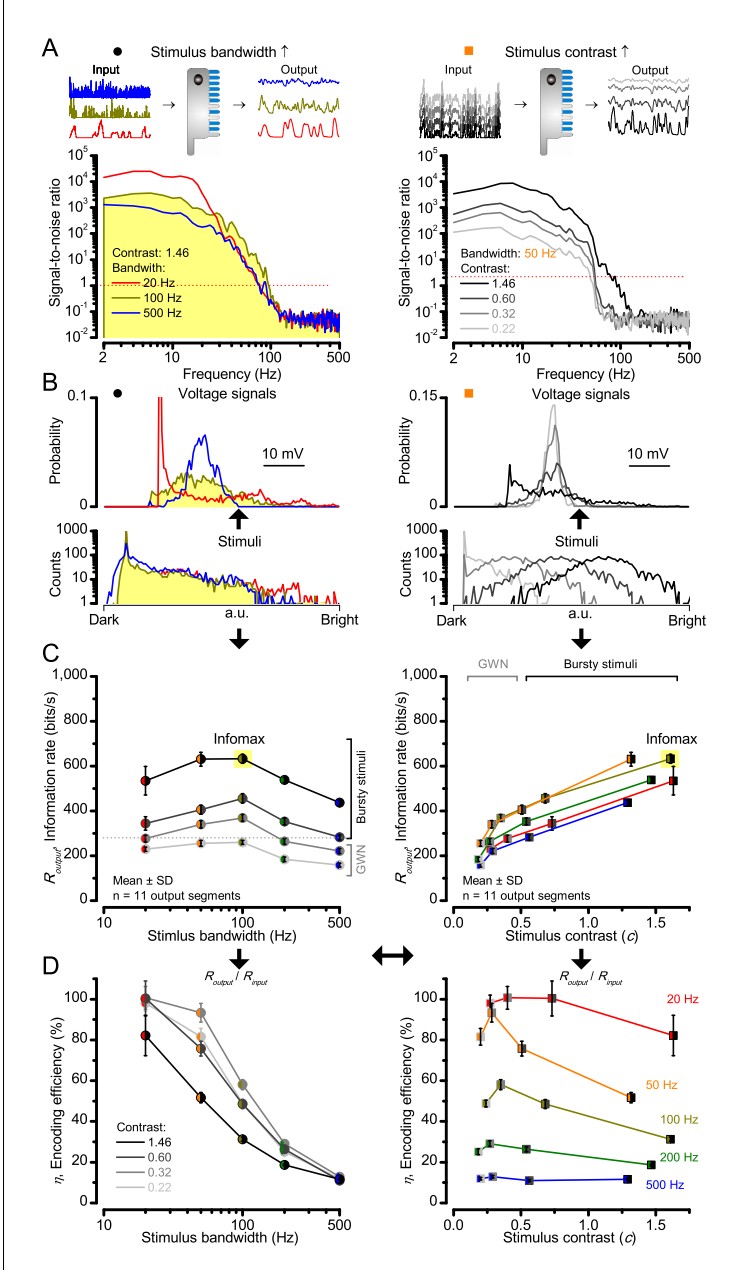

**Figure 4.** Model encodes light information realistically. Encoding capacity of a stochastic photoreceptor model peaks to high-contrast bursts with 100 Hz cut-off, much resembling that of the real recordings (*cf. Figure 2*). (**A**) Inserts show simulations for the bursty input patterns of similar high contrast values (left) that drove its responses (outputs) with maximum information transfer rates. Output signal-to-noise ratios peaked for 20 Hz bursts (red), but was the broadest for 100 Hz bursts (yellow). Signal-to-noise ratio rose with stimulus contrast (right). (**B**) The corresponding probability density functions show that 100 Hz bursts evoked responses with the broadest Gaussian amplitude distribution (yellow). Only responses to low-frequency bursts (20–50 Hz) deviated from Gaussian (skewed). (**C**) Information transfer of the model output reached its global maximum (infomax) of 632.7 ± 19.8 bits/s (yellow) for 100 Hz (left) bursts (right). Corresponding information transfer for Gaussian white-noise stimuli was significantly lower. (**D**) Encoding efficiency peaked for low-frequency stimuli (left), decaying gradually with increasing contrast. For details see Appendix 2.

DOI: https://doi.org/10.7554/eLife.26117.019

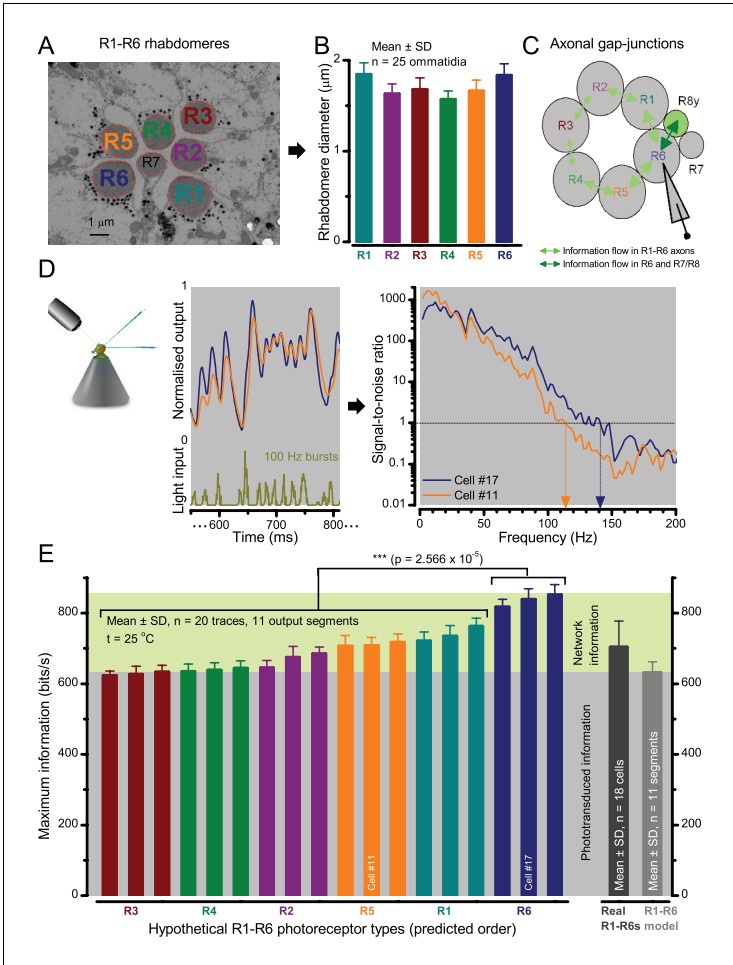

**Figure 5.** Each R1-R6 has a different diameter rhabdomere and network connections, and thus should extract different amounts of information from the same stimulus. (**A**) Electron micrograph of an ommatidium, showing R1-R7 rhabdomeres with characteristic cross-sectional area differences. (**B**) R1 and R6 rhabdomeres are always the largest and R4 the smallest (statistics in Appendix 5, *Appendix 4—table 1*). (**C**) R6 can receive ~ 200 bits/s of network information through axonal gap-junctions from R7/R8 (*Wardill et al., 2012*) in the lamina about local light changes - due to their neural superposition. Gap-junctions between R1-R6 axons and synapses (*Zheng et al., 2006*; *Rivera-Alba et al., 2011*) in the lamina redistribute information (Appendix 2). (**D**) R1-R6s' response waveforms and frequency range varied cell-to-cell; as evidenced by the recording system's low noise and the cells' high signal-to-noise ratios (~1,000). Here, Cell #17 encoded 100 Hz bursts reliably until ~ 140 Hz, but Cell #11 only until ~ 114 Hz. See also *Figure 5—figure supplement 1*. (**E**) Maximum information (for 100 Hz bursts) of 18 R1-R6s, grouped in their predicted ascending order and used for typifying the cells. Because R6s' rhabdomeres are large (**B**), and their axons communicate with R7/R8 (**C**), the cells with the distinctive highest infomax were likely this type (blue). Conversely, R3, R4 and R2 rhabdomeres are smaller and their axons furthest away from R7/R8, and thus they should have lower infomaxes. Notably, our photoreceptor model (*Song et al., 2012*) (grey), which lacked network information, had a similar infomax. The mean infomax of the recordings was 73 bits/s higher than the simulation infomax.

DOI: https://doi.org/10.7554/eLife.26117.020

The following figure supplement is available for figure 5:

**Figure supplement 1.** R1-R6 photoreceptors' response waveforms and frequency range of reliable encoding vary cell-to-cell, and this variation does not reflect recording quality.
DOI: https://doi.org/10.7554/eLife.26117.021

---

high-contrast stimulation (*Video 1*), which maximized photoreceptor information (*Figures 1–2*). Such inputs had increasingly sparse light intensity difference (first derivative) distributions in respect to those of the linear walks or GWN stimulation (*Figure 6D–E*; Appendix 3). Specifically, the saccadic

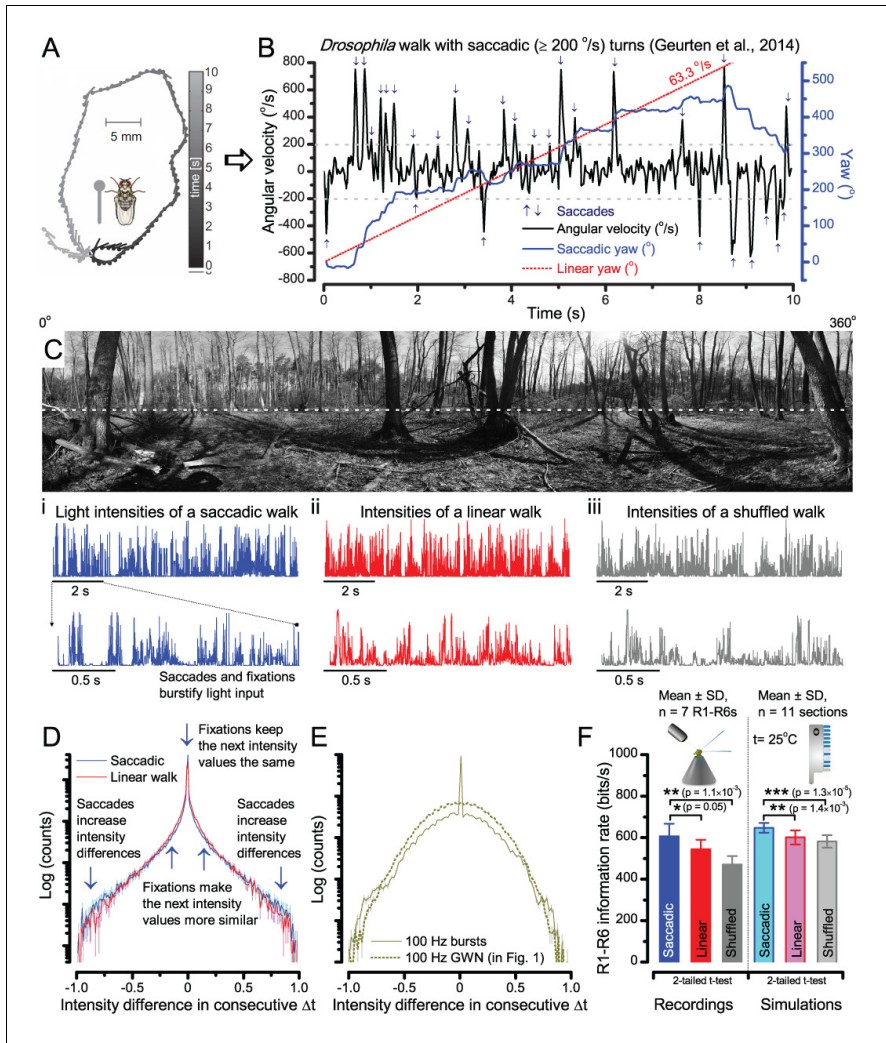

**Figure 6.** A *Drosophila*'s saccadic turns and fixation periods generate bursty high-contrast time series from natural scenes, which enable R1-R6 photoreceptors (even when decoupled from visual selection) to extract information more efficiently than what they could by linear or shuffled viewing. (**A**) A prototypical walking trajectory recoded by *Geurten et al. (2014)* (**B**) Angular velocity and yaw of this walk. Arrows indicate saccades (velocity ≥ |±200| °/s). (**C**) A 360° natural scene used for generating light intensity time series: (i). ) by translating the walking fly's yaw (**A**–**B**) dynamics on it (blue trace), and (ii) by this walk's median (linear: 63.3 °/s, red) and (iii) shuffled velocities. Dotted white line indicates the intensity plane used for the walk. Brief saccades and longer fixation periods 'burstify' light input. (**D**) This increases sparseness, as explained by comparing its intensity difference (first derivative) histogram (blue) to that of the linear walk (red). The saccadic and linear walk histograms for the tested images (Appendix 3; six panoramas each with 15 line-scans) differed significantly: $Peak_{sac}$ = 4478.66 ± 1424.55 vs $Peak_{lin}$ = 3379.98 ± 1753.44 counts (mean ± SD, p=1.4195 × $10^{-32}$, pair-wise t-test). $Kurtosis_{sac}$ = 48.22 ± 99.80 vs $Kurtosis_{lin}$ = 30.25 ± 37.85 (mean ± SD, p=0.01861, pair-wise t-test). (**E**) Bursty stimuli (in *Figure 1*, continuous) had sparse intensity difference histograms, while GWN (dotted) did not. (**F**) Saccadic viewing improves R1-R6s' information transmission, suggesting that it evolved with refractory photon sampling to maximize information capture from natural scenes. Details in Appendix 3.

DOI: https://doi.org/10.7554/eLife.26117.022

The following figure supplement is available for figure 6:

**Figure supplement 1.** *Drosophila* R1-R6 photoreceptors generate responses with higher information transfer rates to saccadic (bursty) naturalist light intensity time series (NS) than to corresponding linear or shuffled stimulation.
DOI: https://doi.org/10.7554/eLife.26117.023

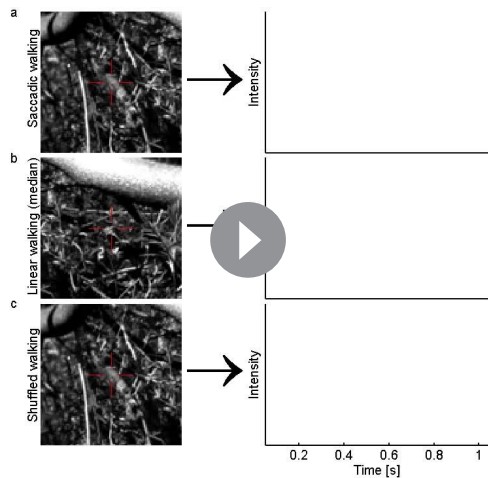

**Video 1.** Using a *Drosophila*'s saccadic walk to extrapolate light input to its photoreceptors from natural scenes. A published recording of a fly's saccadic walk (*Geurten et al., 2014*) was used to sample light intensity values from 360° panoramic images of natural scenes. We collected three types of light stimuli, resulting from: (a) saccadic, (b) median (linear) and (c) shuffled velocities of the walk. DOI: https://doi.org/10.7554/eLife.26117.024

walks contained fixation periods that retained the same light input values for longer durations than the linear walks, which lacked these periods, causing the ~63% higher peak in the saccadic histogram (*Figure 6D*). Saccadic walking also enhanced the proportion of large intensity differences between two consecutive moments, seen as ~18% higher histogram flanks than those for linear walking (p=3.65 × 10⁻⁹, pair-wise t-test for the combined 0.5–1.0 and ⁻0.5-⁻1 ranges). These dynamics drove refractory sampling efficiently (*Song and Juusola, 2014*), enabling a R1-R6 to better utilize its output range, and thus capture more information than through the median or shuffled velocity viewing (*Figure 6F*; *Figure 6—figure supplement 1*; cf. *Figure 2—figure supplement 3*).

Altogether, these results (*Figures 1–6*) imply that saccades and fixations improve a R1-R6's neural representation of the world in time. Furthermore, as behaviors modulate visual inputs in a sensorimotor-loop, bursty spike trains from the brain (*Franceschini et al., 1991*; *Franceschini and Chagneux, 1994*; *Tang and Juusola, 2010*), which direct the gaze through self-motion, may have evolved with photoreceptors' information sampling dynamics to better detect changes in the world. So when a freely-moving fly directs its gaze to visual features that are relevant for its behavior, its R1-R6's information capture may become optimized for the imminent task.

However, visual behaviors should also affect spatial acuity (*Srinivasan and Bernard, 1975*; *Juusola and French, 1997*; *Land, 1997*; *Geurten et al., 2014*). Hence, we next asked how R1-R6s see saccadic light changes over space.

## Testing acuity at saccadic velocities

A *Drosophila*'s head/body-saccades generate fast phasic photoreceptor movements, which ought to blur retinal images (*Srinivasan and Bernard, 1975*; *Juusola and French, 1997*; *Land, 1997*). Moreover, saccades – when dominated by axial rotation - provide little distance information (*Land, 1999*) because objects, near and far, would move across the retina with the same speed. Therefore, it has been long thought that visual information is mostly captured during translational motion and gaze fixation, and less during saccades.

To test this hypothesis, we reasoned that object motion and self-motion shape a photoreceptor's light input the same way. Thus, the influence of eye movements (and motion blur) on a R1-R6's ability to resolve objects could be measured in experiments, where, instead of moving the eye, the objects were moved over its stationary receptive field (*Figure 7A*; Appendixes 4–6).

Using this approach, we recorded individual R1-R6s' voltage responses (*Figure 7B*; black traces) to a pair of bright dots (each 1.7° in size and 6.8° apart, as seen by the fly), moving at constant speed across their receptive field in front-to-back direction. The movements were either fast (205 °/s) or double-fast (409 °/s), both within the head/body saccadic velocity range of a walking *Drosophila* (*Figure 6A–B*: 200–800 °/s) (*Geurten et al., 2014*), and were presented against a dark or lit background (note: during a free flight (*Fry et al., 2003*), saccadic velocities may reach 2,000 °/s). Importantly, the dots' angular separation was less than the half-width of a R1-R6's receptive field (*Figure 7C*) at the two backgrounds (Δρ$_{dark}$ = 9.47 ± 1.57°, n = 19 cells; Δρ$_{light}$ = 7.70 ± 1.27°, n = 6; mean ± SD; *Figure 7—figure supplements 1* and *2*) and 1.5-times the average interommatidial angle (Δφ ~ 4.5°), which should determine *Drosophila*'s visual acuity (*Gonzalez-Bellido et al., 2011*).

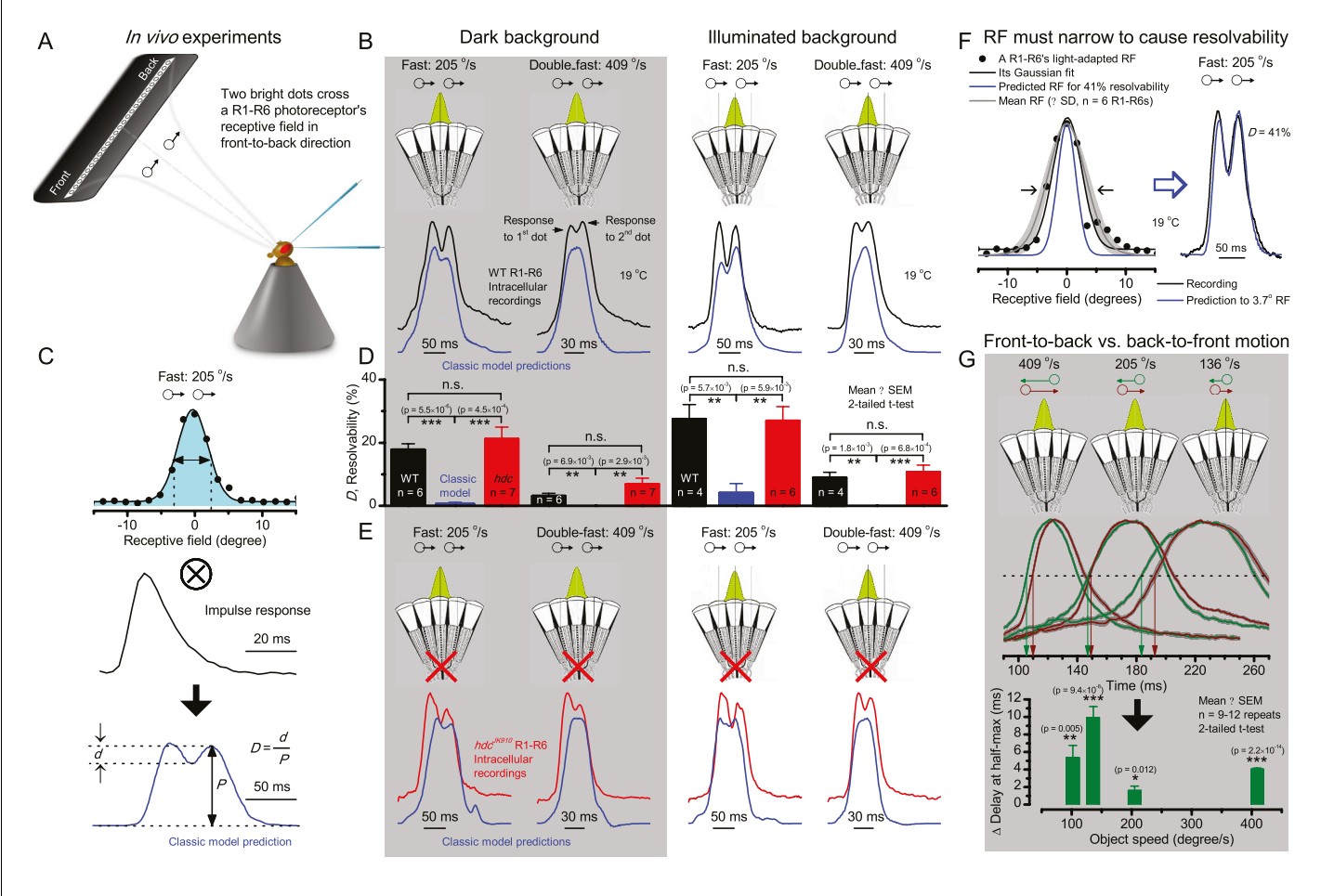

**Figure 7.** Photoreceptors resolve dots at saccadic velocities far better than the classic models. (**A**) 25-light-point stimulus array centered at a R1-R6's receptive field (RF). Each tested photoreceptor saw two bright dots, 6.8° apart, travelling fast (205 °/s) or double-fast (409 °/s) in front-to-back direction. (**B**) Responses (black), both at dark (left) or illuminated backgrounds (right), characteristically showed two peaks. In contrast, the corresponding classic model simulations (blue) rarely resolved the dots. (**C**) In the simulations, each photoreceptor's receptive field (or its Gaussian fit) was convolved with its impulse response (first Volterra kernel). The resolvability, *D*, of the recordings and simulations, was determined by Raleigh criterion. (**D**) Recordings outperformed simulations. (**E**) *hdc*^JK910^ R1-R6s (red), which lacked the neurotransmitter histamine, and so network modulation, resolved the dots as well as the wild-type, indicating that the recordings' higher resolvability was intrinsic and unpredictable by the classic models (Appendix 6). (**F**) To resolve the two dots as well as a real R1-R6 does in light-adaptation, the model's acceptance angle (Δρ) would need to be ≤3.70° (blue trace); instead of its experimentally measured value of 5.73 (black; the narrowest Δρ. The population mean, grey, is wider). (**G**) Normalized responses of a typical R1-R6 to a bright dot, crossing its receptive field in front-to-back or back-to-front at different speeds. Responses to back-to-front motions rose and decayed earlier, suggesting direction-selective encoding. This lead at the half-maximal values was 2–10 ms. See Appendixes 4 and 6.

DOI: https://doi.org/10.7554/eLife.26117.025

The following figure supplements are available for figure 7:

**Figure supplement 1.** Dark-adapted wild-type and *hdc*^JK910^ R1-R6s' acceptance angles differ marginally.
DOI: https://doi.org/10.7554/eLife.26117.026

**Figure supplement 2.** Light-adaptation narrows wild-type and *hdc*^JK910^ R1-R6s' receptive fields similarly.
DOI: https://doi.org/10.7554/eLife.26117.027

Thus, these fast-moving point objects tested the theoretical limit of what a R1-R6 should be able to resolve.

We further estimated each cell's respective impulse response (Appendix 6). Then following the classic theory of compound eyes' resolving power (*Srinivasan and Bernard, 1975*; *Juusola and French, 1997*; *Land, 1997*), we calculated each R1-R6's expected voltage output to the moving dots by convolving its impulse response with its measured dark- or light-adapted receptive field.

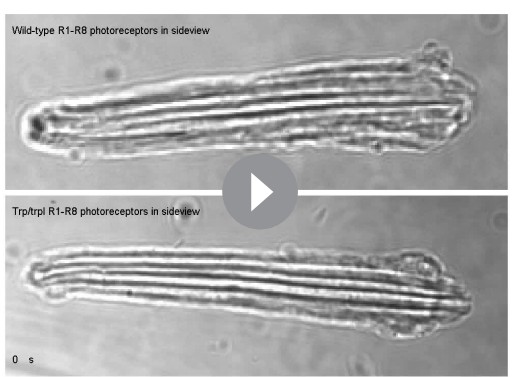

**Video 2.** *Drosophila* R1-R8s in dissociated ommatidia contract photomechanically. Wild-type and *trp/trpl*-mutant R1-R8 photoreceptors contract photomechanically to light flashes. The panels show: top, a sideview of ex vivo wild-type R1-R8 photoreceptors of a single dissociated ommatidium contracting to 1 ms bright light flash; bottom left, R1-R8 of a *trp/trpl* null-mutant, which express normal phototransduction reactants but lack completely their light-gated ion channels, contracting to a similar flash. Notably, *trp/trpl* photoreceptors cannot generate electrical responses to light, with their eyes showing no ERG signal (Appendix 7). Nonetheless, *trp/trpl*-mutant photoreceptors contract photomechanically (but require ~5 min dark-adaptation between flashes to restore their contractility). These observations are consistent with the hypothesis of the light-induced phosphatidylinositol 4,5-bisphosphate (PIP$_2$) cleaving from the microvillar photoreceptor plasma membrane causing the rhabdomere contractions (*Hardie and Franze, 2012*). Video playback slowed down and down-sampled to reveal the contractions, which otherwise would be too fast to see with a naked eye. Each video clip is repeated three times with a running timer giving the time course of the contractions. Notice that the longitudinal contractions reduce the photoreceptor length. Thus, in an intact compound eye, the rhabdomeres would move inwards, away from the lens, likely narrowing their receptive fields (see Appendix 7, *Appendix 7—figure 10* and Appendix 8, *Appendix 8—figure 3*).

DOI: https://doi.org/10.7554/eLife.26117.028

These Volterra-model (*Juusola and French, 1997*) predictions (*Figure 7B–C*; blue) were then compared to the actual recordings (black).

## Eyesight beyond the motion blur-limit

Remarkably in all these tests, the recordings showed distinctive responses to the two dots (*Figure 7B*), as two peaks separated by a trough. The relative magnitude of this amplitude separation was quantified as resolvability, using the Raleigh criterion (*Juusola and French, 1997*) (*Figure 7C*). However, in marked contrast, the model predictions failed to resolve the double-fast dots, instead blurring into one broad response in both adapting states (*Figure 7D*; blue vs. black bars, respectively). The predictions for the fast dots were also poorer than the measured responses. Thus, a photoreceptor's real resolving power was significantly better and less affected by motion blur than predicted by classic theory (Appendix 6).

We next asked whether this better-than-expected resolving power resulted from synaptic interactions (*Zheng et al., 2006*; *Freifeld et al., 2013*) by using *hdc*$^{JK910}$ mutants (*Figure 7E*, red traces), in which photoreceptors lacked their neurotransmitter, histamine (*Burg et al., 1993*) (Appendixes 4–6). Because *hdc*$^{JK910}$ R1-R6s cannot transmit information to their post-synaptic targets (*Dau et al., 2016*) (LMCs, which initiate the motion detection pathways (*Joesch et al., 2010*), and the amacrine cells), neither could these photoreceptors receive any light-driven interneuron feedback modulation (*Dau et al., 2016*). Therefore, if the synaptic interactions improved the wild-type output to the moving dots, then *hdc*$^{JK910}$ R1-R6s, which lacked these interactions, should show diminished resolvability. But this was never observed. Instead, we found that *hdc*$^{JK910}$ R1-R6s resolved the dots at least equally well as the wild-type (*Figure 7D*, red). Thus, high acuity did not result from synaptic inputs but was intrinsic to photoreceptors.

We also calculated Δρ needed to explain the spatial acuity of the recordings. The example (*Figure 7F*) is from a R1-R6, which had the narrowest light-adapted receptive field (Δρ = 5.73°) (*Figure 7—figure supplement 2*). Its response resolved the two fast-moving dots with a 40.5% dip. However, the Volterra model prediction, using its receptive field, only resolved the dots with a 12.5% dip (*cf. Figure 7D*). In fact, for 41.0% resolvability, its Δρ would need to narrow to 3.70° (from 5.73°). Thus, for the prediction to match the recording, the receptive field would have to narrow at least by one-third. Because the required (predicted) acceptance angles of R1-R6s were always much narrower ($\leq$ 4°) than the actual measurements (Δρ$_{dark}$ = 9.47 and Δρ$_{light}$ = 7.70; see above), measurement bias cannot explain this disparity.

We further discovered that R1-R6 recordings often showed phasic directional selectivity (*Figure 7G*), with the responses rising and decaying faster to back-to-front than to front-to-back moving dots. We asked whether these lag-time differences originated from asymmetric

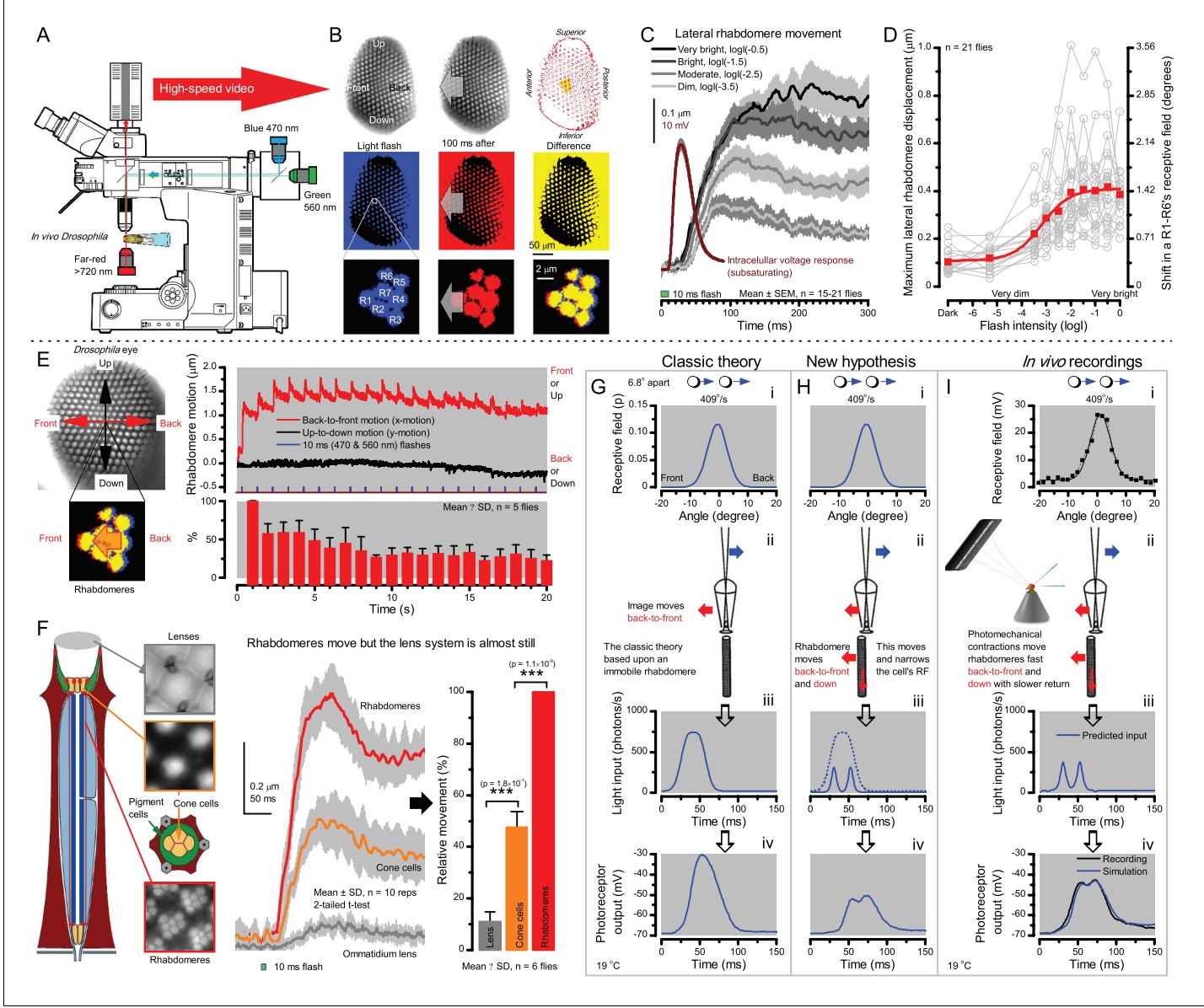

**Figure 8.** Microsaccadic rhabdomere contractions and refractory photon sampling improve visual resolution of moving objects. (**A**) High-speed videos showed fast lateral R1-R7 rhabdomere movements to blue/green flashes, recorded under far-red illumination that *Drosophila* barely saw (*Wardill et al., 2012*). (**B**) Rhabdomeres moved inside those seven ommatidia (up-right: their pseudopupil), which faced and absorbed the incident blue/green light, while the others reflected it. Rhabdomeres moved frontwards 8–20 ms after a flash onset, being maximally displaced 70–200 ms later, before returning. (**C**) Movements were larger and faster the brighter the flash, but slower than R1-R6s' voltage responses. (**D**) Movements followed R1-R6s' logarithmic light-sensitivity relationship. Concurrently, given the ommatidium optics (*Stavenga, 2003b*; *Gonzalez-Bellido et al., 2011*), R1-R6s' receptive fields (RFs) shifted by 0.5–4.0°. (**E**) Rhabdomeres moved along the eye's horizontal (red) axis, with little vertical components (black), adapting to ~ 30% contractions in ~ 10 s during 1 s repetitive flashing. (**F**) Moving ommatidium structures. Cone and pigment cells, linking to the rhabdomeres by adherens-junctions (*Tepass and Harris, 2007*), formed an aperture smaller than the rhabdomeres' pseudopupil pattern. Rhabdomeres moved ~ 2 times more than this aperture, and ~ 10 times more than the lens. (**G–H**) Simulated light inputs and photoreceptor outputs for the classic theory and new 'microsaccadic sampling'-hypothesis when two dots cross a R1-R6's RF (**i**) front-to-back at saccadic speeds. (**G**) In the classic model, because the rhabdomere (**ii**) and its broad RF (**i**) were immobile (**ii**), light input from the dots fused (**iii**), making them neurally unresolvable (**iv**). (**H**) In the new model, with rhabdomere photomechanics (**ii**) moving and narrowing its RF (here acceptance angle, Δρ, narrows from 8.1° to 4.0°), light input transformed into two intensity spikes (**iii**), which photoreceptor output resolved (**iv**). (**I**) New predictions matched recordings (*Figure 8—figure supplement 1*). Details in Appendixes 7–8.

DOI: https://doi.org/10.7554/eLife.26117.029

The following figure supplement is available for figure 8:

*Figure 8 continued on next page*

*Figure 8 continued*

**Figure supplement 1.** Microsaccadic sampling hypothesis predicts realistic voltage output to two bright dots crossing a R1-R6's receptive field in saccadic speeds.

DOI: https://doi.org/10.7554/eLife.26117.030

photomechanical photoreceptor contractions. Namely, atomic-force microscopy has revealed minute (<275 nm) vertical movements on the surface of dissected *Drosophila* eyes, generated by contraction of individual microvilli as $PIP_2$ is hydrolyzed from the inner leaflet of the lipid bilayer (*Hardie and Franze, 2012*). Here, we reasoned that if the ommatidium lenses were effectively fixed and R1-R8s levered to the retinal matrix, the contractions (*Video 2*) might be larger in situ, moving and shaping the photoreceptors' receptive fields along some preferred direction. Such mechanical feedback could then reduce light input to R1-R8s, making it more transient and directional.

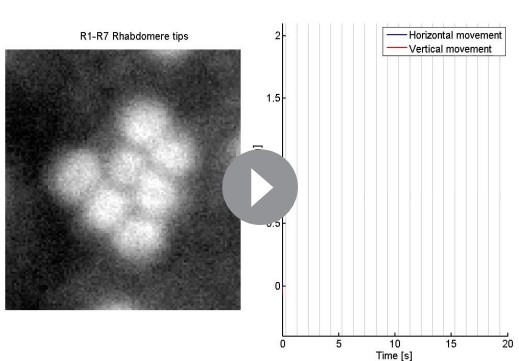

**Video 3.** *Drosophila* R1-R8 photoreceptors contract photomechanically *in vivo*, moving back-to-front inside each observed ommatidium. We utilized the optical cornea-neutralization technique with antidromic deep-red (740 or 785 nm peak) illumination to observe deep pseudopupils (photoreceptor rhabdomeres that align with the observer's viewing axis) in the *Drosophila* eye. High-speed video captures fast rhabdomere movements to bright orthodromic blue-green flashes (470 + 535 nm peaks). The panels show: left, R1-R7 photoreceptor rhabdomere tips moving rapidly back-to-front and returning slower to each 10 ms flash, delivered repeatedly every second; right, the cross-correlated horizontal (blue) and vertical (red) components as the time series of this movement. Grey vertical lines indicate each flash. The rhabdomere movement is caused by the photomechanical photoreceptor contractions (not by muscle activity). These *in vivo* movements are large, here 1.7 µm from dark-adapted rest-state; causing up to five degree transient shift in the R1-R6 photoreceptors receptive fields (Appendix 7). Note average diameter of R1-R6 rhabdomeres is 1.7 µm (Appendix 5). The high-speed video rate was 500 frames/s. Video playback slowed down and down-sampled to reveal the contractions, which otherwise would be too fast to see with a naked eye.

DOI: https://doi.org/10.7554/eLife.26117.031

## Microsaccadic sampling of retinal images

To probe this idea, we recorded *in vivo* high-speed videos of photoreceptor rhabdomeres (viewed by optical neutralization of the cornea) inside the eyes reacting to blue-green light flashes (470 + 560 nm) (*Figure 8A*). The recordings were performed under far-red (>720 nm) illumination, which is nearly invisible to *Drosophila* (Appendix 7).

We found that 8–20 ms after a flash the rhabdomeres, which directly faced the light source at the image center, shifted rapidly towards the anterior side of their ommatidia (*Figure 8B*). These local movements were faster and larger the brighter the flash (*Figure 8C*), and reached their intensity-dependent maxima (0.2–1.2 µm; *Figure 8D*) in 70–200 ms, before returning more slowly to their original positions (Appendix 7 analyses $hdc^{JK910}$-rhabdomere responses). Because the mean R1-R6 rhabdomere tip diameter is ~1.7 µm (*Figure 5B*), a bright flash could shift it more than its half-width sideways. Consequently, the fast rhabdomere movements, whilst still ~3 times slower than their voltage responses (*Figure 8C*, wine), adapted photoreceptors photomechanically by shifting their receptive fields by 0.5–4.0°, away from directly pointing to the light source.

Video footage at different eye locations indicated that light-activated rhabdomeres moved in back-to-front direction along the eye's equatorial (anterior-posterior) plane (*Figure 8E*, red; *Video 3*), with little up-down components (black). Therefore, as each ommatidium lens inverts projected images, the photoreceptors' receptive fields should follow front-to-back image motion. This global motion direction, which corresponds to a forward locomoting fly's dominant horizontal optic flow field, most probably explains the phasic directional selectivity we

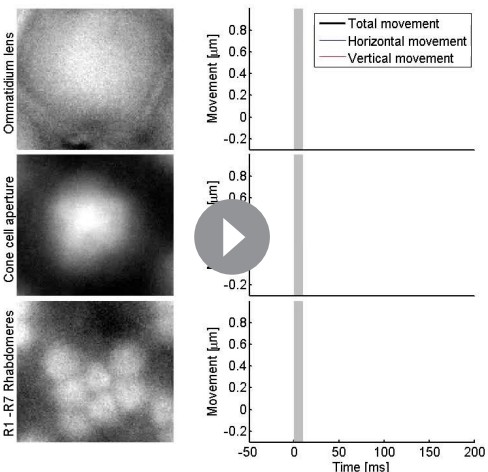

**Video 4.** While R1-R8s contract, the lens above is immobile but a cone-cell aperture, connected to the rhabdomere tips by adherens junctions, moves half as much as the rhabdomeres. We used a z-axis micromanipulator to shift and reposition *Drosophila* in piezo-steps vertically underneath the microscope. This allowed the focused image, as projected on the camera, to scan through each studied ommatidium, providing exact depth readings in μm. We then recorded any structural movements inside the ommatidia to light flashes at different depths; from the corneal lens down to the narrow base, where the cone and pigment cells form an intersection between the crystalline cone and the rhabdomere. The left panels show: up, ommatidium lens; middle, basal cone/pigment cell layer; down, R1-R7 photoreceptor rhabdomeres tips during and after flash stimulation. The right panels show the cross-correlation time series of these high-speed videos: up, the corneal lens and the upper ommatidium structures were essentially immobile), and normally remained so throughout the recordings; Middle, cone cells that connect to the rhabdomere tips with adherens junctions (*Tepass and Harris, 2007*) showed clear light-induced movements; down, R1-R7 rhabdomeres moved half as much as the cone cells above. The high-speed video rate was 500 frames/s. Video playback slowed down and down-sampled to reveal the contractions, which otherwise would be too fast to see with a naked eye.
DOI: https://doi.org/10.7554/eLife.26117.032

found to opposing image motions (*Figure 7F*; Appendix 8). Thus, the responses to back-to-front moving dots were faster because the dots entered and exited each contracting photoreceptor's front-to-back moving receptive field earlier; whereas the dots moving in the opposite direction stayed slightly longer inside each receptive field.

Video analyses further revealed that the first rhabdomere movement was the largest (*Figure 8E*), but 1 s dark intervals, as used in *Figure 7*, could resensitize the photoreceptors for the next (~0.5 μm) movements. Even <100 ms dark periods rescued noticeable motility (*Figure 2—figure supplement 2E*).

To inspect how rhabdomere contractions affected the cornea lens system's image projection, we scanned ommatidia by z-axis piezo steps, with the imaged focal plane travelling down from the lens surface into rhabdomeres (*Figure 8F*; *Video 4*), delivering flashes at predetermined depths. Crucially, we found that the ommatidium lens stayed nearly still, while specific pigment and cone cells, which are connected to the rhabdomere tips by adherens junctions (*Tepass and Harris, 2007*), formed a narrow aperture that moved with the rhabdomeres but only half as much. Thus, as the lens system was immobile but the aperture and sensors (rhabdomeres) underneath swung differentially, the light input to the moving rhabdomeres was shaped dynamically. This implied that, during saccadic image motion, R1-R6s' receptive fields might not only move but also narrow (Appendixes 7–8; *Video 2*).

Essentially, light input to a R1-R6 was modulated by the photoreceptor itself (*Figure 8F*). To estimate how these photomechanics influenced encoding, we implemented them in stochastic model simulations. We then compared how the predicted light inputs of the classic theory (*Figure 8G*) and the new 'microsaccadic sampling'-hypothesis (*Figure 8H*) would drive R1-R6 output during the saccadic dot stimulation.

In the classic theory, the rhabdomere is immobile (ii). Therefore, light input of two moving dots was a convolution of two broad Gaussians (i) that fused together (iii), making them irresolvable to phototransduction (iv); this also flawed the Volterra-models (*Figure 7*).

In the new hypothesis, instead, as microvilli became light-activated (ii), the rhabdomere contracted away from the focal point, and then returned back more slowly, recovering from refractoriness. And because its receptive field moved and narrowed concurrently (its acceptance angle, Δρ, halved to 4.0°), the light input of two moving dots transformed into two intensity peaks (iii), in which time-separation was enhanced by the rhabdomere's asymmetric motion. Crucially, with such input driving the refractory photon sampling model, its output (iv) closely predicted the responses to the two moving dots (*Figure 8I* and *Figure 8—figure supplement 1*). Interestingly, early behavioral

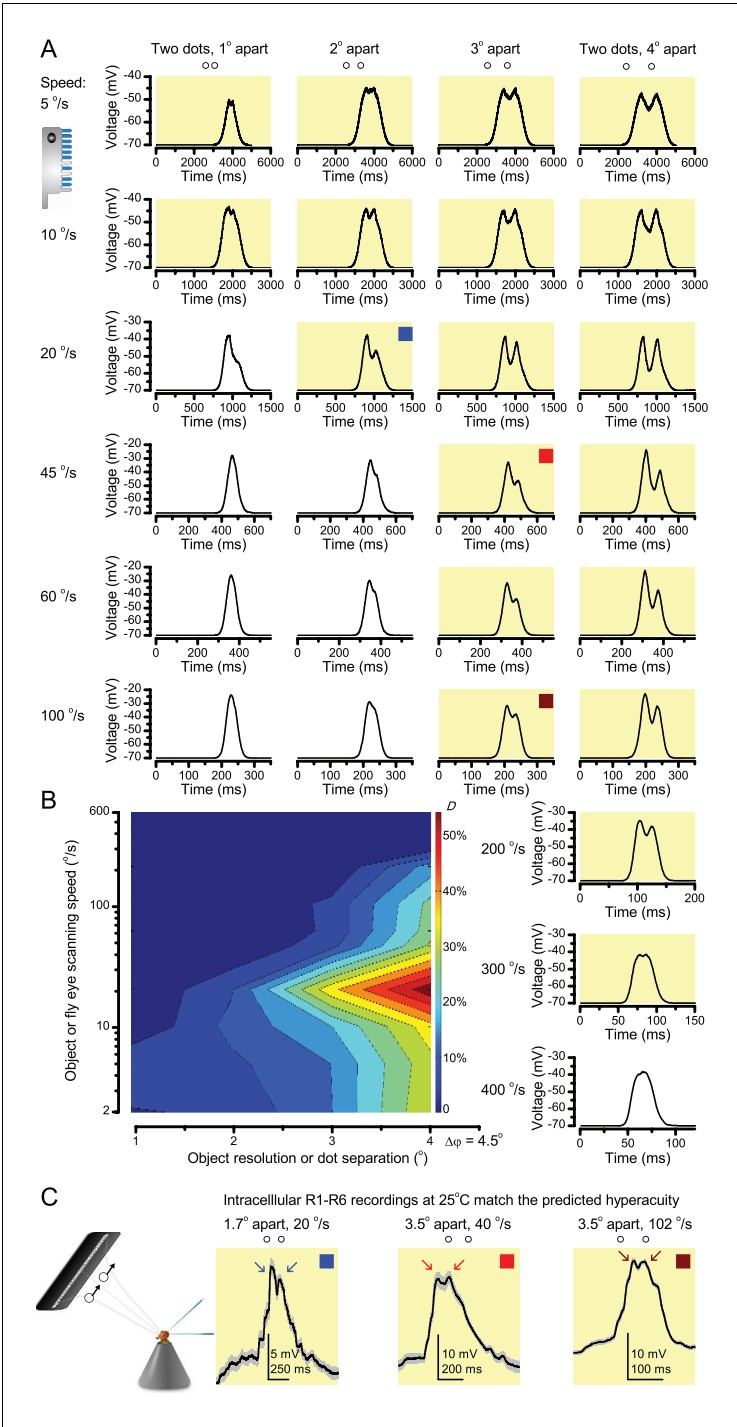

**Figure 9.** 'Microsaccadic sampling' hypothesis predicts visual hyperacuity. (**A**) Simulated R1-R6 output to two dots, at different distances apart, crossing the photoreceptor's receptive field at different speeds. The yellow backgrounds indicate those inter-dot-distances and speeds, which evoked two-peaked responses. The prediction is that the real R1-R6s could resolve (and *Drosophila* distinguish) these dots as two separate objects, whereas those on the white backgrounds would be seen as one object. The simulations were generated with our biophysically realistic R1-R6 model (**Song et al., 2012**; **Song and Juusola, 2014**; **Juusola et al., 2015**), which now included the estimated light input modulation by photomechanical rhabdomere movements (**Figure 8H**). (**B**) The resulting object resolution/speed heat-map, using the Raleigh criterion, *D* (**Figure 7C**), shows the stimulus/behavioral speed regime where *Drosophila* should have hyperacute vision. Thus, by adjusting its behavior (from gaze fixation to saccadic turns) to changing surroundings, *Drosophila* should see the world better than its

*Figure 9 continued on next page*

*Figure 9 continued*

compound eye's optical resolution. (**C**) Intracellular R1-R6 responses resolved the two dots, which were less that the interommatidial angle ($\Delta\varphi$ = 4.5°) apart when these crossed the cell's receptive field at the predicted speed range. Arrows indicate the two response peaks corresponding to the dot separation. Cf. ***Figure 9—figure supplement 1***; details in **Appendixes 7–8**. These results reveal remarkable temporal acuity, which could be used by downstream neurons (***Zheng et al., 2006***; ***Joesch et al., 2010***; ***Behnia et al., 2014***; ***Yang et al., 2016***) for spatial discrimination between a single passing object from two passing objects.

DOI: https://doi.org/10.7554/eLife.26117.033

The following figure supplement is available for figure 9:

**Figure supplement 1.** Encoding space in time - intracellular R1-R6 recordings to two bright dots crossing the receptive field show how their responses convey hyperacute spatial information in time.

DOI: https://doi.org/10.7554/eLife.26117.034

experiments in bright illumination (***Götz, 1964***) suggested similarly narrow R1-R6 acceptance angles (~3.5°).

## From microsaccades to hyperacuity

Because of the close correspondence between R1-R6 recordings and the new hypothesis (Appendixes 6–9), we used it further to predict whether *Drosophila* possessed hyperacute vision (***Figure 9***). We asked whether 'saccade-fixation-saccade'-like behaviors, when linked to refractory photon sampling and photomechanical photoreceptor contractions, allowed encoding in time finer spatial details than the compound eye's optical limit ($\Delta\varphi$ ~4.5°). R1-R6 output was simulated to two bright dots 1-4° apart, crossing its receptive field at different speeds at 25°C.

We found that if the dots, or a *Drosophila*, moved at suitable speed, a photoreceptor should resolve them well (***Figure 9A***), with this performance depending upon the inter-dot-distance. When the dots/eye moved at 10 °/s, a R1-R6 may capture image details at 1° resolution. But with slower movement ($\leq$2.5 °/s), adaptation should fuse the dots together, making them neurally unresolvable. Conversely, 3°-apart-dots should be seen at 5–100 °/s speeds and 4°-apart-dots even during fast saccades (200–300 °/s).

Thus, the 'microsaccadic sampling'-hypothesis implied that *Drosophila* had hyperacute vision over a broad speed range (***Figure 9B***), and through its own self-motion, could adjust the resolution of its neural images. Further comparisons of model outputs with and without refractoriness indicated that it extends the speed range of hyperacute vision (Appendix 8). Again, intracellular recordings corroborated these predictions (***Figure 9C*** and ***Figure 9—figure supplement 1***), demonstrating how acuity could be enhanced by encoding space in time.

These results meant that the unexpectedly fine temporal responses of R1-R6s (***Figures 7–9***) could be used by downstream neurons (***Zheng et al., 2006***; ***Joesch et al., 2010***; ***Rivera-Alba et al., 2011***; ***Wardill et al., 2012***; ***Behnia et al., 2014***), which can have even faster dynamics (***Juusola et al., 1995b***; ***Uusitalo et al., 1995***; ***Zheng et al., 2006***), for spatial discrimination between a single passing object from two passing objects, even if these objects were less than an interommatidial angle apart. The fly brain could then integrate information from hyperacute moving objects and use it for directing behaviors.

## Optomotor behavior confirms hyperacute vision

To test this prediction, we investigated the spatial resolution of *Drosophila* vision through their optomotor behavior in a conventional flight simulator system, which used brightly-lit high-resolution prints for panoramic scenes (***Figure 10***; Appendix 10). We asked whether tethered *Drosophila* possessed motion vision hyperacuity by recording their yaw torque (optomotor response) to vertical black-and-white bar panoramas with <4.5° wavelengths, which slowly rotated (45 °/s) to clockwise and counterclockwise.

We found that every tested fly responded down to ~1° panoramic bar resolution (***Figure 10A*** and ***Figure 10—figure supplement 1***) with their responses becoming smaller the finer its bars (***Figure 10A–C***). Importantly, because these responses consistently followed the rotation direction changes, they were not caused by aliasing. Thus, optomotor behavior verified that *Drosophila* see the world at least in 4-fold finer detail than what was previously thought. Moreover, when a fine-

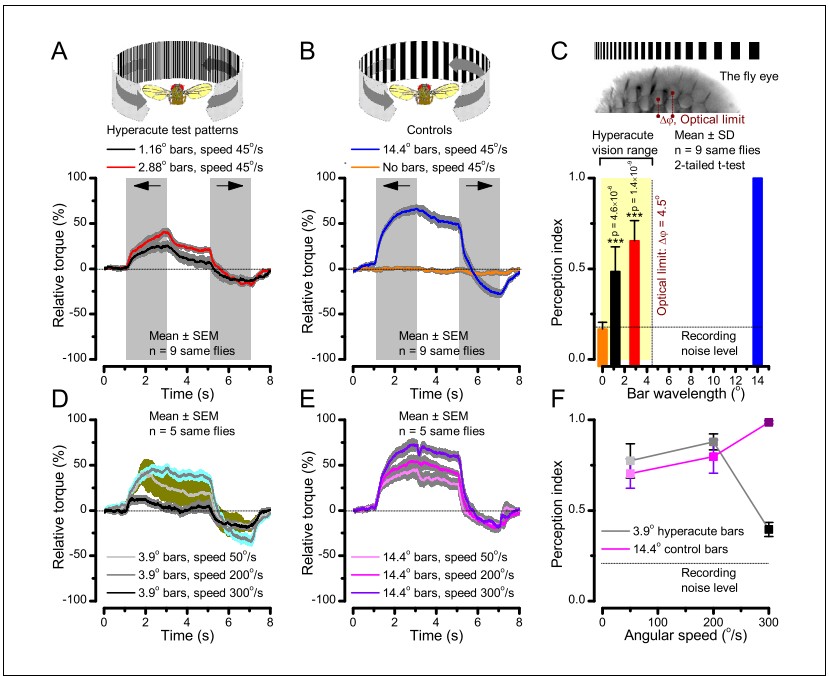

**Figure 10.** Optomotor behavior in a flight simulator system confirms hyperacute vision. Classic open-loop experiments using high-resolution panoramas. (**A**) 360° hyperacute black-and-white bar panorama, with 1.16° or 2.88° wavelengths (= 0.58° and 1.44° inter-bar-distances), rotated counterclockwise and clockwise (grey, arrows) around a tethered fly, with a torque meter measuring its optomotor responses. (**B**) Controls: the same flies' optomotor responses to white (no bars) and wide-bar (14.4° wavelength) rotating panoramas. (**C**) Every *Drosophila* responded to the hyperacute panoramas (wavelength < interommatidial angle, Δφ, yellow area; *Figure 10— figure supplement 1*), but not to the white panorama (orange), which thus provided the recording noise level. The flies optomotor responses were the strongest to the wide-bar panorama (perception index = 1). As the flies' optomotor responses followed the rotation directions consistently, irrespective of the tested bar wavelengths, the hyperacute visual panorama did not generate perceptual aliasing. (**D**) Optomotor responses of five flies to hyperacute panorama with 3.9° wavelength, rotating at 50°/s and saccadic speeds of 200 and 300°/s. (**E**) Control responses of the same flies to 14.4° wavelength panorama at the same speeds. (**F**) The flies' ability to follow hyperacute panorama reduces dramatically when the stimulation approaches the photoreceptors' predicted acuity limit, which for ~4° point resolution is just over 300°/s (*cf. Figure 9A*). Details in Appendix 10.

DOI: https://doi.org/10.7554/eLife.26117.035

The following figure supplement is available for figure 10:

**Figure supplement 1.** Optomotor behavior in a flight simulator system demonstrates that *Drosophila* see hyperacute visual patterns.

DOI: https://doi.org/10.7554/eLife.26117.036

grained (3.9°) panoramic image was rotated faster (*Figure 10D*), the response declined as predicted (*cf.* two dots 4° apart in *Figure 9A*). This result is consistent with photoreceptor output setting the perceptual limit for vision and demonstrates that *Drosophila* see hyperacute details even at saccadic speeds (*Figure 10D–F*).

## Discussion

We have provided deep new insight into spatiotemporal information processing in *Drosophila* R1-R6 photoreceptors and animal perception in general. Our results indicate that the dynamic interplay between saccades and gaze fixation is important for both the maintenance and enhancement of vision already at the photoreceptor level. This interplay, which is commonly observed in locomoting *Drosophila* (*Geurten et al., 2014*), makes light input to photoreceptors bursty.

We showed that high-contrast bursts, which resemble light input during a fly's saccadic behaviors, maximize photoreceptors' information capture in time, and provided evidence that such encoding

involves four interlinked mechanisms. Light input is first regulated by two processes inside photoreceptors: slower screening pigment migration (intracellular pupil, 1–10 s) and much faster photomechanical rhabdomere contractions (0.01–1 s). These modulations have low noise (*Figure 2—figure supplement 2*), enabling refractory photon sampling by microvilli to enhance information intake in phasic stimulus components. Finally, asymmetric synaptic inputs from the network differentiate individual R1-R6 outputs. Remarkably, over space, these mechanisms further sharpen neural resolvability by ~4 fold below the theoretical limit of the compound eye optics, providing hyperacute vision. Further analyses imply that these mechanisms with systematic rhabdomere size variations combat aliasing (Appendixes 2 and 5).

Thus, with microsaccadic sampling, a fly's behavioral decisions govern its visual information/acuity trade-off. To see the finest image details it should scan the world slowly, which probably happens during normal gaze fixation. But gaze fixation cannot be too slow; otherwise, adaptation would fade vision. Conversely, by locomoting faster, in a saccadic or bursty fashion, visual information capture in time is increased (see also: *Juusola and de Polavieja, 2003*), while surprisingly little spatial details about its surroundings would be lost.

This viewing strategy corresponds well with the recent human psychophysics results and modeling of ganglion cell firing (*Rucci and Victor, 2015*), which indicate that microsaccades and ocular drift in the foveal region of the retina actively enhance perception of spatial details (*Rucci et al., 2007*; *Poletti et al., 2013*; *Rucci and Victor, 2015*). Interestingly, here our findings further imply that, in *Drosophila*, the extraction of phasic stimulus features, which characterize object boundaries and line elements in visual scenes, already starts during sampling and integration of visual information in the microvilli, at the first processing stage (rather than later on in the retinal network or in the brain).

Our results make a general prediction about the optimal viewing strategy for maximizing information capture from the world. Animals should fixate gaze on darker features, as this resensitizes photoreceptors by relieving their refractory sampling units (*e.g.* microvilli). And then, rapidly move gaze across to brighter image areas, as saccadic crossings over high-contrast boundaries enhance information intake by increasing photoreceptors' sample (quantum bump) rate changes/time (Appendix 9).

Given the high occurrence of eye/head-saccades in animals with good vision (*Land, 1999*), it seems plausible that their photoreceptors could also have adapted encoding dynamics to quicken response modulation, reducing motion blur. Therefore, if information sampling biophysics in rods and cones were matched to microsaccadic eye movements, this could provide a mechanistic explanation to the old paradox: how saccadic gaze fixation provides stable perception of the world, while curtailing motion blur effects.

## Materials and methods

### Flies

2–10 day old wild-type red-eyed (Canton-S and Berlin) fruit flies (*Drosophila melanogaster*) and $hdc^{JK910}$-mutants were used in the experiments. Other transgenic and mutant *Drosophila* tests and controls are explained in specific **Appendixes**. *Drosophila* were raised at 18°C in a 12 hr/12 hr dark/light cycle and fed on standard medium in our laboratory culture.

### Electrophysiology

Sharp microelectrode recordings from *Drosophila* R1-R6 photoreceptors were detailed before (*Juusola and Hardie, 2001a*; *Juusola et al., 2016*), and we only list the key steps here. Flies were immobilized to a conical holder by beeswax (*Juusola and Hardie, 2001a*) (*Figure 1A*). A small hole, the size of a few ommatidia, was cut in the dorsal cornea for the recording electrode and sealed with Vaseline to prevent tissue from drying. R1-R6s' intracellular voltage responses were recorded to different spatiotemporal light patterns (see below) using sharp filamented quartz or borosilicate microelectrodes (120–220 MΩ), filled with 3 M KCl. A blunt reference electrode, filled with fly ringer, was inserted in the head capsule. The flies' temperature was kept either at 19 ± 1 or 25 ± 1°C by a feedback-controlled Peltier device, as indicated in the figures. The recordings were performed after 1–2 min of dark adaptation, using the discontinuous clamp method with a switching frequency 20–40 kHz. The electrode capacitance was compensated using the head-stage output voltage. To

minimize effects of damage and external noise on the analysis, only stable recordings of low-noise and high sensitivity were chosen for this study (sometimes lasting several hours). Such photoreceptors typically had resting potentials <-60 mV in darkness and >45 mV responses to saturating test light pulses (*Juusola and Hardie, 2001a*).

## Light stimulation

We used a high power 'white' LED (Seoul Z-Power P4 star, white, 100 Lumens) to test individual R1-R6 photoreceptors' encoding dynamics (*Figures 1* and *6F*). It was connected to a randomized quartz fiber optic bundle (transmission range: 180–1,200 nm), fitted with a lens (providing ~3° homogeneous light disk as seen by the flies), and attached onto a Cardan arm system for accurate positioning at the center of each tested cell's receptive field. Its light output was driven by an OptoLED (Cairn Research Ltd, UK), which utilizes a feedback circuitry with a light-sensor. This LED has red component wavelengths, which minimizes prolonged depolarizing afterpotential (PDA) effects. Because long recordings can show sensitivity drifts, attributable to muscle activity (Appendix 4), the stimulus XY-position was regularly tested and, if needed, re-centered between long stimulus runs.

We used a bespoke 25 light-point array to measure individual R1-R6 photoreceptors' receptive fields and responses to moving point objects (bright dots, *Figure 7*; dark dots, Appendix 9). Again, a custom-made Cardan arm system was used to accurately position the array's center light-point (no. 13) at the center of each tested cell's receptive field. The dot size and the minimum inter-dot-distance, as seen by *Drosophila*, was 1.7°. Details of this device and the recording procedures are given in **Appendixes 4** and **6**.

## Stimulus patterns

Single photoreceptors' diurnal temporal encoding gamut was tested systematically over different bandwidth and contrast distributions; using 20 distinct light intensity time series stimuli, which were presented at the center of their receptive fields. The used test stimuli was based upon 5 different 2 s long Gaussian white-noise light intensity time series patterns (generated by Matlab's randn-function), which had 'flat' power spectrum up to 20, 50, 100, 200, or 500 Hz (*Figure 1B*), as low-pass filtered by MATLAB's filter toolbox, and the same peak-to-peak modulation (two units). These were then superimposed on four backgrounds: BG0 (0 units, dark), BG0.5 (0.5 units), BG1 (one unit) or BG1.5 (1.5 units, bright) on a linear intensity scale, giving altogether 20 unique stimulus patterns. As the two lowest backgrounds clipped downwards-modulation, prolonging dark intervals, the resulting stimuli ranged from high-contrast bursts ($c = \Delta I/I \sim 1.46$ at BG0) to low-contrast Gaussian ($c \sim 0.22$ at BG1.5).

As further controls, we tested how well R1-R6 photoreceptors responded to dark contrast bursts of different bandwidths (Appendix 9) and to their bright counterparts. In these experiments, R1-R6s were adapted for 10 s to BG0.5 and BG1 before repeated stimulation. In addition, we recorded the tested cells' responses to naturalistic light intensity time series (*van Hateren, 1997a*; *Song and Juusola, 2014*) (NS), selected from van Hateren natural stimulus collection (*van Hateren, 1997a*) (*Figure 2—figure supplement 3*). We also sampled light intensity time series from panoramic natural images, using three different velocity profiles of a published 10 s *Drosophila* walk (see *Video 1*; details in Appendix 3). These stimuli were then played back to a R1-R6 photoreceptor by the 'white' LED (see above).

In all these experiments,≥25 consecutive responses to each repeated stimulus were recorded.

## Data acquisition

Both the stimuli and responses were filtered at 500 Hz (KEMO VBF/23 low-pass elliptic filter, UK), and sampled together at 1–10 kHz using a 12-bit A/D converter (National Instruments, USA), controlled by a custom-written software system, Biosyst in Matlab (Mathworks, USA) environment. For signal analyses, if need, the data was down-sampled to 1 kHz.

## Analyses

Because of short-term adaptive trends, we removed the first 3–10 responses to repeated stimulation from the analysis and used the most stable continuous segment of the recordings. Information theoretical methods for quantifying responses of approximately steady-state-adapted fly photoreceptors

to different stimuli were described in detail before (*Juusola and Hardie, 2001b*; *Juusola and de Polavieja, 2003*; *Song et al., 2012*; *Song and Juusola, 2014*). Below we list the key approaches used here.

## Signal-to-noise ratio (SNR) and information transfer rate estimates

In each recording, simulation or Poisson light stimulus series (see below), the signal was the mean, and the noise was the difference between individual traces and the signal (*Juusola and Hardie, 2001a*). Therefore, for a data chunk of 20 responses (n = 20 traces), there was one signal trace and 20 noise traces. The signal and noise traces were divided into 50% overlapping stretches and windowed with a Blackman–Harris 4-term window, each giving three 500-points-long samples. Because each trace was 2 s long, we obtained 60 spectral samples for the noise and seven for the signal. These were averaged, respectively, to improve the estimates.

*SNR(f)*, of the recording, simulation, or Poisson light stimulus series was calculated from their signal and noise power spectra, $<|Sf,|^2>$ and $<|Nf,|^2>$, respectively, as their ratio, where $||$ denotes the norm and $<>$ the average over the different stretches (*Juusola and Hardie, 2001a*). To eliminate data size and processing bias, the same number of traces (n = 20) of equal length (2000 points) and sampling rate (1 kHz; 1 ms bin size) were used for calculating the *SNR(f)*, estimates for the corresponding real recordings, photoreceptor model simulations and the simulated Poisson stimuli.

Information transfer rates, *R*, for each recording, simulation, or Poisson light stimulus series were estimated by using the Shannon formula (*Shannon, 1948*), which has been shown to obtain robust estimates for these types of continuous signals (*Juusola and de Polavieja, 2003*; *Song and Juusola, 2014*). We analyzed steady-state-adapted recordings and simulations, in which each response (or stimulus trace) is expected to be equally representative of the underlying encoding (or statistical) process. From *SNR(f)*, the information transfer rate estimates were calculated as follows:

$$R = \int_0^\infty (log_2[SNR(f)+1])df \qquad (1)$$

We used minimum = 2 Hz and maximum = 500 Hz (resulting from 1 kHz sampling rate and 500 points window size). The underlying assumptions of this method and how the number and resolution of spectral signal and noise estimates and the finite size of the used data can affect the resulting Information transfer rate estimates have been analyzed before (*van Hateren, 1992b*; *Juusola and Hardie, 2001b*; *Juusola and de Polavieja, 2003*; *Song and Juusola, 2014*) and are further discussed in Appendix 2. The mean and SD of each photoreceptor recording series (20 × 2000 points) was obtained by estimating *R* from eleven 1,000-point data chunks with 100-point overlaps.

We also tested how the Shannon method's information transfer rate estimates of bursty responses compare with those obtained by the triple extrapolation method (*Juusola and de Polavieja, 2003*) using additional longer recordings. In the triple extrapolation, photoreceptor responses were first digitized (*Figure 2—figure supplement 4A–B*) by dividing these into time intervals, $T_w$, that were subdivided into smaller intervals of $t_w$ = 1 ms. This procedure selects 'words' of length $T_w$ with $T_w/t_w$ 'letters.' The mutual information between the response S and the stimulus is then the difference between the total entropy, $H_s$:

$$H_S = -\sum_i P_S(s_i)log_2 P_S(s_i) \qquad (2)$$

where $P_S(s_i)$ is the probability of finding the *i*-th word in the response, and the noise entropy $H_N$:

$$H_N = -\left\langle \sum_{i=1} P_i(\tau)log_2 P_i(\tau) \right\rangle_\tau \qquad (3)$$

where $P_i(\tau)$ denotes the probability of finding the *i*-th word at a time *t* after the initiation of the trial. This probability $P_i(\tau)$ was calculated across trials of identical 20 Hz bursty stimulation. The values of the digitized entropies depend on the length of the 'words' $T_w$, the number of voltage levels *v*, and the size (as %) of the data file, $H^{T,v,size}$. The rate of information transfer was obtained taking the following three successive limits (*Figure 2—figure supplement 4C–E*, respectively):

$$R = R_S - R_N = \lim_{T_w \to \infty} \frac{1}{T_w} \lim_{v \to \infty} \lim_{size \to \infty} (H_S^{T_w v, size} - H_N^{T_w v, size}) \tag{4}$$

These limits were calculated by extrapolating the values of the experimentally obtained entropies. A response matrix for the analysis contained 2,000 points × 30 trials (note, the 10 first trials from the light onset were removed to minimize any adaptation effects). The total entropy and noise entropy of the responses were then obtained from the response matrices using linear extrapolation within the following parameter ranges: size = 5/10, 6/10,…,10/10 of data; v = 4, 5,…,20 voltage levels; $T_w^{-1}$ = 2, 3,…, 7 points. As adaptation in photoreceptors approaches steady state, their output varies progressively less (*Juusola and de Polavieja, 2003*). Similarly, the entropies of their responses, when digitized to ≤20 voltage levels, ceases to increase with increasing data size, enabling their limits to be extrapolated in control by linear fits (*Figure 2—figure supplement 4C–F*) or Taylor series fits. Consequently, as few as 30 response traces (each 2,000 points long) provided similar information rate estimates to the Shannon method (*Figure 2—figure supplement 4G*) for 20 Hz burst stimulation. All data analyses were performed with Matlab (MathWorks).

## Measuring photoreceptors' visual acuity

We measured dark- and light-adapted wild-type R1-R6 photoreceptors' receptive fields by their acceptance angles, Δρ, using intracellular voltage responses to random light-points in a stimulation array. These measurements were compared to those of *hdc^JK910*-mutants (*Burg et al., 1993*), in which first-order interneurones receive no neurotransmitter (histamine) from photoreceptors and so are incapable of feedback-modulating the photoreceptor output. Both the wild-type and mutant R1-R6 photoreceptors' mean Δρ was about twice the mean interommatidial angle, Δφ. The stimulus apparatus, the method and result details and theoretical electron micrograph comparisons of their mean rhabdomere sizes are explained in **Appendixes 4–5**.

## Spatiotemporal analyses using the classic conventional models

Voltage responses of wild-type and *hdc^JK910* R1-R6s to moving bright dots were evaluated against their respective classic model simulations, in which each recorded receptive field was convoluted by the same cell's impulse response (*Srinivasan and Bernard, 1975*; *Juusola and French, 1997*) (1st order Volterra kernels). The motion blur effects were quantified by comparing the real R1-R6 outputs to their deterministic model predictions. Details of the analysis are given in Appendix 6.

## **Biophysical modeling**

### Time series analyses

We used our recently published biophysically-realistic stochastic photon sampling model (*Song et al., 2012*) of a *Drosophila* R1-R6 photoreceptor to simulate macroscopic voltage response to different repeated light intensity time series patterns from a point source (*Figures 3–4*). The model has no free parameters. Its design and the general aims and details of these simulations are given in **Appendixes 1** and **2**. To eliminate data size bias, the signaling properties and performance of the simulations were quantified and compared to the corresponding recordings by using the same analytical routines on the same-sized data-chunks. The models were run using Matlab in the University of Sheffield computer cluster (Iceberg).

For each stimulus, its mean was adjusted to maximize information of the simulated photoreceptor outputs, mimicking the action of the photomechanical adaptations (intracellular pupil mechanism and rhabdomere contractions; **Appendixes 2** and **7**). This optimization set the effective mean photon rates from $8 \times 10^4$ at BG1.5 to $8 \times 10^5$ photons/s at BG0 (*Figure 3*). Thus, each of these light levels was considered to represent the optimal daylight input (that survived the photomechanical adaptations and was absorbed by a rhabdomere), in which modulation enabled the largest sample (bump) rate changes. Otherwise, more of its sampling units (30,000 microvilli) would be either underutilized or refractory (saturation). The maximum information rates of the simulated photoreceptor outputs closely followed the corresponding mean information transfer rates of the real recordings over the whole tested encoding range (Appendix 2). This implies that the central function of the photoreceptors' combined photomechanical adaptations is to maximize their information transfer, and that the resulting estimates represent realistic maxima.

### Encoding efficiency

A photoreceptor's encoding efficiency, $\eta$, was the ratio between the information rates of its voltage output, $R_{output}$, and the corresponding effective light input, $R_{input}$:

$$\eta = \frac{R_{output}}{R_{input}} \tag{5}$$

with $R_{output}$ and $R_{input}$ estimated by the Shannon formula (*Equation 1*). Details are in Appendix 2.

### Modeling R1-R6 output to moving dots

We developed a new 'microsaccadic sampling'-model to predict how photomechanical rhabdomere contractions (microsaccades) move and narrow *Drosophila* R1-R6 photoreceptors' receptive fields to resolve fast-moving objects. Appendix 8 gives the details of this modeling approach, which combines the stochastic photon sampling model(*Song et al., 2012*) with additional fixed ommatidium optics and photomechanical rhabdomere contraction parameters. The same appendix shows examples of how refractory photon sampling and rhabdomere contractions jointly improve visual acuity.

## High-speed video of the light-induced rhabdomere movements

Cornea-neutralization method with antidromic far-red (>720 nm) illumination was used to observe deep pseudopupils (*Franceschini and Kirschfeld, 1971b*) (photoreceptor rhabdomeres) in the *Drosophila* eye at 21°C. A high-speed camera (Andor Zyla, UK; 500 frames/s), connected to a purpose-built microscope system, recorded fast rhabdomere movements *in vivo* to blue-green light stimuli (470 + 535 nm peaks), which were delivered orthodromically into the eye. The method details, mutant and transgenic *Drosophila* used and the related image analyses are explained in Appendix 7.

## Flight simulator experiments

*Open-loop configuration* was used to test hyperacute motion vision. Wild-type flies were tethered in a classic torque meter (*Tang and Guo, 2001*) with heads fixed, and lowered by a manipulator into the center of a black and white cylinder (spectral full-width: 380–900 nm). A flying fly saw a continuous panoramic scene (360°), which in the tests contained multiple vertical stripes (black and white bars of equal width). The control was a diffuse white background. After viewing the still scene for 1 s, it was spun counterclockwise by a linear stepping motor for 2 s, stopped for 2 s before rotating clockwise for 2 s, and stopped again for 1 s. This 8 s stimulus was repeated 10 times and each trial, together with the fly's yaw torque responses, was sampled at 1 kHz (*Wardill et al., 2012*). Flies followed the stripe scene rotations, generating yaw torque responses (optomotor responses to right and left), the strength of which reflected the strength of their motion perception. The flies did not follow the white control scene rotations. The panoramic scenes had ±360° azimuth and ±45° elevation, as seen by the fly. The stripe scenes had 1.0 contrast and their full-wavelength resolutions were either hyperacute (1.16° or 2.88°) or coarse (14.40°), giving the inter-bar-distances of 0.58°, 1.44° and 7.20°, respectively. The white scene has zero contrast. The tested scene rotation velocities were 45, 50, 200 and 300°/s.

## Transmission electron microscopy

The fly eye dissection, fixation embedding, sectioning and imaging protocols for EM (*Figure 5A*) are described in Appendix 5.

## Statistics

Test responses were compared with their controls by performing two-tailed t-tests to evaluate the difference in the compared data. Welch's t-test was used to accommodate groups with different variances for the unpaired comparisons. In the figures, asterisks are used to mark the statistical significance: ns indicates $p > 0.05$, * indicates $p \leq 0.05$, ** indicates $p \leq 0.01$, and *** indicates $p \leq 0.001$.

## Software code

Custom written simulation and analyses software used in this study can be downloaded under GNU General Public License v3.0 from: https://github.com/JuusolaLab/Microsaccadic_Sampling_Paper. A copy is archived at https://github.com/elifesciences-publications/Microsaccadic_Sampling_Paper.

## Acknowledgements

We thank C-H Lee, E Chiappe, R Strauss and E Buchner for flies, S Tang and M Swann for help with the flight simulator, O List for some illustrations, A Nikolaev, A Lin for discussions, and two anonymous referees for their suggestions. This work was supported by the Open Research Fund of the State Key Laboratory of Cognitive Neuroscience and Learning (MJ), High-End Foreign Expert Grant by Chinese Government (GDT20151100004: MJ), NSFC project (30810103906: MJ), Jane and Aatos Erkko Foundation Fellowships (MJ and JT), The Leverhulme Trust grant (RPG-2012–567: MJ) and the BBSRC grants (BB/F012071/1, BB/D001900/1 and BB/H013849/1: MJ). The authors declare no competing financial interest.

## Additional information

### Funding

| Funder | Grant reference number | Author |
|---|---|---|
| Biotechnology and Biological Sciences Research Council | BB/H013849/1 | Mikko Juusola |
| Leverhulme Trust | RPG-2012-567 | Mikko Juusola |
| Jane ja Aatos Erkon Säätiö | | Mikko Juusola<br>Jouni Takalo |
| High-End Foreign Expert Grant by Chinese Government | GDT20051100004 | Mikko Juusola |
| Biotechnology and Biological Sciences Research Council | BB/F012071/1 | Mikko Juusola |
| Biotechnology and Biological Sciences Research Council | BB/D001900/1 | Mikko Juusola |
| Beijing Normal University | Open Research Fund | Mikko Juusola |
| Biotechnology and Biological Sciences Research Council | BB/M007006/1 | Roger Hardie |
| Biotechnology and Biological Sciences Research Council | BB/J0092531/1 | Roger Hardie |
| National Science Foundation of China | 30810103906 | Mikko Juusola |

The funders had no role in study design, data collection and interpretation, or the decision to submit the work for publication.

### Author contributions

Mikko Juusola, Conceptualization, Resources, Data curation, Software, Formal analysis, Supervision, Funding acquisition, Validation, Investigation, Visualization, Methodology, Writing—original draft, Project administration, Writing—review and editing; An Dau, Software, Validation, Investigation, Visualization, Methodology, Writing—review and editing; Zhuoyi Song, Data curation, Software, Formal analysis, Validation, Investigation, Visualization, Methodology; Narendra Solanki, Diana Rien, Florence Blanchard, Data curation, Investigation; David Jaciuch, Data curation, Investigation, Writing—review and editing; Sidhartha Anil Dongre, Data curation, Formal analysis, Investigation, Writing—review and editing; Gonzalo G de Polavieja, Software, Formal analysis, Visualization, Writing—review and editing; Roger C Hardie, Resources, Data curation, Formal analysis, Investigation, Visualization, Writing—review and editing; Jouni Takalo, Resources, Software, Formal analysis, Validation, Investigation, Visualization, Methodology, Writing—review and editing

### Author ORCIDs

Mikko Juusola http://orcid.org/0000-0002-4428-5330
An Dau http://orcid.org/0000-0002-7802-565X
Zhuoyi Song http://orcid.org/0000-0001-9991-4053

Sidhartha Anil Dongre (iD) http://orcid.org/0000-0003-2575-4362
Gonzalo G de Polavieja (iD) http://orcid.org/0000-0001-5359-3426
Roger C Hardie (iD) https://orcid.org/0000-0001-5531-3264

## Decision letter and Author response

Decision letter https://doi.org/10.7554/eLife.26117.108
Author response https://doi.org/10.7554/eLife.26117.109

---

## Additional files

### Major datasets

The following dataset was generated:

| Author(s) | Year | Dataset title | Dataset URL | Database, license, and accessibility information |
|---|---|---|---|---|
| Juusola M, Dau A, Song Z, Solanki N, Rien D, Jaciuch D, Dongre S, Blanchard F, Polavieja Gd, Hardie R, Takalo J | 2017 | Data from: Microsaccadic sampling of moving image information provides *Drosophila* hyperacute vision | http://datadryad.org/review?doi=doi:10.5061/dryad.12751 | Available at Dryad Digital Repository under a CC0 Public Domain Dedication |

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

## Appendix 1

DOI: https://doi.org/10.7554/eLife.26117.037

# 'Stochastic adaptive visual information sampling'-theory in brief

## Overview

This appendix describes the basic theoretical principles of how fly photoreceptors sample photons, providing central background information for the results presented in the main paper.

## Stochastic adaptive sampling of information by R1-R6 photoreceptors

Many lines of evidence imply that in a *Drosophila* R1-R6 photoreceptor 30,000 individual refractory sampling units (microvilli) integrate exponential photon flux changes (~$10^6$ fold) from the environment into macroscopic voltage responses of biophysically limited amplitude range (~60 mV) and bandwidth (~200 Hz) (*Juusola and Hardie, 2001a*; *Song et al., 2012*; *Song and Juusola, 2014*; *Hardie and Juusola, 2015*). In essence, a light-adapted R1-R6 counts photons imperfectly, which, nonetheless, adds up highly reproducible neural representations of light changes within its receptive field (*Juusola et al., 2015*).

In this study, we quantify such quantal information processing through large-scale experimental and theoretical analyses. Our overriding aim is to analyze R1-R6s' diurnal encoding range systematically; from light bursts to Gaussian white-noise stimulation to point-objects moving across their receptive fields at saccadic speeds. Because of the outstanding stability and signal-to-noise ratio of the intracellular recordings from *in vivo* R1-R6s (*Juusola and Hardie, 2001a*; *Zheng et al., 2006*; *Song and Juusola, 2014*; *Juusola et al., 2016*), providing apparent ergodicity, we can directly compare their voltage responses to those of biophysically realistic R1-R6 model simulations (*Song et al., 2012*; *Song and Juusola, 2014*; *Juusola et al., 2015*; *Song et al., 2016*), in which stochastically operating microvilli sampled similar stimuli (*Appendix 1—figure 1*). The mechanistic knowledge so obtained about the dynamics and limitations of quantal visual information processing provides us with deep new understanding of how well *Drosophila*, and other insect eyes, can see the world.

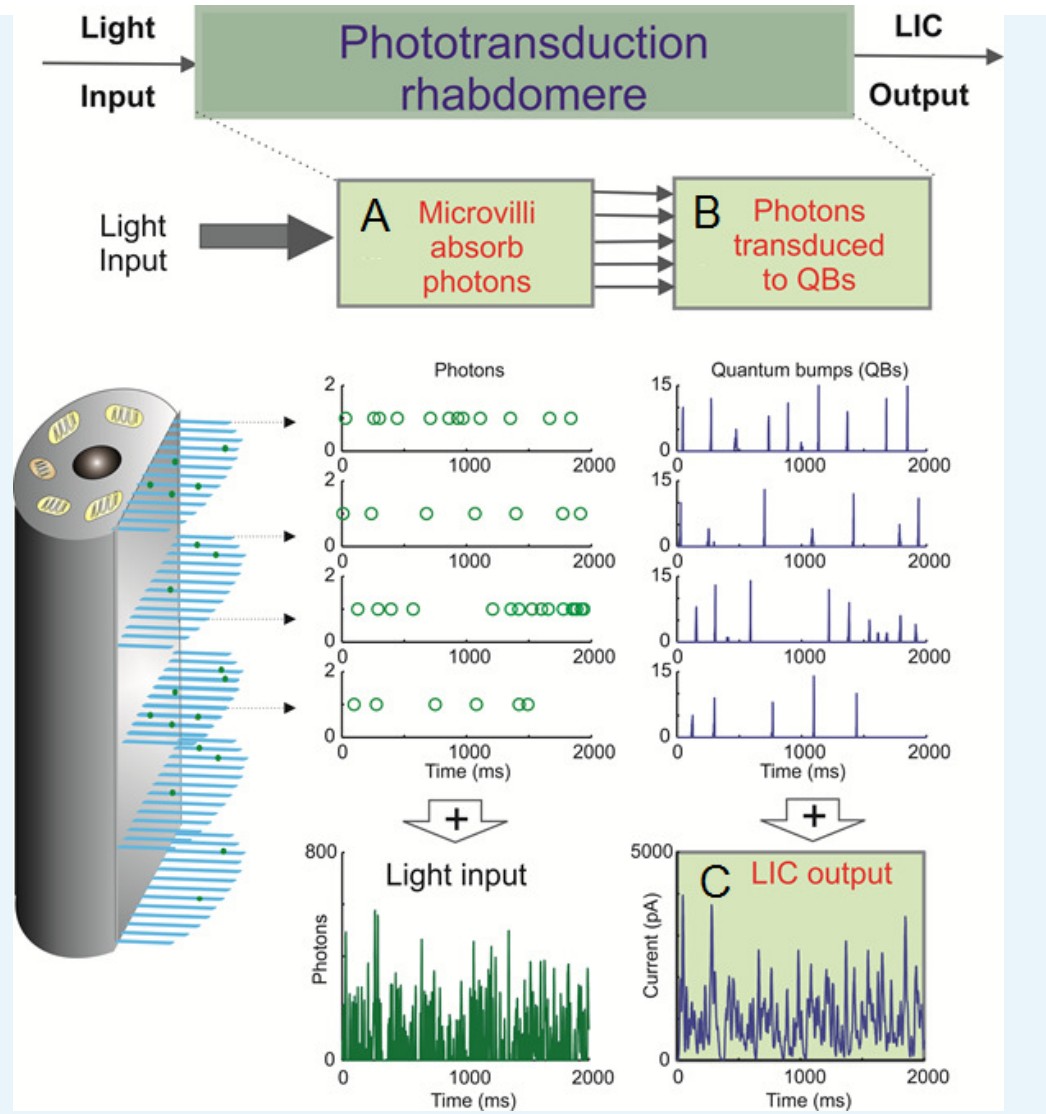

**Appendix 1—figure 1.** Schematic of the biophysically realistic *Drosophila* R1-R6 model, which mimics phototransduction by transducing light input (a dynamic flux of photons) into macroscopic output, light-induced current (LIC). The model is modular, containing four parts, three of which are shown here. Phototransduction occurs within a photoreceptor's light-sensitive part, the rhabdomere, which contains 30,000 photon sampling units, microvilli (blue bristles). Each microvillus contains a full phototransduction cascade reactions, and can transduce single photon energies (green dots) into unitary responses, quantum bumps (QB or samples) of variable amplitudes. (**A**) In the first module, 30,000 microvilli sample incoming photons. The light input, as photons/s, is randomly distributed over them (each row of open circles indicate photons being absorbed by a single microvillus). (**B**) In the second module, the successfully absorbed photons in each microvillus are transduced into QBs (a row of unitary events). In each microvillus, the success of transducing a photon into a QB depends upon the refractoriness of its phototransduction reactions. This means that a microvillus cannot respond to the next photons until its phototransduction reactions have recovered from the previous photon absorption, which takes about 50–300 ms. The photons hitting a refractory microvillus cannot evoke QBs, but will be lost. (**C**) In the third module, QBs from all the microvilli then integrate the dynamic macroscopic LIC. Conversely, the light input (green trace) can be reconstructed by adding up all the photons distributed across the 30,000 microvilli.

DOI: https://doi.org/10.7554/eLife.26117.038

## Main framework

We used our previously published biophysical *Drosophila* R1-R6 model (**Song et al., 2012**; **Song and Juusola, 2014**; **Juusola et al., 2015**; **Song et al., 2016**) to simulate voltage responses to time series of light intensities. The Matlab scripts for this model are downloadable from the repository: https://github.com/JuusolaLab/Microsaccadic_Sampling_ Paper/tree/master/BiophysicalPhotoreceptorModel. The model contains four modules (**Song et al., 2012**; **Song and Juusola, 2014**; **Juusola et al., 2015**; **Song et al., 2016**):

- *Random Photon Absorption Model*: regulates photon hits (absorptions) in each microvillus, following Poisson statistics (**Song et al., 2016**) (**Appendix 1—figure 1A**).
- *Stochastic Bump Model*: stochastic biochemical reactions inside a microvillus capture and transduce the energy of photons to variable quantum bumps or failures (**Appendix 1—figure 1B**). Here, Gillespie algorithm provides discrete and stochastic phototransduction cascade simulations with few reactants as every reaction is explicitly simulated.
- *Summation Model*: bumps from 30,000 microvilli integrate to the macroscopic light-induced current (LIC) response (**Appendix 1—figure 1C**).
- *Hodgkin-Huxley (HH) Model of the photoreceptor plasma membrane* (**Niven et al., 2003**; **Vahasoyrinki et al., 2006**): transduces LIC into a voltage response (**Appendix 1—figure 2**).

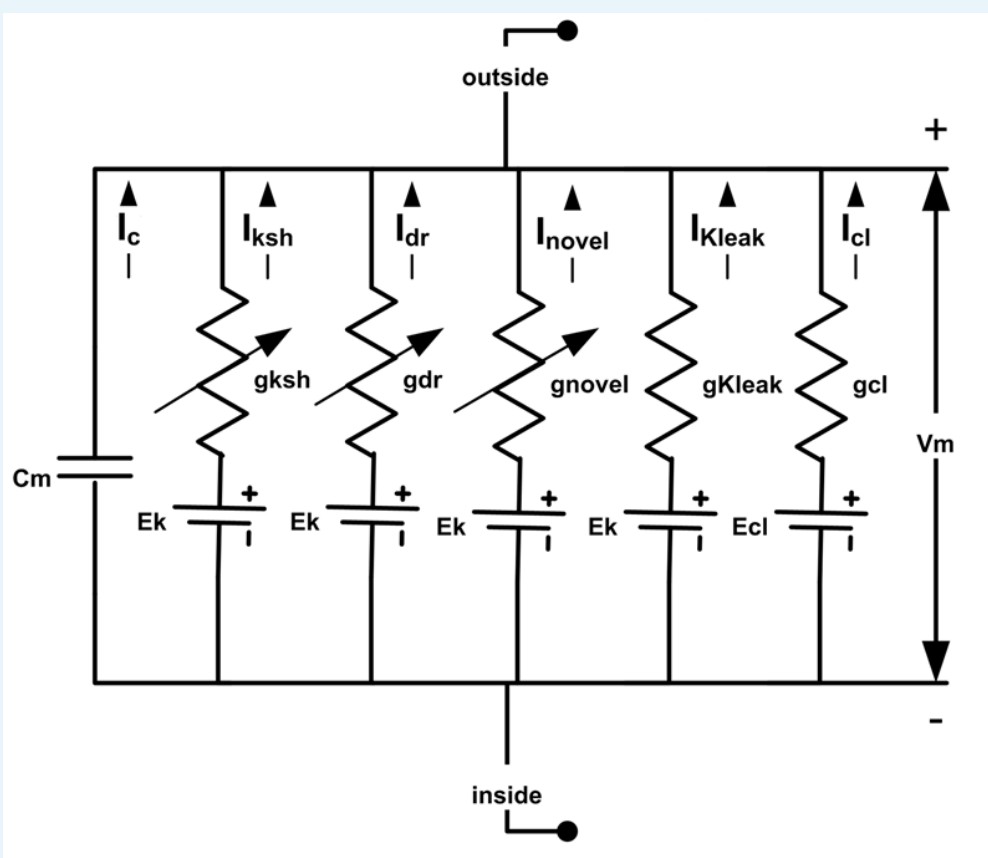

**Appendix 1—figure 2.** *Drosophila* R1-R6 photoreceptor membrane's electrical circuit (HH-model). A photoreceptor's membrane potential, $V_m$, is the difference between the negative inside (intracellular) and positive outside (extracellular) voltages. $V_m$ can be calculated, using Hodgkin-Huxley formalism, whereupon, a photoreceptor membrane is modelled as a capacitor, $C_m$, voltage-gated channels as voltage-regulated conductances, g, leak channels as fixed conductances, reversal potentials for different ion species as DC-batteries. Abbreviations: $I_{ksh}$: Shaker; $I_{dr}$, delayed rectifier; $I_{novel}$, novel $K^+$; $K_{leak}$: $K^+$ leak; $I_{cl}$, chloride leak

currents. The used deterministic *Drosophila* photoreceptor HH-model is adapted from (**Niven et al., 2003**; **Vahasoyrinki et al., 2006**).
DOI: https://doi.org/10.7554/eLife.26117.039

The formalism, assumptions and many tests of the biophysical photoreceptor model, which has no free parameters, are given in our previous publications (**Song et al., 2012**; **Song and Juusola, 2014**; **Juusola et al., 2015**). All its parameter values can be found and downloaded from: http://www.sciencedirect.com/science/article/pii/S0960982212006343

## Stochastic quantal models supersede empirical black-box approaches

### Stochastic photon sampling fly photoreceptor model

- Has no free parameters
- Is general - predicts realistic responses to any light stimulus pattern
- Is transferable - predicts realistic responses of different fly photoreceptors
- Provides deep mechanistic understanding to light information sampling

We have shown before that this quantal stochastic modeling approach is general and transferable, and therefore directly applicable to quantify photoreceptor functions in different light conditions and fly species (**Song et al., 2012**; **Song and Juusola, 2014**; **Juusola et al., 2015**). Importantly, it does not require full knowledge of all molecular players and dynamics in the phototransduction to generate realistic responses (**Song et al., 2012**; **Song and Juusola, 2014**; **Juusola et al., 2015**). From a computational viewpoint, the exactness of the simulated molecular interactions is not critical. As long as the photoreceptor model contains the right number of microvilli (*e.g.* 30,000 in a *Drosophila* and 90,000 in a *Calliphora* R1-R6 photoreceptor), each of which is a semiautonomous photon sampling unit, and the dynamics of their photon-triggered unitary responses (quantum bumps [QBs] or samples) approximate those in the real recordings, it will sample and process information much like a real photoreceptor. Conversely, this further means that by knowing the number of microvilli and their average QB waveform, latency distribution, and refractory period distributions, we can closely predict a fly photoreceptor's macroscopic response to any given light intensity time series stimulus (*cf.* **Figures 3–4**). The same model can then be applied to estimate how well the real photoreceptor output resolves moving objects (see **Figures 8G–I and 9**; **Appendix 8** gives the details of this approach).

### Deterministic empirical fly photoreceptor models

- Fit parameters to specific stimulus sets
- Predict less accurately responses to new stimuli of different input statistics (to which the models have not been tuned to before)
- Cannot provide deep mechanistic understanding of how photoreceptors sample light information

In the conventional empirical 'black-box' approaches, the photoreceptor models' filters, such as linear and nonlinear kernels, and static nonlinearities are adjusted to minimize the difference between the recorded responses and the model output to a specific stimulus set (light condition) (**French et al., 1993**; **Juusola et al., 1995b**; **Juusola and French, 1997**; **Friederich et al., 2009**). However, because such *models are not built upon the real cells' physical quantal information sampling constraints*, which change from one stimulus statistics to another (**Song and Juusola, 2014**), they struggle to respond accurately to new stimulus statistics. Explicitly, the models lack intrinsic structural information of how quantum bump dynamics and microvilli refractoriness must differ during different stimuli. For example, Volterra (**French et al., 1993**) kernels estimated for Gaussian white-noise stimulation will predict less accurately responses to bursty light inputs. This is because during bursty light stimulation the fly photoreceptors' quantal information sampling dynamics rapidly adapt to a different regime, where their microvilli (sampling units) are less refractory. Hence, the real

photoreceptors now integrate macroscopic responses from larger sample (QB) rate changes of enhanced rise and decay dynamics. To appropriately approximate these new dynamics, the empirical models would need to generate new kernels of different temporal profiles, which is impossible without retuning the model parameters. Accordingly, without the biophysical knowledge being implemented in their mathematical structure, the classic dynamic photoreceptor models fail to predict how well the real photoreceptors resolve moving objects (see *Figure 7*, Appendix 6 and Appendix 8).

## Appendix 2

DOI: https://doi.org/10.7554/eLife.26117.040

# Information maximization by photomechanical adaptations and connectivity

## Overview

This appendix describes how the 'stochastic adaptive visual information sampling'-theory (Appendix 1) predicts and explains the roles of R1-R6 photoreceptors' photomechanical adaptations and network connections in information maximization at different stimulus conditions, as shown in *Figures 1–4*.

In this appendix:

- We test the hypotheses that photomechanical adaptations (intracellular pupil and rhabdomere contractions; see Appendix 7) and network connections in the *Drosofila* eye contribute importantly to optimizing the capture and representation of visual information.
- We first estimate through simulations how a R1-R6 photoreceptor's intracellular pupil and rhabdomere contractions are jointly optimized for maximal information sampling by its 30,000 microvilli. The simulations predict that these mechanisms' optimal combined photon throughput in bright conditions (to be absorbed by an average R1-R6 photoreceptor) should be different for different stimuli.
- We then compare the model predictions to corresponding intracellular recordings and find a comprehensive agreement between the theory and mean experiments for all the tested stimuli.
- This striking correspondence enables us to further estimate how the lamina network shapes information transfer of individual R1-R6 photoreceptors.
- Remarkably, our data and analyses strongly suggest that voltage output is different in each R1-R6, which are brought together in neural superposition during development to sample light changes from a small local visual area.
- These results are consistent with the hypothesis that the variability in the retinal sampling matrix dynamics and topology minimizes aliasing and noise, enabling its parallel processing to generate reliable and maximally informative neural estimates of the variable world (*Barlow H, 1961*; *Yellott, 1982*; *Song and Juusola, 2014*; *Juusola et al., 2015*).
- Finally, we explain how to calculate a photoreceptor's encoding efficiency for different light stimuli, highlighting the assumptions and limits of this method.

## Fly photoreceptors' pupil mechanism

In a fly photoreceptor, intracellular screening pigments form its pupil mechanism (*Appendix 2—figure 1*). The pupil protects a photoreceptor's sampling units (30,000 microvilli in a *Drosophila* R1-R6) from saturation (*Howard et al., 1987*; *Song and Juusola, 2014*). At bright light exposure, screening pigments migrate to narrow the aperture they form collectively (*Franceschini and Kirschfeld, 1971b*), shielding off excess light from reaching the microvilli. This is important because midday sunshine on a photoreceptor may contain $10^{6-8}$ photons/s, and without the pupil mechanism would deteriorate the encoding function of its finite microvillus population (*Howard et al., 1987*; *Song and Juusola, 2014*). The pupil opening and closing seem modulated by light-driven intracellular $Ca^{2+}$-concentration changes (*Hofstee and Stavenga, 1996*), and show reasonably fast dynamics (from fully open to fully closed within 15 s) (*Franceschini and Kirschfeld, 1976*) for adapting its light throughput to ambient changes. Although our biophysical (stochastically operating) *Drosophila* photoreceptor model (*Song et al., 2012*; *Juusola et al., 2015*) lacks the pupil mechanism and any other photomechanical adaptations (*cf.* Appendix 7), their joint effects can be predicted through simulations; by assuming that their objective function is to maximize the photoreceptor's information capture.

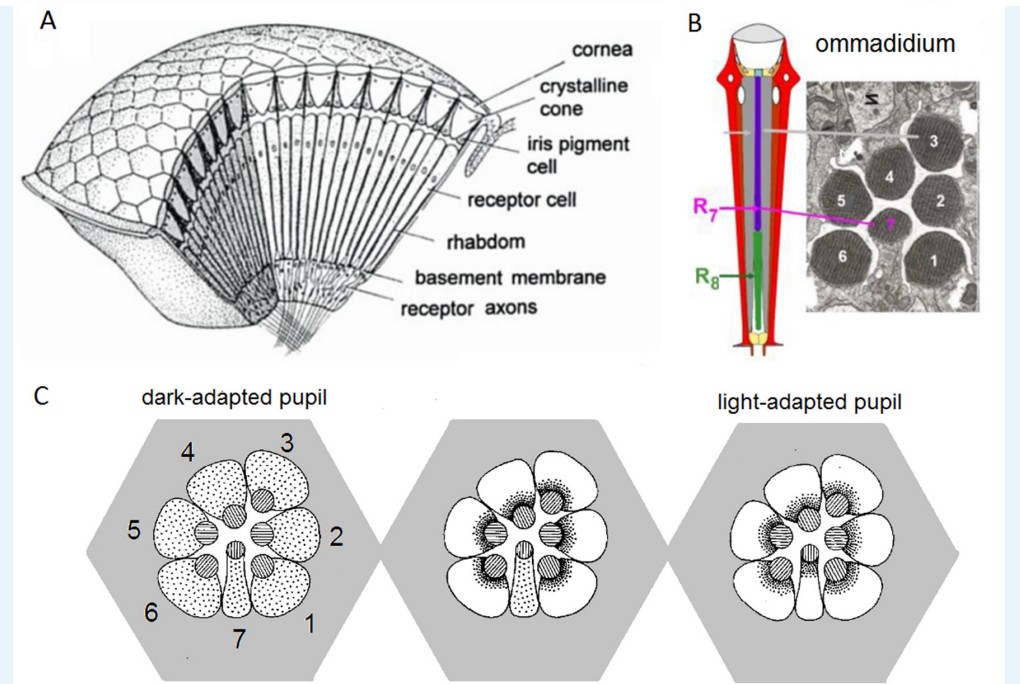

**Appendix 2—figure 1.** Compound eye and photoreceptors' intracellular pupil mechanism. (**A**) *Drosophila* eyes are composed of about 800 modular units, ommmadia. (**B**) Each ommatidium contains a lens system and underneath it eight photoreceptor cells: the outer receptors, R1-R6, and the inner receptors, R7/R8. In the electron micrograph, which numbers each cell's rhabdomere (light sensitive part), R8 is not shown because it lies directly below R7. (**C**) Schematic of the intracellular pupil mechanism. Left: During dark-adaptation, screening pigments (small dots) are scattered in the R1-R7 somata. Middle: R1-R6 light-adapted. Blue-green bright light drives the screening pigment migration towards the R1-R6 rhabdomeres (central discs, containing 30,000 microvilli, photon sampling units, depicted as stripes in the discs), which express blue-green-sensitive Rh1-rhodopsin, as their phototransduction rises intracellular $Ca^{2+}$-concentration. With the pupil closing (seen as the dark rims around the rhabdomeres), light input to the microvilli reduces. Note that R7, which expresses UV-rhodopsin, is not light-adapted and its screening pigments remain scattered. Right: All photoreceptors light-adapted. Bright UV-light closes all pupils because R7s express UV-sensitive Rh3- and Rh4-rhodopsins, and in R1-R6s' Rh1-rhodopsin is electrochemically coupled to UV-sensitive sensitizing pigment. Redrawn and modified from (*Franceschini and Kirschfeld, 1976*; *Elyada et al., 2009*).

DOI: https://doi.org/10.7554/eLife.26117.041

## Photomechanical light-screening hypothesis

We hypothesize that the intracellular pupil, besides affecting a photoreceptor's angular and spectral sensitivity (*Stavenga, 2004a*) (see Appendix 4), participates in maximizing a photoreceptor's information sampling by optimizing light input intensity to its microvilli in time. Specifically in this context, it works together with all other photomechanical adaptations within an ommatidium, including the much faster light-induced rhabdomere contractions (*Videos 2–4*), in protecting the microvilli from saturation. Thus, collectively, we consider the photomechanical adaptations (*Appendix 2—figure 2*) as a biological manifestation of a mathematical information maximization function.

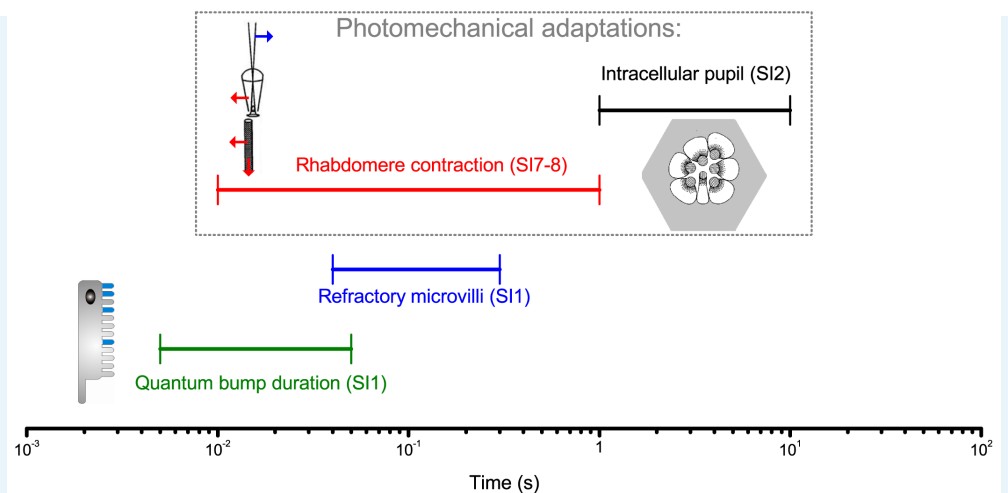

**Appendix 2—figure 2.** Different temporal ranges of a *Drosophila* R1-R6 photoreceptor's intrinsic light adaptation mechanisms. Photomechanical adaptations, such as light-induced intracellular screening pigment migration (pupil mechanism, black; see *Appendix 2—figure 1*) and rhabdomere contractions (red; see Appendix 7), operate with refractory photon sampling by 30,000 microvilli (blue) and their quantum bump dynamics (sample duration and jitter, green; see Appendix 1) in modulating light input to a photoreceptor, and consequently its voltage output. Together these mechanisms, which work to eliminate excess photons, enable efficient encoding of behaviorally important visual information at daylight conditions (*Figures 1–2*) by covering a broad range of temporal light changes; with the pupil and rhabdomere contractions being slower than the photon sampling dynamics.
DOI: https://doi.org/10.7554/eLife.26117.042

To function optimally, the photomechanical adaptations need to regulate input from the ambient illumination so that the temporal light changes they let through would cause maximal sample (quantum bump) rate changes (*Song and Juusola, 2014*). The higher the photoreceptor's sample rate changes, the higher its rate of information transfer (*Song and Juusola, 2014*). Too bright light would saturate microvilli because most of them would be rendered refractory, reducing their dynamic sample counts. Conversely, too dim light would not utilize microvilli population fully, producing low (suboptimal) sample counts. Therefore, the optimal light intensity throughput for maximum information capture is somewhere between. This value is expected to depend upon five factors:

- Light intensity time series structure (we consider all the 20 light patterns tested in *Figure 1B*)
- Number of microvilli (~30,000 in a R1-R6 *Drosophila* photoreceptor)
- Refractory period distribution of the microvilli (full range: 50–500 ms in a R1-R6 *Drosophila* photoreceptor)
- Quantum bump waveform (sample duration)
- Quantum bump latency distribution ('sample jitter')

Detailed tests and descriptions of why and how these factors contribute to encoding in fly photoreceptors are given in our previous publications (*Juusola et al., 1994*; *Henderson et al., 2000*; *Juusola and Hardie, 2001b*; *Juusola and Hardie, 2001a*; *Juusola and de Polavieja, 2003*; *Song et al., 2012*; *Song and Juusola, 2014*; *Juusola et al., 2015*).

## Hypothesis testing and verification

To test the photomechanical light-screening hypothesis, we simulated voltage responses to 20 different light patterns (*Figure 1B*) at 15 different light intensity (or brightness) levels, which ranged from $5 \times 10^4$ to $1 \times 10^6$ photons/s (5, 6, ..., 9 $\times 10^4$; 1, 2,. .., 9 $\times 10^5$; 1 $\times 10^6$). In each simulation, the stochastic photoreceptor model generated 20 independent responses to the given 2-seconds-long (2000 points) light pattern of a given intensity, following the

published procedures (*Song and Juusola, 2014*). These 20 responses were used to estimate the model's rate of information transfer for that specific stimulus pattern (1/20) at that specific light level (1/15). So all together, we could have simulated 20 repeated photoreceptor outputs to 300 (20 × 15) different 2,000-points-long stimulus patterns. But because the model's maximum information transfer rate estimates turned out to be relatively straightforward to determine for many light patterns (*cf. Appendix 2—figure 3*), the total number of simulations never reached this limit. Nevertheless, being computationally expensive, the stochastic simulations took months to complete.

Crucially, in all the simulations, the photoreceptor model was exactly the same. Its stochastic bump production dynamics (waveform, latency and refractory distributions) were governed by light-adapted values with every single parameter fixed, and these parameter values were unchanged in each simulation. The supplement of the reference (*Song et al., 2012*) lists these parameter values, which were collected from intracellular experiments or logically extrapolated to be biophysically realistic for light-adapted *Drosophila* photoreceptors. This supplement is downloadable from: http://www.sciencedirect.com/science/article/pii/S0960982212006343

*Figure 3B* shows the simulated voltage responses (traces above) that carried the maximum information transfer rates for the 20 tested light patterns (traces below) and the corresponding intensity levels (as effective photons/s) that evoked them. The simulations match the overall size, appearance and dynamics of the real recordings astonishingly well (*Figure 1B* and *Figure 1—figure supplement 1B*), indicating that the photoreceptor model, with its photomechanics optimizing light input intensity, samples and integrates light information much like its real-life counterparts. Notice that the optimal light intensity is the same for the different bandwidth (20, 50, 100, 200 and 500 Hz) stimuli within one BG. But for each BG (BG0, BG0.5, BG1 and BG1.5) this optimum is different. For example, for BG0, which results in bursty (high-contrast) stimulation, the optimal light intensity level is $\sim 8 \times 10^5$ photons/s. Whereas for BG1.5 of low contrast Gaussian white-noise stimulation, this is 10-times lower ($\sim 8 \times 10^4$ photons/s).

We express these intensity levels in units of *effective* photons/s. This is because, theoretically, we have deduced the mean photon throughput that effectively fluxes into microvilli for a *Drosophila* photoreceptor to sample the best estimates of the given light stimuli. In other words, if the photomechanical screening mechanisms set the light input intensity for maximal information capture, as is our hypothesis, then these light intensity values should also closely approximate the actual photon absorption changes that drive phototransduction in the real experiments (as recorded intracellularly from wild-type R1-R6 photoreceptors, which have the normal pupil mechanism and photomechanical rhabdomere contractility; *Figure 1* and *Figure 1—figure supplement 1B*). Note that as the preceding photomechanical light screening mechanisms eliminate photons, a photoreceptor's photon absorption rates will always be considerably lower than the photon emission rates from the light source.

Because of the remarkable dynamic correspondence between the experiments (*Figures 1– 2*) and the theory (*Figures 3–4*) over the whole tested encoding space, we now judge that this hypothesis must be largely true. Importantly, this realization opens up new ways to analyze photoreceptor function. For example, by making the general assumption that the input - photon absorptions (and light emission from our LED light source) - follows Poisson statistics, we could further estimate the lower information transfer rate bound for each tested light intensity pattern (as absorbed by an average R1-R6 photoreceptor), and consequently the upper bound for the *Drosophila* photoreceptor's encoding efficiency (*e.g. Figure 2D*, *Figure 2—figure supplement 1D* and *Figure 4D*). More details about this assumption and the analysis are given at the end of this appendix (Appendix 2).

## New insight into maximal visual encoding of different stimulus statistics

The reasons why and how the optimal light intensity input (that drives a photoreceptor's information transfer maximally) is different for bursts and Gaussian white-noise stimulation are summarized in *Appendix 2—figure 3*. Here we assess both cases using data from the stochastic *Drosophila* R1-R6 photoreceptor model simulations, starting with light bursts.

### Bursts

(*Appendix 2—figure 3A-–D*). These light intensity time series characteristically contain periods of longer dark contrasts, intertwined with brief and bright contrast events, as shown for 100 Hz bandwidth stimulation (*Appendix 2—figure 3A*, dark-yellow trace). Based on our previous analyses (*Song and Juusola, 2014*), longer dark contrasts help to recover more refractory microvilli than equally-bright stimuli without these features, improving neural information capture. This makes it more difficult for bursty stimuli to saturate the photoreceptor output. By increasing the stimulus intensity 8-fold, here from $1 \times 10^5$ to $8 \times 10^5$ effective photons/s, simply evoked larger macroscopic responses. These, thus, integrated more samples (bumps); as indicated by the larger (black) and smaller (blue) trace, respectively.

Because noise changes little in light-adapted photoreceptor output (*Juusola et al., 1994*; *Juusola and Hardie, 2001b*, Juusola and Hardie, 2001a*Juusola and Hardie, 2001a*; *Song et al., 2012*; *Song and Juusola, 2014*) (*Figure 2—figure supplement 2*), the larger responses to brighter bursts have higher and broader signal-to-noise ratio, $SNR_{output}(f)$, (*Appendix 2—figure 3B*). This, in turn, results in higher information transfer rate estimates, $R_{output}$ (*Appendix 2—figure 3C*), following Shannon's equation (*Shannon, 1948*):

$$R_{output} = \int_0^\infty (log_2[SNR_{output}(f) + 1])df \qquad \text{(A2.1)}$$

Note that with 1 kHz sampling rate used in every experiment, this estimation did not integrate information rate for frequencies from 0 to infinite, but from 2 to 500 Hz instead. However, the limited bandwidth would not considerably affect estimation results because: (i) high-frequency components have SNR <<1 and therefore contain mostly noise. (ii) Whereas even a high $SNR_{output}(f)$ contains little information in its low-frequency components, below 2 Hz. Note also that we have previously shown the generality of Shannon's information theory for estimating information transfer rates of continuous (analogue) repetitive responses, irrespective of their statistical structure (*Juusola and de Polavieja, 2003*; *Song and Juusola, 2014*). That is, for sufficient amount of data, Shannon's equation and triple extrapolation method, which is free of signal and noise additivity and Gaussian distribution assumptions, give comparable rate estimates. Thus, these estimates should evaluate the simulations' relative information rate differences truthfully; *i.e.* consistently with only small errors.

Markedly, a photoreceptor's performance is systematically better to the brighter bursts (black line) than to the less bright ones (blue line), irrespective of their bandwidth (*Appendix 2—figure 3C*). Thus, for the brighter bursts, more microvilli are dynamically activated, generating larger sample rate changes. These bumps sum up larger (and more accentuated – see [*Song and Juusola, 2014*]) macroscopic responses, packing in more information than the corresponding responses to the less bright bursts.

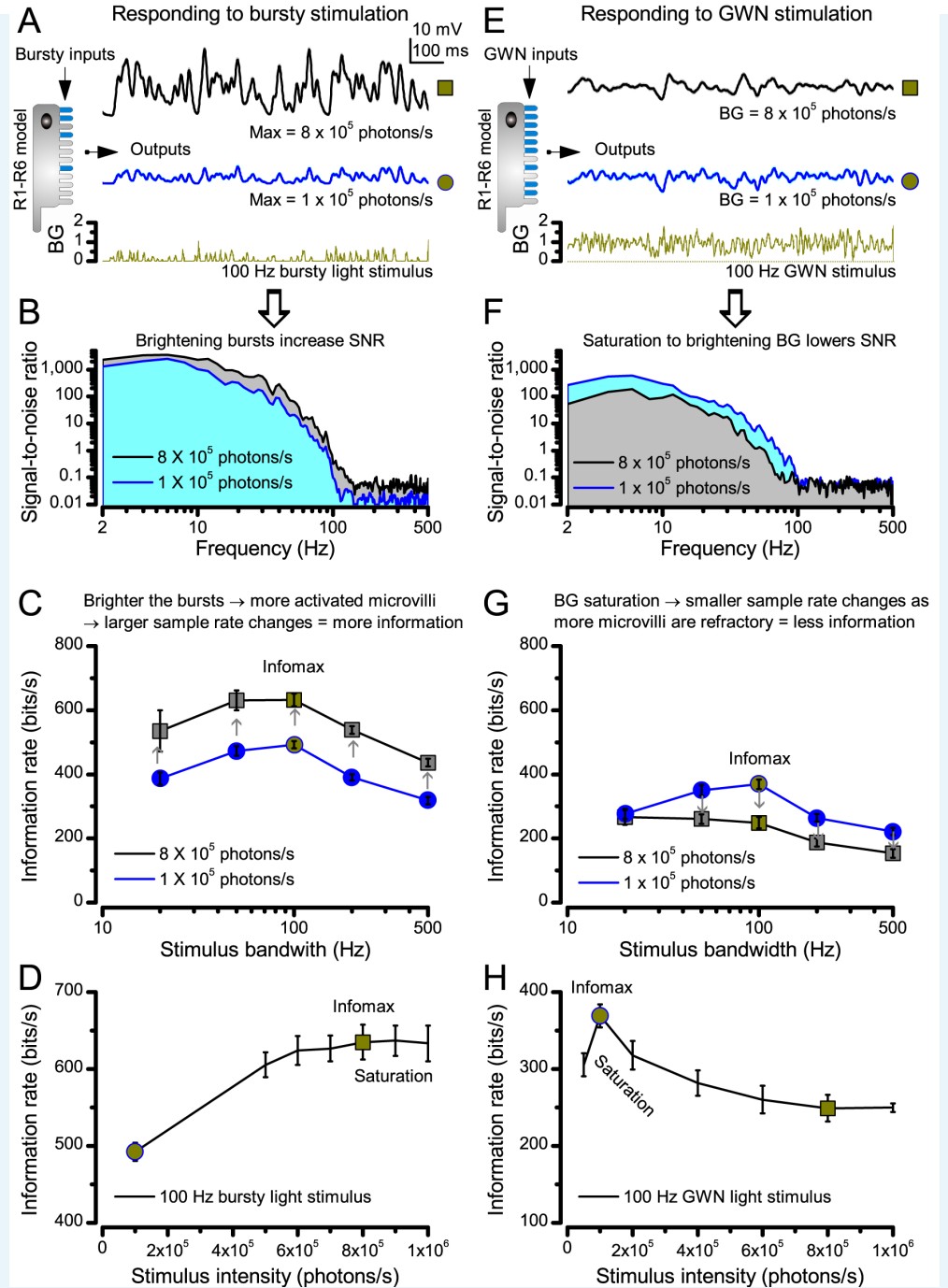

**Appendix 2—figure 3.** Estimating optimal light intensity for 100 Hz high-contrast bursts and Gaussian white-noise (GWN) stimuli for a R1-R6 photoreceptor's maximal information transfer. We hypothesize that the role of photomechanical adaptations, which include the intracellular pupil and contracting rhabdomere (Appendix 7) of a photoreceptor, is to maximize information capture of microvilli by dynamically adjusting the light input falling upon them. The left side of the figure shows how encoding of light bursts (**A–D**) depends upon light intensity; the right side shows the same for GWN (**E–H**). (**A**) Owing to sufficient dark periods, a photoreceptor's sampling units (microvilli) have enough time to recover from their refractoriness even after they have responded to very bright bursts. This enables a photoreceptor to maintain a large pool of available microvilli to sum up high sample (bump) rate changes to any new incoming input, generating larger macroscopic responses to the brighter bursts ($8 \times 10^5$ photons/s, black) than to the less bright bursts ($10^5$ photons/s, blue).

(B) Macroscopic responses with larger sample rate changes (black trace, grey area) have higher and broader signal-to-noise ratios (*Song and Juusola, 2014*). (C) Correspondingly, as the sample sizes (bumps) are similar to both stimuli (*cf. Figure 2—figure supplement 2B*), the larger responses carry a higher information transfer rate (*Song and Juusola, 2014*), irrespective of the tested stimulus bandwidth. (D) Therefore, a photoreceptor's information transfer rate to bursty inputs increases with light intensity, until the sample rate changes eventually saturate at $8 \times 10^5$ photons/s; when most of 30,000 microvilli become refractory (*i. e.* more microvilli are refractory than available to be light-activated). (E) A R1-R6 generates similar size responses to the brighter ($8 \times 10^5$ photons/s, black) and the less bright ($10^5$ photons/s, blue) GWN inputs. But the response to the less bright input shows more high-frequency modulation. (F) Consequently, the response to the less bright input (blue area) has higher and broader signal-to-noise ratio than the response to the brighter input (grey area). (G) This is reflected also in the photoreceptor's information transfer rate, regardless of the GWN bandwidth. (H) Information transfer rate in macroscopic photoreceptor output to GWN stimulation saturates at 8-times less bright intensity levels than to bursts (D), reaching its maximum at $10^5$ photons/s.

DOI: https://doi.org/10.7554/eLife.26117.043

However, because a *Drosophila* photoreceptor has a finite amount of microvilli, each of which - once activated by a photon's energy - stays briefly refractory, its sample rate changes and thus signaling performance first increases monotonically until about $6 \times 10^5$ photons/s, before gradually saturating, and eventually decreasing, with increasing burst brightness (*Appendix 2—figure 3D*). The photoreceptor model's maximum information transfer rate estimate ($R_{max} = 631 \pm 31$ bits/s; marked by a square) for 100 Hz bright bursts is reached at the optimal stimulus intensity of $8 \times 10^5$ effective photons/s. In other words, this is the amount light the photomechanical adaptations, including the intracellular pupil mechanism and rhabdomere contractions (see Appendix 7), should let through (to be absorbed) in bright daylight for the fly to see bursty real-world events best. The corresponding performance estimate with the less bright bursts ($10^5$ effective photons/s) is $493 \pm 12$ bits/s (circle).

## Gaussian white noise

(GWN, *Appendix 2—figure 3E–H*). Because GWN lacks long dark contrasts, refractory microvilli have fewer chances to recover (*Song and Juusola, 2014*). Consequently, photoreceptor output to GWN begins to show signs of saturation at lower light intensity levels. *Appendix 2—figure 3E* shows responses to 100 Hz bandwidth GWN with the mean intensity of $1 \times 10^5$ (blue) or $8 \times 10^5$ (black) effective photons/s, respectively. Both responses are about the same size, but the one to the brighter stimulation carries less high-frequency modulation. As more microvilli become refractory, smaller sample rate changes (modulation) map light changes into macroscopic responses. (How refractoriness dynamically modulates bump counts and macroscopic response waveforms was analyzed in detail recently (*Song and Juusola, 2014*), and thus is not repeated here). Hence, the response to the brighter GWN (black/grey) has lower and narrower signal-to-noise ratio (*Appendix 2—figure 3F*) over the frequency range than the responses to the 8-times less bright GWN (blue). Naturally, the same holds true for the photoreceptor's information transfer rate estimates (*Appendix 2—figure 3G*); the less bright GWN gives consistently a better performance (blue), irrespective of the used stimulus bandwidth.

Again, the amount of microvilli and their refractoriness curb a photoreceptor's signaling performance. But to minimize their impact on encoding GWN, the photomechanical screening needs to be more restrictive, letting in less light. With brightening 100 Hz GWN (*Appendix 2—figure 3H*), the model's information transfer rate first steeply increases until its peak ($R_{max} = 369 \pm 15$ bits/s; marked by a circle) at $10^5$ photons/s, and then swiftly declines as progressively more microvilli become refractory and fewer samples are being produced. The corresponding transfer rate estimate for $8 \times 10^5$ photons/s GWN is $249 \pm 17$ bits/s (square). Notice, however, that although these results quantify the optimal photon absorption rate for generating maximally informative responses to GWN, such performance is far from the

models' estimated information capacity of 631 bits/s (*cf.* **Appendix 2—figure 3D** and **Figure 4C**).

## Simulations' maximal information transfer largely match those of recordings

We next compare the maximum information transfer rate estimates (squares) of the model simulations to those of corresponding *in vivo* recordings (circles) for all BG0 (bursts) and BG1 (GWN) stimuli (**Appendix 2—figure 4**). The simulated performance is very close to the measured mean performance for all the tested stimuli, typically falling within the standard deviation of the recordings' information transfer.

In further inspection, two interesting observations can be drawn from this data. First, for the GWN stimuli, irrespective of their bandwidth, the maximum information transfer rate estimates of the model (dotted line) are just a few bits/s (1–10%) higher than the corresponding mean estimates of the real recordings (continuous line). These small differences are probably caused by recording noise. Second, the simulations to bursty stimuli carry less information than the corresponding best recordings, and the recordings show variations in their information transfer.

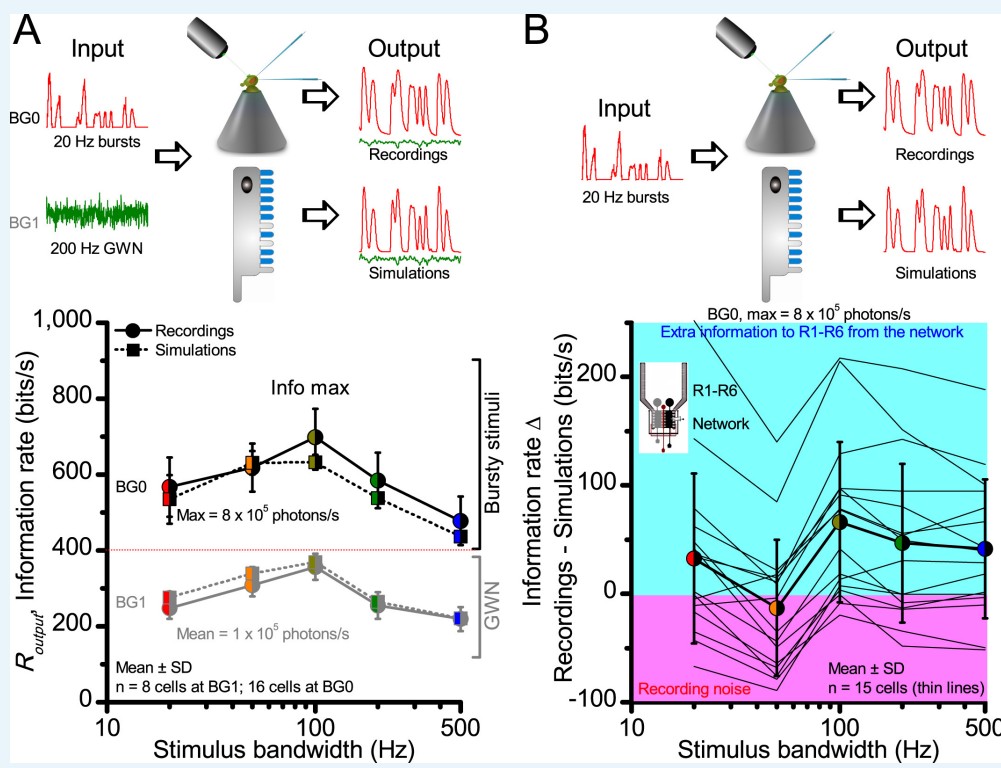

**Appendix 2—figure 4.** Information transfer rate estimates, $R_{output}$, of *in vivo* recordings and model simulations show similar encoding dynamics. (**A**) Comparison of corresponding information transfer rates of R1-R6 recordings and stochastic model simulations to light bursts and Gaussian white noise (GWN) stimuli of different bandwidths. The recorded and simulated information transfer estimates correspond closely over the whole tested encoding space (*cf.* **Figure 2—figure supplement 1** and **Figure 4**). (**B**) Their differences to light bursts help to identify extra information in the recordings, which likely comes from the lamina network (through gap-junctions (**Wardill et al., 2012**) and feedback synapses [**Zheng et al., 2006**]) to individual photoreceptors. The clear variability between different recordings from individual cells (continuous thin lines) indicates that some R1-R6s may receive up to 200–250 bits/s of information from the network, whereas others receive less (cyan background). Some recordings likely contained more instrumental/experimental noise (pink background), which could render their information transfer rates (in particular to low-frequency bursts) less than

that of the model; some of this noise likely comes from low-frequency eye and photoreceptor movements (*cf. Figure 2—figure supplement 2*). Thick line and error bars give the average information transfer rate difference between the recordings and the model (~0–50 bits/s). The data implies that the extra network information to R1-R6s in vivo is mostly at high burst frequencies (100–500 Hz).

DOI: https://doi.org/10.7554/eLife.26117.044

## Each R1-R6 receives different amounts of information from the network

In *Appendix 2—figure 4B*, the difference in information rate estimates between the corresponding recordings and simulations to the bursty stimuli is plotted for each complete recording series (thin lines) of individual cells. The thick line gives the mean difference to all these cells' performance. Most noticeably, some photoreceptor cells carry ~100–200 bits/s more information from the bursty stimuli, but many other cells also show information rates that surpass the model's performance (see also *Figure 5*). Any information surplus (cyan background) presumably comes from the lamina network (*Zheng et al., 2006*; *Wardill et al., 2012*); through gap-junctions and feedback synapses from the cells that sample information from the same small visual area (due to neural superposition [*Vigier, 1907b*; *Vigier, 1907a*; *Agi et al., 2014*]). The photoreceptor model lacks this network information.

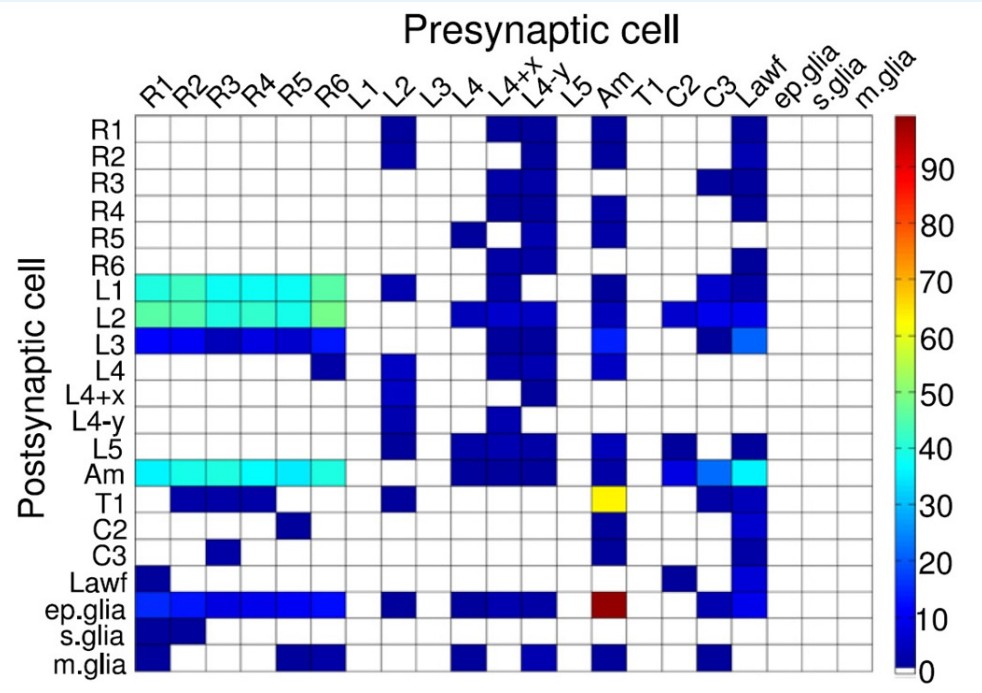

**Appendix 2—figure 5.** Synaptic connectivity between neurons within a lamina cartridge. Synapse numbers color-coded as indicated by the column on the right. Large monopolar cells, L1-L5; amacrine cell, Am; C2 and C3 are retinotopic centrifugal fibers from the next synaptic processing layer, medulla. Image from (*Rivera-Alba et al., 2011*).

DOI: https://doi.org/10.7554/eLife.26117.045

In fact, each of the six R1-R6s, which pool their inputs in the same lamina cartridge for feed-forward synaptic transmission, should show different information transfer rates. This is because the lamina connections are asymmetric (*Appendix 2—figure 5*). Electron micrographs have shown that R1, R2, R3, R4, R5 and R6 make different amounts of feedback synapses with the lamina interneurons (*Meinertzhagen and O'Neil, 1991*; *Rivera-Alba et al., 2011*). Most feedbacks are provided by neurons belonging to the L2/L4 circuits (*Meinertzhagen and O'Neil, 1991*; *Rivera-Alba et al., 2011*). Whilst same-cartridge connections are selectively from L2 to R1 and R2 and from L4 to R5, all R1-R6s receive feedback signals from L4 of

neighboring cartridges. There are further connections from Am to R1, R2, R4 and R5, and glia are also synaptically connected to the network (*Meinertzhagen and O'Neil, 1991*; *Rivera-Alba et al., 2011*), but only R6 makes direct gap-junctions (*Shaw et al., 1989*) with R7 or R8. These asymmetric functional connections (*Zheng et al., 2006*) may largely explain the variability in photoreceptor output (*Figure 1—figure supplement 1*) and information rates (*Figure 2—figure supplement 1C*).

Our recent work (*Wardill et al., 2012*) further showed that during naturalistic stimulation R6 can receive up to ~200 bits/s of information from R8, as channeled through gap-junctions between these cells. Therefore, we infer here that the recordings with the highest information transfer rates (~850 bits/s) were probably of R6-type, which directly receive extra information from its R8y and R7y neighbors (*Shaw, 1984*; *Shaw et al., 1989*; *Wardill et al., 2012*) (*Appendix 2—figure 6*). Conversely, the recordings, in which information rates were lower than those of the simulations (*Appendix 2—figure 4B*, pink background), carried presumably more recording/experimental noise, with one potential source being minute retinal movements (see Appendixes 4, 6–9).

Our intracellular recordings establish that during bright light stimulation, the voltage output of an individual photoreceptor is highly repeatable (*cf.* *Figure 1*). Consequently, our recording system could be used to study variability among individual R1-R6 photoreceptors of the fly eye. We discovered that for the same stimuli the characteristic output waveforms and frequency distributions of one particular cell are typically different to those of another photoreceptor (*Figure 5—figure supplement 1*), even when recorded from the neighboring cells in the very same eye (by the same microelectrode). Because the signal-to-noise ratios of the recordings were very high (*Figure 2*), sometimes over 6,000, it was evident that the observed cell-to-cell variability had little to do with the quality of the recordings. Hence, in the *Drosophila* retina, R1-R6s show intercellular variability that is far greater than the observed small intracellular variability.

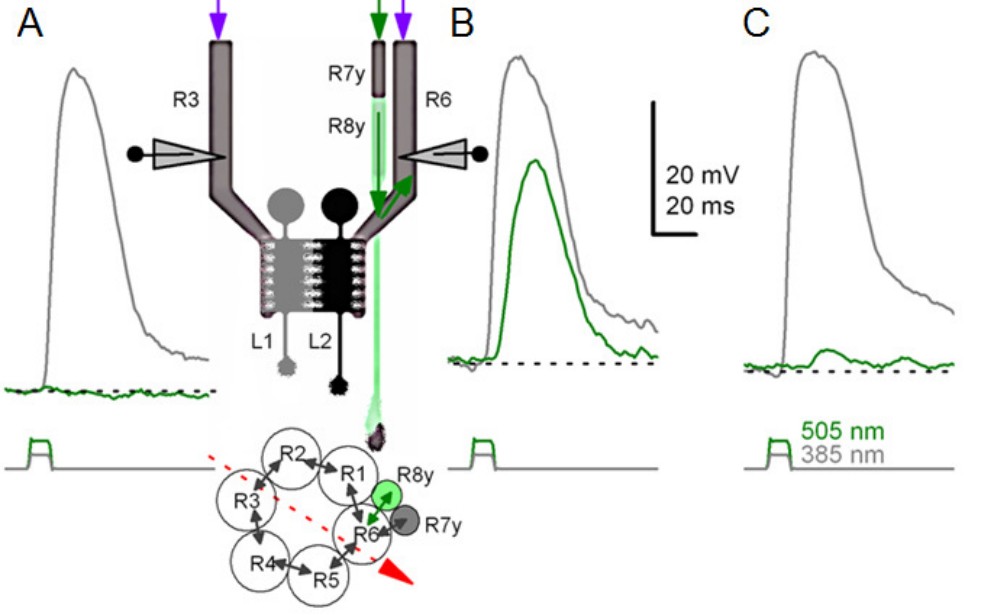

**Appendix 2—figure 6.** Gap-junction spread information. Because of axonal gap-junctions between R6 and R7/R8 photoreceptors in the lamina (*Shaw, 1984*; *Shaw et al., 1989*), R1-R6s that have been genetically engineered to express UV-sensitive Rh3-rhodopsin ('UV-flies') can still respond to green light by different degrees (*Wardill et al., 2012*). This flow of extra 'color'- information can be readily identified in intracellular responses of different R1-R6 photoreceptors in the same 'UV-fly' to very bright UV (385 nm) and green-yellow (505 nm) flashes. (**A**) First cell responded to UV but not to green. (**B**) Next cell (likely R6 in the same or neighboring neuro-ommatidium) responded to both UV and green. This cell cannot be R7y/p, which are less green-sensitive, or R8y/p, which are less UV-sensitive. Inset highlights a

hypothetical recording path, somewhere close to the retina/lamina border (red arrow), and gap-junctions (black arrows) between photoreceptor axons. Histaminergic L1 and L2 cells receive visual information from R1-R6 photoreceptors' output synapses in the same neuro-ommatidium. (**C**) Another cell responded to UV and weakly to green-yellow. Modified from (*Wardill et al., 2012*).

DOI: https://doi.org/10.7554/eLife.26117.046

Collectively, these results strongly suggest that every R1-R6, which is pooled in one lamina cartridge under the developmental neural superposition principle (*Agi et al., 2014*) to transmit information about light changes in a small area of visual space to visual interneurons (L1-L3 and Am) (*Meinertzhagen and O'Neil, 1991*; *Zheng et al., 2006*; *Zheng et al., 2009*; *Rivera-Alba et al., 2011*), has, in fact, its own unique output. Besides asymmetric connectivity within a neuro-ommatidium, some of the observed response variations may also reflect different recording locations. For example, *Drosophila* R1-R6s in the front of the eye might show different responsiveness to those at the back, as already shown for localized polarization-sensitivity differences (*Wernet et al., 2012*). Compound eyes of many insects exhibit structural adaptations that alter their lens sizes and shapes locally, such as bright or acute zones for increasing sensitivity or resolution, respectively (*Land, 1998*). Furthermore, electrophysiological recordings in some fly species suggest that their photoreceptor output vary across the eyes and could be tuned to the spatial and temporal characteristics of the light environment (*Hardie, 1985*; *Laughlin and Weckström, 1993*; *Burton et al., 2001*).

## Variable sampling matrix protects from aliasing, improving vision

With each R1-R6 having variable 'network-tuned' (and possibly 'location-tuned') encoding properties and output, and with each image pixel being sampled through variable size rhabdomeres (see Appendix 5, *Appendix 5—figure 1*) and ommatidial lenses (the photoreceptors' receptive fields vary; see Appendix 4, *Figure 7—figure supplement 1* and interommatidial angles change progressively from front to back [*Gonzalez-Bellido et al., 2011*]), the *Drosophila* eye should generate reliable neural estimates of the variable world. This is because a sampling matrix made out of variable pixels (neuro-ommatidia), in which size and sensitivity show random-like constituents:

- *prevents aliasing of image information* (*Appendix 2—figure 7*); see also (*Yellott, 1982*; *Dippé and Wold, 1985*; *Juusola et al., 2015*).
- mixes color information to the R1-R6 motion vision channel, whitening its spectral sensitivity (*Appendix 2—figure 7E*) (*Wardill et al., 2012*), which is a prerequisite for an optimal motion detector (*Srinivasan, 1985*).

Aliasing effects are reduced by sampling faster and/or with finer spatial resolution, and eliminated by sampling more than twice over the highest stimulus frequency (*Cover and Thomas, 1991*). The Nyquist–Shannon sampling theorem establishes a sufficient condition for a sample rate that enables a discrete sequence of samples to capture all the information from a continuous-time signal of finite bandwidth. Specifically, it only applies to a class of mathematical functions having a Fourier transform that is zero outside of a finite region of frequencies. This condition, however, cannot be fully realized in sensory systems, which show finite spatiotemporal sampling resolution and evolved around $1/f^n$-stimulus (*Field, 1987*; *van Hateren, 1997b*) distributions of the real-world objects and events. Because any physical transformation affects signal and noise equally (data processing theorem [*Shannon, 1948*; *Cover and Thomas, 1991*]) and because real-world low-pass filters, such as a lens, cannot cut-off sharply at an exact point, but instead gradually eliminate frequency components and exhibit a fall-off or roll-off slope, aliasing effects would not be removed completely in an ordered sampling matrix. Therefore, to prevent phantom sensations of aliased signals fooling the brain and perception of physical reality, sampling matrixes of sensory systems must entail stochastic variations.

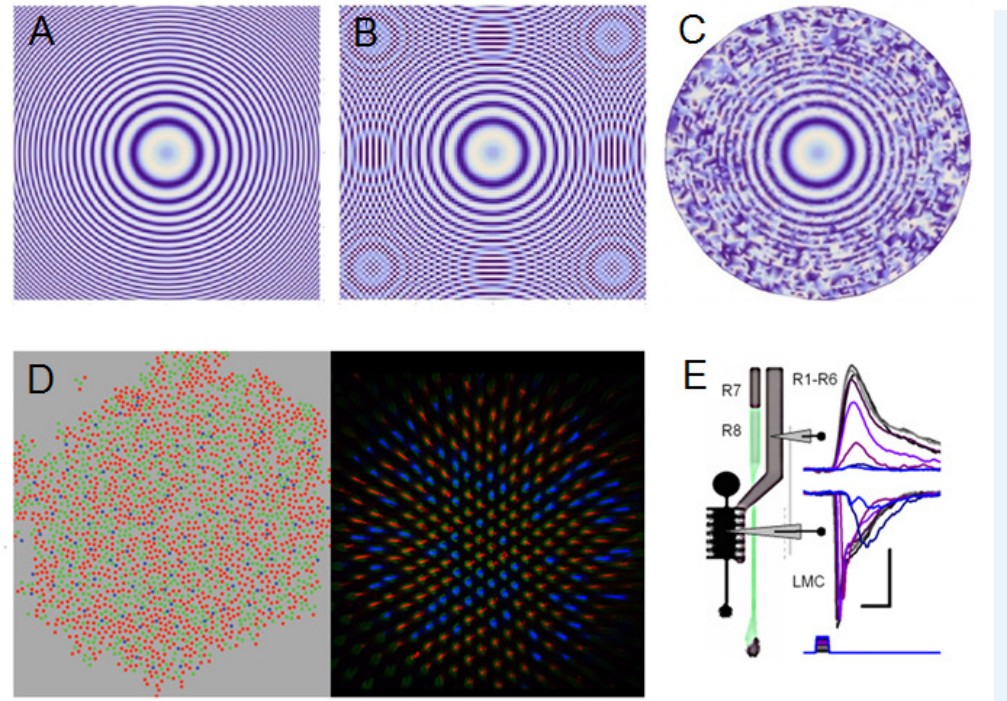

**Appendix 2—figure 7.** Projected image of $Sin(x^2 + y^2)$ function is used to illustrate the effect of aliasing and how stochastic variability in the sampling matrix combats aliasing effectively. (**A**) $Sin(x^2+y^2)$ is plotted with 0.1 resolution. (**B**) Under-sampling the same image by an ordered matrix leads to aliasing: ghost rings appear when the image (of the function) is sampled with 0.2 resolution. Aliasing is a critical problem, as the nervous system cannot differentiate the fake image rings from the original real image. (**C**) Sampling the image (**A**) with a random matrix may lose some of its fine resolution, due to broadband noise, but such sampling is anti-aliasing; sampling with random points at 0.2 resolution. (**D**) Color photoreceptor distributions across macaque (*Field et al., 2010*) (red, green and blue cones; left) and *Drosophila* retina (*Vasiliauskas et al., 2011*) (R7y and R7p receptors; right) show random-like sampling matrixes, suggesting that this sampling matrix sensitivity randomization would have an anti-aliasing role. (**E**) Crucially, by integrating and redistributing R1-R6 outputs with additional gap-junctional inputs from randomized R7/R8 color channels (*Wardill et al., 2012*) (**D**) and *Appendix 2—figure 6*) for each image pixel during synaptic transmission to LMCs, any broadband sampling noise should be much reduced and the R1-R6 (motion) channel's spectral range whitened (*Wardill et al., 2012*). Note how LMC output peaks before the corresponding R1-R6 output. Scale bars: 10 mV / 20 ms. Sub-figure (**E**) is modified from (*Wardill et al., 2012*).

DOI: https://doi.org/10.7554/eLife.26117.047

Whilst temporal and topological sampling matrix variations in retinae combat aliasing (*Yellott, 1982*; *Juusola et al., 2015*), their trade-off is broad-band noise (*Dippé and Wold, 1985*) (*Appendix 2—figure 7C*; the Python script for these simulations is downloadable from: https://github.com/JuusolaLab/Microsaccadic_Sampling_Paper/tree/master/AntialiasingByRandomisation). This noise, however, is much reduced (or nearly eliminated) by parallel sampling of the same information (*Song and Juusola, 2014*; *Juusola et al., 2015*). For example, noise reduction occurs naturally in the fly eye - both in time and in space. In every R1-R6 photoreceptor, 30,000 microvilli sample discrete information stochastically in time, generating virtually aliasing-free macroscopic responses of very high signal-to-noise (*Figure 5—figure supplement 1*). Whereas, across the lamina neuro-ommatidia of variable connectivity and spectral sensitivity, neural superposition integrates local R1-R8 signals of overlapping information from each pixel (a small largely aligned area in the visual space) to improve the signal-to-noise ratio of the sampled images (see Appendix 5). Such images should provide the brain reliable and maximally informative estimates of the environment.

## Estimating a R1-R6's encoding efficiency

A photoreceptor's encoding efficiency, $\eta$, is the ratio between the information rates of the voltage output, $R_{output}$, and the effective light input (photon absorptions), $R_{input}$, that drove it:

$$\eta = \frac{R_{output}}{R_{input}} \qquad (A2.2)$$

As we already had determined the maximum photoreceptor output information rates, $R_{output}$ (*Equation A2.1*; *Figure 2C*, *Figure 2—figure supplement 1C* and *Figure 4C*), only the corresponding information rates of the effective light stimuli, $R_{input}$, needed to be worked out. Because the output simulations' maximum information rates matched well the corresponding mean rates of the real recordings (*Appendix 2—figure 6A*), we had extrapolated successfully each effective light intensity (photon absorption) time series that drove the voltage response. Therefore, we could now estimate the rate of information transfer of the effective light input by making the following assumptions:

- Photon emission from the light source (LED) follows Poisson statistics; this may or may not be true (see the discussion below).
- But if true, the effective photons, which survived photomechanical adaptations (*Appendix 2—figure 2*) and were absorbed by a photoreceptor and used for calculating $R_{input}$, should also follow Poisson statistics (*Song et al., 2012*; *Song and Juusola, 2014*; *Juusola et al., 2015*; *Song et al., 2016*).

Photons are thought to be emitted by the light source, such as the LEDs, at random, exhibiting detectable statistical fluctuations (shot noise). Such dynamics can be modelled by Poisson statistics (*Song and Juusola, 2014*). Therefore, as each light stimulus trace differs from any other, with their mean equaling their variance, we could estimate through simulations their average signals and noise, and signal-to-noise ratios, $SNR_{input}(f)$. The corresponding information transfer rates, $R_{input}$, could then be estimated by Shannon's equation (*Equation A2.1*). For each tested stimulus pattern, this was done by using the same amount of simulated input data as with the output data (2,000 points x 20 repetitions) to control estimation bias. More details and examples about Poisson stimulus simulation procedures are given in (*Song and Juusola, 2014*).

- Notice that currently there are no manmade sensors more efficient than the biological photoreceptors themselves for measuring the photon emissions from the LED light source. Therefore, we had no good direct methods to measure the LED's photon rate changes at the same level of accuracy as the photoreceptor output that it evoked. Accordingly, calculating **mutual information** directly between the less accurate light input estimate and the more accurate photoreceptor output would be both impractical and erroneous.

For the simulated inputs and outputs, the data processing theorem (*Shannon, 1948*) dictates that $R_{input} \geq R_{output}$; thus $\eta \leq 1$ ($\leq 100\%$). If not, then one or both estimates are biased or incorrect; information cannot be created out of nothing. However, for the efficiency estimates based on the real recordings, it is quite possible that $R_{output} > R_{input}$, and thus $\eta > 1$ ($>100\%$), because R1-R6s receive extra information from the network (*Appendix 2—figure 5* and *Appendix 2—figure 6*) that is missing from the $R_{input}$ estimates of an average R1-R6 photoreceptor's photon absorptions (*cf. Appendix 2—figure 4B*).

We recognize that there are methodological limitations and unknowns, which may affect the accuracy and consistency of these estimates:

- Experimental and theoretical evidence suggests that photon output of some light sources might be sub-Poisson (*Teich et al., 1984*); meaning, not maximally random. If this were true for our LED, then our approach would slightly underestimate $R_{input}$, used in the experiments, and consequently overestimate *Drosophila* photoreceptors' encoding efficiency.
- Shannon's equation can bias information transfer rate estimates for any corresponding light input and photoreceptor output differently. This is because the signal and noise components of the input and output may deviate from the expected Gaussian by different amounts. Even

though we used systematically the same amount of data for both estimates (20 × 2,000 data points), in the cases where light distribution is skewed (bursty stimuli) but the photoreceptor output is more Gaussian, it is possible that Shannon's equation would underestimate input but not (or less so) output information, causing us to overestimate efficiency.

- Small data chunks limit analyses. In the past, we have compared information transfer rate estimates, as obtained by Shannon's equation to those estimated through the triple extrapolation method (*Juusola and de Polavieja, 2003*), which is directly derived from Shannon's information theory. For ergodic data of different distributions, and when appropriately applied, both methods provided similar estimates (*Figure 2—figure supplement 4*) (*Juusola and de Polavieja, 2003*; *Song and Juusola, 2014*). However, the triple extrapolation method works best with large sets of data; preferably containing ≥30 responses to the same stimulus (*Juusola and de Polavieja, 2003*). In the current study, because of the practical limitations (to map a photoreceptor's whole encoding space within a reasonable time), all the selected recordings and simulations consisted responses to 20 stimulus repetitions. This data size was deemed insufficient for an accurate estimate comparison between the two methods and was not done here. In the analyses, to provide fair comparison between simulations and recordings in all tested conditions, all the data chunks (for the recordings, simulations and stimuli) were exactly the same size (20 × 2,000 points) and they were processed systematically in the same way (apart from the two exceptions we discuss next). Therefore, the data-size bias should be under control and the results comparable within these limits.

- Implementation of Shannon's equation (*Equation A2.1*) in digital computers typically requires windowing of the data chunks (for signal and noise) before calculating their power spectra though Fast Fourier Transfer (FFT). Windowing combats spectral leakage (smearing), but this affects especially low-frequency signals, in which information content is low. So this trade-off can be considered reasonable, and its effect on most performance estimates is marginal. But here as the input and output information transfer is calculated separately, windowing affects more 20 Hz GWN light input than its corresponding photoreceptor output. This is because windowing clips lower frequency power from 20 Hz GWN input, whereas in the simulated and real voltage responses much of this power is nonlinearly translated (through adaptation) to higher frequencies, including those over 20 Hz. The simulated light input, of course, carries no information on frequencies > 20 Hz, but now it has also lost in windowing some of its low-frequency modulation, which the photoreceptors could translate into high-frequency voltage modulation (note, photomechanical phase enhancement can further contribute to this nonlinearity, see Appendix 3). For the two lowest intensity levels only, we judge that because of this methodological bias, the efficiency estimates for the 20 Hz GWN input-output data became unrealistic by a small margin of 10–40 bits/s, implying that $R_{output} > R_{input}$. Therefore, for data to these two stimuli only, we applied box-car windowing (instead of the normal Blackman-Harris type), to retain its low frequency information content, and so to reduce this bias.

Because of all these possible error and bias sources, a *Drosophila* photoreceptor's encoding efficiency ($\eta$) estimates given in this publication must be considered as upper bounds. Nonetheless, for real photoreceptors, it is realistic to expect their maxima to approach 100%, and in some cells (likely R6s) be beyond, for the tested low-frequency stimuli (20 Hz). This is because of the extra information from the network, which is missing from the simulated mean photon absorption estimates. (Note that in the *in vivo* experiments, the light source emits at each moment 100–10,000-times more photons than what can be absorbed by the tested photoreceptor. Thus, the light source's $R_{input}^{emitted}$ always exceeds a photoreceptor's $R_{input}^{absorbed}$).

Overall, the maximum $\eta$ values are slightly higher but consistent with our previous (conservative) estimates of 90–95%, in which the light input intensity to microvilli was inferred by comparing the wild-type photoreceptor performance to that of white-eye mutants, lacking the intracellular pupil. Therefore, we conclude (again conservatively) that the error margin of these new encoding efficiency estimates may reach ±5%.

## Appendix 3

DOI: https://doi.org/10.7554/eLife.26117.048

# Similarities and differences in encoding bursts and other stimuli

## Overview

This appendix explains why the encoding of high-contrast bursts is both highly informative and reliable, proving additional insight to the results presented in *Figures 1–6*. It also explains how we generated the different light intensity time series from panoramic images of natural scenes, following a *Drosophila*'s saccadic walking patterns (*Figure 6*), and how these stimuli were analyzed and used in the experiments.

## Why do bursty responses carry the most information?

*Figure 2—figure supplement 3* shows that R1-R6 photoreceptors can sample more information from high-contrast bursts than from naturalistic light intensity time series ($1/f^n$-stimuli). We can explain this performance difference by the bursty stimuli' proportionally more, and more evenly distributed, long dark contrasts. Such events enable refractory microvilli to recover efficiently from their previous light-activation so that large numbers of them are continuously available to sample ongoing light changes; *i.e.* to transduce photons to quantum bumps. This leads to larger sample (quantum bump) rate changes and, thus, to a higher rate of information transfer (see also: *Song and Juusola, 2014*).

Moreover, fast high-contrast events survive the slower intracellular pupil mechanism (Appendix 2, *e.g. Appendix 2—figure 2*) and photomechanical rhabdomere contractions (Appendix 7) well (*Figure 8C*). And consequently, fewer photons are being filtered out (lost) from high-contrast bursts than from naturalistic stimulation (*Figure 2—figure supplement 3B*, pink trace) or Gaussian white-noise (grey), which adapt the photoreceptors more continuously to the given light background. Therefore, at the level of the light source, bursty stimuli can have much lower power than the other two stimuli to drive photon-to-quantum bump sampling efficiently by 30,000 microvilli.

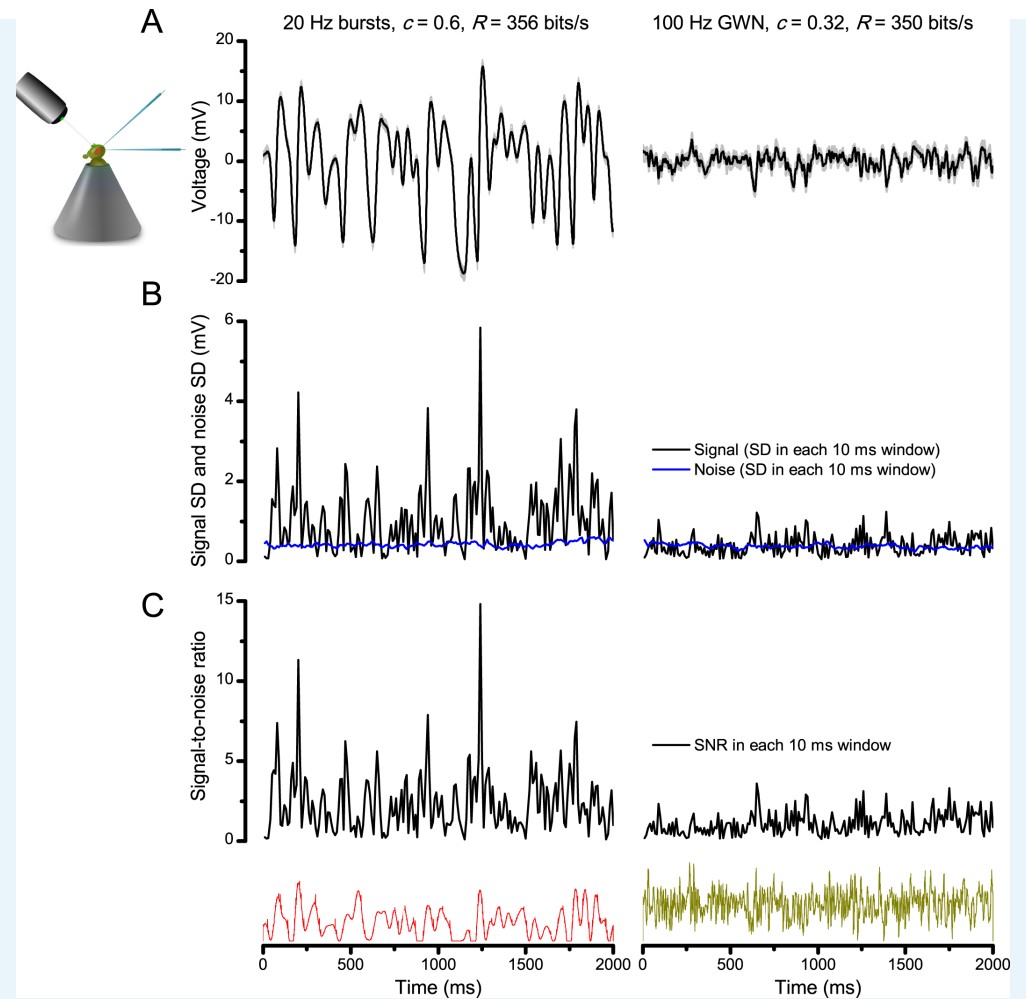

**Appendix 3—figure 1.** Bursts are informative. Whilst having similar information transfer rates, the responses to bursty stimuli (left) show higher signal-to-noise ratio within brief (10–100 ms) perceptually relevant time windows than the responses to Gaussian white-noise (GWN, right). The data are from the same cell. (**A**) 25 intracellularly recorded voltage responses to repeated bursty and GWN light stimuli. Individual responses (superimposed) are shown in light grey and their mean in black traces. Both of these responses series carry similar information contents (~350 bits/s), as estimated by Shannon's equation (see *Equation A2.1*). (**B**) The signal (average response) standard deviation (SD) in 10 ms windows to bursty stimulation vary much more than that to white-noise. The noise variability (blue traces) in the two sets of responses (SD in 10 ms time windows) is similar. (**C**) Signal-to-noise ratio is much greater in the responses to light bursts than to white-noise stimulation; it was calculated as the ratio between signal SD and noise SD, using 10 ms time resolution.

DOI: https://doi.org/10.7554/eLife.26117.049

## Relevance of bursty responses for seeing germane visual patterns

From an information theoretic point of view, the amount of visual information that is encoded by a photoreceptor can be similar for bursty (phasic) and Gaussian (tonic) signals (*Figure 2C*; *cf*. 20 Hz 0.6-contrast bursts and 100 Hz 0.32-contrast GWN). However, when signal-detection-theoretic measures are applied, bursts outperform 'tonic' GWN signals in indicating visual 'things'; *i.e.* the occurrence of perceptually relevant changes in light input. Bursts appear somewhat like all-or-none events (*Appendix 3—figure 1*), having much higher local signal-to-noise ratios in the rising or decaying phases of a photoreceptor's voltage responses (*Zheng et al., 2006*). They tower over the background noise, making their detection much easier than for the tiny blips of the GWN signals. Accordingly, photoreceptors' voltage bursts

support robust transmission of behaviorally relevant visual information and should further improve the reliability of synaptic transmission and perception (*Zheng et al., 2006*; *Zheng et al., 2009*).

## Light intensity time series based on a *Drosophila*'s walk

By combining a fly's movement during free walking (*Geurten et al., 2014*) with natural image statistics, we estimated light intensity stimuli, which a *Drosophila* R1-R6 photoreceptor would face during locomotion through different natural scenes (*Figure 6* and *Video 1*).

To reproduce a fly's saccadic movements and fixations during its 10 s walk (*Figure 6A*), we used the published angular velocity data (*Geurten et al., 2014*) (*Figure 6B*; *Appendix 3—figure 2C*). This was traced from *Figure 1D* in (*Geurten et al., 2014*), using the unobscured section between 0.05–9.95 s, and re-sampled with 1 ms steps. The velocity was integrated over time to give the yaw signal (yaw(t)). Both the velocity and the yaw matched the original published data. Finally, the yaw was wrapped between 0° and 360°.

For generating the light intensity time series stimuli, which a walking fly would experience in different surroundings, we used six different 360° panoramic images (high-density digital photographs of natural scenes), taken from the internet (*Appendix 3—table 1*). These natural scenes were arbitrarily chosen from Google image search results, and we do not know how representative their image statistics are, for example, in respect to the van Hateren database (*van Hateren, 1997a*). Each image's left (0°) and right (360°) side were stitched together to enable continuous viewing over a full rotation. The images were preprocessed in the following way:

- Because stitching can cause errors (distortions), the lower and upper quarters of the images were discarded.
- The color images (*Appendix 3—figure 2A*) were reduced to gray scale, and their gamma correction was removed, enabling us to use their raw intensity values (*Appendix 3—figure 2B*).
- For each image, light intensity values were collected from 15 horizontal line scans taken in regular intervals (from top to bottom; *Appendix 3—figure 2B*).

**Appendix 3—table 1.** The used six panoramic high-resolution digital images of natural scenes were downloaded from:

https://commons.wikimedia.org/wiki/File:Swampy_forest_panorama.jpg

https://commons.wikimedia.org/wiki/File:2014-08-29_11_51_08_Full_360_degree_panorama_from_the_fir-e_tower_on_Apple_Pie_Hill_in_Wharton_State_Forest,_Tabernacle_Township,_New_Jersey.jpg

https://en.wikipedia.org/wiki/File:Helvellyn_Striding_Edge_360_Panorama,_Lake_District_-_June_09.jpg

https://commons.wikimedia.org/wiki/File:Schleienl%C3%B6cher_Hard_360%C2%B0_Panorama.jpg https://farm3.staticflickr.com/2820/9296652749_7c502de9e7_o.jpg

http://www.bodenstab.org/panorama/images/Green%20Valley/panorama.jpg

DOI: https://doi.org/10.7554/eLife.26117.050

As each image spanned 360° horizontally, we could calculate the degree-value between 0° and 360° for each pixel (intensity) in the horizontal line scans (Intensity(angle)). The pixel intensity values (within the chosen horizontal line) were then sampled at the corresponding yaw positions (*Appendix 3—figure 2D*; in degrees) of the fly's walk for each 1 ms time-bin (Intensity(yaw(t))). This generated unique light intensity time series (*Appendix 3—figure 2E*), which mimicked walk-induced photoreceptor stimulation from the given scene. In the sampling, each yaw value was automatically rounded to the closest pixels angle value (*Appendix 3—figure 2C*, blue). Note: this process assumes that, during a free walk, the fly head would not rotate or move vertically. The Matlab script is in: https://github.com/JuusolaLab/Microsaccadic_Sampling_Paper/tree/master/PanoramicIntensitySeries.

We also generated two sets of control light intensity time series data from the same images. first control: to compare saccadic movements to linear movements (named *linear*), we used the same walk's median yaw velocity of 63.3 °/s (median(abs(angularvel(t)))) (*Appendix 3—figure 2D*, red traces). second control: we shuffled the angular velocity trace values (named *shuffled*; gray traces); this removed all time correlations in the velocity trace without affecting its histogram. *Appendix 3—figure 2E* shows two examples of the saccadic (test) and the two control light intensity time series, taken from two different horizontal scan lines (scans 8 and 15 in *Appendix 3—figure 2B*).

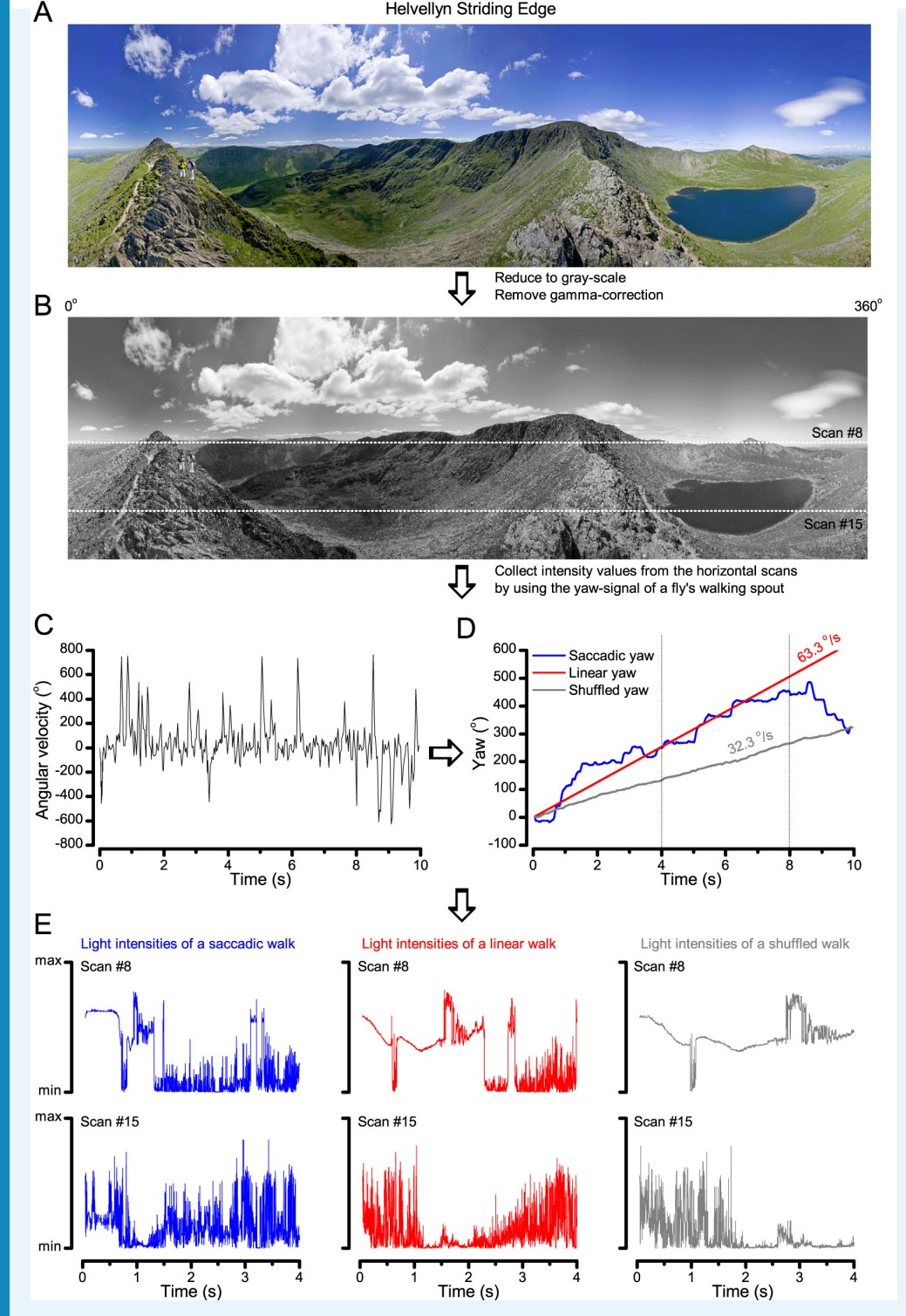

**Appendix 3—figure 2.** Image processing steps. (**A**) 360° panoramic natural images were downloaded from the internet. (**B**) The images were reduced to gray-scale and their gamma-correction was removed to expose their underlying intensity differences more accurately. We then used 15 evenly spaced horizontal (x-axis) line scans to sample their relative intensity values at different vertical (y-coordinate) position. The white dotted lines show two of these scan lines. (**C**) Angular velocities during a free fly's walk, from (*Geurten et al., 2014*). (**D**) These velocities were translated to a yaw signal (degree values) over time (named *saccadic*: blue trace). Red trace shows the *linear* (median) yaw signal, which corresponds to a fly

walking in one direction with the fixed speed of 63.3 °/s. The *shuffled* yaw (gray trace) is generated by randomly selecting angular velocity values from the recorded walk (in **C**). (**E**) These three different yaw signals (°). were then used to sample intensity values from the linear line scans (in **B**; here shown for #8 and #15) at each 1 ms time-bin, generating unique light intensity time series from the panoramic image. Here the corresponding traces are shown for the first 4 s to highlight how differences in locomotion cause large differences in temporal light stimulation (*i.e.* light input to photoreceptors). *Video 1* shows how these three different walking (or locomotion) dynamics (*saccadic*, *linear* and *shuffled*) affect the image stream to the eyes, using the panoramic 'swamp forest' scene (*Figure 6C*).

DOI: https://doi.org/10.7554/eLife.26117.051

## Light intensity time series analysis – differential histograms

Intensity changes for *saccadic*, *linear* and *shuffled* locomotion dynamics were calculated by subtracting two neighboring points in each intensity series (using Matlab '*diff*'-function). The corresponding 'intensity change'-histograms were calculated from all 15 traces per each image. Differential histogram was calculated as the mean of the 'intensity change'-histograms.

We found (predictably) that the differential histograms of the saccadic light intensity time series were sparser than those of the corresponding linear light intensity time series stimuli (of the same median velocity; *Figure 6D*). This was true for all the tested panoramic images (*Appendix 3—figure 3*). Saccadic sampling (blue traces) 'burstified' light input. This was because it increased the proportion of rare large intensity differences between two consecutive moments in comparison to linear sampling (red); *i.e.* saccades made the histogram flanks to reach out further. Furthermore, the fixation periods made it more likely that light intensity over the neighboring moments remained similar or the same (higher proportion of zero values). These features are obvious in *Video 1*.

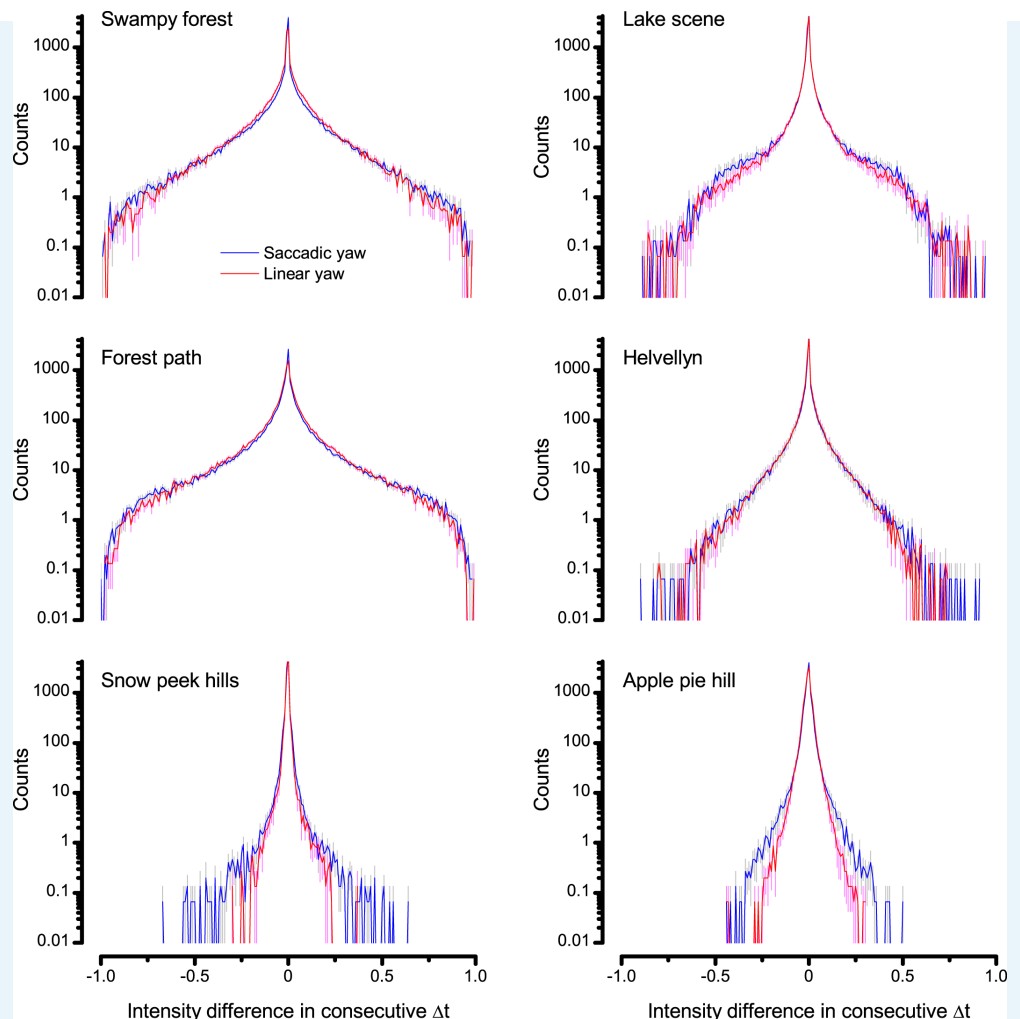

**Appendix 3—figure 3.** Difference histograms of the six panoramic images of natural scenes (used in this study) as scanned by *saccadic* (blue) and *linear* (red) yaw signals of the same median velocity. Saccadic viewing increased the burstiness (*Video 1*) and, thus, sparseness in the difference histograms beyond that of the linear viewing. This was because saccades, proportionally, generated more large light intensity variations; seen by the extended flanks of the histograms. Conversely, fixation periods prolonged the periods of similar light intensities. Thus, the likelihood that the light intensity at one moment would be similar or the same at the next moment was increased; seen by the histograms' higher counts for zero difference.

DOI: https://doi.org/10.7554/eLife.26117.052

## Selecting the stimulus patters and their playback velocity

There are important factors to consider when selecting the stimulus series and their playback velocity for testing how self-motion affects R1-R6 photoreceptors ability to encode naturalistic stimulation.

A fly photoreceptor's information transfer rate is limited by (**i**) the number of its photon sampling units (*Howard et al., 1987*; *Song et al., 2012*; *Song and Juusola, 2014*; *Juusola et al., 2015*) (30,000 and 90,000 microvilli in a typical *Drosophila* and *Calliphora* R1-R6, respectively) and (**ii**) the speed, (**iii**) reliability (jitter) and (**iv**) recoverability (refractoriness) of their phototransduction reactions (*Juusola and Hardie, 2001b*; *Juusola and Hardie, 2001a*; *Song et al., 2012*; *Song and Juusola, 2014*; *Juusola et al., 2015*). In general, the more efficiently the light stimulus utilizes the available microvilli population in generating the larger sample (quantum bump) rate changes, the higher the photoreceptor's information

transfer rate (*Song and Juusola, 2014*) (see Appendix 2). Consequently, the efficiency of photon sampling depends upon the stimulus speed and statistics (*Juusola and de Polavieja, 2003*; *Zheng et al., 2009*; *Song et al., 2012*; *Song and Juusola, 2014*) (*Appendix 3—figure 4*). For naturalistic light intensity time series stimulation (NS), we have previously shown that:

- A R1-R6 photoreceptor's information transfer increases with stimulus playback velocity until saturation, when most of its microvilli likely become refractory for most of the time (*Juusola and de Polavieja, 2003*; *Song et al., 2012*; *Song and Juusola, 2014*; *Juusola et al., 2015*) (*Appendix 3—figure 4A–C*). This information increase results from the increased entropy rate in photoreceptor output (*Appendix 3—figure 4B*), as it reliably packs in more sample rate changes in a given time unit. The corresponding noise entropy rate (*Juusola and de Polavieja, 2003*), similar to noise power (*Figure 2—figure supplement 2A*), remains practically invariable.
- Sample rate changes in R1-R6 output further depend upon the stimulus structure (*Juusola and de Polavieja, 2003*; *Song et al., 2012*; *Song and Juusola, 2014*; *Juusola et al., 2015*) (the distribution of its dark and bright contrasts). For example, R1-R6 output information peaks at lower playback velocity (10 kHz) for $NS_1$, which had fewer long dark-contrast periods (to recover refractory microvilli) than for $NS_2$ (>20 kHz), which had more and more evenly spaced dark-contrasts (*Appendix 3—figure 4*).

For testing how naturalistic *saccadic*, *linear* and *shuffled* locomotion patterns affect *Drosophila* R1-R6s' encoding performance (*Figure 6F*), we used the three corresponding light intensity time series stimuli (first 8,000-points) from 'swampy forest' panorama (line scan #8). These stimulus sequences were selected because each of them carried high-contrast modulation. In the intracellular experiments and model simulations, these stimuli were repeatedly presented one after another to each tested photoreceptor with 4 kHz playback velocity. This stimulus speed was chosen because:

- It would cover well the broad velocity range of natural visual inputs to photoreceptors, including both the slower walking and the faster saccadic flight behaviors.
- Each stimulus could be presented in 2 s, enabling us to collect 30 responses in 1 min and the three different sets of data in 3 min, keeping the recording conditions under control.
- It evokes high R1-R6 information transfer, which for many high-contrast NS sequences approaches their maxima (*Juusola and de Polavieja, 2003*; *Zheng et al., 2009*) (*cf. Appendix 3—figure 4D–E*). Theoretically, *Drosophila* R1-R6 output information rate cannot exceed ~850 bits/s (*Figure 2C*), which was evoked by 100 Hz bursts. Such stimulus entailed the right mix of bright and dark contrasts to optimally utilize a R1-R6's frequency and amplitude ranges.

These stimuli evoked comparable responses (of high information rates) both from R1-R6 photoreceptors *in vivo* and the biophysically realistic R1-R6 model (Appendix 1). The recordings and the simulations showed consistently that the voltage responses to saccadic (*i. e.* the most bursty) light intensity time series had the highest information transfer rate (*Figure 6F*; *Figure 6—figure supplement 1*).

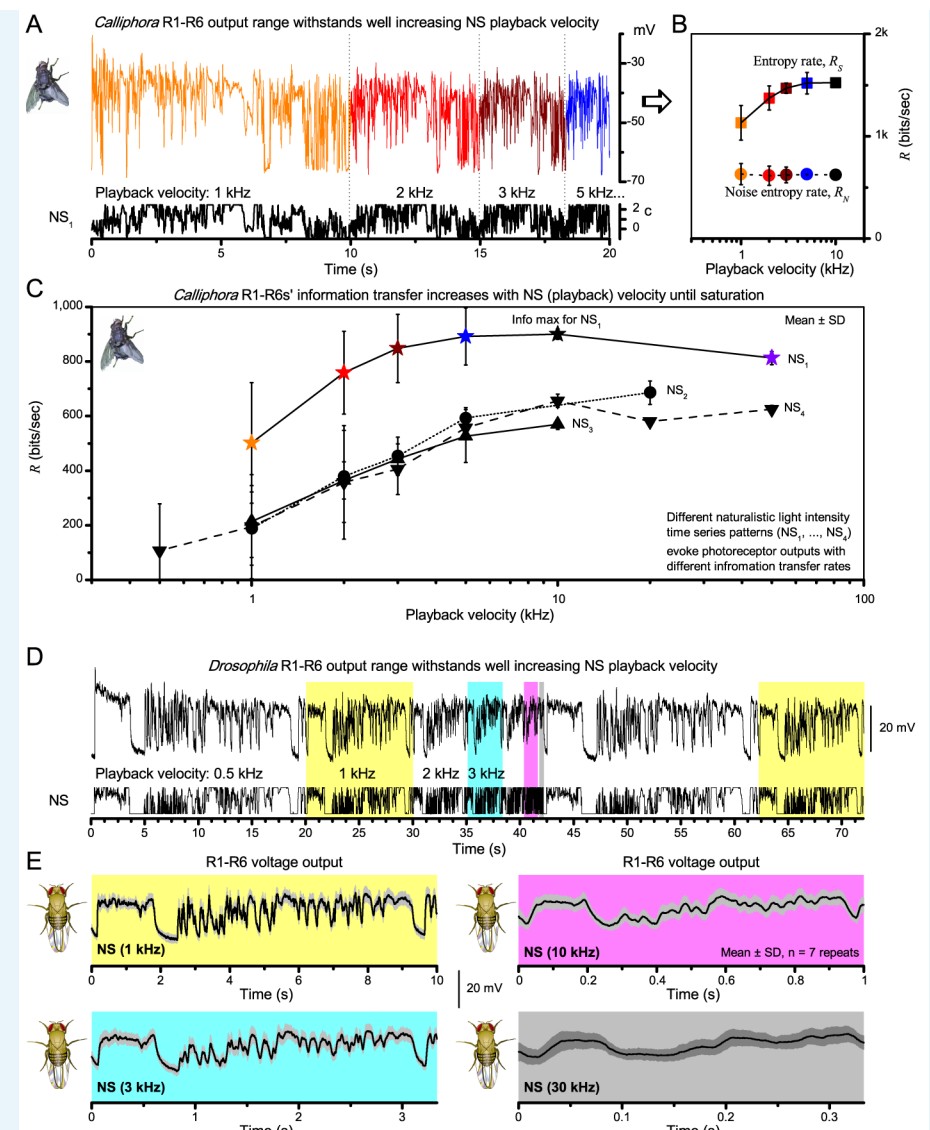

**Appendix 3—figure 4.** Photoreceptor output information rate depends on the speed and temporal structure of naturalistic stimulation (NS). (**A**) Intracellular voltage responses of a blowfly (*Calliphora vicina*) R1-R6 to a NS sequence repeated at different playback velocities. (**B**) The entropy rate, $R_S$, of photoreceptor responses increases with the playback velocity until saturation, whereas the noise entropy rate, $R_N$, remains virtually unchanged (*cf.* photoreceptor noise power spectra in *Figure 2—figure supplement 2A*). This improves the photoreceptor's encoding performance. (**C**) Information transfer rate (*Shannon, 1948*; *Juusola & de Polavieja, 2003*) ($R = R_S \, RN$) increases with playback velocity for four different NS sequence until saturation. Such dynamics resemble information maximization in *Drosophila* photoreceptor output by stimulus bandwidth broadening (*Figure 2C*). However, because *Calliphora* R1-R6s generate quicker responses than *Drosophila* R1-R6s, their information transfer saturates at considerably higher stimulus frequencies (*Juusola et al., 1994*; *Juusola and Hardie, 2001b*; *Juusola and Hardie, 2001a*; *Gonzalez-Bellido et al., 2011*; *Song et al., 2012*; *Song and Juusola, 2014*), suggesting superior encoding performance at high saccadic velocities. (**A–C**) Data is from (*Juusola and de Polavieja, 2003*) (*Figure 5*). (**D**) *Drosophila* R1-R6 output shows relative scale-invariance to NS pattern speed. NS was repeated at different playback velocities and the corresponding intracellular responses of a R1-R6 are shown above. Responses to four NS velocities are highlighted (yellow: 1 kHz, 10 s window; cyan: 3 kHz, ~3.3 s window; magenta: 10 kHz, 1 s window; gray:

30 kHz, ~0.3 s window). (**E**) The time-normalized shapes of R1-R6 output emphasize similar aspects in NS, regardless of the used playback velocity (here from 0.5 to 30 kHz). R1-R6s integrate voltage responses of a similar size for the same NS pattern, much irrespective of its speed. Mean ± SD shown, n = 7 traces. (**D–E**) Data is from (*Zheng et al., 2009*) (*Figure 4*).
DOI: https://doi.org/10.7554/eLife.26117.053

Finally, we note that it is possible that in scenes with different spatial structure (particularly lower spatial frequency structure), flies would use different turn velocities to bring contrast features into ideal sampling range (irrespective of saccades making sampling shorter). Future studies need to explore whether such a match with saccade statistics exists.

## Appendix 4

DOI: https://doi.org/10.7554/eLife.26117.054

# Spatial resolution (visual acuity) of the *Drosophila* eye (conventional measure)

## Overview

This appendix describes in detail a new method to measure a *Drosophila* photoreceptor's receptive field, and provides important background information about the experimental and theoretical results presented in *Figures 7–9*.

In this appendix:

- We test the hypothesis that in the *Drosophila* eye visual information is integrated laterally in dim conditions, and fed back synaptically to its photoreceptors, contributing to their spatial responsiveness.
- We measure dark- and moderately light-adapted wild-type R1-R6 photoreceptors' receptive fields by their acceptance angles, $\Delta\rho$, using intracellularly recorded voltage responses to light flashes, delivered from randomized positions of an orthogonal stimulation array.
- We compare these measurements to those of histamine-mutants (*Burg et al., 1993*; *Melzig et al., 1996*; *Melzig et al., 1998*) ($hdc^{JK910}$), in which first-order interneurons are blind (receiving no neurotransmitter from photoreceptors) and so incapable of feedback-modulating photoreceptor output.
- We show that the average acceptance angles of dark-adapted wild-type photoreceptors are 10.9% broader than those of the mutant, while light-adapted cells show no such difference.
- We characterize slow spontaneous retinal movements in the *Drosophila* eye and show how this activity can influence intracellular photoreceptor recordings.
- Our results suggest that in dim conditions spatial information is pooled in the lamina and fed-back to wild-type photoreceptors. Such excitatory lateral synaptic modulation, which is missing in the mutant, increases spatial sensitivity, broadening the photoreceptors' receptive fields.

## Optical limits of the fly compound eyes' visual acuity

Visual acuity is defined as the minimum angle that the eye can resolve. In the fly compound eye, if each ommatidium constitutes a sampling point in space, then the eye's maximal spatial resolution is set by the density of its ommatidial array (*Snyder et al., 1977*). Suppose a regular pattern of black and white stripes is presented to a fly. The maximum spatial frequency that the fly can resolve, $\nu_s$, is achieved when one ommatidium points to a black stripe and its adjacent ommatidium points to the next white stripe (*Appendix 4—figure 1A*). Thus, the interommatidial angle (*Snyder and Miller, 1977*), $\Delta\varphi$, is the key parameter in determining $\nu_s$. For the compound eyes with hexagonal layout, as is the case of most flies, the effective interommatidial angle, $\Delta\varphi_e$ (*Appendix 4—figure 1B*), can be calculated as:

$$\Delta\varphi_{e=} \cos\left(30°\right)\Delta\varphi = \frac{\sqrt{3}}{2}\Delta\varphi \tag{A4.1}$$

Thus, the upper limit of the fly eye's visual acuity is given by:

$$\nu_s = \frac{1}{2(\Delta\varphi_e)} = \frac{1}{\sqrt{3}(\Delta\varphi)} \tag{A4.2}$$

Whether this limit is achieved or not depends upon the spatial performance of a photoreceptor (*Snyder, 1977*). However, when estimating a photoreceptor's receptive field, which is quantified by its width at half-maximum, or acceptance angle (*Warrant and McIntyre, 1993*), $\Delta\rho$, we need to consider several contributing factors.

Firstly, since the ommatidium lens and a photoreceptor's rhabdomere are very small, optical quality is strongly affected by light diffraction, of which airy pattern (the point-spread function) depends upon light wavelength, $\lambda$, lens diameter, $D$, rhabdomere diameter, $d$, and focal distance, $f$ (**Appendix 4—figure 1C**). Theoretically, the blurring functions at the ommatidium lens and rhabdomere tip are both Gaussian and therefore can be combined (**Snyder, 1977**) to yield a simple approximation of $\Delta\rho$:

$$\Delta\rho = \sqrt{\left(\frac{\lambda}{D}\right)^2 + \left(\frac{d}{f}\right)^2} \tag{A4.3}$$

However, owing to the rather complex waveguide properties of small-diameter rhabdomeres, this formula is somewhat inaccurate. Van Hateren (**van Hateren, 1984**) and Stavenga (**Stavenga, 2003b**; **Stavenga, 2003a**) found that along a fly photoreceptor's rhabdomere only a limited number of light patterns (modes) could be formed and that this number depends upon the incident angle of light, leading to a smaller actual $\Delta\rho$ than what **Equation A4.3** implies.

Another contributing factor is the spatial cross talk, in which a photon escapes the rhabdomere it first travels in and enters an adjacent rhabdomere (**Horridge et al., 1976**). Such an effect is likely to happen when the cross-talk index of the ommatidia/rhabdomere structure is less than three (**Wijngaard and Stavenga, 1975**). This was indeed reported for *Drosophila* (**Gonzalez-Bellido et al., 2011**), suggesting that its neural images might have lower resolution than theoretically calculated from the optics.

Lastly, the intracellular pupil mechanism further affects $\Delta\rho$ estimation. Inside each photoreceptor cell, there are tiny pigment granules that migrate toward its rhabdomere boundary upon light adaptation (see Appendix 2, **Appendix 2—figure 1**). These pigments absorb and scatter light that travels inside the rhabdomere, reducing the light influx absorbable by its rhodopsin molecules (**Kirschfeld and Franceschini, 1969**; **Boschek, 1971**; **Roebroek and Stavenga, 1990**). Consequently, the pupil mechanism shapes a photoreceptor's angular and spectral sensitivity (**Stavenga, 2004a**). Moreover, in Appendix 2, we show by experiments and theory that it further helps to maximize a photoreceptor's information transfer; by optimizing the light intensity passing into the rhabdomere.

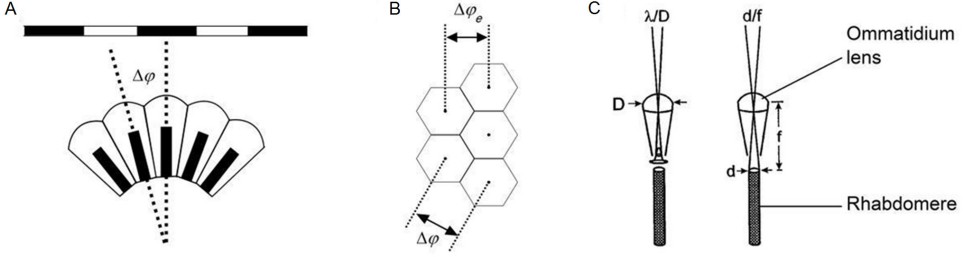

**Appendix 4—figure 1.** Classic theories of compound eye optics. (**A**) It is assumed that the minimum angle a compound eye can resolve is its interommatidial angle, $\Delta\varphi$. (**B**) The effective interommatidial angle of an eye with a hexagonal layout, $\Delta\varphi_e$, is smaller than its actual interommatidial angle, $\Delta\varphi$. **Equation A4.1** gives their geometrical relation. (**C**) Light diffraction at the ommatidial lens and the rhabdomere tip strongly affects the optical quality of the image pixel that a photoreceptor samples. D = ommatidial lens diameter; d = rhabdomere tip diameter; f = focal length; $\lambda$ = wavelength. Redrawn from (**Land, 1997**).
DOI: https://doi.org/10.7554/eLife.26117.055

## Rationale for investigating wild-type and *hdc* R1-R6s' receptive fields

The fly compound eyes are small and size-constrained, presumably to save energy and improve survival (**Land, 1998**; **Laughlin et al., 1998**; **Niven et al., 2007**). This puts their sensitivity/acuity trade-off under intense evolutionary pressure (**Snyder et al., 1977**; **Laughlin, 1989**; **Nilsson, 1989**; **Warrant and Mcintyre, 1992**). While an increase in

ommatidium size would improve photon capture, it would also result in fewer sampling points (image pixels) in the eye, lowering the resolution of its neural responses (neural images of the world). In dim conditions, where photon noise is relatively large compared to available information (signal), the task to enhance visual reliability and sensitivity becomes challenging. Optical mechanisms, including the widening of photoreceptor receptive fields by pupil opening (**Williams, 1982**; **Laughlin, 1992**; **Nilsson and Ro, 1994**; **Stavenga, 2004a**), can increase the amount of light collected in each ommatidium only to some extent. Yet under dim illumination, insect visual behaviors appear remarkably robust (**Pick and Buchner, 1979**; **Warrant et al., 1996**; **Honkanen et al., 2014**), suggesting that their eyes' neural mechanisms could successfully overcome the apparent shortfall in photon supply (**Warrant, 1999**).

Sensitivity can be increased neurally at the cost of decreasing acuity by (**i**) increasing photoreceptors' voltage/light intensity gain (**Laughlin and Hardie, 1978**; **Matić and Laughlin, 1981**; **Song et al., 2012**), (**ii**) increasing their integration time (temporal summation) (**Skorupski and Chittka, 2010**; **Song et al., 2012**) and (**iii**) spatially summing information, or reducing lateral inhibition from neural neighbors (**Srinivasan et al., 1982**; **van Hateren, 1992c**; **1993b**). Experimentally, spatial summation has been shown to occur in the directionally-selective motion-detecting (DSMD) neurons of the fly lobula plate (**Srinivasan and Dvorak, 1980**), but it is possible that such signals might also reflect upstream processing in the earlier optic neuropiles (the lamina, medulla and lobula).

Although the fly photoreceptor biophysics for adapting temporal summation are well characterized (**Juusola et al., 1994**; **Juusola and Hardie, 2001b**, Juusola and Hardie, 2001a**Juusola and Hardie, 2001a**; **Song et al., 2012**; **Song and Juusola, 2014**; **Hardie and Juusola, 2015**; **Juusola et al., 2015**) (see Appendixes 1–3), the neural substrate for spatial summation is less well understood. In the lamina, electrical couplings by gap-junctions were found only between the photoreceptor axons that share the same optical axis (**Ribi, 1978**; **Shaw, 1984**; **van Hateren, 1986**; **Shaw et al., 1989**) in neural superposition. Hence, these presynaptic connections probably cannot distribute spatial information. Nonetheless, postsynaptically, the evidence is more suggestive. Intracellular responses of the histaminergic interneurons (large monopolar cells, LMCs) to narrow (point source) and wide-field light stimuli match well the theoretical predictions of spatiotemporal summation (**Dubs et al., 1981**; **van Hateren, 1992a**, **1992b**). This notion was further advocated by the structural study in the nocturnal bee *Megalopta genalis* (**Greiner et al., 2005**) lamina, which revealed extensive synaptic connections between adjacent cartridges. Finally, early functional studies of the housefly (*Musca domestica*) photoreceptors (**Dubs et al., 1981**) indicated that quantum bumps, recorded to dim light at the behavioral threshold, contain additional small-amplitude events. These were judged not to be generated by the impaled cells but by single photon captures in their neighbors; with the receptive fields being wider than what were expected from the optics alone (**Dubs, 1982**).

In this appendix, we test whether or how spatial information is integrated laterally and fed back synaptically to photoreceptors, contributing to their acceptance angles, Δρ. The tight coupling between feed-forward and feedback pathways in the photoreceptor-lamina circuit is known to have crucial roles in maintaining robust adaptation and temporal coding efficiency (**Zheng et al., 2006**; **Nikolaev et al., 2009**; **Zheng et al., 2009**). Theoretically, similar spatial information regulation should further improve fly vision. Specifically here, we take advantage of *Drosophila* genetics and compare R1-R6 photoreceptor outputs of wild-type and $hdc^{JK910}$ mutant. Synaptic transmission from $hdc^{JK910}$ photoreceptors is blocked, making their interneurons effectively blind (**Dau et al., 2016**). Therefore, feedback from the mutant LMCs (and possibly from amacrine cells (**Zheng et al., 2006**; **Hu et al., 2015**), Am, which also receive histaminergic input from photoreceptors) to R1-R6s cannot contain any lateral modulation, neither inhibitory nor excitatory.

We show that the dark-adapted wild-type R1-R6 photoreceptors' mean acceptance angles are 10.9% broader than in the mutant, while no significant differences are found between the light-adapted cells. We further show how stimulus history and retinal movements affect the receptive fields in the *Drosophila* eye. Our results suggest that in dim conditions spatial information is pooled in the lamina and channeled back to R1-R6 photoreceptors in the form

of excitatory synaptic modulation, which increases spatial responsiveness by broadening the cells' receptive fields.

## Measurement and calculation of a R1-R6's receptive field

A photoreceptor's receptive field can be estimated electrophysiologically by measuring its intracellular response amplitudes, $V_n$, to a light flash intensity, $I_n$, at varying angular positions, $\alpha_n$. From all these $V_n$, $I_n$, and $\alpha_n$ values generated by a complete scan, the receptive field width can be computed by three different methods as comparatively reviewed below.

### Method 1

$V_n$ is clamped to a constant value in a closed-loop system, which accordingly vary $I_n$ for each tested light source positions (**Smakman et al., 1984**; **Smakman and Stavenga, 1987**). Sensitivity at each position, $S_n$, is then defined by:

$$S_n = \frac{I_o}{I_n} \tag{A4.4}$$

where $I_0$ is the intensity required from a point source at the center of the receptive field. The definition of sensitivity can be equivalently expressed as the light source at an off-axis position. The off-axis light intensity needs to be $\frac{1}{S_n}$ fold brighter than the axial one to stimulate responses of the same amplitude.

After corresponding $S_n$ was computed for every $S_n$, the sensitivity-angle relation is fitted by a Gaussian function. The width at the half-maximum of this Gaussian curve is called the *acceptance angle*. This is the conventional parameter, $\Delta\rho$, for quantifying the receptive field width.

### Method 2

in which the same light flash intensity $I_n$ is tested at many different angular positions, is the most widely used (**Wilson, 1975**; **Horridge et al., 1976**; **Hardie, 1979**; **Mimura, 1981**; **Gonzalez-Bellido et al., 2011**). Initially, the $V/\log(I)$ relation of the impaled photoreceptor is determined at the center of its receptive field by presenting logarithmically intensified flashes from a point-like light source (through scaled neutral density filters). $V_n$ elicited by the light at each off-axis angle, $V_n$, is then substituted into the $V/\log(I)$ function to estimate, $I_a$, the light intensity that was effectively absorbed by the cell's photopigments. Angular sensitivity, $S_n$, is given by **Equation A4.5**:

$$S_n = \frac{I_a}{I_n} \tag{A4.5}$$

Gaussian fitting and the acceptance angle calculation are performed as in Method 1.

Method two is based upon the same principle as Method 1, which is to assess the light intensity necessary to elicit a criterion response (**Warrant and Nilsson, 2006**). When the light-point with intensity $I_0$ is exactly at the optical axis of the cell, sensitivity is the highest with the response amplitude $V_0$. To evoke $V_n = V_0$ by a light source located at an angular position, $\alpha_n$, it is required that the effective intensity $I_a$ equals to $I_0$. Given the angular sensitivity formula:

$$S_n = \frac{I_a}{I_n} = \frac{I_o}{I_n} \tag{A4.6}$$

the necessary intensity $I_n$ would be $\frac{1}{S_n}$ fold brighter than $I_o$.

Though Method 2 does not require a closed-loop system and is, therefore, less experimentally challenging, its effective intensity, $I_a$, estimation has drawbacks. Fitting $V/\log(I)$ function to a small number of maximum amplitude values, which are adaptation-dependent, can introduce scaling errors. Whilst its underlying assumptions, that the voltage/(effective intensity) relation is static and independent of light source position, neglect possible dynamic and lateral interactions between neighboring cells.

Nonetheless, the outcomes of both methods are theoretically independent of photoreceptor biophysics and the test flash intensity. Hence, they enable electrophysiological receptive field measurements to be compared with those derived from optical, morphological and waveguide theories.

## Method 3

Similar to Method 2, each tested photoreceptor is stimulated by the same light intensity flashes at different angular positions around its optical axes. Response amplitude $V_n$ to a light flash coming from an angle $\alpha_n$ is then normalized to the maximum response evoked by the on-axis light source, $V_0$. The receptive field is determined by the Gaussian fitting of the relation between ratios $V_n/V_0$ and incident angle $\alpha_n$. This may yield wider half-maximum widths ($\Delta\rho$ values) than the acceptance angles (*Washizu et al., 1964*; *Burkhardt, 1977*) estimated by using Method 1 and Method 2.

We used Method 3 to estimate R1-R6 photoreceptors' receptive fields from intracellular recordings, despite its disadvantages; the results would depend on the flash intensity and would not be fully comparable to other approaches and the previous studies in *Drosophila*. Our main rationale was that this method characterizes 'how well the flies see' most directly and reliably, without making any assumptions about lateral interactions between photoreceptors and LMC feedbacks. Moreover, the method's limitations should not compromise our objectives to compare the receptive fields in different genotypes and to report how these are affected by different light conditions and stimulation history. And importantly, these receptive field estimates could be directly used in further calculations to assess the same photoreceptors' theoretical acuity to detect moving objects, as shown in Appendixes 6–8. Experimentally, it was also unfeasible to expand our recording set-up either with a closed-loop system (as in Method 1) or with easily exchangeable neutral density filter sets for characterizing $V/log(I)$ function (as in Method 2).

## 25 LED light-point array and LED pads

A R1-R6 photoreceptor's receptive field was scanned by using an array of 25 light-points, mounted on a Cardan arm (*Appendix 4—figure 2*). Each light-point (small light guide end) subtended an angle of 1.7° as seen by the fly, transmitting its light output from its specific LED (1/25). The system was controlled by 2 channels, both with voltage inputs ranging from 0 V to 10 V. Channel 0 was used to select the light-point while Channel 1 was used to linearly set its intensity.

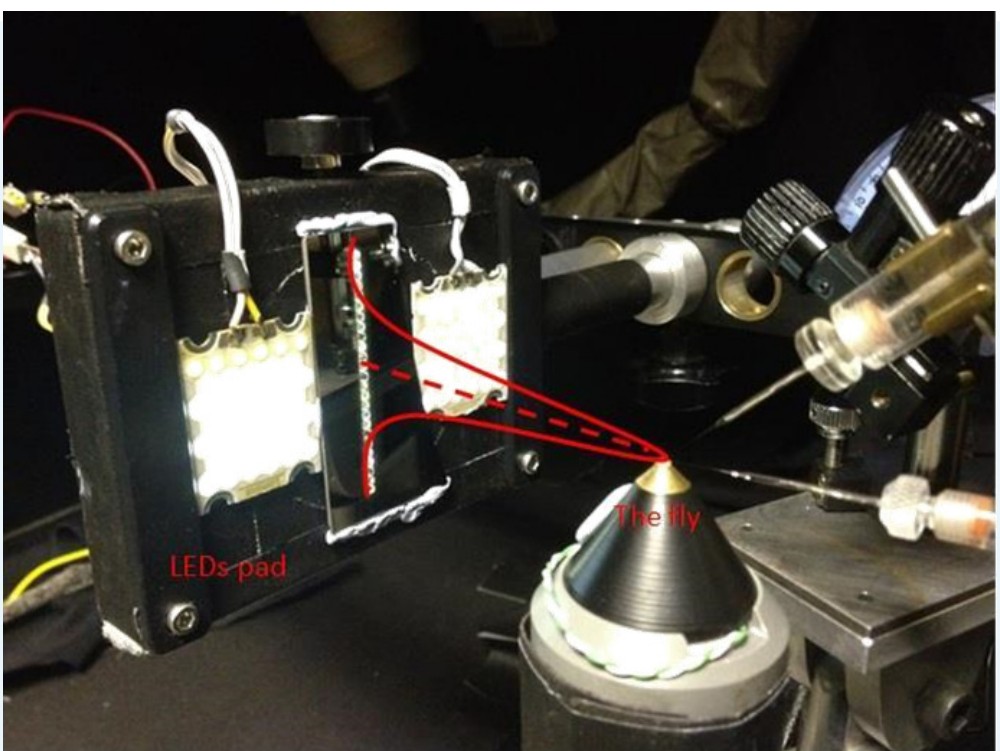

**Appendix 4—figure 2.** 25 light-point stimulus array. Each tested dark-adapted photoreceptor's receptive field (red Gaussian) was assessed by measuring its intracellular responses to successive flashes from 25 light-points. In light-adaptation experiments, two 39-LED pads, (on both sides of the vertical stimulus array) provided background illumination. The intact fly was fixed inside the conical holder, which was placed upon a close-looped Peltier-element system, providing accurate temperature control (at 19°C). The rig was attached on a black anti-vibration table, inside a black-painted Faraday cage, to reduce noise and light scatter.
DOI: https://doi.org/10.7554/eLife.26117.056

*Appendix 4—figure 3A* shows a typical light-point's spectral density measured at the light-guide end. Each light-point had a narrow spectral **Peak1** at ~450 nm and a broader **Peak2** at ~570 nm, in which intensities and wavelengths are listed in *Appendix 4—table 1*. Based on their relatively small variations, all 25 light-points provided reasonably uniform light input, except the 4-times brighter No.22 (see below). This is because a fly photoreceptor's response amplitudes differ only marginally until light intensity changes several-fold, as defined by the sigmoidal $V/\log(I)$ relationship (*Matić and Laughlin, 1981*). Standard light flashes, containing ~$2 \times 10^6$ photons/s at **Peak1** and ~$3 \times 10^6$ photons/s at **Peak2**, were produced by setting Channel 1 to an input value of 2 V. Here, the estimated photon counts are given at the light source, not at the level of photoreceptor sampling. Moreover, in the experiments, to evoke subsaturating responses, we used a neutral density filter plate to cut the light-point intensity by 100-fold.

**Appendix 4—table 1.** Light flash peak wavelengths and intensities delivered by each of the 25 light-points in the stimulus array The given light intensities were measured at the light source by Hamamatsu Mini C10082CAH spectrometer, before 100-fold neutral density filtering. Thus, these values are estimated to be $10^{2\text{-}3}$-times higher than the corresponding effective photon rates at the level of R1-R6 sampling (see *Appendix 2—figure 3*). Accordingly at the optical axis, the center LED (No.13), with the neutral density filter on it, evoked subsaturating (~20–35 mV) responses from *Drosophila* R1-R6 photoreceptors (*Appendix 4—figure 4*).

| Light-Point | Peak1 wavelength (nm) | Peak1 intensity ($10^6$ photons/s) | Peak2 wavelength (nm) | Peak2 intensity ($10^6$ photons/s) |
|---|---|---|---|---|
| No.1 | 448 | 2.720 | 571 | 2.805 |
| No.2 | 452 | 1.790 | 570 | 2.818 |
| No.3 | 448 | 2.618 | 565 | 2.900 |
| No.4 | 451 | 1.840 | 576 | 3.020 |
| No.5 | 451 | 2.570 | 575 | 3.710 |
| No.6 | 451 | 2.214 | 572 | 3.640 |
| No.7 | 452 | 1.430 | 575 | 2.020 |
| No.8 | 446 | 2.203 | 570 | 2.990 |
| No.9 | 453 | 1.465 | 571 | 2.350 |
| No.10 | 451 | 3.300 | 578 | 5.100 |
| No.11 | 453 | 1.877 | 575 | 3.080 |
| No.12 | 455 | 1.763 | 575 | 3.020 |
| No.13 | 451 | 2.334 | 575 | 3.440 |
| No.14 | 451 | 2.009 | 576 | 2.634 |
| No.15 | 454 | 2.400 | 568 | 4.480 |
| No.16 | 452 | 2.390 | 572 | 4.165 |
| No.17 | 455 | 3.190 | 566 | 3.010 |
| No.18 | 452 | 1.990 | 578 | 3.320 |
| No.19 | 455 | 1.958 | 578 | 3.336 |
| No.20 | 454 | 1.745 | 569 | 2.670 |
| No.21 | 450 | 2.642 | 573 | 2.314 |
| No.22 | 452 | 9.520 | 572 | 13.300 |
| No.23 | 452 | 2.420 | 575 | 3.380 |
| No.24 | 452 | 2.658 | 573 | 3.284 |
| No.25 | 452 | 1.670 | 570 | 2.750 |

DOI: https://doi.org/10.7554/eLife.26117.057

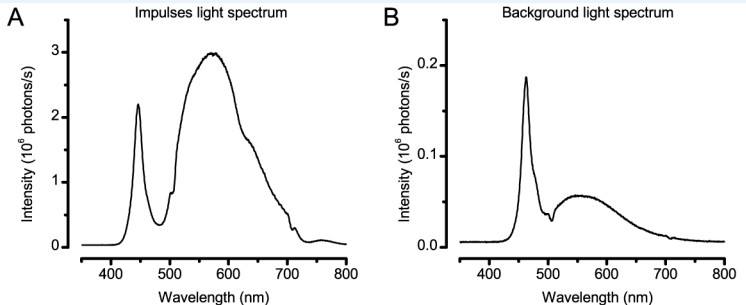

**Appendix 4—figure 3.** Spectral properties of the light stimuli. (**A**) Typical spectral density of the light impulses delivered by the 25-point array. Note the spectra has two prominent peaks, named **Peak1** (~450 nm) and **Peak2** (~570 nm). (**B**) Spectral density of a single LED on the two Lamina pads, which were used to provide ambient background illumination during light-adaptation experiments. These spectral intensities (photon counts) were measured by a spectrometer (Hamamatsu Mini C10082CAH, Japan).

DOI: https://doi.org/10.7554/eLife.26117.058

Light-point No.22 was 4-fold brighter than the others. However, no attempt was made to correct its intensity for three reasons. Firstly, it was located at the tested receptive fields' periphery, and thus had limited influence on the measurements. Secondly, because this

'error' occurred stereotypically in every experiment, the brighter No.22 would not bias the comparative studies (see below). Lastly, having one brighter light-point was beneficial for other experiments, as will be shown in Appendix 6.

Two Lamina LED pads, each with 39 similar LEDs, provided ambient illumination to moderately light-adapt the tested photoreceptors. The pads were located at the outer half and outside each tested cell's receptive field (*Appendix 4—figure 2*). Thus, a large portion of their light projected onto the neighboring photoreceptors. But the pads also illuminated the whole recording chamber, revealing its spatial structure and possibly inducing spatial processing in the lamina network. Light from each of the pad's LEDs peaked at 460 nm (*Appendix 4—figure 3B*), delivering estimated ~$2 \times 10^5$ photons/s.

## Pseudo-random receptive field scans

Before recording intracellular voltage responses from a R1-R6 photoreceptor, we located its receptive field center. This was done by flashing the light-point No.13 (at the center of the array) and moving the array (with the Cardan arm along its XY-axes) until the maximum response amplitude was elicited. The array was then locked at this position.

In the dark-adaptation experiments, the photoreceptor faced darkness (*Appendix 4—figure 4A*) for 30–60 s before its receptive field was measured. In the light-adaptation experiments, preselected background illumination (using the LED pads: *Appendix 4—figure 2*) was turned on 30–60 s before the corresponding receptive field measurement.

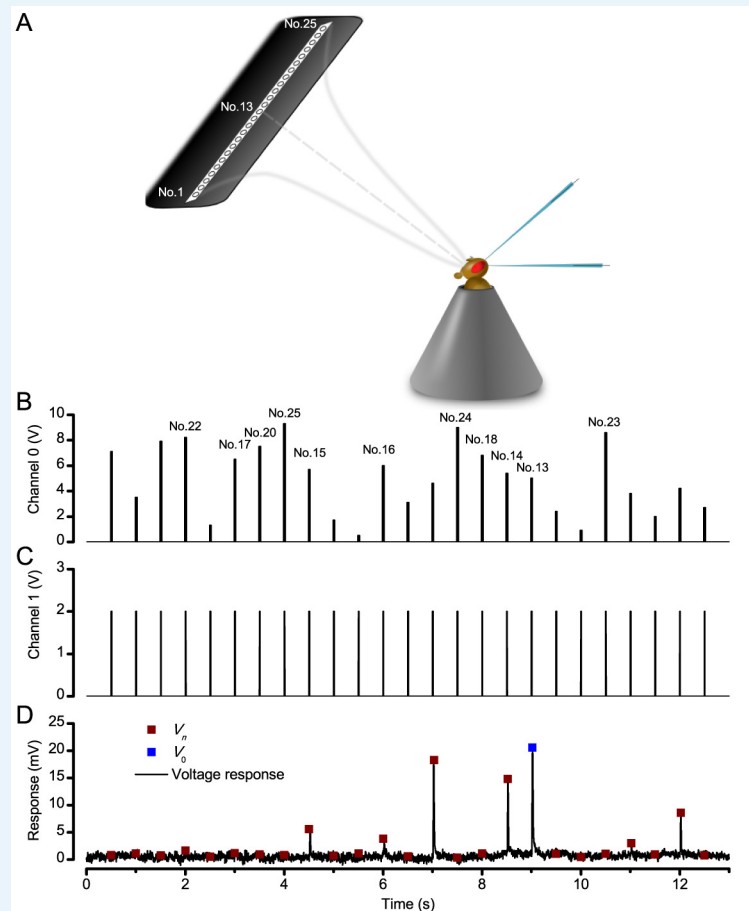

**Appendix 4—figure 4.** Measuring a R1-R6 photoreceptor's receptive field with intracellular recordings. (**A**) Schematic Image of how the LED stimulus array - seen as 25 light-points (light-guide-ends in a row) was centered by a Cardan arm system in respect to the studied photoreceptor and the fly eye. (**B**) Channel 0 input was used to select the LED (light-point) to be turned on. (**C**) Channel one input defined light intensity of the selected LED. A standard light impulse (flash) was produced by a 2 V input, which lasted 10 ms. (**D**) A photoreceptor's

intracellular voltage responses to a complete receptive field scan. These sub-saturating responses were recorded at 19°C. Amplitude $V_n$ of each flash response was the local maxima. $V_0$ is the amplitude of the response to a light flash at the center of the receptive field (on-axis).

DOI: https://doi.org/10.7554/eLife.26117.059

A complete scan of a photoreceptor's receptive field comprised 25 subsaturating flashes from all the light-points, one after another in a pseudo-random order (*Appendix 4—figure 4B*). Each flash lasted 10 ms, and was 490 ms apart from the next one (*Appendix 4—figure 4C*). Although this inter-flash-interval largely rescued the photoreceptor sensitivity, spatiotemporal adaptation might have still affected their responses. For instance, a flash near the center of the receptive field would light-adapt the cell more than one at the periphery, possibly causing the response to the next flash to be artificially smaller. Therefore, by randomizing the order of flash positions - with the Matlab command *randperm(25)* - we could reduce this kind of potential adaptation effects.

Channel 0 input was turned on only when Channel one was set to zero V; that is, in the resting period when all the light-points were off. Otherwise, the transitions of Channel 0 input values would generate running dot images. Each tested photoreceptor's responses to 2–5 repetitions of pseudorandom scans were averaged (*Appendix 4—figure 4D*) before the acceptance angle (or half-maximum width), Δρ, of its receptive field was determined (*Appendix 4—figure 5*).

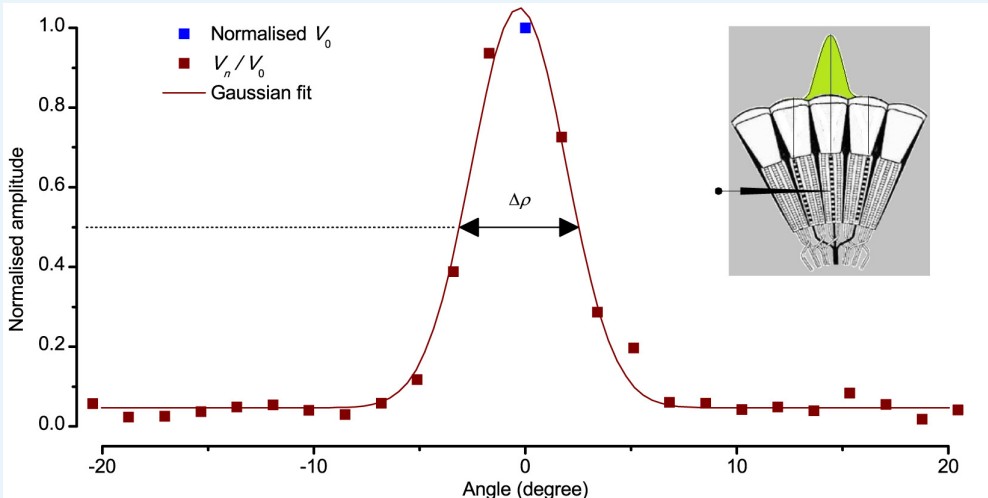

**Appendix 4—figure 5.** Estimating a dark-adapted *Drosophila* R1-R6 photoreceptor's receptive field and its half-width. Flash response amplitudes $V_n$ were initially normalized to $V_0$, the maximum response elicited by an on-axis light-point. A Gaussian curve was then fitted to these normalized values, yielding an estimate of the receptive field. Half-maximum width of this Gaussian function, Δρ, defined the tested photoreceptor's acceptance angle. The schematic fly eye inset clarifies how a single photoreceptor integrates light from the world spatially through its receptive field (green area), whilst being bounded by the ommatidial lens system. For a standard measurement, each tested photoreceptor's intracellular responses to 2–5 repetitions of pseudorandom scans (as shown in *Appendix 4—figure 4*) were averaged.

DOI: https://doi.org/10.7554/eLife.26117.060

## Gaussian white-noise (GWN) stimuli

Light-point No.13 intensity was controlled by setting Channel 0 input to 5 V and modulating Channel one input with a white-noise time series, which had a mean value of 2.5 V and cut-off frequency of 200 Hz. With these settings, light-point No.13 emitted $2.5 \times 10^6$ photons/s at **Peak1** (451 nm) and $3.75 \times 10^6$ photons/s at **Peak2** (575 nm) on average (*Appendix 4—*

*table 1*). But, as in other experiments, these intensities were reduced 100-fold by neutral density filtering

## Receptive fields of dark-adapted photoreceptors

In every experiment, we first assessed the recorded photoreceptor's receptive field after dark-adaptation (*Figure 7—figure supplement 1A*). Wild-type R1-R6s' mean acceptance angle, measured as their receptive fields' half-maximum width, $\Delta\rho$, was 9.47 ± 0.36° (± SEM, n = 19 cells), ranging from 7.00° to 11.65°. Interestingly, $hdc^{JK910}$ R1-R6's receptive fields were 10.9% narrower (p =0.0397, two-tailed t-test). Their mean, minimum and maximum acceptance angles were 8.44 ± 0.32° (n = 18 cells), 6.18° and 11.50°, respectively.

Because each photoreceptor's flash response amplitudes, $V_n$, were directly used to estimate its receptive field (*Figure 7—figure supplement 1B*), rather than being converted to angular sensitivities (see above), the obtained $\Delta\rho$ metric depended upon the cell's output/input characteristics and the test flash intensity. To ensure that wild-type and $hdc^{JK910}$ photoreceptors' $\Delta\rho$ comparison was unbiased by variable on-axis light sensitivities, we also compared their maximum response amplitudes, $V_0$, and $V_0/\Delta\rho$ relations.

The corresponding $V_0$ values were very similar (*Figure 7—figure supplement 1B*) and mostly within 20–35 mV sub-saturated linear range of the photoreceptors' V/log(I) curves (*Dau et al., 2016*). Moreover, in both wild-type and the *hdc* mutant, the linear correlations between $V_0$ and $\Delta\rho$ reflected only a weak trend of more sensitive photoreceptors (larger $V_0$) having wider receptive fields (larger $\Delta\rho$) (*Figure 7—figure supplement 1C,D*).

Together, these findings indicate that the narrower $\Delta\rho$ of dark-adapted $hdc^{JK910}$ photoreceptors neither resulted from altered phototransduction nor was an artefact of this measurement method.

## Receptive fields of light-adapted photoreceptors

A photoreceptor's $\Delta\rho$ measured under light-adaptation (at specific ambient illumination; *Appendix 4—figure 2*) should be smaller than during dark-adaption. There are four reasons for this difference:

- Light-adaptation steepens a photoreceptor's V/log(I) function (*Laughlin and Hardie, 1978*; *Matić and Laughlin, 1981*; *Eguchi and Horikoshi, 1984*). This reduces the difference between $I_0$ and $I_{a50}$ (or the effective intensity that could evoke response amplitude $V_0/2$ ), which in turn leads to a smaller $\alpha_{50}$ (or the corresponding angular position of $I_{a50}$) and thus to a smaller $\Delta\rho$, as reported by the chosen method.
- Because the test flash intensity was kept unchanged, their contrast would be lower during light-adaptation than in the dark-adaption experiments, further reducing the $I_o/I_{\alpha50}$ ratio, $\alpha_{50}$ and $\Delta\rho$.
- Light adaptation activates screening pigment migration, narrowing the intracellular pupil (*cf.* Appendix 2, *Appendix 2—figure 1*). The narrower pupil effectively reduces the amount of light from off-axis angles that can be absorbed by rhodopsin-molecules in the rhabdomere (*Hardie, 1979*; *Smakman et al., 1984*; *Stavenga, 2004a*; *Stavenga, 2004b*), reducing $\Delta\rho$.
- Theoretical studies and some experimental data suggest that in dim conditions neural signal summation from neighboring cells may enhance sensitivity. But in bright conditions, lack of summation, increased lateral inhibition or both improve image resolution (*Srinivasan et al., 1982*; *van Hateren, 1992c*; *1993a*; *Warrant, 1999*; *Klaus and Warrant, 2009*).

To quantify how moderate ambient light affects the fly eye's spatial responsiveness, we analyzed the receptive fields of six wild-type (n = 6) and eight $hdc^{JK910}$ (n = 8) photoreceptors both at their dark- and light-adapted states.

At the dark-adapted state, the acceptance angles of wild-type and $hdc^{JK910}$ R1-R6s' receptive fields, $\Delta\rho$, were 9.65 ± 1.06° and 8.16 ± 0.62° (mean ± SEM), respectively (Box 4.8A). However, owing to the smaller test and control group sizes than in *Figure 7—figure*

supplement 1A, this average difference (15.44%), though similar, was now statistically insignificant (p=0.258, two-tailed t-test).

At the light-adapted state, under the given ambient illumination (**Figure 7—figure supplement 2B**), the corresponding Δρ values were 7.70 ± 0.52° for wild-type photoreceptors and 6.98 ± 0.46° for their $hdc^{JK910}$ counterparts. Thus, light-adaptation significantly reduced Δρ values from dark-adaptation (p = $3.49 \times 10^{-4}$, paired two-tailed t-test). Switching from the dark- to light-adapted states, wild-type photoreceptors' receptive fields narrowed down by 18.44 ± 3.5% (**Figure 7—figure supplement 2C**), slightly more than those of mutants, which changed by 13.68 ± 3.37%. Yet, none of these parameters differed significantly between the wild-type and mutant photoreceptors.

## R1-R6 acceptance angles are much broader than the theoretical prediction

Based on the ommatidium dimensions, as extracted from histological images of fixed/non-living retinae, and the waveguide optic theory, Stavenga (**Stavenga, 2003b**) calculated that the acceptance angles or dark-adapted *Drosophila* R1-R6 photoreceptors should be from 3.8 to 5.0° (as amended for 16.5 µm diameter ommatidium lens). Yet, our current (Δρ = 9.47°; **Figure 7—figure supplements 1** and **2**) and earlier (**Gonzalez-Bellido et al., 2011**) measurements (Δρ = 8.23°; using **Method two** above) clearly showed that their acceptance angles *in vivo* are in fact about twice as large. What kind of physical mechanism(s) could explain this discrepancy between the theory and measurements?

We briefly introduce here some key points of the new 'microsaccadic sampling hypothesis', which is examined in detail in **Appendixes 7–8**.

- Living R1-R6 photoreceptors are not still but transiently contract to light (**Hardie and Franze, 2012**) (**Video 3**). We show in **Figure 8** (see also **Appendixes 7–8**) that this causes considerable horizontal rhabdomere movements (up to 1.4 µm, peaking ~60–150 ms after a flash onset and returning back slower). As the lens system stays practically put, the rhabdomere tips shift away from the central axis, skewing the light input and narrowing the photoreceptors' receptive fields dynamically.
- Rhabdomere contractions also move their tips axially; transiently down the focal plane (**Video 2**).
  - In a dark-adapted state, rhabdomere tips are elongated upwards (closer to the ommatidium lens), and possibly out of focus, collecting light through the lens system over a wider space. Thus, R1-R6s' acceptance angles would be broader for light flashes spaced by normal (500 ms) intervals, which recover refractory microvilli, returning rhabdomeres to their old positions. But during a bright passing light stimulus, which progressively contracts R1-R6s, their acceptance angles dynamically narrow as the rhabdomeres draw a bit deeper in the retina.
  - At a moderate light-adaptation state (**Figure 7—figure supplement 2**), the intracellular pupil mechanism has reduced light input and the rhabdomere lengths should occupy a position (or set-point), which allows further contractions to light increments (*cf.* hair cells in the inner ear [**Howard et al., 1988**]). However, here, R1-R6s' acceptance angles would still be broader for flashes with 500 ms intervals. Thus, the rhabdomeres would return to their pre-flash positions, which are closer to the lens than during a bright passing light stimulus.
- Photomechanical R1-R6 contractions are partially levered by the adherence junctions (**Tepass and Harris, 2007**) from their rhabdomeres to the above cone cells and epithelial pigment cells, which form the inner wall of the ommatidium underneath the lens (**Figure 8F**). Thus, as the photoreceptors contract to light, their adherence junctions appear to pull the pigments cells, moving and narrowing the aperture (and the light beam) in the front of their rhabdomere tips (**Video 4**).

## Slow *Drosophila* retina movements

Despite a *Drosophila*'s head and thorax being immobilized to the conical holder (*Appendix 4—figure 4A*), its eyes could still move affecting the electrophysiological recordings (*Kirschfeld and Franceschini, 1969*). Retinal movements, caused by spontaneous intraocular muscle activity, have been described in larger flies (*Franceschini et al., 1991*; *Franceschini and Chagneux, 1994*; *Franceschini et al., 1995*; *Franceschini and Chagneux, 1997*; *Franceschini, 1998*) and treated by different methods, including cooling, anesthesia and fixing their slightly pulled-out antennae (*Smakman et al., 1984*).

Here we report possibly related but slower retinal movements in fixed *Drosophila* preparations *in vivo*. *Appendix 4—figure 6A* shows an example, in which the optical axis of a R1-R6 photoreceptor moved between two consecutive receptive field scans. In the first scan, the receptive field center, which was localized by the largest flash response, corresponded to the light-point No.12. However, the second receptive field scan indicated that the cell's optical axis pointed toward the light-point No.10. This displacement corresponds to an angular movement of ~3°. We found that about 50% of photoreceptors, in which receptive fields were scanned more than once (8/18 wild-type and 8/16 mutant cells), displayed similar retinal movements in the range of 1–3.5°. Moreover, these movements occurred in both front-to-back and back-to-front directions, validating that they were not equipment related artefacts; for example, not caused by gravitational drift in the 25 light-point stimulus array (*Appendix 4—figure 2*).

It has been suggested that recordings from damaged fly photoreceptors may result in (i) extraordinarily wide acceptance angles, (ii) diminishing sensitivity (*Wilson, 1975*), or (iii) markedly asymmetrical receptive fields, attributed to artificial electrical coupling between neighboring cells (*Smakman and Stavenga, 1987*). To ensure that high-quality Δρ measurements were presented in this study, we only considered data from photoreceptors in which intracellular responses were stable and repeatable, and their receptive fields reasonably symmetrical. However, we acknowledge that *Drosophila* eye movements can inadvertently affect the receptive field assessment accuracy.

The *slow* retinal movements and drifts that shift R1-R6 photoreceptors' receptive fields are likely driven by eye muscle (*Hengstenberg, 1971*) activity. These movements can modulate light input to photoreceptors, causing spontaneous dips and peaks in their output during continuous repetitive stimulation, as is sometimes seen during long-lasting intracellular recordings (*Appendix 4—figure 6B*). Because this additional input modulation seems largely occur in the timescale of seconds, it should reduce mostly the low-frequency signal-to-noise ratio in the R1-R6 output. In this study, to obtain as good estimates as possible of the *Drosophila* photoreceptors' encoding capacity, we only used data from the very best (most stable) recording series. These recordings (*Figures 1–2*; *Figure 1—figure supplement 1*; *Figure 2—figure supplement 1*) showed very little or no clear signs of such perturbations.

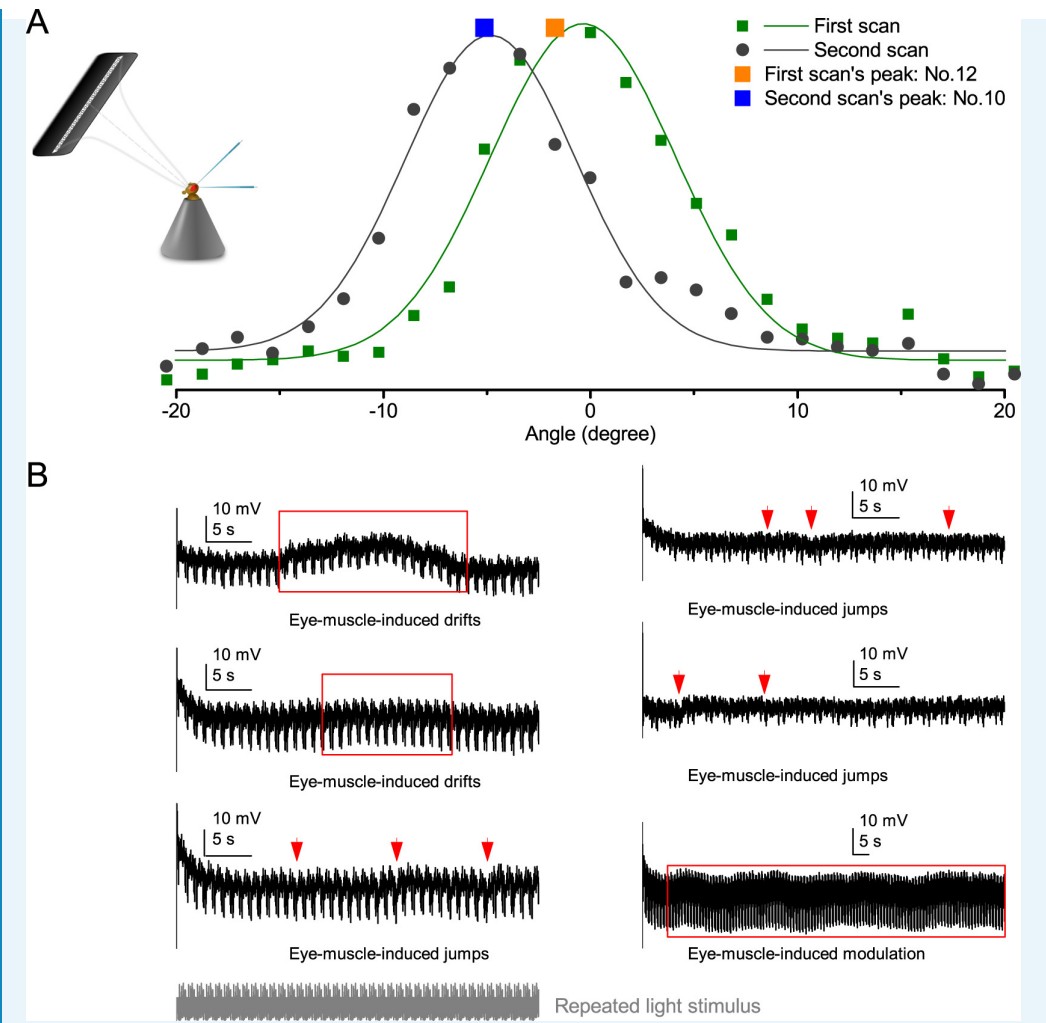

**Appendix 4—figure 6.** Retinal movements in the *Drosophila* eyes shift photoreceptors' receptive fields, modulating their light input and hence the transduced voltage output. (**A**) A R1-R6 photoreceptor's receptive field shifted between consecutive measurements to 25 light-point stimuli. The first scan showed that the receptive field center was closest to light-point No.12; while in the second scan, the peak response was evoked by the light-point No.10. The difference between the optical axes, as indicated by the two scans, was ~3°. (**B**) Examples of R1-R6 photoreceptors' intracellular responses that show slow spontaneous voltage drifts, saccades, jumps or modulation (red arrows and boxes) during repeated light intensity time series stimulation from a fixed point source. Characteristically, these perturbations do not correlate with the light stimuli but are erratically superimposed on the responses' normal adapting trends. Because they occur in the time scale of seconds and show variable rhythmicity, they are likely caused by intrinsic eye muscle activity. Before the recordings, the light stimulus source (3° light-guide-end, as seen by the fly) was carefully positioned at the center of each tested photoreceptor's receptive field.

DOI: https://doi.org/10.7554/eLife.26117.061

Furthermore, in **Appendixes 7–8**, we quantify how all *Drosophila* photoreceptors exhibit *fast* light-triggered photomechanical micro-saccades, peaking ~100 ms after the stimulus onset and lasting 0.2–3 s, depending upon the stimulus intensity. These microsaccades directly modulate light input from the moving objects and consequently photoreceptor output, and we show through simulations and recordings how they can improve the spatiotemporal resolution of neural images (*Figures 8* and *9*; *Figure 8—figure supplement 1* and *Figure 9—figure supplement 1*).

## Conclusions

In this appendix, we described how receptive fields (spatial responsiveness) and acceptance angles ($\Delta\rho$) of wild-type and synaptically-blind $hdc^{JK910}$ R1-R6 photoreceptors were estimated, using intracellular voltage responses to new light-point array stimulation. Their characteristics in the dark- and moderately light-adapted states, and after prolonged light stimulation, were compared to test the hypothesis that spatial information summation in the lamina contributes to *Drosophila* photoreceptor function. We found that the dark-adapted wild-type R1-R6s show wider receptive fields. But in steady illlumination, the two photoreceptor groups' $\Delta\rho$-estimates adapted similarly.

Could the difference between the dark-adapted wild-type and $hdc^{JK910}$ receptive field widths result from recording artefacts? It is well known that LMCs' hyperpolarizing intracellular responses to light pulses depolarize the surrounding extracellular space (**Shaw, 1975**), seen as the on-transient in the electroretinogram (ERG) recordings (**Heisenberg, 1971**). Theoretically, these signals could be picked up by recording microelectrodes, adding artificial components to the intracellularly measured $V_n$, and so making wild-type photoreceptors' receptive fields appear wider (**Hardie et al., 1981**). However, we can essentially rule out this notion because the Gaussian functions, which quantified the actual recordings, were fitted to their real flank amplitudes rather than to zero (**Appendix 4—figure 5**). Consequently, any extra DC component would not affect the resulting $\Delta\rho$-estimates. Furthermore, our past comparative study, which included different synaptic mutant flies, failed to find clear signs for ERG contamination in high-quality intracellular wild-type photoreceptor recordings (**Zheng et al., 2006**). Whereas by lacking neurotransmitter histamine, $hdc^{JK910}$ LMCs cannot respond to light (**Burg et al., 1993**; **Melzig et al., 1996**; **Melzig et al., 1998**), and thus $hdc^{JK910}$ photoreceptor recordings cannot be ERG contaminated. Hence, we conclude that ERG signals could only have marginal contribution to our results at most.

In fact, the dark-adapted photoreceptors' $\Delta\rho$-estimates constituted the most reliable data in this appendix, with the largest number of samples ($n_{wild-type}$ = 19, $n_{hdc}$ = 18 cells) obtained through consistent recording protocols. In every experiment, the receptive field assessment in the dark-adapted state was strictly the first examination, following the standard stimulus centering procedure. This minimized any potential downgrade in the recording quality or variation in the stimulation history. Notably, our estimate of WT *Drosophila* R1-R6 photoreceptors' average receptive field half-width in the dark-adapted state, $\Delta\rho$ = 9.47 ± 0.36° (n = 19 cells; see **Method 3**, above), is reasonably similar to the previous estimate of 8.23 ± 0.54° (n = 11 cells) (**Gonzalez-Bellido et al., 2011**), which was obtained through a less stationary recording apparatus/method with more assumptions (see **Method 2**, above).

The dominant factors determining a fly photoreceptor's receptive field are optical, waveguide properties and, particularly for the chosen measurement method, the phototransduction characteristics (**Snyder, 1977**; **Land, 1997**; **Stavenga, 2003b**, **2003a**). For blowflies, it has been shown that the receptive field shape can be largely derived from the optical structure dimensions with the waveguide theory (**Smakman et al., 1984**). Given the $hdc^{JK910}$ mutants' seemingly normal ommatidial and rhabdomere optics, as seen *in vivo* (Appendix 7) and under electron microscopy (Appendix 5; R1-R6 rhabdomere diameters, $d$, were ~96% of the wild-type), and their wild-type like photoreceptor voltage/intensity relations to brief light pulses (**Dau et al., 2016**), it is reasonable to expect that their $\Delta\rho$-estimates would be close to wild-type. This should especially be true in light-adaptation, which is predicted to make photoreceptor output more independent of its neighbors (**Atick, 2011**; **van Hateren, 1992c**, **1992b**). And indeed, we found $\Delta\rho$-estimates of the light-adapted wild-type and $hdc^{JK910}$ photoreceptors alike. But because of this conformity, the 10.9% difference between their dark-adapted acceptance angles requires an additional explanation. Although at the flanks this difference increases (**Figure 7—figure supplement 1A**), it still may seem rather small when compared to the measured cell-to-cell variation within each genotype, with the respective maxima being 66% and 88% wider than the

minima, and its statistical significance becomes less with fewer samples (*Figure 7—figure supplement 2A*). Nevertheless, the finding is conceptually important as it supports an expansion in the classic spatial vision paradigm; from the optical constraints to spatial information summation in the network (*Stöckl et al., 2016*).

The dark-adapted wild-type photoreceptors' wider receptive fields are consistent with what we know about how synaptic inputs are channeled from lamina interneurons to R1-R6 axons (*Meinertzhagen and O'Neil, 1991*; *Sinakevitch and Strausfeld, 2004*; *Zheng et al., 2006, 2009*; *Abou Tayoun et al., 2011*; *Rivera-Alba et al., 2011*; *Hu et al., 2015*). Thus, the corresponding narrower receptive fields of mutant photoreceptors seem most sensibly attributed to the missing excitatory feedback modulation from their interneurons (*Zheng et al., 2006*; *Nikolaev et al., 2009*; *Zheng et al., 2009*; *Dau et al., 2016*). As shown by intracellular recordings (*Zheng et al., 2006*; *Dau et al., 2016*), feedforward and feedback signals dynamically contribute to photoreceptor and interneuron outputs *in vivo*. When the probability of light saturation is low, the stronger synaptic transmission in both pathways helps to amplify their response amplitudes.

Therefore, taken together with the findings of *Dubs et al. (1981)* and *Dubs (1982)*, these results suggest that under dim illumination, lateral excitation spreads synaptically within the lamina of the fly visual system. Spatial information summation is likely implemented by the first-order interneurons and fed back to photoreceptors through connections (*Meinertzhagen and O'Neil, 1991*; *Rivera-Alba et al., 2011*) that utilize excitatory neurotransmitters (*Zheng et al., 2006*; *Hu et al., 2015*). This model is further supported by our anatomical observations (Appendix 5 and Appendix 7), which imply that there are no major developmental defects in $hdc^{JK910}$ retina and that their lens systems and rhabdomere sizes are broadly wild-type-like.

Appendix 5

DOI: https://doi.org/10.7554/eLife.26117.062

## R1-R6 rhabdomere sizes, neural superposition and hyperacuity schemes

### Overview

This appendix shows how R1-R6 rhabdomere diameters vary systematically and consistently in *Drosophila* ommatidia and provides supporting background information for the results presented in *Figures 7–9* and Appendix 4.

In this appendix:

- We measure wild-type and *hdc^JK910* histamine-mutant (**Burg et al., 1993**; **Melzig et al., 1996**; **Melzig et al., 1998**) R1-R6 photoreceptors' rhabdomere sizes from the electron micrographs of their retinae.
- We compare these measurements to their electrophysiologically measured receptive field estimates (Appendix 4).
- We show that the mean wild-type R1-R6 rhabdomere diameter, $d_{R1-R6}$, is only ~4.1% wider than in *hdc^JK910* mutant eyes. This difference may contribute in part to their 10.9% wider average acceptance angle ($\Delta\rho$) in a dark-adapted state (Appendix 4), but cannot fully explain it (see also Appendix 7).
- We further show that in both phenotypes R1, R3 and R6 rhabdomeres are systematically larger than R2 and R4 rhabdomeres. As for the maximum difference, the mean R1 rhabdomere diameter ($d_{R1}$ = 1.8433 ± 0.1294 μm, mean ± SD) is about 18% wider than that of R4 ($d_{R4}$ = 1.5691 ± 0.0915 μm, n = 25, p = 2.3419×10$^{-11}$, 2-tailed t-test).
- These findings imply that each R1-R6 photoreceptor should have a different receptive field size (coinciding with their considerable $\Delta\rho$ variation seen in Appendix 4, *Figure 7—figure supplement 1C*).
- Our results further suggest an asymmetric information integration model, in which the neural superposition of different-sized overlapping receptive fields (of the neighboring R1-R6s) has a potential to contribute in enhancing *Drosophila*'s visual acuity beyond the presumed optical limits of its compound eyes (as revealed by behavioral experiments in Appendix 9).

### Readjusting the current theoretical viewpoint

Neural superposition eyes provide more samples from local light intensity changes for each image pixel, represented by large monopolar cell (L1 and L2) outputs. In the conventional viewpoint, each pixel's signal-to-noise improves by $\sqrt{6}$ because its L1 and L2 receive similar inputs from six 'physiologically identical' R1-R6 photoreceptors, which sample information from the same small area in the visual space. Thus, the conventional assumptions and limits for neural superposition performance are:

- Each R1-R6 functions virtually identically, generating similar outputs to the same light input
- Each R1-R6 in superposition has an identical receptive field size and shape
- The receptive fields in superposition overlap perfectly
- Each image pixel represents sampling and processing within its inter-ommatidial angle
- Inter-ommatidial angle sets the visual resolution of the *Drosophila* eye

This appendix presents anatomical and theoretical evidence that these assumptions and limits are overly simplistic and suggests ways the real neural images could be sharpened beyond them to improve *Drosophila* vision.

## Electron micrographs

### Fixation

Flies were cold anaesthetized on ice and transferred to a drop of pre-fixative (modified Karnovsky's fixative (**Shaw et al., 1989**): 2.5% glutaraldehyde, 2.5% paraformaldehyde in 0.1 M sodium cacodylate buffered to pH 7.3) on a transparent agar dissection dish. Dissection was performed using a shard of a razor blade (Feather S). Flies were restrained on their backs with insect pins through their lower abdomen and distal proboscis. Their heads were severed, probosces excised, and halved. The left half-heads were collected in fresh pre-fixative and kept for 2 hr at room temperature under normal lighting conditions.

After pre-fixation, the half-heads were washed (2 × 15 min) in 0.1 M Cacodylate buffer, and then transferred to a 1 hr post-fixative step, comprising Veronal Acetate buffer and 2% Osmium Tetroxide in the fridge (4°C). They were moved back to room temperature for a 9 min wash (1:1 Veronal Acetate and double-distilled $H_2O$ mixture), and serially dehydrated in multi-well plates with subsequent 9 min washes in 50%, 70%, 80%, 90%, 95% and 2 × 100% ethanol.

Post-dehydration, the half-heads were transferred to small glass vials for infiltration. They were covered in Propylene Oxide (PPO) for 2 × 9 min, transferred into a 1:1 PPO:Epoxy resin mixture (Poly/Bed 812) and left overnight. The following morning, the half-heads were placed in freshly made pure resin for 4 hr, and placed in fresh resin for a further 72 hr at 60°C in the oven. Fixation protocol was kindly provided by Professor Ian Meinertzhagen at Dalhousie University, Canada.

### Sectioning and staining

Embedded half-heads were first sectioned (at 0.5 μm thickness) using a glass knife, mounted in an ultramicrotome (Reichert-Jung Ultracut E, Germany). Samples were collected on glass slides, stained using Toluidine Blue and observed under a light microscope. This process was repeated and the cutting angle was continuously optimized until the correct orientation and sample depth was achieved; stopping when approximately 40 ommatidia were discernible. The block was then trimmed and shaped for ultra-thin sectioning. The trimming was necessary to reduce cutting pressure on the sample-block and resulting sections, thus helping to prevent 'chattering' and compression artefacts. Ultra-thin sections (85 nm thickness) were cut using a diamond cutting knife (DiATOME Ultra 45°, USA), mounted and controlled using the ultramicrotome. The knife edge was first cleaned using a polystyrol rod to ensure integrity of the sample-blocks. The cutting angles were aligned and the automatic approach- and return-speeds set on the microtome. Sectioning was automatic and samples were collected in the knife water boat.

Sections were transferred to Formvar-coated mesh-grids and stained for imaging: 25 min in Uranyl Acetate; a double-distilled $H_2O$ wash; 5 min in Reynolds' Lead Citrate (**Reynolds, 1963**); and a final double-distilled $H_2O$ wash.

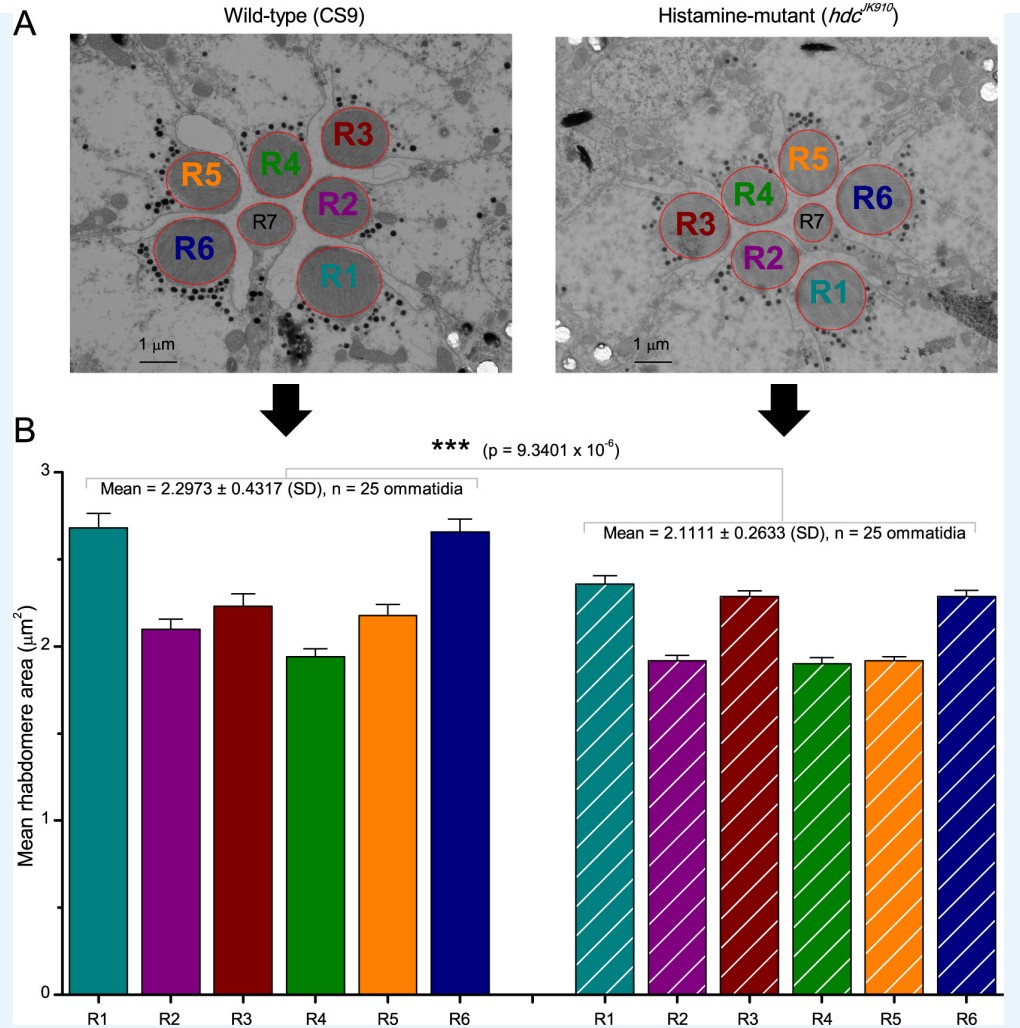

**Appendix 5—figure 1.** R1-R6 photoreceptors' rhabdomere sizes differ consistently. (**A**) Electron micrographs of a characteristic wild-type (left) and *hdc*[JK910] (right) ommatidia. Each ommatidium contains the outer receptors, R1-R6, and the inner receptors, R7/R8, which can be identified by their rhabdomeres' relative positions. Here R8s are not visible because these lie directly below R7s. Markedly, both wild-type and *hdc*[JK910] R1-R6 photoreceptor rhabdomere sizes vary systematically. (**B**) The mean rhabdomere sizes measured from 25 ommatidia from 10 flies. *hdc*[JK910] R1-R6 rhabdomere cross-sectional areas are smaller than those of the wild-type cells, but show similar proportional variations. Error bars show SEMs.

DOI: https://doi.org/10.7554/eLife.26117.063

## Rhabdomere measurements

Transmission EM images for R1-R6 rhabdomere size comparisons were taken below the rhabdomere tips, as sectioned 25 μm down from the corneal surface of the ommatidium lens. 25 wild-type and *hdc*[JK910] ommatidia (n = 25) from 10 flies of each phenotype were used to estimate R1-R6 rhabdomere sizes. The images were processed with ImageJ software. Because the rhabdomere cross-sectional area is often better approximated by an ellipse than a circle (*Appendix 5—figure 1A*), for greater accuracy, its circumference was fitted with an ellipse.

For an EM rhabdomere area, *A*, its mean diameter was then taken: $d = 2 \times \sqrt{\frac{A}{\pi}}$. Note that the obtained mean rhabdomere diameter estimates are somewhat smaller than the previous estimates, which measured the minimum and maximum rhabdomere diameters from edge-to-edge (*Gonzalez-Bellido, 2009*; *Gonzalez-Bellido et al., 2011*). Here, instead, we standardized the measurement protocol to reduce *d*-estimation bias between the wild-type and *hdc*[JK910]

electron micrographs to obtain straightforward statistical comparisons of their means. Nevertheless, both these and the previous (*Gonzalez-Bellido, 2009*) *d*-estimates indicated systematic rhabdomere size differences.

## R1-R6 rhabdomere sizes differ

*Drosophila* R1-R6 photoreceptors' rhabdomere sizes vary systematically and consistently in each ommatidium (*Appendix 5—figure 1B*). R1 and R6 rhabdomere cross-sectional areas are always the largest and R4 rhabdomeres the smallest (*Appendix 5—table 1*).

**Appendix 5—table 1.** Statistical comparison of wild-type Canton-S R1-R6 rhabdomere cross-sectional areas. The table gives the differences as p-values, calculated for 25 ommatidia of 10 flies at the same retinal depth. Red indicates statistically significant difference. R4 rhabdomeres are smaller than the other rhabdomeres, whereas R1 and R6 are always the largest.

| Significance t-test 2-tail | wild-type R1 | wild-type R2 | wild-type R3 | wild-type R4 | wild-type R5 | wild-type R6 |
|---|---|---|---|---|---|---|
| wild-type R1 | N/A | $5.3626 \times 10^{-7}$ | $1.3033 \times 10^{-4}$ | $3.6882 \times 10^{-10}$ | $1.5564 \times 10^{-5}$ | 0.8282 |
| wild-type R2 | $5.3626 \times 10^{-7}$ | N/A | 0.1531 | 0.0371 | 0.3760 | $3.4906 \times 10^{-7}$ |
| wild-type R3 | $1.3033 \times 10^{-4}$ | 0.1531 | N/A | 0.0011 | 0.5736 | $1.3187 \times 10^{-4}$ |
| wild-type R4 | $3.6882 \times 10^{-10}$ | 0.0371 | 0.0011 | N/A | 0.0047 | $1.2193 \times 10^{-10}$ |
| wild-type R5 | $1.5564 \times 10^{-5}$ | 0.3760 | 0.5736 | 0.0047 | N/A | $1.3479 \times 10^{-5}$ |
| wild-type R6 | 0.8282 | $3.4906 \times 10^{-7}$ | $1.3187 \times 10^{-4}$ | $1.2193 \times 10^{-10}$ | $1.3479 \times 10^{-5}$ | N/A |

DOI: https://doi.org/10.7554/eLife.26117.064

Theoretically, a fly photoreceptor's receptive field size depends upon its rhabdomere diameter, *d* (*Equation A4.3*). This relationship was recently supported experimentally by comparing the rhabdomere diameters and acceptance angle estimates, Δρ, of *Drosophila* R1-R6 photoreceptors to those of killer fly (*Coenosia attenuata*), both of which eyes have rather similar ommatidial lens sizes (16–17 vs 14–20 μm) and focal lengths (21.36 vs 24.70 μm) (*Gonzalez-Bellido et al., 2011*), *cf. Equation A4.3*. In ♀ *Drosophila*, maximum *d* was ~2 μm and in ♀ *Coenosia* ~1 μm, while *Drosophila*'s mean Δρ-estimate was 8.23 ± 0.54° and *Coenosia*'s 2.88 ± 0.07° (*Gonzalez-Bellido et al., 2011*). Thus, the wider rhabdomere tip correlates strongly with the wider receptive field.

Accordingly, with each *Drosophila* ommatidium hosting R1-R6 rhabdomeres of distinct size differences (*Appendix 5—figure 1B*, left; *Appendix 5—table 1*), the receptive fields of the neighboring R1-R6, which are pooled together in neural superposition, should differ in size and overlap broadly. These observations and analysis concur with the broad variability of the intracellularly measured wild-type R1-R6s' receptive field widths (Appendix 4:*Figure 7—figure supplement 1A,C*).

However, although collectively the mean wild-type R1-R6 rhabdomere cross-sectional areas (*Appendix 5—tables 2,3*) are larger than their *hdc*$^{JK910}$ counterparts (*Appendix 5—figure 1B*), their corresponding diameter differences are small. The average wild-type R1-R6 rhabdomere diameter, (WT $d_{R1-R6}$ = 1.70 ± 0.15 μm; mean ± SD, n = 150 rhabdomeres; *Figure 5*) is only ~4.1% wider than in *hdc*$^{JK910}$ mutant eyes (*hdc*$^{JK910}$ $d_{R1-R6}$ = 1.64 ± 0.10 μm; n = 150 rhabdomeres), as extrapolated from the ellipsoid rhabdomere area fits. Thus, *hdc*$^{JK910}$ photoreceptors' smaller average rhabdomere diameters can contribute to their ~ 10.9% narrower Δρ (Appendix 4:*Figure 7—figure supplement 1A,D*), but cannot fully explain it (*Appendix 5—figure 1B*, right).

**Appendix 5—table 2.** Statistical comparison of $hdc^{JK910}$ R1-R6 rhabdomere cross-sectional areas. The table gives the differences as p-values, calculated for 25 ommatidia of 10 flies at the same retinal depth. Red indicates statistically significant difference. R2, R4 and R5 rhabdomeres are smaller than R1, R3 and R6 rhabdomeres.

| Significance t-test 2-tail | $hdc^{JK910}$ R1 | $hdc^{JK910}$ R2 | $hdc^{JK910}$ R3 | $hdc^{JK910}$ R4 | $hdc^{JK910}$ R5 | $hdc^{JK910}$ R6 |
|---|---|---|---|---|---|---|
| $hdc^{JK910}$ R1 | N/A | $4.7387 \times 10^{-10}$ | 0.2281 | $6.3122 \times 10^{-10}$ | $1.0125 \times 10^{-10}$ | 0.2335 |
| $hdc^{JK910}$ R2 | $4.7387 \times 10^{-10}$ | N/A | $7.9300 \times 10^{-11}$ | 0.7006 | 0.9897 | $2.0471 \times 10^{-10}$ |
| $hdc^{JK910}$ R3 | 0.2281 | $7.9300 \times 10^{-11}$ | N/A | $1.9466 \times 10^{-10}$ | $5.0272 \times 10^{-12}$ | 0.9921 |
| $hdc^{JK910}$ R4 | $6.3122 \times 10^{-10}$ | 0.7006 | $1.9466 \times 10^{-10}$ | N/A | 0.6830 | $4.3275 \times 10^{-10}$ |
| $hdc^{JK910}$ R5 | $1.0125 \times 10^{-10}$ | 0.9897 | $5.0272 \times 10^{-12}$ | 0.6830 | N/A | $1.6775 \times 10^{-11}$ |
| $hdc^{JK910}$ R6 | 0.2335 | $2.0471 \times 10^{-10}$ | 0.9921 | $4.3275 \times 10^{-10}$ | $1.6775 \times 10^{-11}$ | N/A |

DOI: https://doi.org/10.7554/eLife.26117.065

**Appendix 5—table 3.** Statistical comparison of wild-type and $hdc^{JK910}$ R1-R6 rhabdomere cross-sectional areas. The table gives the differences as p-values, calculated for 25 ommatidia of 10 flies at the same retinal depth. Red indicates statistically significant difference. The wild-type and mutant R3 and R4 rhabdomeres are the same size (highlighted in bold); the other wild-type rhabdomeres are larger than their respective counterparts.

| Significance t-test 2-tail | wild-type R1 | wild-type R2 | wild-type R3 | wild-type R4 | wild-type R5 | wild-type R6 |
|---|---|---|---|---|---|---|
| $hdc^{JK910}$ R1 | 0.0014 | 0.0012 | 0.1434 | $1.0091 \times 10^{-7}$ | 0.0302 | 0.0015 |
| $hdc^{JK910}$ R2 | $1.8211 \times 10^{-11}$ | 0.0078 | $1.5520 \times 10^{-4}$ | 0.6952 | $7.1690 \times 10^{-4}$ | $3.5405 \times 10^{-12}$ |
| $hdc^{JK910}$ R3 | $5.1489 \times 10^{-5}$ | 0.0073 | 0.4818 | $1.8478 \times 10^{-7}$ | 0.1415 | $3.8432 \times 10^{-5}$ |
| $hdc^{JK910}$ R4 | $1.7483 \times 10^{-11}$ | 0.0051 | $1.0904 \times 10^{-4}$ | 0.5015 | $4.9066 \times 10^{-4}$ | $3.7421 \times 10^{-12}$ |
| $hdc^{JK910}$ R5 | $9.2505 \times 10^{-12}$ | 0.0056 | $1.0272 \times 10^{-4}$ | 0.6710 | $4.8280 \times 10^{-4}$ | $1.5313 \times 10^{-12}$ |
| $hdc^{JK910}$ R6 | $5.6987 \times 10^{-5}$ | 0.0083 | 0.4903 | $2.8960 \times 10^{-7}$ | 0.1480 | $4.3539 \times 10^{-5}$ |

DOI: https://doi.org/10.7554/eLife.26117.066

## Theoretical models for spatial hyperacuity

In the *Drosophila* compound eye, R1-R6 photoreceptors' mean acceptance angle ($\Delta\rho$ ~9.5°) is on average about twice as wide (see Appendix 4) as its mean inter-ommatidial angle ($\Delta\varphi$ ~4.5°) (*Gonzalez-Bellido et al., 2011*). Such an eye design could potentially facilitate spatial hyperacuity through neural image processing. In the vein of previous suggestions for manmade optoelectrical systems (*Luke et al., 2012*) and retinae of other species (*Zurek and Nelson, 2012*), we consider three alternative hypothetical scenarios based upon neighboring R1-R6s' overlapping receptive fields (RFs) (*Pick, 1977*).

The first case (*Appendix 5—figure 2*, left) considers RF variations of the photoreceptors in neural superposition. An image of an object (here, a thin 1° vertical bar, grey) stimulates simultaneously eight photoreceptors (R1-R8), which gather light information about the same area in the visual space by their overlapping RFs (for clarity only R1-R3' RFs are shown; orange Gaussians). However, because the rhabdomere diameters, *d*, vary considerably and

consistently between them (*e.g.* R1 and R6 rhabdomere diameters areas are ~18% wider than those of R4s; *Appendix 5—figure 1*), so could also be their RF sizes (see Appendix 4, *Equation A4.3*), causing overlaps. Six of these inputs (R1-R6) are pooled in the lamina. Consequently, even a small (1°) displacement of the vertical bar stimulus (grey) could lead to variable but specific light intensity modulations in each of the six converging input channels. Their signals to LMCs (L1-L3), which in high signal-to-noise conditions transforms input modulations into phasic responses (*Zettler and Järvilehto, 1972*; *Järvilehto and Zettler, 1973*; *Zheng et al., 2006*; *Zheng et al., 2009*; *Wardill et al., 2012*), could therefore be different before and after the bar displacement and possibly neurally detectable.

In the second case (*Appendix 5—figure 2*, middle), six photoreceptor cells of the same type (say R2s) in the neighboring ommatidia gather light information from the neighboring small visual areas (~4.5° apart, as separated by the interommatidial angle), but their receptive fields (~9.5° half-widths) overlap; for clarity only three R2s are shown: orange, blue and red. Lateral connections (L4, Lawf and Am cells) between lamina neuro-ommmadia could then be used by LMCs to compare their outputs, enhancing the spatial resolution of each neural channel (see Appendix 2, *Appendix 2—figure 5*).

In the third case (*Appendix 5—figure 2*, right), R1-R6 photoreceptor cells in the same ommatidium gather light information from the neighboring small visual areas, but their receptive fields overlap; again for clarity only three: orange (R1), blue (R2) and red (R3) are shown. Lateral connections (L4, Lawf and Am cells) between adjacent lamina neuro-ommatidia may enable LMCs to compare their outputs, enhancing the spatial resolution of each neural channel.

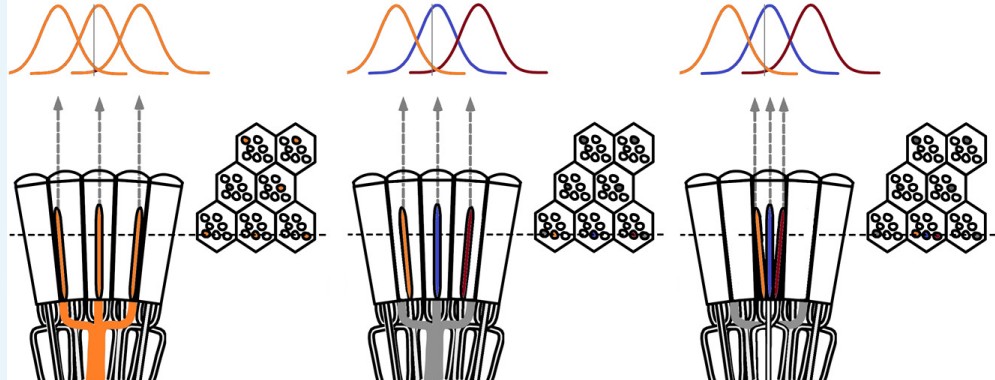

**Appendix 5—figure 2.** Theoretical ways to neurally improve the optical image resolution in the fly compound eyes. **Left**, neural superposition with overlapping photoreceptor receptive fields. **Middle**, same type photoreceptors (here R2s) in the neighboring ommatidia ($\Delta\varphi \sim 4.5$°) collect light from neighboring visual areas, but their receptive fields (RFs) overlap, having twice as large acceptance angles, $\Delta\rho \sim 9.5$°. **Right**, neighboring photoreceptors in the same ommatidium collect light from neighboring visual areas with their RFs overlapping. By comparing the resulting variable photoreceptor outputs from a small visual object (0.5° vertical grey bar), neural circuitry may resolve objects finer than the inter-ommatidial angle ($\Delta\varphi$).

DOI: https://doi.org/10.7554/eLife.26117.067

All these circuit models could theoretically contribute to motion vision hyperacuity, which occurs when a stationary *Drosophila* eye resolves object motion finer than its ~4.5° inter-ommatidial angle (the average sampling point or photoreceptor spacing). And crucially, in Appendix 10, we use a flight simulator system to demonstrate and quantify *Drosophila* hyperacute optomotor behavior to sub-interommatidial stimuli (1-4°). However in Appendix 7-8, we further provide decisive evidence that the spatiotemporal resolution of the early neural images is improved by synchronized and coherent photomechanical rhabdomere contractions (*Hardie and Franze, 2012*), which move and *narrow* R1-R6s' receptive fields. Together with refractory information sampling, this enables photoreceptors to encode space

in time. Therefore, as the RFs *narrow* with moving stimuli, this reduces the overlap between the neighboring RFs and, consequently, affects the potential resolving power of the circuit models in *Appendix 5—figure 2*.

We conclude that **at the photoreceptor level** the overlapping RFs provide neither necessary nor sufficient mechanistic explanations for the *Drosophila* spatiotemporal hyperacuity. However, **at the lamina interneuron level**, such connectivity arrangements may further enhance hyperacute vision. It is also possible that any kind of retinal image enhancement would be further coordinated centrally to match the visual needs of a locomoting fly. In walking blowflies, intraocular muscles in both the left and right eye seem to contract synchronously with increasing rates, causing vergence eye movements (*Franceschini et al., 1991*; *Franceschini and Chagneux, 1994*; *Franceschini et al., 1995*; *Franceschini and Chagneux, 1997*; *Franceschini, 1998*). Finally, we note that in lobula plate motion sensitive cells, responses to moving visual stimuli increase with locomotion (*Chiappe et al., 2010*; *Haag et al., 2010*; *Maimon et al., 2010*; *Tang and Juusola, 2010*), although there is currently no evidence to relate this phenomenon to hyperacuity, as characterized here.

## Possible dynamics arising from connectivity and variable rhabdomere sizes

Temporal output modulation through gap-junctions between R1-R6 and R7-R8 photoreceptor axons could further contribute, as a possible network mechanism, to the acuity improvements that we report in this publication (*Figures 7–8*; see Appendix 6). *Drosophila* R1-R6s have larger rhabdomeres, each with ~30,000 microvilli, whereas those of R7s and R8s contain only ~15,000. Owing to superposition in each neuro-ommatidium, both of these photoreceptor classes integrate photons from the same small area in space, but for given light changes, the macroscopic R1-R6 output rises and decays faster than those of R7s and R8s. This is because R1-R6 rhabdomeres integrate twice as many samples (quantum bumps) from the same stimulus and their membranes have likely smaller time constant (*Anderson and Laughlin, 2000*). Therefore, at each moment in time, R1-R6 and R7-R8 outputs carry a dynamic phase difference. If their responses were to antagonize each other through the gap-junctions between R6 and R7/R8 axons (*Shaw, 1984*; *Shaw et al., 1989*; *Wardill et al., 2012*), similar to the crosstalk between *Calliphora* R7 and R8 outputs (*Hardie, 1984*), phasic R1-R6 output components could be enhanced even further.

## Appendix 6

DOI: https://doi.org/10.7554/eLife.26117.068

# Neural images of moving point-objects (R1-R6 recordings vs. classic predictions)

## Overview

This appendix describes a new method to measure *Drosophila* photoreceptor output to moving dots, and shows how these responses provide much higher visual resolution and motion blur resistance than what is predicted by the classic theories, supporting the results in *Figures 7–9*.

In this appendix:

- We measure how well dark- and moderately light-adapted wild-type *Drosophila* R1-R6 photoreceptors resolve bright dots (point-objects), which cross their receptive fields at different speeds.
- We compare these intracellular recordings to those of histamine-mutants, $hdc^{JK910}$ (*Burg et al., 1993*; *Melzig et al., 1996*; *Melzig et al., 1998*), in which first-order interneurons are blind (receive no neurotransmitter from photoreceptors) and thus incapable of feedback-modulating the photoreceptor output.
- We further record voltage responses of blowfly (*Calliphora vicina*) R1-R6 photoreceptors to moving point-objects, as an additional test of our experimental setup, stimulus paradigm and mathematical analyses, validating this method.
- We evaluate the wild-type and mutant recordings against their respective classic model simulations, in which each recorded receptive field is convolved by the same cell's impulse response.
- Our results indicate that both wild-type and $hdc^{JK910}$ R1-R6s resolve moving dots about equally well, and significantly better than the corresponding classic model simulations.
- These findings demonstrate that the classic deterministic photoreceptor models (*Srinivasan and Bernard, 1975*; *Juusola and French, 1997*; *Land, 1997*) for resolving moving objects grossly underestimate the visual resolving power of real photoreceptors.
- Consequently, the classic theory overestimates the effects of motion blur on *Drosophila* vision during saccadic behaviors.

## Retinal limitations and capacity to resolve moving objects

A single photoreceptor's voltage responses can give important insight into neural image processing behind a fly's ability to detect small objects, whether in high-speed chasing flights or against a cluttered background (*Burton and Laughlin, 2003*; *Brinkworth et al., 2008*). Theoretically, when a point-object moves uniformly across an array of photoreceptors, each cell would produce a similar response. But these responses would be displaced in time, $t$, which it takes for the object to travel between two adjacent cells' receptive field centers (*Srinivasan and Bernard, 1975*). Thus, the response of the whole array, as a collective neural representation of the moving point-object, would be a travelling pattern with a mirrored waveform. Mathematically, this can be extrapolated from a single photoreceptor's response. We now apply this classic approach to simplify the questions about neural images of moving point-objects and compare its predictions to R1-R6s' spatiotemporal responses.

First, we consider the problem of neural latency compensation in motion perception. This is related to the flash-lag effects observed in humans (*Krekelberg and Lappe, 2001*; *Nijhawan, 2002*). Due to the inevitable phototransduction delays, every sighted animal should encounter this problem and *Drosophila* is not an exception. At 19°C, its photoreceptors' voltage responses rise ~10 ms from the light flash and peak 15–30 ms later (*Juusola and Hardie, 2001b*). Given that a fly's saccadic turning speed (*Fry et al., 2003*) in flight can exceed 1000 °/s, such delays need compensating by network computations. Otherwise, neural

images of its surroundings could lag behind their actual positions by more than 25°, making fast and accurate visual behaviors seemingly infeasible.

Such compensations have been shown to occur early on in vertebrate eyes. In the tiger salamander and rabbit retina, ganglion cells' firing rates lag behind flashing but not moving bars (**Berry et al., 1999**). Whether similar processing happens in insect eyes is unknown. By analyzing *Calliphora*, wild-type *Drosophila* and *hdc$^{JK910}$* R1-R6 output to a point-object crossing their receptive fields, we find their time-to-peak values broadly similar to those evoked by light flashes. Therefore, these lag times are not, or at most weakly, compensated by the signal spread between photoreceptors.

Second, we examine whether R1-R6 output to point-object motion displays directional preference, as suggested by the asymmetric synaptic feedback to them (**Meinertzhagen and O'Neil, 1991**; **Rivera-Alba et al., 2011**). While both L1 and L2 monopolar cells mediate a major neural pathway in the lamina, L2s show richer connectivity. Only L2 projects feedback to R1-R6 and have reciprocal connections with L4, which in turn connects to L4s of the neighboring neural cartridges (**Braitenberg and Debbage, 1974**), providing further feedback to photoreceptors (**Meinertzhagen and O'Neil, 1991**; **Rivera-Alba et al., 2011**). We find that whilst the photoreceptors' peak responses show no clear directional preference, their rise and delay time courses to front-to-back and back-to-front moving point-objects often differ significantly. Although it is still possible that these differences may in part be augmented by the asymmetric network feedback, we show in **Appendixes 7–8** that they actually originate from photomechanical R1-R8 contractions (**Hardie and Franze, 2012**).

Third, we investigate whether network regulation affects R1-R6 cell's spatiotemporal acuity. By using the classic approaches (**Srinivasan and Bernard, 1975**; **Juusola and French, 1997**; **Land, 1997**), we estimate the theoretical blur effects and the eye's ability to resolve bright dots moving at certain speeds. Furthermore, we apply the Volterra series, a widely used 'black-box' modeling method (**Marmarelis and McCann, 1973**; **Eckert and Bishop, 1975**; **Gemperlein and McCann, 1975**; **Juusola et al., 1995b**; **Korenberg et al., 1998**), to simulate wild-type and *hdc$^{JK910}$* R1-R6 output to these point-objects (**Juusola et al., 2003**; **Niven et al., 2004**). Remarkably, the simulations make it clear that these models cannot predict the recordings accurately (see Appendix 1 and **Figures 7** and **8G–I**).

Spatiotemporal resolution of the eye is thought to be determined by two components with special characteristics: the static spatial resolution of light input, as channeled through the optics (**Srinivasan and Bernard, 1975**; **Hornstein et al., 2000**), and the temporal response dynamics of photoreceptors. These characteristics are further influenced by the photoreceptors' adaptation state and synaptic feedback. The classic approaches suggest that because wild-type R1-R6s' acceptance angles ($\Delta\rho$) are 10.9% wider than those of *hdc$^{JK910}$* but their response dynamics are similar (see Appendix 4), they should produce blurrier images (of wider spatial half-width, *S*). Moreover, as *hdc$^{JK910}$* R1-R6s lack synaptic feedback modulation (**Dau et al., 2016**), their predicted higher acuity should reflect differences in spatiotemporal photon sampling dynamics. Our recordings show, however, that both wild-type and *hdc$^{JK910}$* R1-R6s resolve moving dots at least twice as well what the classic theory predicts, and that any resolvability difference between these cells largely disappears against a lit background. Thus, in dim conditions, lateral summation within the network may sensitize R1-R6 output by trading-off acuity, whereas in bright conditions more independent photoreceptor output sharpens neural images. Nonetheless, the classic theory cannot account for these dynamics, as it greatly overestimates the effect of motion blur on photoreceptor output.

We later demonstrate in **Appendixes 7–8** how and why the model simulations (of this appendix) differ from the corresponding recordings. Essentially, this is because the classic theoretical approaches do not incorporate two interlinked biophysical mechanisms that are critical for high acuity. (**i**) Rapid photomechanical photoreceptor contractions (**Hardie and Franze, 2012**) accentuate light input dynamically by shifting (front-to-back) and narrowing the cell's receptive field as moving bright point-objects enter in its view. While (**ii**) stochastic refractory photon sampling by microvilli accentuates the temporal dynamics in R1-R6 output. These mechanisms work together to improve the acuity and resolvability of moving objects far beyond the predictions of the classic models.

## Moving visual stimuli

The 25 light-point array and LEDs pads, which we used for creating images of moving objects and providing ambient illumination, respectively, are described in Appendix 4. In the *Drosophila* experiments, the 25 light-point array was placed 6.7 cm away from the fly, subtending an angle of 40.92°. This gave each light-point (dot) 1.7° size and minimum inter-dot-distance. In the *Calliphora* experiments, these parameters were 17 cm (distance) and 16.73° (viewing angle).

Images of one moving point-object (dot) were produced by briefly turning each light-point on and off, one after another in an incremental (for front-to-back direction) or decremental order (for back-to-front direction). Accordingly, Channel 0 input was driven with increasing or decreasing 'ramp' (*Appendix 6—figure 3A*), while Channel one input was set to 2 V. The travelling time of an object, or duration of the 'ramp', was between 50 ms and 2 s, resulting in object speeds within naturalistic range (*Schilstra and Hateren, 1999*; *Hateren and Schilstra, 1999*; *Fry et al., 2003*): from 20 to 818 °/s for *Drosophila* and 8 to 334 °/s for *Calliphora*.

The ability to resolve two moving point-objects of *Drosophila* photoreceptors were tested in dark-and light-adaptation conditions. In light-adaptation experiments, two 39-LED pads, on both sides of the 25 light-point array, provided background illumination (see Appendix 4, *Appendix 4—figure 2*). The two dots in the 25 light-point array were separated by 6.8° (four dark points in between) and moved together at different speeds; typically, 205, 409 or 818 °/s. Each stimulus was presented 8–10 times to the fly and the resulting photoreceptor responses were averaged before being analyzed.

## Gaussian white-noise (GWN) stimuli

To evaluate how well the classic theory of fly compound eye optics/function (*Srinivasan and Bernard, 1975*; *Juusola and French, 1997*; *Land, 1998*) explains single *Drosophila* R1-R6 photoreceptors ability to resolve moving dots, we needed to estimate each photoreceptor's linear impulse response (the first Volterra kernel) separately. The cell's voltage response to moving dots could then be predicted by convolving each recorded receptive field by the same cell's impulse response

Light-point No.13 intensity was controlled by setting Channel 0 input to 5 V and modulating Channel one input with a Gaussian white-noise (GWN) time series, which had the mean value of 2.5 V and cut-off frequency of 200 Hz. With these settings, light-point No.13 delivered $2.5 \times 10^6$ photons/s at **Peak1** (451 nm) and $3.75 \times 10^6$ photons/s at **Peak2** (575 nm) on average (*cf*. *Appendix 4—Table 1*). Finally, these intensities were reduced 100-fold by neutral density filtering

## Volterra series model of each tested *Drosophila* R1-R6

The principal assumptions of the Volterra series method are that the system has finite memory and is time-invariant (*Schetzen, 1980*). That is, (**i**) the relationship between output (photoreceptor voltage response) $y(t)$ and input (light stimuli) $u(t)$ is characterized by an unchanging impulse response and (**ii**) $y(t)$ depends only on current and past values of $u(t - \tau) \rightarrow u(t)$ with limited regression time, $\tau$. The continuous form of this input/output relationship is described by the following equation:

$$y(t) = k_0 + \int_0^T k_1(\tau)u(t-\tau)d\tau + \int_0^T \int_0^T k_2 u(t-\tau_1)u(t-\tau_2)d\tau_1 d\tau_2 \qquad \text{(A6.1)}$$

where $k_0$, $k_1$ and $k_2$ are the zero-, first- and second-order time-invariant kernels, which define the system's impulse response. $T$ is the finite system memory limit.

Note that the model order is not limited, as expressed only up to second-order in *Equation A6.1*, but instead could be extended arbitrarily further. However, it has been shown

that a light-adapted fly photoreceptor's response to GWN light intensity time series stimulation, as used in these experiments, can be approximated well by the linear terms (**Juusola et al., 1994**; **Juusola et al., 1995a**).

Therefore, the estimation of system output was simplified to a linear convolution of input with zero- and first-order kernels. Each measurement of photoreceptor voltage response and light stimuli could be fitted into the discrete and simplified form of **Equation A6.1** as:

$$
\begin{aligned}
y(n) &= k_0 + k_1(0)u(n) + k_1(1)u(n-1) + \ldots + k_1(T)u(n-T) \\
y(n-1) &= k_0 + k_1(0)u(n-1) + k_1(1)u(n-2) + \ldots + k_1(T)u(n-1-T) \\
&\quad\vdots \\
y(n-N) &= k_0 + k_1(0)u(n-N) + k_1(1)u(n-N-1) + \ldots + k_1(T)u(n-n-T)
\end{aligned}
\tag{A6.2}
$$

The group of **Equation A6.2**, which approximates $N$ values of photoreceptor output, was then re-arranged into matrix form:

$$
\begin{pmatrix}
y(n) \\
y(n-1) \\
\vdots \\
y(n-N)
\end{pmatrix}
=
\begin{pmatrix}
1 & u(n) & u(n-1) & \cdots & u(n-T) \\
1 & u(n-1) & u(n-2) & \cdots & u(n-1-T) \\
& & \vdots & & \\
1 & u(n-N) & u(n-N-1) & \cdots & u(n-N-T)
\end{pmatrix}
\times
\begin{pmatrix}
k_0 \\
k_1(0) \\
k_1(1) \\
\vdots \\
k_1(T)
\end{pmatrix}
\tag{A6.3}
$$

Equivalently, **Equation A6.3** could be symbolized as:

$$
Y = P\theta
\tag{A6.4}
$$

where vector $Y$ contained a sequence of $N$ output values, $P$ was the regression matrix constructed from the lagged input values, and the column $\theta$ elements were the kernel values. The problem of determining a photoreceptor's Volterra series model was hence broken down to designing input stimuli $u(t)$ and measuring output values $y(t)$ to construct matrices $P$ and $Y$ of **Equation A6.4**, and estimating $\theta$.

$u(t)$ was a GWN series with 200 Hz bandwidth, which thus tested the whole frequency range of photoreceptor output. Initially, each tested photoreceptor was steady-state-adapted to the chosen light background (the average brightness of the GWN stimuli) for 30–60 s (see also Appendix 4). Input was then delivered from the light-point No.13, by setting Channel 0 to 5 V and modulating Channel one input by the GWN series around the mean value of 2.5 V. Each time series was 3-second-long and was repeated 8–10 times before the responses were averaged. The first 1.5 s of the recorded data was used to estimate the kernel values.

Photoreceptor output $y(t)$ was sampled at 10 kHz. It was then preprocessed by removing the mean value and trends, and down-sampled.

Once the matrices $Y$ and $P$ of **Equation A6.4** are constructed, there are several approaches to estimate $\theta$ with minimal error, such as the least squares regression by using Gram-Schmidt orthogonalisation (**Korenberg et al., 1988**; **Korenberg and Paarmann, 1989**) or Meixner functions (**Asyali and Juusola, 2005**). Here, $\theta$ was approximated by the single value decomposition method (**Golub and Reinsch, 1970**; **Lawson and Hanson, 1974**), in which the factorization of matrix $P$ and the calculation of its Moore-Penrose pseudoinverse matrix, $P^+$, were carried out by the command *pinv(P)* in MATLAB. The linear least-squares estimation of $\theta$, $\theta$, was given by:

$$
\theta = P^+ Y
\tag{A6.5}
$$

The computed kernels and the second half of GWN stimuli were then substituted to **Equation A6.2** to yield the model prediction of the photoreceptor response. The accuracy, or fitness, $F$, of the prediction was quantified by the complement of its mean squared error:

$$F = 1 - MSE = 1 - \frac{(y' - \bar{y})^2}{(y - \bar{y})^2} \qquad \text{(A6.6)}$$

where $y$ were the actual data measured from the photoreceptor voltage response and $y'$ were the values simulated by the mathematical model.

## Conventional simulation of intracellular responses to object motion

After determining and testing the Volterra model, we next approximated the light stimuli (input) delivered by the point-objects crossing a photoreceptor's receptive field. Since a photoreceptor's voltage response was assumed to be linearly correlated to light input, the response amplitudes to the subsaturating light flashes during the receptive field scans (see Appendix 4) were also considered linear measurements of the effective intensity from each light-point. Therefore, $u_m(t)$, created by one moving point-object, was modelled as 25 intensity steps, in which amplitudes were proportional to their corresponding flash responses (*Appendix 6—figure 1*). The temporal width of each step was calculated according to the object speed. Similarly, $u'_m(t)$ of two moving point-objects was constructed from the superimposition of $u_m(t)$ and $u_m(t + \Delta)$, where $\Delta$ was calculated according to the point-objects' speed and their separation angle.

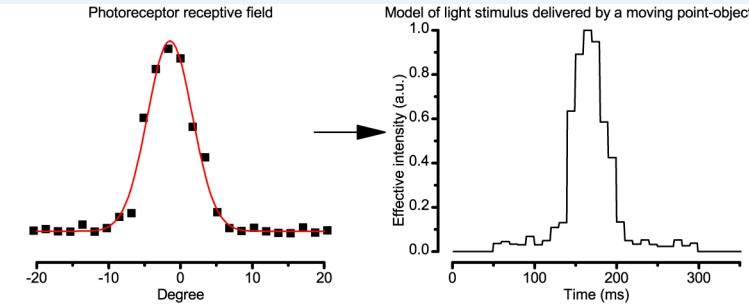

**Appendix 6—figure 1.** Estimating the light input from a moving object. The effective light stimulus, which was delivered to a photoreceptor by a moving point-object, was estimated by a linear transformation of the cell's receptive field. The width of each intensity step was calculated according to the point-object speed.

DOI: https://doi.org/10.7554/eLife.26117.069

Lastly, a photoreceptor's voltage response to point-object motion was simulated by substituting input $u_m(t)$ or $u'_m(t)$ and the kernels values to the zero- and first-order terms of *Equation A6.2*.

## Maximal responses to moving dots lag behind their actual positions

Neural images in the fly retina, lamina and medulla are generated by retinotopically mapping the surrounding light intensity distribution. This means that light coming from each point in space is sampled and processed by one neural cartridge (or neuro-ommatidia) (*Meinertzhagen and O'Neil, 1991*). While a stationary object might be seen by several photoreceptors belonging to neighboring ommatidia due to their large acceptance angles (Appendix 4) and overlapping receptive fields (Appendix 5), the object position is almost certainly perceived on the photoreceptor's optical axis. This position, along its corresponding lamina/medulla cartridge below, produces the largest/fastest intracellular responses (*Appendix 6—figure 2*; see also Appendix 4) as it channels the maximum light influx into the rhabdomere (see Appendix 2).

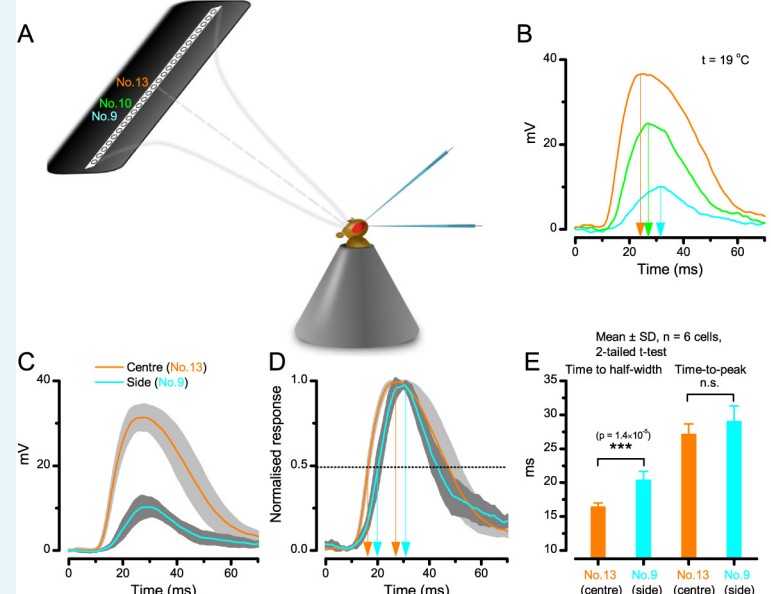

**Appendix 6—figure 2.** A R1-R6 photoreceptor's sensitivity is the highest at the center of its receptive field. (**A**) Schematic showing how subsaturating isoluminant light pulses were delivered at different locations within each tested photoreceptor's receptive field. (**B**) The center stimulus (light-point No.13) typically evoked a response (orange trace) with the larger amplitude and faster rise-time than any stimulation at the flacks (light-points No.9 and No.10, respectively). (**C**) Average voltage responses of six photoreceptors to corresponding center and side stimulation. (**D**) Normalized responses make it clear that the responses to the center stimulus rise faster. (**E**) Time to the half-width response is significantly briefer (p = $1.39 \times 10^{-5}$) with the center stimulus. However, time-to-peak of the responses to center and side light-points shows more variability between individual cells (p = 0.125). Mean ± SD; two-tailed t-test; $n_{wild-type}$ = 6 cells.

DOI: https://doi.org/10.7554/eLife.26117.070

Furthermore, it is customarily assumed that a moving point-object's position would be associated with the peak of its neural image. However, this does not necessarily mean that a R1-R6's response maximum would indicate the object position. Rather, it is more plausible - especially during high signal-to-noise ratio conditions (bright stimulation) - that the lamina circuitry (*cf.* Appendix 2, **Appendix 2—figure 7**) would be amplifying more the photoreceptor signal derivative. This is because the large monopolar cells (LMCs) are then more tuned to responding to the rate of light changes (**Zettler and Järvilehto, 1972**; **van Hateren, 1992b**; **Juusola et al., 1995a**; **Zheng et al., 2006**).

In fact, neural latency might be compensated at the subsequent processing stages in the visual system, so that the peak of the travelling network response wave would closely follow the object's actual position, as is the case of the vertebrate ganglion cells (**Berry et al., 1999**). Theoretically, if such 'correction' occurred maximally at the single neuron level, it would mark the coincidence of two events. (**i**) The neuron's response would peak as (**ii**) the object passes its receptive field center.

To examine whether, or to what degree, neural images of moving objects are compensated for latency at the first processing stage in the fly R1-R6 photoreceptors, we measured their intracellular voltage responses while presenting the fly with a bright dot moving at different speeds.

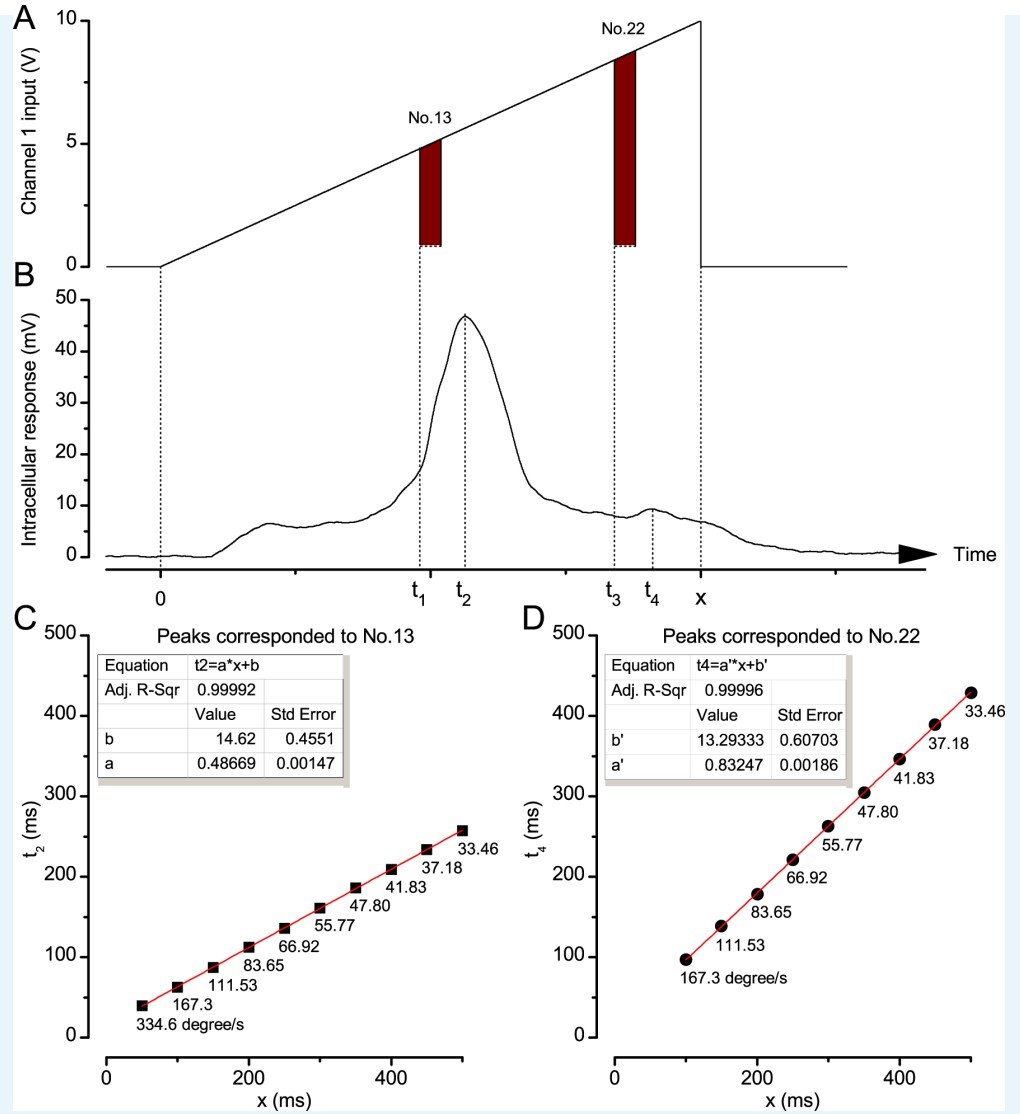

**Appendix 6—figure 3.** Photoreceptor response maxima lag moving objects. (**A**) Channel 1 input was driven by an incremental ramp to create an image of a bright dot (point-object) moving from the No.1 light-point to the No.25 (front-to-back). Similar decremental ramps were used to produce back-to-front motion. (**B**) Intracellular responses of *Calliphora* photoreceptors to a moving point-object showed two response peaks: a large peak at $t_2$, which corresponded to the moment it travelled pass the cell's optical axis at $t_1$, and a smaller peak at $t_4$ caused by the exceptional brightness of light-point No.22, which was turn on at $t_3$. $x$ was the object's travelling time. (**C**) An example of the linear correlation between $t_2$ and $x$. Below each data point is its corresponding stimulus (dot) velocity. (**D**) An example of the linear correlation between $t_4$ and $x$. Again, the corresponding dot velocities are shown.

DOI: https://doi.org/10.7554/eLife.26117.071

*Appendix 6—figure 3B* depicts a typical response waveform of a blowfly (*Calliphora vicina*) R1-R6 to a bright moving dot (the point-object in *Appendix 6—figure 3A*). Let $x$ be the time needed for it to travel through the 25 light-point array, $t_1$ be the moment when the dot pass the cell's optical axis, *i.e.* the corresponding light-point is turned on, and $t_2$ be when the intracellular response peaks. With varying $x$, and thus the dot speed, $t_1$, can be computed as:

$$t_1 = a \times x \tag{A6.7}$$

where the coefficient $a$ is a constant. The aim was to align the 25 light-point array so that the point No.13 lies at the tested photoreceptor's receptive field center. Therefore in theory, $a$ is approximately 0.48. However, the light-point No.13 might, in fact, be off-axis. For example, the cell's receptive field center could lie in between No.13 and No.12, causing inaccuracy in the calculation of $a$, $t_1$ and lag time $b$, which is given by:

$$b = t_2 - t_1 \qquad (A6.8)$$

To overcome this ambiguity, we plotted $t_2$ against $x$, given that:

$$t_2 = t_1 + b = a \times x + b \qquad (A6.9)$$

**Appendix 6—figure 3C** illustrates an example of the relationship between $t_2$ and $x$ obtained from a *Calliphora* R1-R6. The two parameters fitted exceedingly well to a linear relationship (adjusted R-squared >0.9999), in which coefficient $a$ and lag time $b$ were found as 0.486 and 14.62 ms, respectively. These data show that in this particular case, indeed light-point No.13 was close to the center of the cell's receptive field and that lag time $b$ was virtually unchanged for different object speeds. The on-axial position of the light-point No.13 was later confirmed by 11 receptive field scans, all of which indicated that response elicited by a light flash from No.13 was the largest (data not shown).

The same photoreceptor was also stimulated by repeating a light flash, to which its voltage response peaked 15 ms later. The small difference between the lag time of motion response and the response time-to-peak to a flash does not readily imply neural network latency compensation. In case of the moving stimuli, photoreceptor was stimulated when the dot entered its receptive field, causing its intracellular voltage to start depolarizing before the dot reached the cell's optical axis. Conversely, the photoreceptor's response to light impulse was only elicited after the stimulus onset, making its time-to-peak slightly longer than the lag time.

In the classic flash-lag psychological experiment, where a flashing bar and another uniformly illuminated one travelled together, the former was perceived to be trailing (**Nijhawan, 1994**; **Brenner and Smeets, 2000**). Thus, it is probable that at some stage in the visual system, the voltage response peak (maximum) caused by moving object would display shorter delay than those elicited by increasing light intensity. In the present study, the light-point No.22 was 4-fold brighter than the others, as discussed in Appendix 4, and indirectly played the role of the flashing bar, causing a 'local peak' in a photoreceptor's voltage response (**Appendix 6—figure 3B**). Hence, to further examine neural latency of R1-R6 output, we next assessed the lag time, $b'$, corresponding to this peak in the response. Given $t_3$ is the moment when No.22 was turned on, which can be calculated as:

$$t_3 = a' \times x \qquad (A6.10)$$

and is the time of the local response peak (**Appendix 6—figure 3**), lag time is defined as their difference:

$$b' = t_4 - t_3 \qquad (A6.11)$$

The relationship between $t_4$ and $x$ could also be described by linear fitting with almost zero residue (**Appendix 6—figure 3D**), yielding $a'$ and $b'$ values of 0.83 and 13.29 ms, respectively.

These data exemplify that a photoreceptor's response maxima, no matter whether caused by a point-object moving across its receptive field or by an unexpected increase in light intensity, show similar lag time characteristics. Both $b$ and $b'$ were independent of the object speed and comparable to the response time-to-peak, as induced by a comparable flash. These features were reproducible and general; observed in all seven tested *Calliphora* photoreceptors, without exception (**Appendix 6—table 1**).

**Appendix 6—table 1.** Response latency to dot motion analyses in *Calliphora* R1-R6s (Mean ± SD). Intracellular recordings were performed at 19°C. The tested moving dot (point-object) velocities were: 334.6, 167.3, 111.53, 83.65, 66.92, 55.77, 47.8, 41.83, 37.18 and 33.46 °/sec.

| Animal | Flash response time-to-peak (ms) | Peaks corresponding to the receptive field center | | Peaks corresponding to the light-point No.22 | |
|---|---|---|---|---|---|
| | | Lag-time $b$ (ms) | Adj. R-Sqr | Lag-time $b'$ (ms) | Adj. R-Sqr |
| *Calliphora* | 14.85 ± 0.78 | 14.6 ± 0.64 | 0.99985 ± 0.00012 | 13.9 ± 3.59 | 0.99985 ± 0.00017 |
| | | n = 7 | | n = 5 | |

DOI: https://doi.org/10.7554/eLife.26117.072

*Drosophila* photoreceptors' voltage responses did not clearly exhibit the 'local peak' to the light-point No.22; possibly owing to their slower temporal dynamics. Nevertheless, the temporal position of the 'global peak', measured by $t_2$ of wild-type (n = 12) and $hdc^{JK910}$ (n = 3) photoreceptor outputs, consistently showed linear correlation to $x$ and comparable lag-time/time-to-peak values (*Appendix 6—table 2*). Their corresponding maxima showed slightly larger lag time variations, and thus the linear fits were not as error-free as with the *Calliphora* data. Nonetheless, overall, the mathematical relation between their peak response lag time and the object speed appeared similar. The comparable wild-type and $hdc^{JK910}$ R1-R6 output maxima to the tested point-object velocities, as recorded from their somata, suggests that their response dynamics mostly reflect similar phototransduction processing, with possibly only marginal influence from the lamina network.

**Appendix 6—table 2.** Response latency to dot motion analyses in *Drosophila* wild-type and hdc$^{JK910}$ R1-R6s (Mean ± SD). Intracellular recordings were performed at 19°C. The tested point-object velocities were: 818.4, 409.2, 272.8, 204.6, 163.68, 136.4, 116.91, 102.3, 90.93 and 81.84 °/s. Note, these statistics are collected from individual recordings, not from paired data.

| Animal | Flash response time-to-peak (ms) | Front-to-back | | Back-to-front | |
|---|---|---|---|---|---|
| | | Lag-time $b$ (ms) | Adj. R-Sqr | Lag-time $b$ (ms) | Adj. R-Sqr |
| Wild-type *Drosophila* | 23.81 ± 1.41 | 21.41 ± 4.5 | 0.99649 ± 0.0065 | 22.54 ± 4.15 | 0.99378 ± 0.007 |
| | | n = 12 | | n = 5 | |
| hdc$^{JK910}$ | 24.4 ± 1.08 | 21.82 ± 1.36 | 0.9992 ± 0.0008 | 23.79 ± 5.72 | 0.9978 ± 0.003 |
| | | n = 3 | | | |

DOI: https://doi.org/10.7554/eLife.26117.073

Altogether, these data imply that fly phototransduction machinery (see **Appendixes 1–2**) samples intensity changes and object motion much the same way. Because its peak responses lag behind the actual positions of the moving objects, the neural latency of moving objects is most likely compensated downstream by image processing within the interneuron networks, starting with the LMCs (*cf.* Appendix 2, *Appendix 2—figure 7E*).

## Response rise and decay to object motion show directional selectivity

As summarized in *Appendix 6—table 2*, *Drosophila* photoreceptors' maximum responses to a front-to-back or back-to-front moving bright dot did not exhibit clear signs of *latency compensation*, as indicated by their similar time-to-peak durations (estimated from the population means of individual unpaired recordings). Interestingly, in the paired recordings, however, the response rise and decay time-courses often showed considerable *latency modulation*.

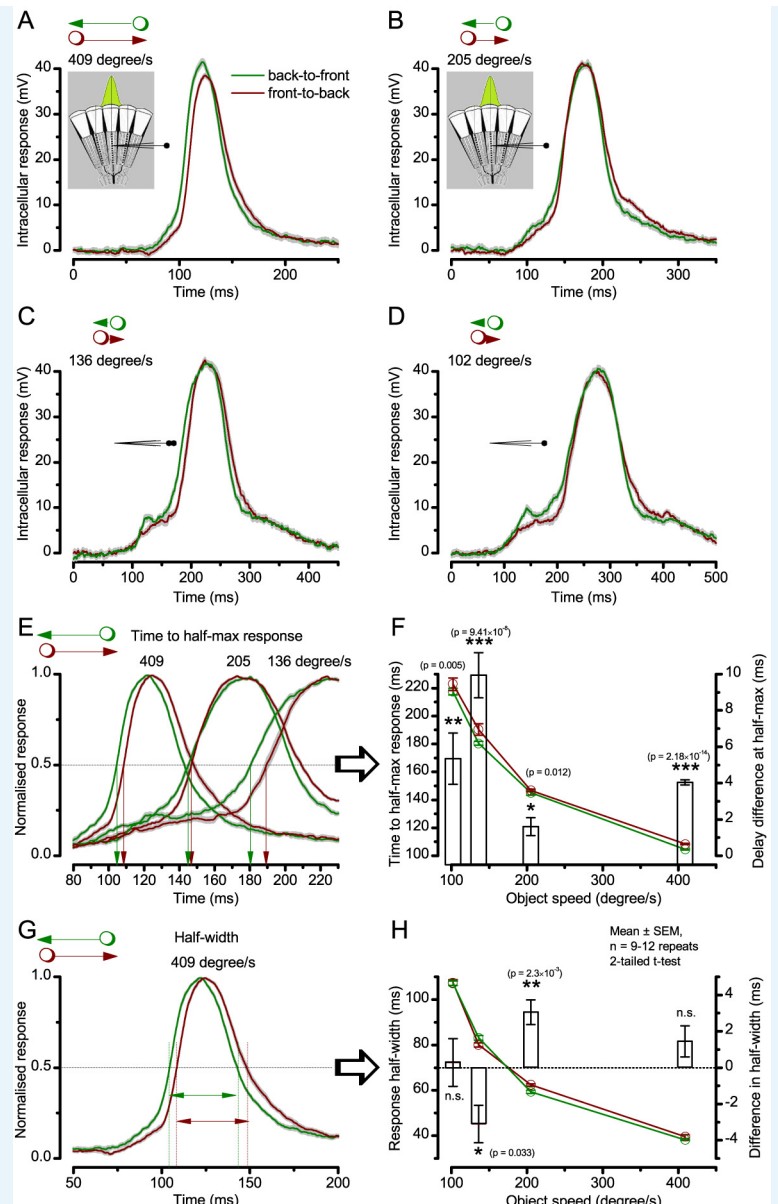

**Appendix 6—figure 4.** Photoreceptor output's time-to-peak is insensitive to object motion direction but its waveforms rise and decay faster to back-to-front motion. Examples of intracellular voltage responses of the same dark-adapted *Drosophila* photoreceptor to a bright dot (point-object), which crosses its receptive field in back-to-front (green) or front-to-back (brown) directions (Mean ± SEM). The insets show a schematic of the compound eye structure, with the green Gaussian representing a R1-R6's receptive field. (**A–D**) Voltage responses to 409, 205, 136 and 102 °/s object speeds, respectively, plotted over the whole duration of the corresponding stimuli. (**E**) Whilst the similarly timed respective response peaks showed no clear directional preference, the response waveforms, nevertheless, systematically rose and decayed earlier to back-to-front (green) than to front-to-back motion (brown), irrespective of the object speed. (**F**) The response rise-time (measured by time to half-maximal response), decreased with increasing object motion but it was always less to corresponding back-to-front (green) than front-to-back (brown) motion. This delay difference (white bars) was significant for all the tested object speeds and varied between 2 and 10 ms (mean ± SEM, $0.01 \leq p \leq 2.18 \times 10^{-14}$, $9 \leq n \leq 12$ trials, two-tailed t-test). (**G**) Because the response rise and decay times changed in unity, the resulting response half-widths to corresponding back-to-front and front-to-back object motion were largely similar. (**H**)

Accordingly, the response half-width decreased with increasing object motion, but showed only small (<4 ms) inconsistent differences (bars) between the corresponding back-to-front and front-to-back object speeds. These results are consistent with the fast light-induced photoreceptor contractions, which move their receptive fields in front-to-back direction, as observed directly in high-speed video recordings *in vivo* (Appendix 7).
DOI: https://doi.org/10.7554/eLife.26117.074

When a R1-R6 contracts to light input, its receptive field moves front-to-back (**Appendixes 7–8**). Thus, with the ommatidium lens inverting images, its responses to the back-to-front dot motion raise systematically slightly earlier (*Appendix 6—figure 4E,F*). This rise-time lag reduces because the dot moves against the receptive field motion, whilst the rise-time lag increases during comparable front-to-back motion when the dot moves along the receptive field motion. In other words, a dot stayed a bit longer within a R1-R6's receptive field during front-to-back motion than back-to-front motion. Such phasic differences were consistently observed in most recordings over the tested speed range. For example, 205 °/s back-to-front movement evoked narrower temporal response half-widths in 8/10 R1-R6s than the opposite movement. Similarly, 409 °/s back-to-front movement evoked narrower temporal response half-widths in 6/10 R1-R6s (in 2 cells, these were identical; and wider in 2).

*Appendix 6—figure 4* depicts intracellular responses to a moving dot, passing a photoreceptor's receptive field front-to-back and back-to-front at (**A**) 409, (**B**) 205, (**C**) 136 and (**D**) 102 °/s.

Although their time-to-peak values appeared similar, the response rise and decay dynamics showed clear differences (*Appendix 6—figure 4E–H*), which correlated with the dot speed and motion direction. We shall later show in Appendix 7, using high-speed video recordings of photoreceptor rhabdomeres, that their photomechanical contractions (*Hardie and Franze, 2012*) occur in back-to-front direction. Light input modulation by these directional microsaccades can much explain the phasic differences in photoreceptor output to different directional point-object motions.

In summary, a prominent feature of R1-R6s' voltage responses to opposing object motion directions is their similar time-to-peak values. This was found in all somatic recordings of *Calliphora* and *Drosophila* photoreceptors. Intriguingly, though, we further identified small (2–10 ms) but significant differences in the response rise and decay to front-to-back and back-to-front object motion. These phasic differences in photoreceptor output can be largely explained by each cell's directional photomechanical contractions, and we later show how these contribute to improving the fly's visual acuity (**Appendixes 7–8**). It is plausible that these directional preferences would be further enhanced downstream at the level of network processing. During bright stimulation, LMCs respond most strongly to the rate of change in photoreceptor output (*Juusola et al., 1995a*; *Zheng et al., 2006*; *Zheng et al., 2009*; *Wardill et al., 2012*), with the rich connectivity of the optic lobes proving further possibilities for the required phase coding (*Meinertzhagen and O'Neil, 1991*; *Rivera-Alba et al., 2011*; *Wardill et al., 2012*; *Behnia et al., 2014*).

## Classic theory greatly overestimates motion blur in R1-R6 output

We showed in Appendix 4 that dark-adapted *hdc^{JK910}* photoreceptors have narrower receptive fields (acceptance angles) than their wild-type counterparts but broadly similar response dynamics (*Figure 7—figure supplement 1*). Thus, the prediction is that *hdc^{JK910}* R1-R6s should produce slightly sharper neural images than their wild-type counterparts after dark-adaptation. Classic theoretical approaches have been used to predict how the spatial and temporal factors might jointly affect visual acuity (*Srinivasan and Bernard, 1975*; *Juusola and French, 1997*). Accordingly here, we first predict with them the motion blur effects on wild-type and *hdc^{JK910}* R1-R6 outputs. Later on, we test the ability of these cells to distinguish two dots moving together, separated by less than the cell's acceptance angles.

Since a fast moving bright dot can stimulate several photoreceptors virtually at the same time (*Appendix 6—figure 5A*), theoretically, it should not be perceived as a single point but

a streak, of which length is a function of object speed. This motion blur effect is classically quantified by the spatial half-width $S$ of object's neural image. Because the spatial response in the retina has a similar waveform with the temporal response of a single photoreceptor (**Srinivasan and Bernard, 1975**; **Juusola and French, 1997**), $S$ can be calculated as:

$$S = w \times T_h \tag{A6.12}$$

where $w$ is the object speed and $T_h$ (cf. **Appendix 6—figure 4G**) is the temporal half-width of a single photoreceptor response.

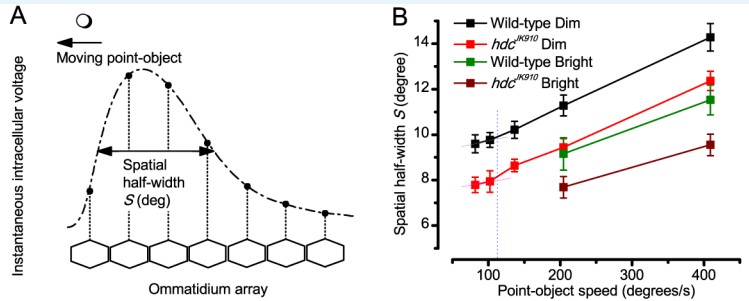

**Appendix 6—figure 5.** Object motion blur according to the classic theory, applied to the neural images in the *Drosophila* eye as it was thought to affect them in the past. (**A**) Hypothetical spatial pattern of an instantaneous voltage response at an ommatidial array produced by a moving point-object. Figure redrawn from (**Srinivasan and Bernard, 1975**). (**B**) Spatial half-width of neural image of a moving point-object as a function of its speed in Dim and Bright conditions. For all the tested speeds, $S_{hdc}$ were significantly smaller than $S_{wild-type}$ (p = 0.0036–0.049, t-test), except for 205°/s in Bright condition, where the statistical test yielded p=0.137. S was calculated using data at 19°C. Mean ± SEM, $n_{wild-type}$ = 4–15, $n_{hdc}$ = 3–16, two-tailed student test.
DOI: https://doi.org/10.7554/eLife.26117.075

 **Appendix 6—figure 5B** illustrates the predicted relationship between the object speed and neural image resolution in wild-type and $hdc^{JK910}$ *Drosophila*. These estimates imply that the spatial half-width of wild-type neural images should be 1-2° wider than that of the $hdc^{JK910}$ during both the dim and bright conditions, reflecting wild-type R1-R6s' wider acceptance angles (Δρ) (see Appendix 4, **Figure 7—figure supplements 1A** and **2B**). This prediction agrees with the previous theoretical works (**Srinivasan and Bernard, 1975**; **Juusola and French, 1997**), which used similar methods to indicate two distinct regions of image resolution. Thus, theoretically, at low object speeds, visual acuity should be mostly determined by a photoreceptor's spatial receptive field, but at high speeds, the motion-blur effect should increase rapidly, becoming the dominating factor. The corresponding trend differences (as separated by a thin dotted line) suggest that the point-object speed threshold dividing the two regions would be about 100–120 °/s.

 Remarkably, however, we next demonstrate how these theoretical predictions greatly overestimate motion blur effects in R1-R6 output, and that these cells can, in fact, resolve image details finer than the half-width of their receptive fields, even at very high saccadic speeds.

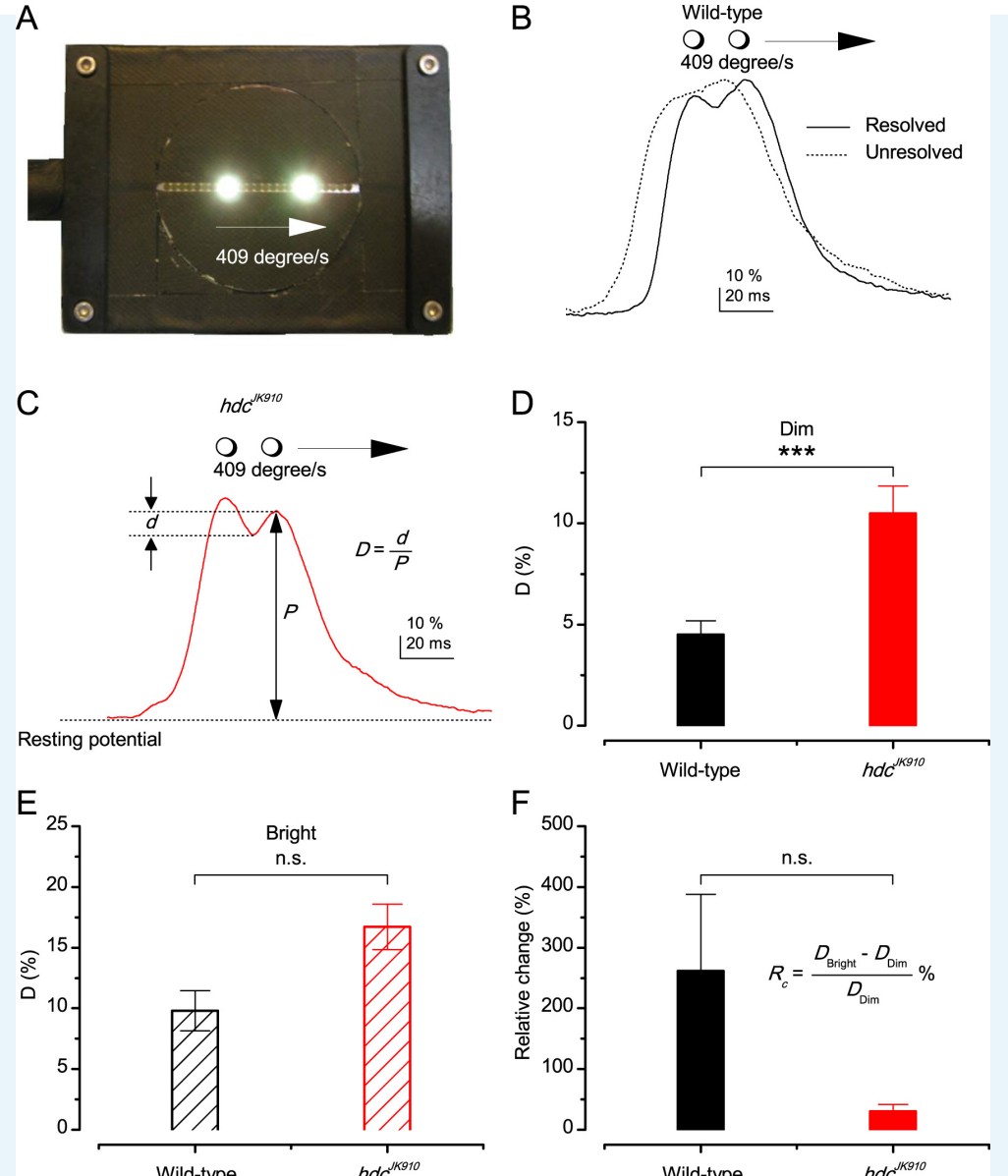

**Appendix 6—figure 6.** Resolving two moving point-objects. (**A**) Two light-points (bright dots) moving together were presented in the tested photoreceptor's receptive field at 19°C. The dots were 6.8° apart and travelled at 409°/s in the front-to-back direction. For clarity, neutral density filters and the LEDs pads used for background illumination are not shown in this picture. (**B**) In Dim, 12 out of 18 wild-type photoreceptors could resolve them, showing the response waveform depicted by the continuous line, with the larger trailing peak. While six other could not distinguish the two objects; showing the waveform of by the dotted line. In Bright, 5 out of 6 examined wild-type photoreceptors resolved the two dots. (**C**) Response waveform of $hdc^{JK910}$ R1-R6s exhibited two distinct peaks, with the leading one was the larger. In Dim, 15 out of 16 tested mutant photoreceptors could resolve the two objects and 14 of them displayed this waveform. In Bright, all of 8 recorded $hdc^{JK910}$ photoreceptor responses resolved the objects. D-values were calculated from the amplitude of the smaller peak and the dip in between. (**D**) In Dim, D-values of hdc$^{JK910}$ R1-R6 responses were significantly larger than their wild-type counterpart. $D_{wild-type}$ = 4.51 ± 0.67%, $D_{hdc}$ = 10.5 ± 1.35%, p = 0.00075, t-test, $n_{wild-type}$ = 12, $n_{hdc}$ = 15. (**E**) In Bright, the difference of D-values of the two photoreceptor groups was statistical insignificant. $D_{wild-type}$ = 9.81 ± 1.65%, $D_{hdc}$ = 16.71 ± 1.86%, p = 0.117, t-test, $n_{wild-type}$ = 5, $n_{hdc}$ = 8. (**F**). )

Changing from Dim to Bright condition, D-values of wild-type photoreceptors appeared to exhibit larger changes than those of mutant photoreceptors. However, the difference was not statistically significant due to the large cell-to-cell variation. $R_{C \text{ wild-type}} = 262 \pm 126\%$, $R_{C\ hdc} = 31 \pm 11\%$, p = 0.164, $n_{\text{wild-type}} = 4$, $n_{hdc} = 7$. (**D–F**) Mean ± SEM, two-tailed student test. DOI: https://doi.org/10.7554/eLife.26117.076

In the second type of experiment, two bright dots, which were less than the half-width of a R1-R6's receptive field (6.8°) apart, crossed its receptive field at 409 °/s (**Appendix 6— figure 6A**). Each tested photoreceptor's ability to distinguish the dots was assessed whether its response showed two clear peaks (**Appendix 6—figure 6B**, solid line) or only one (dotted line). Quite unexpectedly, even at the low room-temperature of 19°C, where phototransduction is slower than at the flies' preferred temperature of 25°C (**Sayeed and Benzer, 1996**; **Juusola and Hardie, 2001b**), 12/18 of wild-type R1-R6 photoreceptors and 15/16 $hdc^{JK910}$ R1-R6s could clearly resolve the two dots. (Note that at 25°C, every tested R1-R6 resolved them well; **Figure 9—figure supplement 1E–F**). Amongst the wild-type responses, the trailing peak was often larger than the leading one (**Appendix 6—figure 6B**, solid line), whereas all but one $hdc^{JK910}$ R1-R6 had the larger leading peak (**Appendix 6— figure 6C**). This observation suggests that excitatory synaptic feedback modulation, which $hdc^{JK910}$ photoreceptors lack, may enhance the second peak in the wild-type responses. Resolvability was further quantified by D-values:

$$D = \frac{d}{P}\%$$ (A6.13)

Where $P$ is the amplitude of the smaller peak and $d$ is the depth of the response dip between the two peaks (**Appendix 6—figure 6C**). In darkness, D-values measured from the mutant photoreceptors were, on average, more than double those of wild-type ($D_{\text{wild-type}} = 4.51 \pm 2.33\%$, $D_{hdc} = 10.5 \pm 5.23\%$), indicating that $hdc^{JK910}$ R1-R6s resolve the two points more clearly than their wild-type counterpart (**Appendix 6—figure 6D**). Under light-adaptation (ambient illumination), while R1-R6s of both genotypes exhibited significant improvements in their image resolution ($D_{\text{wild-type}} = 9.81 \pm 3.7\%$, $D_{hdc} = 14.85 \pm 6.95\%$), the difference between the two groups decreased and was at the margin of statistical significance (**Appendix 6—figure 6E**; p=0.058, t-test). Taking into account only the cells in which D-values were measured in both dim and bright conditions, the enhancement of D-values to the ambient light change was quantified by their relative change:

$$R_C = \frac{D_{\text{Bright}} - D_{\text{Dim}}}{D_{\text{Dim}}}\%$$ (A6.14)

On average, wild-type D-values improved by 262%, ranging from 43% to 604%. These changes appeared to be markedly larger than those observed in $hdc^{JK910}$ photoreceptors, which varied from 4% to 89% and averaged as 31%. Yet, the difference between the two groups was not statistically significant because of the large individual variations (**Appendix 6—figure 6F**).

In **Figure 7**, we further analyze the resolvability of those high-quality wild-type and $hdc^{JK910}$ R1-R6s, from which we recorded the impulse response and receptive field measurements at the two adapting backgrounds - dim and bright at 25°C, as well as responses to both 205 and 409 °/s moving dots. Such data allowed us to compare the classic theory to the real recordings even more thoroughly.

Together, these results show that the theoretical spatial half-width, $S$, grossly underestimates R1-R6 photoreceptors' image resolution. Recordings clarify that two bright dots that travel 409 °/s can, in fact, be resolved by a single photoreceptor, even when the dots (6.8° separation) are less than the photoreceptor's acceptance angle ($\Delta \rho = 9.5°$) apart. Therefore, a R1-R6 photoreceptor's real spatial half-width for the same high (saccadic) speed must be less than half of the theoretical estimate (~15°; **Appendix 6—figure 5B**). In other words, the classic theory overestimates the role of motion blur on *Drosophila* vision, as its R1-R6 photoreceptors resolve fast-moving dots beyond the predicted motion blur limit.

The recordings further indicate, consistent with $hdc^{JK910}$ R1-R6s' marginally narrower acceptance angles (Appendix 4, *Figure 7—figure supplement 1* and *2*), that their spatiotemporal resolution is somewhat better than that of wild-type photoreceptors, both in dim and moderately bright conditions. However, when ambient light intensity was changed, the spatiotemporal resolutions of wild-type R1-R6s improved more. Here, possible contributing factors include:

- Slight (~4%) differences in the photoreceptors' rhabdomere diameters (see Appendix 5)
- Dynamic and homeostatic regulation of $[Ca^{2+}]_i$, membrane properties and synaptic feedback (*Dau et al., 2016*)
- Intracellular pupil (see Appendix 2 and Appendix 4)
- Differences in photomechanical rhabdomere contractions (see Appendixes 7–8)
- Electrical coupling between the cells

Their potential roles are further discussed in Appendixes 7–8

## Modeling R1-R6 output by the Volterra series method

Volterra kernels of each photoreceptor model were computed from the first half (1.5 s) of GWN data (https://github.com/JuusolaLab/Microsaccadic_Sampling_Paper/tree/master/VolterraModelOfPhotoreceptor), before the other half of recorded light stimuli and voltage responses were used to validate the model. Because the output simulation accuracy depends upon input statistics and the model computation specifications, the system identification process was optimized by selecting suitable parameters.

Firstly, to test whether the selected 200 Hz input bandwidth was appropriate (*Appendix 6—figure 7A*), we analyzed the resulting signal-to-noise (SNR) ratio of photoreceptor output (*Appendix 6—figure 7B*). SNR decayed below 1 at around 66 Hz; at which point photoreceptor response contained more noise than signal. Thus, the GWN stimuli predictably activated a R1-R6 photoreceptor's whole frequency range.

Secondly, we assessed different sampling rates. According to the Nyquist-Shannon sampling theorem, a signal without frequencies higher than $B$ Hz can be perfectly sampled and reconstructed (*Shannon, 1948*) by sampling rate $F_s$ of $2B$ Hz. Because the bandwidth of interest was 0–66 Hz, the data could be processed, in theory, at any sampling rate from 132 Hz to the recorded rate of 10 kHz, without compromising its information content.

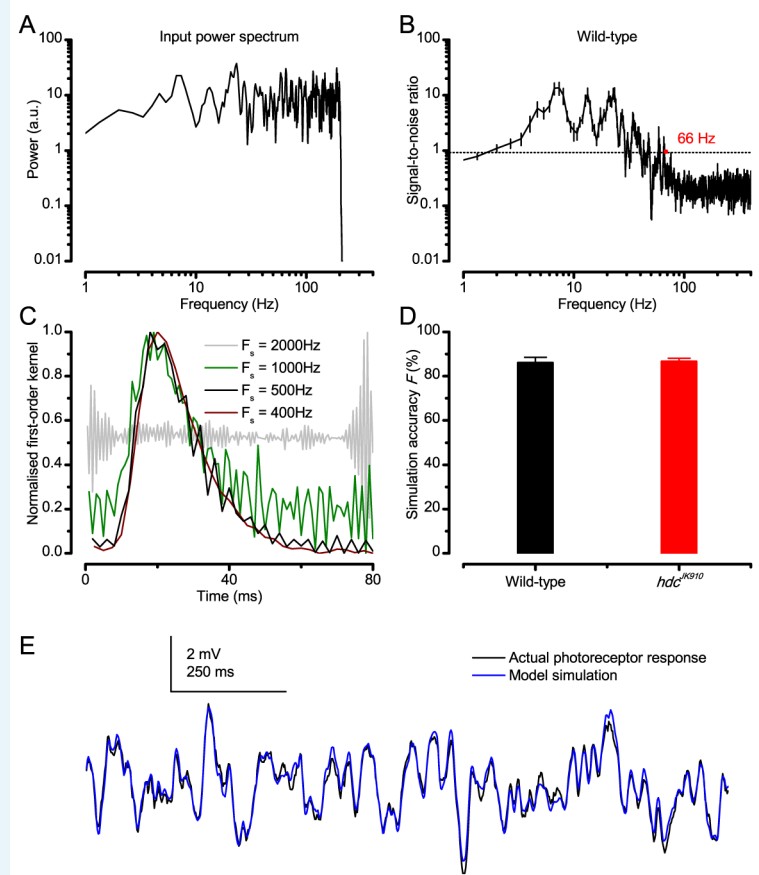

**Appendix 6—figure 7.** Predicting responses to Gaussian white noise (GWN). (**A**) GWN light stimulus power spectrum with 200 Hz cut-off frequency. (**B**) Signal-to-noise ratio (*SNR*) of a wild-type R1-R6 photoreceptor's voltage response at 19°C. Here, because the mean stimulus intensity was kept well within the subsaturating range (100-times lower than in the experiments of the main paper), having 200 Hz bandwidth, $SNR_{max}$ of the photoreceptor output was ~ 20 (consistent with high-quality recordings for this specific stimulus condition) (*Juusola and Hardie, 2001a*). Noise power exceeded signal power at around 66 Hz. *$hdc^{JK910}$* R1-R6 outputs exhibited similar characteristics (data not shown for clarity). (**C**) Examples of kernels computed at different sampling rates from the same raw data. (**D**) Accuracy of GWN response simulation by Volterra series models for wild-type and *$hdc^{JK910}$* R1-R6s. $F_{wild-type}$ = 86 ± 2.5%, $F_{hdc}$ = 86.6 ± 1.6%. (**E**) Simulations of *Drosophila* R1-R6 responses to GWN stimuli matched the actual data closely. (**B, D**) Mean ± SEM, $n_{wild-type}$ = 9, $n_{hdc}$ = 8.

DOI: https://doi.org/10.7554/eLife.26117.077

We found that higher sampling rates yielded models, which predicted R1-R6 output with slightly higher accuracy. However, their kernels also exhibited larger fluctuations, and the kernels did not decay to zero over time, most likely due to high-frequency noise. For $F_s$ = 1,000 Hz and higher, such fluctuations undermined the physiological meaning of the Volterra first-order kernel (*Appendix 6—figure 7C*), which is the photoreceptor's impulse response (*Victor, 1992*). Thus, the kernels computed from too richly-sampled data would be useful only for response prediction to this particular GWN stimulus.

On the other hand, while computations performed with lower $F_s$ data would produce smoother kernels, a low sampling rate would also limit the model's other applications. For example, Volterra series models were used to simulate photoreceptor response to the image of moving objects created by the 25 light-point array. For an object moving at 409 °/s, its travelling time across the array was 100 ms, or 4 ms per light-point. As the simulation required at least two data-points per light-point, $F_s$ was chosen to be 500 Hz, at which rate reasonably smooth kernels could still be produced (*Appendix 6—figure 7C*). Moreover,

because the first-order kernel values decayed to zero at 50–60 ms, it was deemed that a 80 ms kernel length was sufficient for the computations.

Volterra series models, computed from data sampled at 500 Hz, could consistently predict response of *Drosophila* photoreceptors to GWN stimuli (for example, see *Appendix 6—figure 7E*). On average, the model simulation accuracy, given by *Equation A6.6*, was ~86% for both wild-type and *hdc^{JK910}* photoreceptors (*Appendix 6—figure 7D*). These high *F* values confirmed that a linear Volterra series model could approximate light-adapted *Drosophila* photoreceptor output to the test stimulation appropriately.

## Classic theory underestimates how well R1-R6s resolve fast moving dots

By approximating the light input directly from the receptive field measurements (*Appendix 6—figure 1*) and the corresponding R1-R6 output by Volterra series (*Appendix 6—figure 7C*), we could estimate each tested *Drosophila* photoreceptor's responses to moving bright dots. This was done by convolving the extrapolated light stimuli with the corresponding impulse responses.

The model predictions for a single moving stimulus were far less consistent than those for GWN stimuli. *Appendix 6—figure 8A,B* show representative simulations with broadly acceptable and clearly unacceptable accuracies, respectively, together with the corresponding intracellular recordings.

From both the recordings and simulations, we further calculated the theoretical dot motion effects on the neural image resolution, or spatial half-width, *S* (*Appendix 6—figure 8C*). As explained above (*cf. Appendix 6—figure 5*), the classic theory can only broadly suggest the relative differences between wild-type and mutant performances. Here, its application to the simulations further underestimated the spatial half-width predictions of wild-type recordings and over-estimated those of the *hdc* mutant.

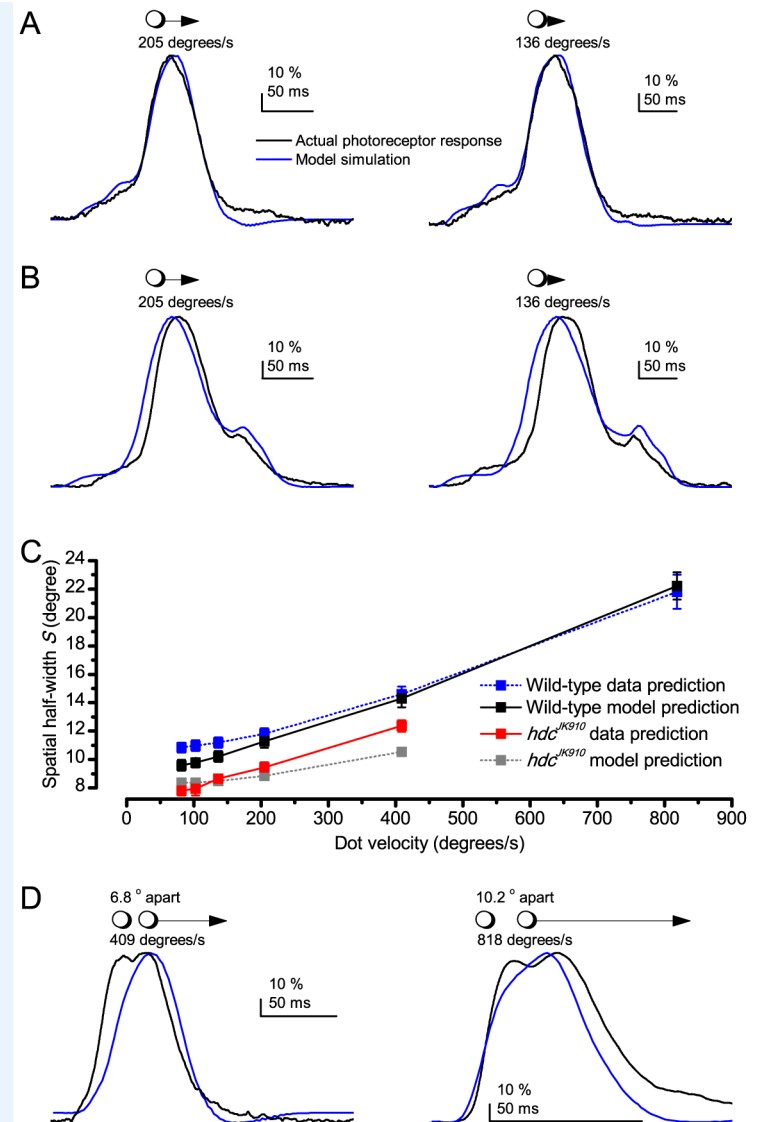

**Appendix 6—figure 8.** Prediction accuracy of the Volterra series photoreceptor models to moving point-objects varies considerably. (**A**) Two examples of model simulations, which were reasonably close to the actual intracellular recordings to the tested dot motion. (**B**) Two examples of simulations that clearly differed from the recordings. (**C**) Theoretical predictions of photoreceptor output spatial half-width, calculated from the recordings and simulations as a function of the point-object speed. Mean ± SEM, $n_{wild-type}$ = 9, $n_{hdc}$ = 8. The theoretical spatial half-widths of the simulations differ from those of the recordings. *E.g.* the wild-type recordings (black) predicted consistently narrower S than the corresponding simulations (blue). The predicted resolvability of the resulting neural image, or spatial half-width (**S**), was consistently lower for the simulations than for the recordings. (**D**) Crucially, Volterra series models failed to predict how well the actual photoreceptor output resolves two close objects moving together very fast (shown for 409 and 818 °/s). For the 818 °/s prediction, we used here the fastest impulse response, recorded from another cell, but even so, the model still could not resolve the two dots. Thus, the actual spatial half-width of R1-R6s, limiting *Drosophila*'s resolving power at high image velocities, is about half of that estimated in (**C**). (Note, the dynamic biophysical mechanisms causing this difference – both in light input and photoreceptor output - are explained in Appendix 8. Recordings and simulations were at 19°C.

DOI: https://doi.org/10.7554/eLife.26117.078

Most critically, however, Volterra series models consistently failed to predict the fast phasic components of the recorded voltage responses, and thus their real resolvability, to two fast moving dots (*Appendix 6—figure 8D*; see also *Figure 7*). The model simulations, and hence its underlying classic theory, always predicted lower resolvability than what we saw in the actual recordings. Further investigations (*Appendixes 7–8*) revealed that this discrepancy reflected the missing biophysical mechanisms of the empirical black-box models (see Appendix 1). Specifically, the used photoreceptor models lacked: (**i**) the photomechanical rhabdomere movements, which shift and narrow a R1-R6's receptive fields, and (**ii**) the refractory sampling, which allows many microvilli (after the first dot) to recover from refractoriness (*Song et al., 2012*; *Song and Juusola, 2014*; *Juusola et al., 2015*) to respond to the second dot. Moreover, by tuning and validating the Volterra kernels to GWN stimuli at relative steady-state, we inadvertently limited its exposure to (**iii**) excitatory dynamic synaptic feedback modulation (*Zheng et al., 2006*; *Dau et al., 2016*), which accentuates sudden changes in photoreceptor output. For these reasons, the responses of real R1-R6s, which naturally utilize the given mechanisms, showed systematically larger widths and second-peaks, providing higher resolvability.

We show in *Figure 8G–I*, *Figure 8—figure supplement 1* and *Appendixes 7–8* that a new biophysically realistic microsaccadic sampling model, which allows for realistic refractory quantal phototransduction (*Song and Juusola, 2014*) and photomechanical rhabdomere contractions (*Hardie and Franze, 2012*), yields (significantly better) theoretical predictions that closely approximate the real R1-R6 output. Importantly, in Appendix 9, we further show that R1-R6 output to two *dark* dots, moving at saccadic speeds, has the same relative resolution as their output to the corresponding two *bright* dots. Collectively, our results demonstrate that *Drosophila* photoreceptors resolve fast moving objects far better than what was believed previously.

## Conclusions

In this appendix, we used intracellular recordings and classic theoretical approaches to study how fly photoreceptors encode moving bright dots. Model simulations about each tested R1-R6s' spatiotemporal responses were compared to the actual recordings to the same stimuli. We found that both wild-type and $hdc^{JK910}$ photoreceptors resolved moving dots nearly equally well, and significantly better than the corresponding deterministic simulations. These findings demonstrate that the classic dynamic photoreceptor models (*Srinivasan and Bernard, 1975*; *Juusola and French, 1997*; *Land, 1997*), which lack knowledge about the underlying phototransduction biophysics and photomechanics, grossly underestimate the spatiotemporal resolution of the real cells.

Animals counter self-motion blur effects by moving their eyes. This compensates for head and body movements by keeping the neural image position near stationary as long as possible (*Land, 1999*). Interestingly here, a fly photoreceptor's response to two moving point-objects represents an opposite case where image motion, in fact, improves acuity. In the classic theory, to resolve two stationary objects, at least three photoreceptors are required so that the intensity difference in between can be detected. Because *Drosophila* photoreceptors' interommatidial angles (*Land, 1997*; *Gonzalez-Bellido et al., 2011*) vary from 3.4° to 9.0° and their average acceptance angle is ~9.5° (Appendix 4), its eye should not resolve two point-objects 6.8° apart. Nevertheless, responses of single photoreceptors, during even very fast (saccadic-speed) (*Geurten et al., 2014*) movements, show large enough dips (in their temporal dynamics) to indicate that the objects are resolved neurally. In the classic theory instead, the outputs of several adjacent photoreceptors had to be processed together to distinguish two moving point-objects from one stationary object, in which brightness changes over time. This example highlights the inseparability of spatiotemporal information processing and acuity.

The unique advantages of the present study were the bespoke equipment and stimulus paradigm. These allowed high-quality photoreceptor recordings with precisely controlled moving point-objects stimulation. Therefore, we could directly test and compare the

theoretically predicted relationship between the neural image resolution and the object speed (*Srinivasan and Bernard, 1975*; *Juusola and French, 1997*) to the experimental data. However, the equipment also had limitations to be improved in future research. Wider object speed range is necessary, especially for testing insect eyes with fast responses. Owing to the long transient time, each light-point now required 2 ms switching period. Consequently, the minimum travelling time was 50 ms and the object speed limit in *Calliphora* experiments was 334 °/s, which is far slower than observed during the flies' saccadic flight behaviors (*Schilstra and Hateren, 1999*; *Hateren and Schilstra, 1999*) (2,000–4,000 °/s). Whilst positioning the light-point array closer to the fly eye would increase object angular speed, it would compromise resolution as fewer light-points would then lie within a tested cell's receptive field.

In Appendixes 7–9, we show how both the enhanced resolvability of moving point-objects and the phasic modulation of their rising and decaying phases, as was shown here, emerge from the joint contributions of photomechanical rhabdomere contraction and its refractory information sampling.

## Appendix 7

DOI: https://doi.org/10.7554/eLife.26117.079

# Photomechanical microsaccades move photoreceptors' receptive fields

## Overview

This appendix describes a new powerful high-speed video recording method to measure photomechanical rhabdomere movements *in situ*, and provides important experimental and theoretical background information for the results presented in *Figures 8–9*.

In this appendix:

- We utilize the optical cornea-neutralization technique (*Franceschini and Kirschfeld, 1971b*; *Franceschini and Kirschfeld, 1971*) with antidromic deep-red (740 or 785 nm peak) illumination to observe deep pseudopupils (photoreceptor rhabdomeres that align with the observer's viewing axis) in the *Drosophila* eye. We use an ultra-sensitive high-speed camera with a purpose-built microscope system to record fast rhabdomere movements across the compound eyes, while delivering blue-green stimuli (470 + 535 nm peaks) orthodromically into the eye.

- We show that light-activation moves rhabdomeres (*Video 3*) side-ways (horizontally) both in dark- and light-adapted eyes. This movement starts after a 8–20 ms delay from the light stimulus onset, and reaches its peak in about 70–150 ms. Because these movements have fast onset and light intensity-dependency, which are similar to those of the R1-R6 photoreceptors' intracellular voltage responses to comparable stimuli, they must result from individual photoreceptors' photomechanical contractions; see (*Hardie and Franze, 2012*).

- We show that *trp/trpl*-mutant photoreceptors, which have normal phototransduction reactions but lack the light-gated ion channels, also contract to light. Since these photoreceptors cannot produce electrical responses and thus communicate electrically or synaptically with other cells, including eye muscles, their *contractility cannot be caused by eye muscle activity but must be intrinsic*, supporting the earlier hypothesis (of phototransduction reactants interacting locally with the plasma membrane) (*Hardie and Franze, 2012*).

- We show that light moves rhabdomeres fast in the back-to-front direction, while darkness returns them back to their original positions slower. Because the ommatidium lens inverts images, R1-R8 photoreceptors' receptive fields move in the opposite direction - front-to-back after light and back-to-front after darkness. Therefore, when front-to-back moving bright dots cross the eyes, the photoreceptors' receptive fields move along. But when bright dots cross the eyes in the back-to-front direction, the photoreceptors' receptive fields move against them (*cf.* Appendix 6).

- At the level of rhabdomere tips, the horizontal movements can be up to 1.4 μm, as measured occasionally in light-adapted eyes. Therefore, given the known optical dimensions, these photomechanical microsaccades can rapidly shift R1-R6 photoreceptors' receptive fields by ~5°. Remarkably, such a large image pixel displacement reaches the average interommatidial angle, $\Delta\varphi$ ~4.5-5°, in the *Drosophila* eye (*cf.* Appendix 4; *Appendix 4—figure 1*).

- We show that the light stimulus also contracts rhabdomeres axially (*Video 2*; inwardly: 0.5–1.7 μm), down away from the lens. This transient increase in focal length should contribute in narrowing R1-R6's receptive fields dynamically. We further show that specific cone- and pigment-cells inside each ommatidium form an aperture, which is connected to the rhabdomere tips. During light stimulation, this aperture moves laterally with the rhabdomeres but only half as much (*Video 4*). And since the ommatidium lens remains practically immobile, the light beam falling upon the rhabdomeres is shaped dynamically. These observations mean that a R1-R6's receptive field must both move and narrow during dynamic light stimulation.

- We show that rhabdomeres of $hdc^{JK910}$ histamine-mutant (*Burg et al., 1993*; *Melzig et al., 1996*; *Melzig et al., 1998*) R1-R6 photoreceptors, in which visual interneurons are blind (receive no neurotransmitter from photoreceptors), have broadly wild-type-like contraction

dynamics, again refuting the role of eye muscle activity in the data. But interestingly, their light-sensitivity is about 10-fold reduced, similar to their voltage responses (*Dau et al., 2016*). In part, this may reflect $hdc^{JK910}$ photoreceptors' smaller size. Given that $hdc^{JK910}$ rhabdomere diameters are ~4% smaller than in wild-type (Appendix 5), their length should also be reduced in the same proportion. As the average wild-type R1-R6 is ~100 μm, $hdc^{JK910}$ R1-R6s should be ~4 μm shorter. And indeed we find in situ that $hdc^{JK910}$ rhabdomere tips are ~4 μm further away from the lens than the wild-type tips. In addition, the higher $[Ca^{2+}]_i$, caused by tonic excitatory synaptic feedback overload (*Dau et al., 2016*), may further affect their mobility, possibly retaining them in a slightly more contracted state.

## Rapid adaptation caused by light–induced R1-R8 contractions

Atomic force microscopy (AFM) at the dissected *Drosophila* eyes' corneal surface (*Hardie and Franze, 2012*) has shown up to 275 nm radial movements to brief light pulses, caused by transient *photomechanical* R1-R8 photoreceptor contractions. Such movements are too small and fast to see with the naked eye, and were initially considered: (i) only to participate in gating photoreceptor's transduction-channels, and (ii) possibly too small to affect fly vision in general. In this appendix, we use high-speed video microscopy to show that *in vivo* the underlying photomechanical rhabdomere (light sensor) movements are larger both laterally (horizontally: 0.3–1.4 μm) and axially (inwardly: 0.5–1.7 μm). Because these movements are also synchronous, ubiquitous, robust and reproducible, they influence how the fly eyes sample visual information about the world.

## High-speed video recordings of light-induced rhabdomere movements

Dark-adapted dissociated photoreceptors rapidly contract to light (*Hardie and Franze, 2012*) (*Video 2*). It has been suggested that this contraction results from light-induced phosphatidylinositol 4,5-bisphosphate ($PIP_2$) cleaving, which modulates their rhabdomere membrane volume and so participates in gating the phototransduction-channels (*trp* and *trpl*) (*Hardie and Juusola, 2015*).

Here, we directly test the hypotheses that (i) the photomechanical photoreceptor contractions occur also in intact flies in normal stimulus conditions, and (ii) these movements serve the purpose of modulating light input to photoreceptors and thus photoreceptor output. We do this by recording high-speed video of how *Drosophila* photoreceptor rhabdomeres move to different light stimuli *in vivo*, and by analyzing and characterizing how these movements affect R1-R6s' receptive fields. Later on, in Appendix 8, we include their light input parameter changes in biophysically-realistic mathematical models to predict R1-R6 voltage output to moving visual stimuli.

## Imaging setup for recording photomechanical rhabdomere contractions

We used the optical cornea-neutralization method to monitor how light stimuli evoke *Drosophila* photoreceptor rhabdomere movements. The imaging system was constructed upon an upright microscope (Olympus BX51), secured to a XY-micrometer stage on an anti-vibration table (MellesGriot, UK) (*Appendix 7—figure 1*). To minimize light pollution in the recordings, the system was light-shielded inside a black Faraday cage with black lightproof curtains in the front, and the experiments were performed in a dark room. For collecting and recording deep pseudopupil images, the system was equipped with a 40x water immersion objective (Zeiss C Achroplan NIR 40x/0.8 w, ∞/0.17, Germany) and an ultra-sensitive high-speed camera (Andor Zyla, UK), respectively.

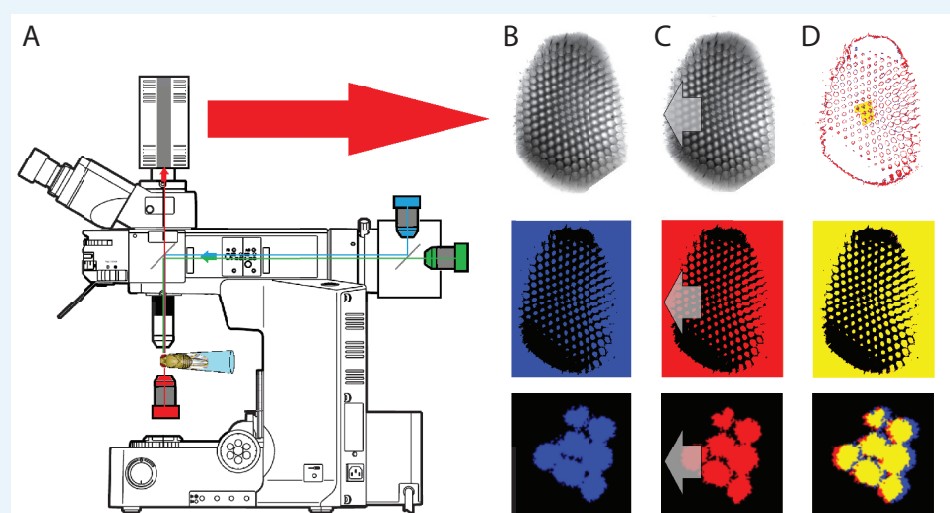

**Appendix 7—figure 1.** Microscope system for high-speed video recording of light-induced photoreceptor movements. (**A**) High-speed camera (Andor Zyla, UK) recorded images of deep pseudopupils in the eye of an intact living *Drosophila* under deep-red antidromic illumination (here 740 nm LED + 720 nm long-pass edge filter underneath the fly head). Each studied fly was immobilized inside a pipette tip. (**B**) A 10 ms blue-green light flash, delivered through the microscope system (orthodromically) into the left fly eye (above), was used to excite R1-R8 photoreceptors; the inset below shows R1-R7 rhabdomeres (blue) of one ommatidium just before the flash. (**C**) Light caused the rhabdomeres to twitch photomechanically in back-to-front direction (arrows) after 8–16 ms delay, with the photoreceptors being maximally displaced in ~ 100 ms from the stimulus onset. Invariably, this was seen as a sudden jump in the recorded rhabdomere position (red). (**D**) The difference in the rhabdomere position (displacement) before and after the flash, depended upon the light intensity, ranging between 0.3–1.4 μm; note a typical R1-R6 rhabdomere diameter is about 1.7 μm (Appendix 5). The frame subtraction (before and after the light flash) indicates that only the rhabdomeres that aligned directly with the blue/green light source moved (within the seven central ommatidia; yellow area), while the rest of the eye remained immobile. Accordingly, the difference image shows little ommatidial walls, as these and other immobile eye structures became mostly subtracted away. In contrast, eye muscle activity, which is every so often seen with this preparation (Appendix 4, *Appendix 4—figure 6*) occurs more gradually and moves all the eye structures together.

DOI: https://doi.org/10.7554/eLife.26117.080

A *Drosophila* was gently fastened to an enlarged fine-end of a 1 ml pipette tip by puffing air from a 100 ml syringe at the large end until the fly head and ~1/5 of the thorax emerged outside (*Appendix 7—figure 1A*). The head and thorax were carefully fixed (from the proboscis and cuticle) to the pipette wall in a preferred orientation by melted beeswax, without touching the eyes. The fly was then positioned with a remote-controlled XYZ-fine resolution micromanipulator (Sensapex, Finland) underneath the water immersion objective, using both visual inspection and live video stream on a computer monitor.

Antidromic illumination (through the fly head) revealed the deep pseudopupils of the fly eyes. It was provided with a high-power deep-red light source (740 nm LED with 720 nm high-pass edge-filter; or 785 nm LED with ±10 nm bandpass filter), driven by a linear current LED driver (Cairn OptoLED, UK). Note that very bright deep-red illumination, which is a prerequisite for good signal-to-noise ratio high-speed video imaging, activates R1-R8 photoreceptors only marginally. This is because their different rhodopsins' absorbance maxima are at much lower wavelengths (*Britt et al., 1993*; *Wardill et al., 2012*). The photoreceptors' near insensitivity to >720 nm red light was confirmed *in vivo* by ERG recordings (*Appendix 7—figure 2*).

Orthodromic light stimulation (through the 40x objective into the eye), which evoked the photoreceptor contractions, was delivered by two high-power LEDs: 470 nm (blue) and 545 nm (green), each separately controlled by its own driver (Cairn OptoLED, UK). These peak wavelengths were selected to activate R1-R6s' rhodopsin (Rh1) and its meta-form near maximally, and so through joint stimulation to minimize desensitization by prolonged depolarizing after-potentials (PDA) (*Minke, 2012*). Simultaneous stimuli from the two LEDs were merged into one focused beam by a 495 nm dichroic mirror and low-pass-filtered at 590 nm. Pseudopupil signals of the observed fly eye (left or right) were split spectrally by another dichroic mirror (600 nm), and essentially only red image intensity information ($\geq$600 nm) was picked up by the high-speed camera.

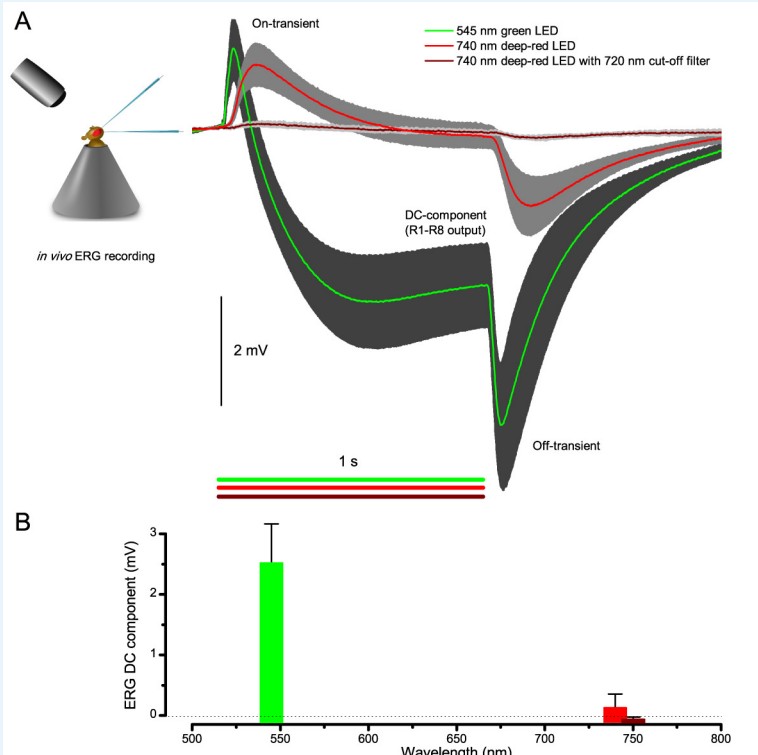

**Appendix 7—figure 2.** Testing R1-R8 photoreceptors sensitivity to deep-red illumination by electroretinogram (ERG) recordings. (**A**) ERGs were recorded in intact *Drosophila*, placed inside a conical holder, to 1 s long very bright green (545 nm), red (740 nm) and deep-red (740 nm LED with 720 nm high-performance long-pass filter) pulses. A normal ERG contains a large but slow photoreceptor (DC) component and the faster on- and off-transients (*Heisenberg, 1971*), which signal histaminergic transmission to lamina interneurons. (**B**) The very small ERG response (wine) to the deep-red pulse indicated very little R1-R8 photoreceptor activation. Mean ± SD shown, n = 6–7 wild-type flies.

DOI: https://doi.org/10.7554/eLife.26117.081

## Recording procedures

Light stimulus generation was performed by a custom-written Matlab (MathWorks, USA) program (Biosyst; M. Juusola, 1997–2015) (*Juusola and Hardie, 2001a*; *Juusola and de Polavieja, 2003*) with an interface package for National Instruments (USA) boards (MATDAQ; H. P. C. Robinson, 1997–2008). The length of the light stimuli (including the continuous deep-red background and the blue/green stimulus patterns) was made to match the number of frames to be acquired by the Andor camera, and the stimuli were externally triggered by the camera software (Solis). During the recordings, the frames were first buffered in the RAM in high-speed and then transferred on the computer's hard drive. Light stimulus intensity could

be attenuated by a neutral density filter set (Thorn Labs, USA), covering a 5.3 log intensity unit range.

## Key observations from unprocessed high-speed footage

High-speed video microscopy (*Appendix 7—figure 1* e.g. *Video 3*) from intact wild-type *Drosophila* eyes (n >> 100 flies) showed repeatedly and unequivocally that:

- Full-field light flashes evoked rapid local R1-R7 rhabdomere movements within those seven ommatidia, which at the center of the imaged view, faced the blue/green stimulus source directly (*Appendix 7—figure 3A*, orange area). Rhabdomeres in few other neighboring ommatidia also moved marginally (yellow area), but not obviously in other ommatidia. This meant that only the ommatidia that aligned with the blue/green stimulation absorbed the incident light, while those to one side reflected it. This local area, which showed photomechanical rhabdomere movements, closely matched *Drosophila*' normal pseudopupil (*Appendix 7—figure 3B*).
- The rhabdomere movement was in the back-to-front direction (*Appendix 7—figure 1B–D*), whilst in darkness, the rhabdomeres returned in front-to-back to their original positions more slowly. These dynamics and their directions were similar in both the left and right eye.
- Because light always moved the rhabdomeres back-to-front, the corresponding neural images of the left and the right eye comprise left-right mirror symmetry; *i.e.* against the vertical (sagittal) plane, the rhabdomeres in the left and right eye display mirror symmetric motion. We show later in Appendix 8 how this symmetry may allow *Drosophila* photoreceptors to encode orientation information during saccades or image rotation.
- The rhabdomere movement directions seemed homogeneous (at least in the first approximation) across each tested eye, appearing similar in its different regions: whether measured at its up, down, front or back ommatidia. Such 'pixel interlocking' across the whole eye's visual field may help to preserve, or enhance, the neural images' spatial resolution of the world.

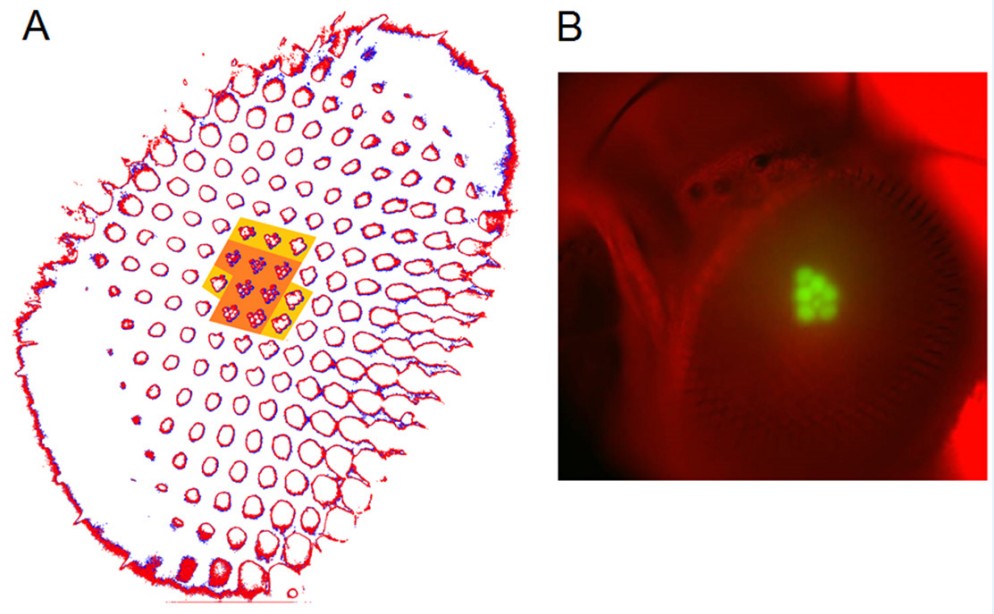

**Appendix 7—figure 3.** Photomechanical rhabdomere movements were localized inside those seven ommatidia that form the normal pseudopupil. (**A**) High-speed video recordings in the *Drosophila* eye showed clear rhabdomere displacement before (marked blue) and after (marked red) a blue/green flash only within seven ommatidia (orange area). This was revealed by subtracting the corresponding frame contours. The rhabdomeres of these seven ommatidia aligned directly with the blue/green stimulus, which was carefully centered above in the microscope port (*Appendix 7—figure 1*). Marginal rhabdomere movements were further

detected in six other neighboring ommatidia (yellow area). These results meant that only the rhabdomeres that faced the centered Orthodromic blue/green stimulus absorbed its light and contracted, while the rest of the eye reflected this stimulus and remained immobile. Note that this local rhabdomere activation pattern was restricted by the same eye design principle that causes the insect eye pseudopupil. (**B**) The *Drosophila* eye, in which photoreceptors were made to express green-fluorescence, displayed a green pseudopupil only form those seven ommatidia that directly faced the observer (and the blue light source through the microscope lenses). This happened because these ommatidia (their rhabdomeres) both absorbed the incident blue light and their GFP-molecules released green light back to the observer's eye/camera, while the other ommatidia around reflected the blue light.

DOI: https://doi.org/10.7554/eLife.26117.082

## Image analysis

To accurately quantify the size and direction of the observed rhabdomere movements in time we devised specific image-analysis procedures. First, the stored images (raw data frames of each recording) in the hard drive were exported as a tiff stack, in which pixel intensity range was set by the frames' minimum and maximum values, using the camera software (Andor SOLIS). ImageJ software was then used to convert the tiff-stack into a single tiff-file. The Matlab scripts to process and analyze the images are downloadable from the repository: (https://github.com/JuusolaLab/Microsaccadic_Sampling_Paper/tree/master/AnalyzeRhabdomereMovement). These methods included (***Appendix 7—figure 4A***):

i.    Loading the image stack.
ii.   Subtracting the median and mean from each frame and setting its negative values to (0. 0) to remove the dark noise background. This process was repeated for every frame.
iii.  Calculating 2D cross-correlation between each frame and the reference frame.
iv.   iv. Selecting the cross-correlation values, which were ≥95% of the maximum (peak) value. This was repeated for every frame.
v.    Calculating the weighted average position of the peak by using all the positions of the previous selection and using the cross correlation values as weights both in x- and y-direction. This was repeated for every frame.
vi.   Subtracting the reference frame position from every frame.

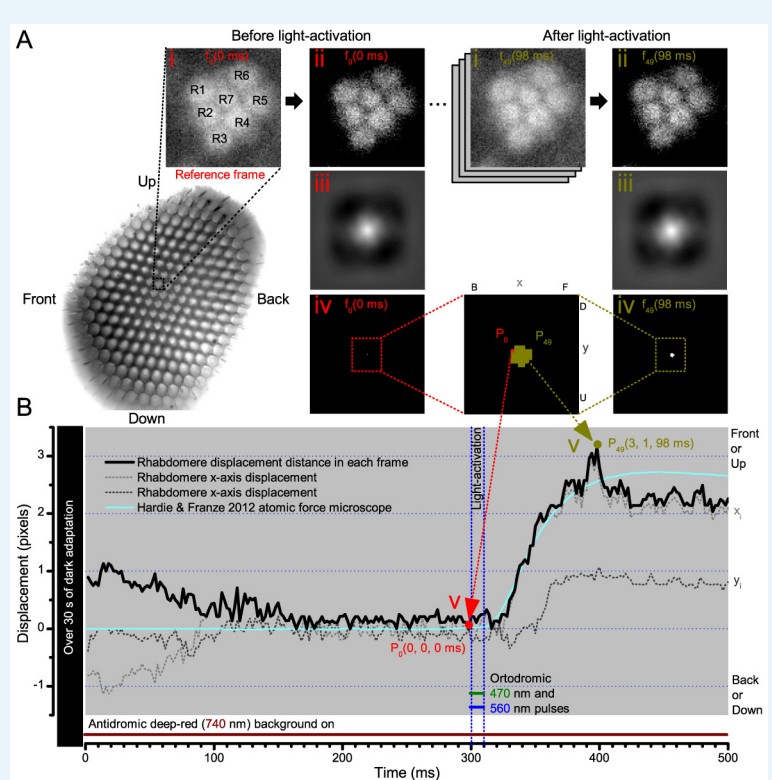

**Appendix 7—figure 4.** Cross-correlation image analysis to estimate photomechanical R1-R7 rhabdomere movements. (**A**) Analytical steps are shown for the reference frame at time zero ($f_0$), 2 ms before the 10 blue/green light stimulus pulse (red), and for the frame at the maximum rhabdomere displacement ($f_{49}$), 98 ms after (dark yellow). High-speed camera images of rhabdomeres were recorded using 750 nm red light. (i) Image stacks were uploaded, and (ii) the median of each frame was subtracted to remove its noise background. (iii) 2D cross-correlation was calculated for each frame, and (iv) the values within 5% of their peak value were selected. (v). ) The weighted mean peak positions gave each frame's x- and y-positions at its specific time point, and their distance, sqrt($x^2+y^2$), the total rhabdomere displacement (in pixels) against the reference frame position. Notice that the 2D cross-correlation images have flipped x- and y-axis directions (up, U, appears down, D; front, F, appears back, (**B**). (**B**) The resulting rhabdomere displacement distance and the corresponding x- and y-positions are plotted for each frame in time at 2 ms resolution (500 frames/s), against the reference frame position, $P_0(0, 0, 0)$. A comparable (inverted) atomic force microscopy data (cyan) closely matches the rise-time dynamic of the cross-correlation rhabdomere displacement estimate, validating our analytical approach. The analysis also implies that well dark-adapted photoreceptors may respond weakly to deep-red (740 nm) light onset (black trace 0–100 ms). Note R8 rhabdomere, which lies directly below R7, likely contracts too.

DOI: https://doi.org/10.7554/eLife.26117.083

## Quantifying rhabdomere travels and their receptive field shifts

In this study, the rhabdomere displacement measurements are given in microns (μm) and the resulting receptive field movements in degrees (°). *Appendix 7—figure 5A* shows a whole image of a *Drosophila*'s left eye, as focused upon its rhabdomeres in the center and magnified by the microscope system to use the camera's full 2048 × 2,048 pixel range. By placing a high-resolution μm-graticule on the same focal plane (*Appendix 7—figure 5B*), we calibrated that the whole image is 303 × 303 μm. After converting the recorded rhabdomere displacements from pixels (*Appendix 7—figure 4*) into microns, we then used the published parameters (*Stavenga, 2003b*) about the *Drosophila* ommatidium optics (*Appendix 7—figure*

5C) to translate these measurements into corresponding receptive field movements in degrees.

*Drosophila* ommatidium optical parameters were described by (*Stavenga, 2003b*) and (*Gonzalez-Bellido et al., 2011*). Its biconvex facet lens focuses light to a rhabdomere (grey rectangle) tip. The outer and inner lens curvatures, $r_1 = -r_2$, are 11 µm, and its thickness, $l_1$, is 8 µm. Distance from lens to the rhabdomere, $l_2$, is 15 µm. Reflective indices, $n$, for the object space, lens and image space, respectively, are: $n_1 = 1$, $n_2 = 1.45$ and $n_3 = 1.34$.

We used standard ray transfer matrix analysis (*Laufer, 1996*) to determine optical properties between the lens surface, $P_1$, and the rhabdomere tip, $P_2$. Both of these are represented as vectors of their positions, **y**, and angles, $\theta$: $P_1 = \begin{bmatrix} y_1 \\ \theta_1 \end{bmatrix}$ and $P_2 = \begin{bmatrix} y_2 \\ \theta_2 \end{bmatrix}$. Then, the optical system of the facet lens follows equation $P_2 = M\, P_1$, where

$$M = \begin{bmatrix} 0.23 & 1.61 * 10^{-5} \\ -3.63 * 10^{4} & 0.71 \end{bmatrix},$$ obtained from the ray transfer matrix analysis.

The transform matrix clarifies that the distance, $y_2$ (at $P_2$), mostly depends upon the angle $\theta_1$ (of $P_1$). Thus, 1 µm movement gives $1 \times 10^{-6}/1.61 \times 10^{-5} = 0.0621$ (rad) angular change, which is 3.56 °/µm. This movement is an inverse of the visual field movement. Note that by using the comparable optical parameter values of (*Gonzalez-Bellido et al., 2011*) (and considering the normal lens f-value variation across the *Drosophila* ommatidia), gives practically the same movement ratio (±5% error).

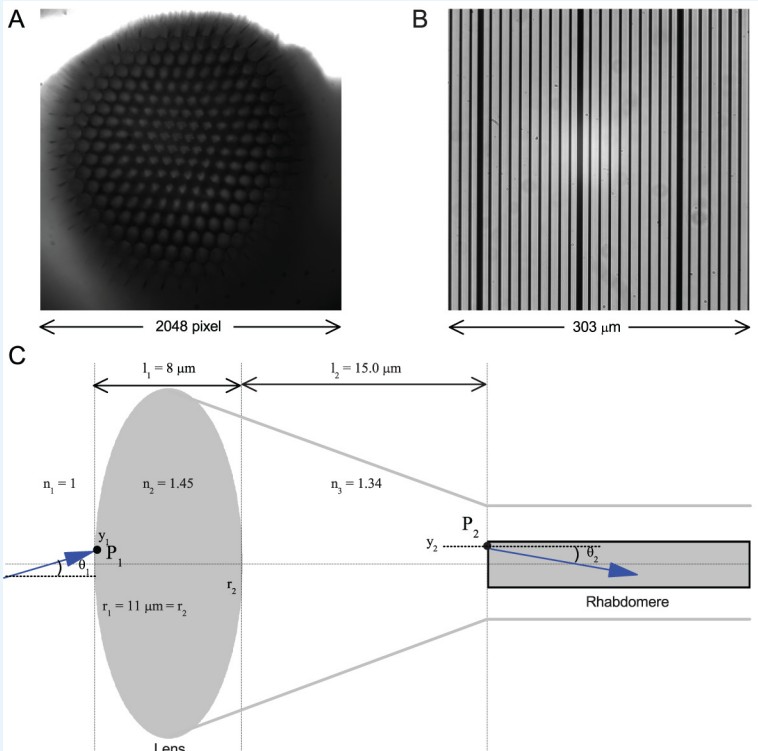

**Appendix 7—figure 5.** Calibrating the rhabdomere displacements in microns and their receptive field movements in degrees. (**A**) A whole image of a *Drosophila*'s left eye, the camera chip's full 2048 × 2,048 pixel range. (**B**) a high-resolution graticule placed at the same focal plane as the image (**A**) gives the full image size of 303 × 303 µm. Thus one pixel ~ 0.1479 µm. (**C**) A schematic of the main optical components in a normal *Drosophila* ommatidium. Its optical properties indicate that a 1 µm rhabdomere displacement shifts its receptive field by 3.56°.
DOI: https://doi.org/10.7554/eLife.26117.084

### Trp/trpl-mutants confirm the contractions' photomechanical origin

We then tested whether the rhabdomere contractions were generated by the photoreceptors themselves (photomechanically) or by eye muscle activity. This was done by recording in *trp/trpl* null-mutants, which express normal phototransduction reactants but lack completely their light-gated ion channels. Consequently, these photoreceptors did not generate electrical responses to light, and their eyes showed no ERG signal (*Appendix 7—figure 6A*). Nonetheless, high-speed video recordings revealed that *trp/trpl*-mutant photoreceptors contracted photomechanically (*Appendix 7—figure 6B*; see also *Video 2*). These observations are consistent with the hypothesis of the light-induced phosphatidylinositol 4,5-bisphosphate (PIP$_2$) cleaving from the microvillar photoreceptor plasma membrane causing the rhabdomere contractions (*Hardie and Franze, 2012*).

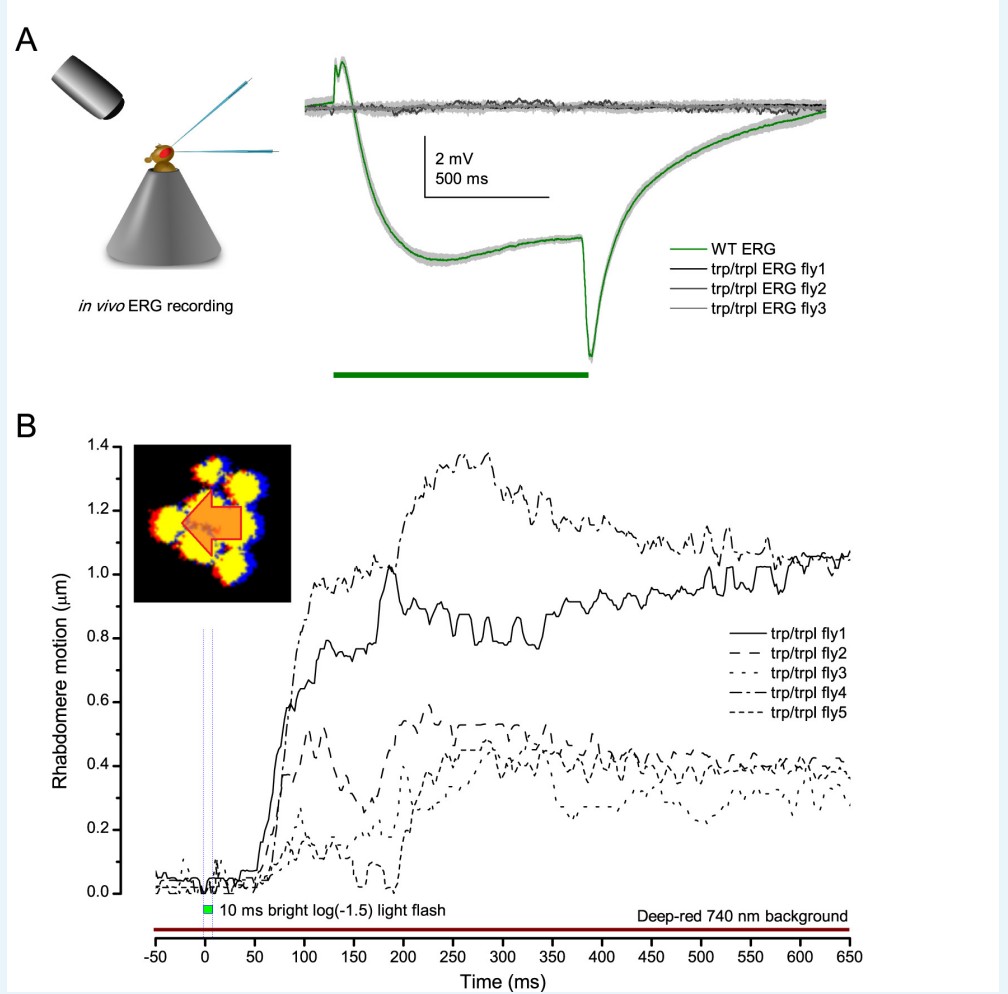

**Appendix 7—figure 6.** Photoreceptors of blind *trp/trpl* null-mutants flies show photomechanical contractions. (**A**) ERGs of *trpl/trpl* mutants show no electrical activity indicating that these flies are profoundly blind. (**B**) High-speed video recordings at their rhabdomeres show light-induced lateral movements, indicating that (i) these photoreceptors contact photomechanically and (ii) these movements cannot involve eye muscle activation.

DOI: https://doi.org/10.7554/eLife.26117.085

### When rhabdomeres move, the ommatidium lens system above stays still

Using the high-speed video microscopy, we next tested whether the *Drosophila* lens system or any other ommatidium structures moved during the rhabdomere movements (*Appendix 7—figure 7*). In the experiments, a z-axis micromanipulator (Sensapex, Finland) was used to shift

and reposition *Drosophila* in piezo-steps vertically. This allowed the focused image, as projected on the camera, to scan through each studied ommatidium, providing exact depth readings in µm. We then recorded any structural movements inside the ommatidia at different depths; from their corneal lens down to the narrow base, where the cone and pigment cells form an intersection between the crystalline cone and the rhabdomere tips (*Tepass and Harris, 2007*).

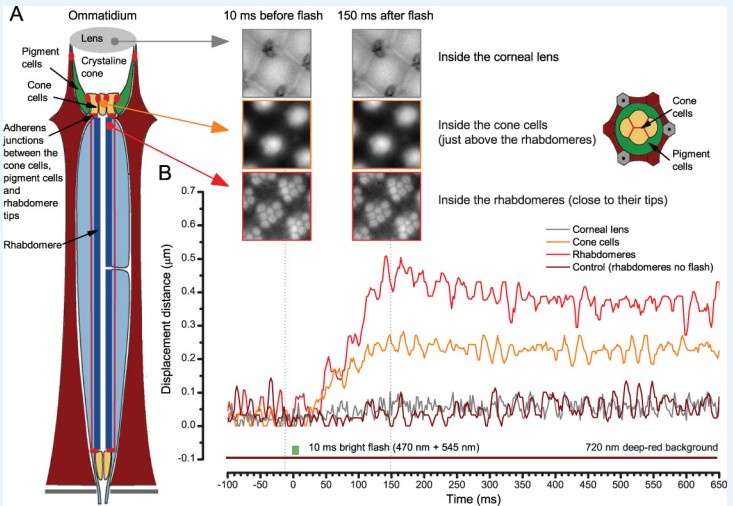

**Appendix 7—figure 7.** When the rhabdomeres move the ommatidium lens stays still. (**A**) High-speed video recordings at different depths inside ommatidia before and after a bright light flash. (**B**) Cornea (ommatidium) lens and the optical structures to the narrow base of the crystal cone remained virtually immobile. Below these, the cone cells showed movement that was half of that seen in the rhabdomeres. In the schematic, red dots and lines indicate adherens junctions that link the photoreceptors to the pigment and cone cells.
DOI: https://doi.org/10.7554/eLife.26117.086

We found that when the rhabdomeres moved photomechanically (*Appendix 7—figure 7B*, red trace; *Video 4*) the corneal lens and the upper ommatidium structures were essentially immobile (grey), and normally remained so throughout the recordings. However, clear stimulus-induced movements were detected at the basal cone/pigment cell layer (orange; also inset) that connects to the rhabdomere tips with adherens junctions (*Tepass and Harris, 2007*). Although it is less clear how much these structures reflected the rhabdomere motion underneath or were pulled by it, it is quite certain that they formed an aperture in the light path, which moved less than the light-sensors (rhabdomeres) below (orange vs. red). In Appendix 8, we analyze how such an interaction might dynamically narrow the R1-R6s' receptive fields to visual motion.

Overall, these results further verified that in normal stable recordings the used blue/green light flash was not evoking eye muscle activity, which would otherwise move the whole eye.

Light intensity-dependence of rhabdomere movements (*in vivo* dynamics)

Through a wide-ranging testing regime, we further discovered (*Appendix 7—figure 8* and *Appendix 7—figure 9*) that:

- Light-induced R1-R7 rhabdomere movements were robust and repeatable. *Appendix 7—figure 8A* shows 10 trials (thin grey traces) and their mean (black) to a 10 ms bright flash measured from the same ommatidium. Between each flash, the eye was dark-adapted for 30 s. Characteristically, the rhabdomeres contracted to every light flash without a failure. Whilst these movements showed amplitude variations, their dynamic behavior was similar. Here, they reached their peak (mean = 0.806 µm, *Appendix 7—figure 8B*) in about 140 ms and then decayed back to the baseline slower, mean $\tau_r$ ~190 ms (*Appendix 7—figure 8C*).
- Dark-adapted R1-R7 rhabdomeres' maximum movement range (*Appendix 7—figure 9A–I*) was considerably larger (0.3–1.2 µm) than the displacement range measured ex vivo by atomic force microscope (*Appendix 7—figure 9B*, $AFM_{max}$ ≤0.275 µm) (*Hardie and Franze,*

2012) on the corneal surface. This difference is hardly surprising. AFM measures axial (inward) cornea displacements, presumably resulting from a large number of simultaneous photoreceptor contractions underneath, whereas our high-speed video microscopy method measures orthogonal (horizontal) rhabdomere movements locally at their source. Owing to the slight excitation caused by the bright 740 nm red-light background needed for *in vivo* imaging (*Appendix 7—figure 9B*), the actual rhabdomere movements in full dark-adapted conditions could be even larger.

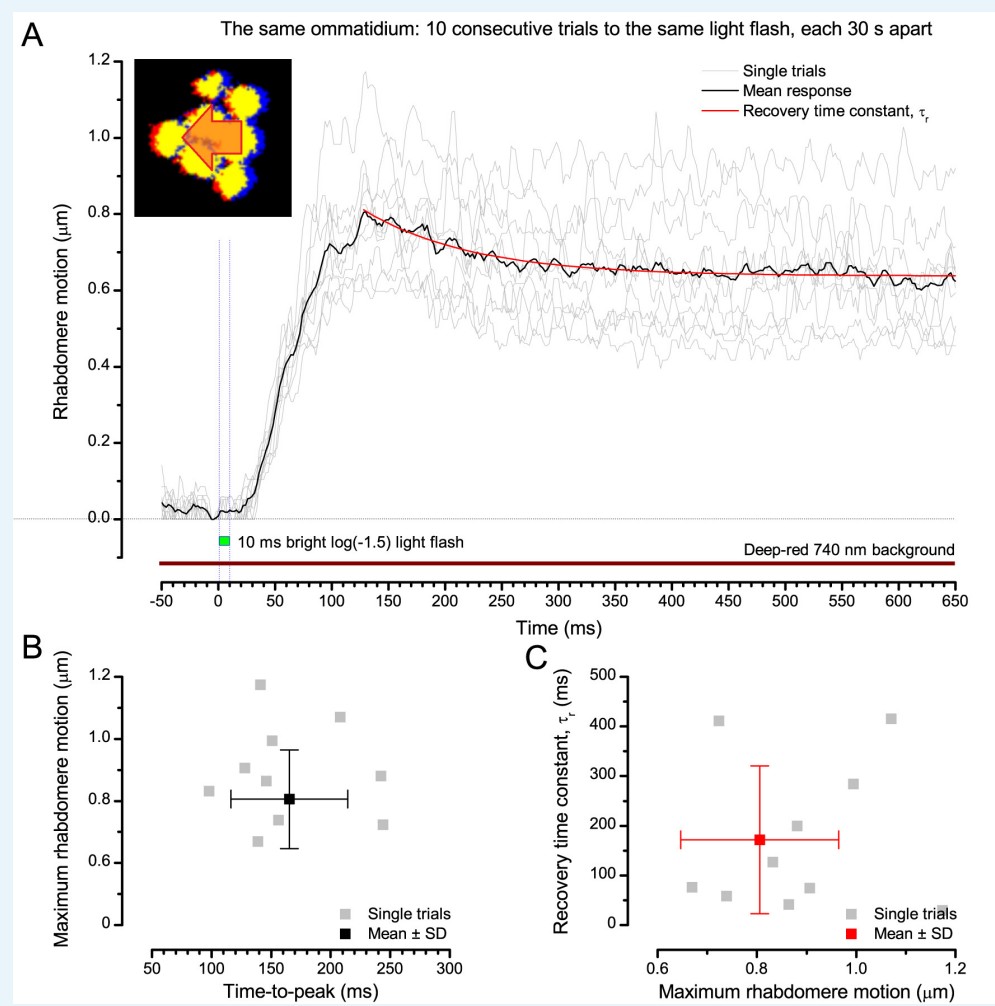

**Appendix 7—figure 8.** Photomechanical rhabdomere motion is robust and repeatable. (**A**) Consecutive rhabdomere motions (grey thin traces) of the same ommatidium and their mean (black) to 10 stimulus repetitions (bright flash). The mean response recovery is fitted with an exponential, $\tau_r$. (**B**) Each response maximum is shown against its time delay (time-to-peak) with the mean and SD. (**C**) Each response recovery time constant is plotted against its maximum with the mean and SD. Recording at t = 20°C.

DOI: https://doi.org/10.7554/eLife.26117.087

- The rhabdomere movement recordings (thin grey traces) from single ommatidia of individual flies vary more than the corneal AFM data (black traces). The sizable variations in their movement range and fine dynamics (such as minor oscillations) imply both considerable trial-to-trial (*Appendix 7—figure 8*) and fly-to-fly variability (*Appendix 7—figure 9*). Much of this is clearly physiological, as rhabdomere movement sizes and waveforms to specific stimuli were similar in one fly but often slightly different to those seen in another fly. However, because of the extreme sensitivity of our method (*Appendix 7—figure 4*, providing subpixel movement

resolution), some of the variations clearly reflected experimental noise. Such noise included microscopic mechanical vibrations in the recording system, minute spontaneous eye muscle activity (see Appendix 4, *Appendix 4—figure 6*), and Poisson-noise, in which the image signal-to-noise ratio - as captured by the camera's CMOS sensor – reduced the more the faster the sampling. Appropriately, the average responses (red) to different intensity flashes were smoother, yet still remained much larger than in the AFM data.

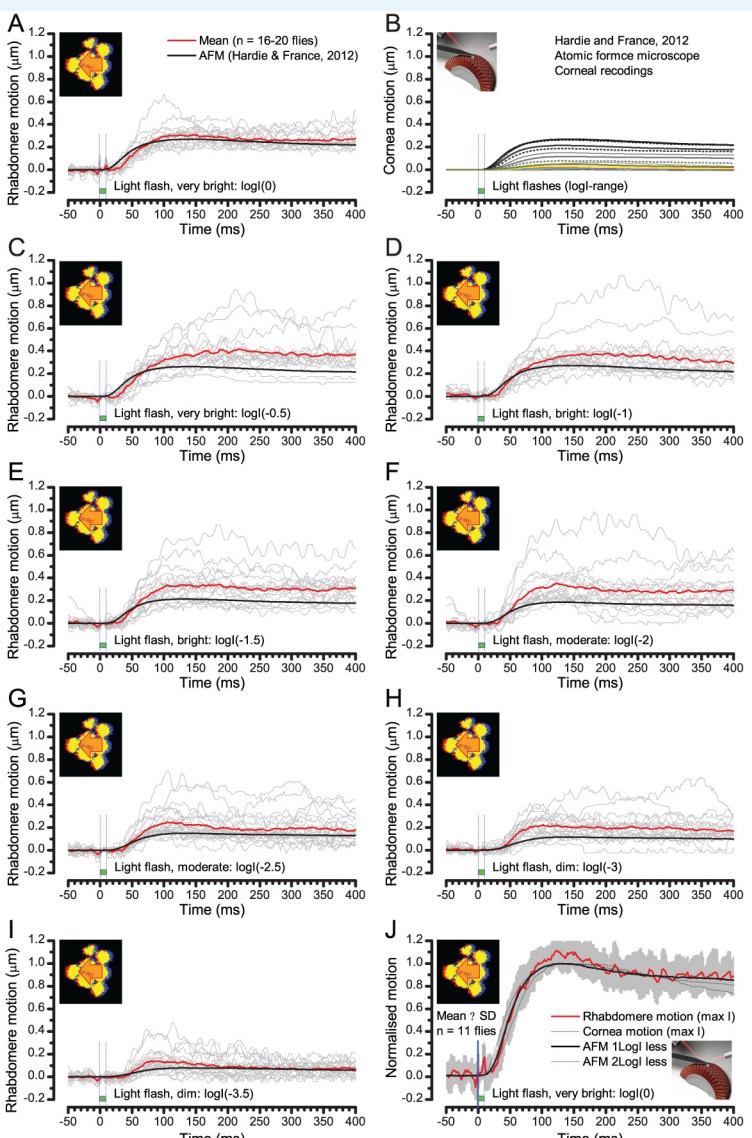

**Appendix 7—figure 9.** Comparing optically resolved wild-type rhabdomere movements to corresponding atomic force microscope (AFM) recordings from the corneal surface. To ease the comparisons, the AFM data is inverted. (**A**) Rhabdomere motion within individual ommatidia (grey thin traces) and their mean (red) evoked by the brightest 10 ms test flash are plotted against the largest AFM recording (black) to the brightest 5 ms test flash (data from *Hardie and Franze, 2012*). The rhabdomere movement range is larger than what the AFM data suggests. (**B**) AFM recordings to a broad logarithmic light flash intensity range. (**C–I** Rhabdomere movements vs. AFM recordings to light flashes of broadly comparable diminishing intensities. Notice that some individual rhabdomere movement recordings show minor oscillations that could be related to recording noise or physiological activity. (**J**) the mean and SD of normalized rhabdomere movements to the brightest test flash are compared

with the normalized AFM recordings to three different test flash intensities. All these AFM recordings fall within the SD of the given rhabdomere recordings.

DOI: https://doi.org/10.7554/eLife.26117.088

- The average rise and decay time courses of the normalized rhabdomere movement recordings followed closely those of the normalized AMF recordings (*Appendix 7—figure 9J*). Such dynamic conformity strongly suggests that both the methods capture accurately photo-mechanical R1-R8 photoreceptor contractions in their fast natural time resolution. But these observations also provided further evidence that *ex vivo* AFM data underestimate the actual magnitude of rhabdomere movements within ommatidia. In fact, it seems possible that to maximize neural images' spatial resolution, the eye's architectural design dampens the lens system movement, while its sensors (rhabdomeres) contract. This would inadvertently impede the AFM signal (axial movement), and any horizontal lens shift (*Appendix 7—figure 7*), measured on the corneal surface.

## Estimating light intensity falling upon the rhabdomeres

Only in 2 out of 21 tested *Drosophila* eyes, the rhabdomeres moved unmistakably (twitched) to a very dim 10 ms blue/green LED flash, in which intensity was reduced ~200,000 fold by neutral density filters. Therefore, in these two positive occasions: (**i**) the resulting response must have been quantal with (ii) the 10 ms flash maximally containing ~1–3 absorbed photons. Furthermore, because this flash only succeeded in ~1/10 eyes, its average maximum intensity could only be $\leq$3/10 photons/10 ms, *i.e.* $\leq$0.03 photons/ms. This means that the brightest flash (logI(0)), which was not filtered, could maximally contain $\leq$6,000 photons/ms, or $\leq$6 million photons/s, making the used light intensity range natural and directly comparable to that used for the intracellular recordings (*Figures 1–2* and *6–9*).

This reasoning is in line with the similar LED driver settings used in all the experiments, and the similar V/log(I) and (μm)/log(I) functions, which resulted from these experiments.

## In vitro rhabdomere movements

As further controls, we measured photomechanical R1-R8 rhabdomere contractions of freshly dissociated ommatidia (*Hardie and Franze, 2012*) to green (480 nm) light flashes using high-speed video recordings with infrared 850 nm background illumination (*Appendix 7—figure 10*). The benefit of this in vitro method was that it provided a clear side-view of the tested wild-type and mutant rhabdomeres, enabling us to estimate their axial (longitudinal inward) contraction component; or how much the rhabdomere tip moved away from the ommatidium lens (*Video 2*). *In vivo*, such fast lengthwise light-sensor movements should contribute to R1-R6 photoreceptors' transiently narrowing receptive fields (see Appendix 8).

We found that after dark-adaptation bright flashes could evoke 0.8–1.7 μm longitudinal rhabdomere contractions. These were characteristically accompanied by synchronous (about equally large) crosswise movement (or twist), which likely forms the basis of the sideways rhabdomere displacement; seen during the *in vivo* recordings (*e.g. Appendix 7—figure 1*).

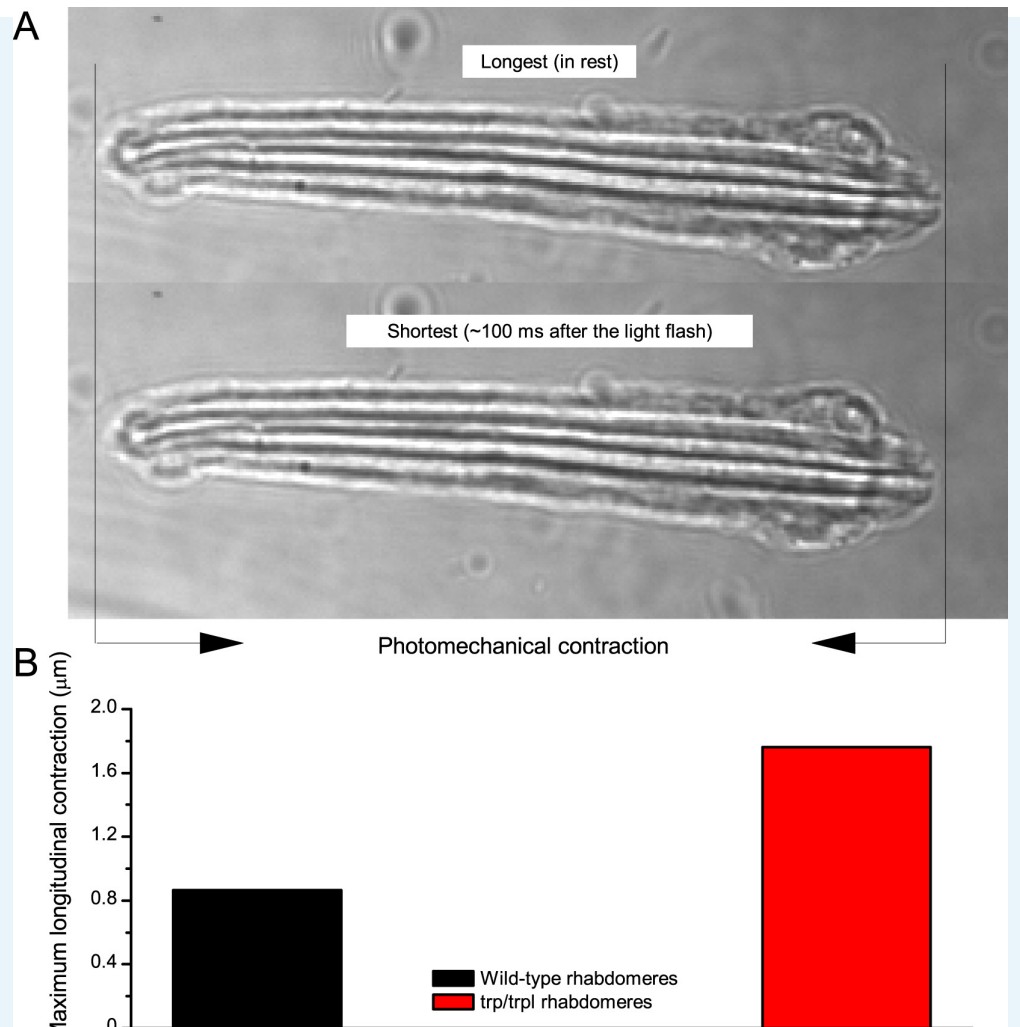

**Appendix 7—figure 10.** Photomechanical rhabdomere contractions in dissociated ommatidia. (**A**) Two frames of high-speed video footage of contracting R1-R8 photoreceptor rhabdomeres in vitro, evoked by a bright flash. The size-view imaging reveals the size and dynamics of their photomechanical lengthwise changes. (**B**) Characteristic maximum longitudinal rhabdomere contractions range from 0.8 to 1.7 µm. Thus, in an intact eye, during contractions the rhabdomeres would move inwards, away from the lens. This movement is likely to move the rhabdomere tips into the ommatidium lens' focal point, narrowing the photoreceptors' acceptance angles (see Appendix 8, *Appendix 8—figure 3*). Notably, many rhabdomeres also twist during these contractions, providing additional crosswise movements. See *Video 2*.

DOI: https://doi.org/10.7554/eLife.26117.089

## Photomechanical rhabdomere movements vs. R1-R6s' voltage responses

We next compared intracellular wild-type and $hdc^{JK910}$ R1-R6 voltage outputs, evoked by 10 ms light flashes after brief dark-adaptation, to their characteristic rhabdomere movements (*Appendix 7—figure 11*). R1-R6 output and rhabdomere motion exhibited broadly comparable delays (or dead-time), but overall the rhabdomeres moved considerably slower than how their voltage was changing (*cf.* the thick black traces in A–B, and the thick red traces in C–D for similar flash intensities).The voltage responses peaked 25–30 ms from the light onset, and the rhabdomere movements 40–120 ms later. At this point, the photoreceptors had almost repolarized back to their dark resting potential (indicated by zero ordinate).

Moreover, the recordings suggested that for a given flash $hdc^{JK910}$ rhabdomeres typically moved less and returned faster to their original positions than their wild-type counterparts.

To further characterize their light-dependent differences, we plotted the maximum wild-type and $hdc^{JK910}$ rhabdomere movements of many flies against the corresponding flash intensities over the whole tested light range. The analysis revealed that:

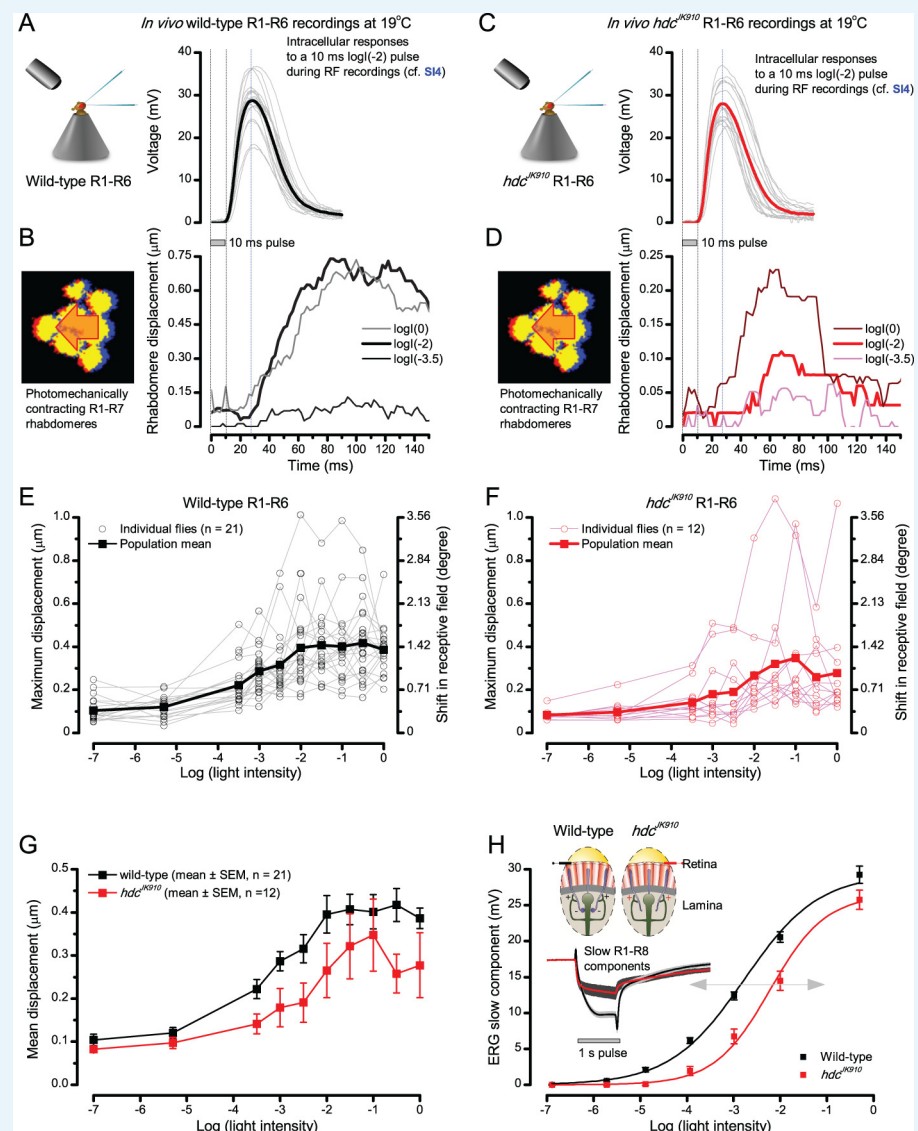

**Appendix 7—figure 11.** Comparing dark-adapted wild-type and $hdc^{JK910}$ R1-R6s' photome-chanical rhabdomere contractions to their corresponding electrophysiological responses at 19°C. (**A**) Intracellular voltage responses of dark-adapted wild-type R1-R6s to a 10 ms subsaturating light pulse, delivered at the center of their receptive field. (**B**) Characteristic non-averaged photomechanical rhabdomere movements (displacement in time) of a typical wild-type fly, as quantified by the cross-correlation analysis (**Appendix 7—figure 4**) to very dim (log(−3.5)), moderate (log(−2)) and very bright (log(0)) 10 ms light flashes. (**C**) Intracellular responses of dark-adapted $hdc^{JK910}$ R1-R6s to a similar stimulus (as in a). (**D**) $hdc^{JK910}$ rhabdomere movements to very dim, moderate and very bright 10 ms flashes are characteristically slightly smaller than the corresponding wild-type recordings in (**B**). (**E**) The wild-type and (**F**) $hdc^{JK910}$ rhabdomere displacements increase with logarithmic light intensity. (**G**) Mean wild-type rhabdomere movement was larger than that of $hdc^{JK910}$ over the tested intensity range, with the $hdc^{JK910}$ photoreceptors' apparent right-shift indicating a 10-fold

reduced sensitivity. (H) This right-shift is broadly similar to these photoreceptors' V/LogI characteristics, measured from their ERG slow components (*Dau et al., 2016*).
DOI: https://doi.org/10.7554/eLife.26117.090

- Both wild-type and $hdc^{JK910}$ rhabdomere movements increased with flash intensity (*Appendix 7—figure 11E–F*), following a characteristic sigmoidal displacement/logI-relationship.

- The average $hdc^{JK910}$ rhabdomere movements were smaller than those in wild-type eyes. In part, this could result from $hdc^{JK910}$ R1-R6s being smaller than their wild-type counterparts, and embedded deeper inside the ommatidia. As their rhabdomere diameters are ~4% smaller than those of the wild-type (Appendix 5), their length should also be reduced proportionally; with the average wild-type R1-R6 being ~100 μm tall, $hdc^{JK910}$ R1-R6s should be ~4 μm shorter. Accordingly, by using the z-axis micromanipulator (see above), we measured in situ (rest) that $hdc^{JK910}$ rhabdomere tips were 3.5 μm further away from the lens than the wild-type tips ($hdc^{JK910}$: 27.4 ± 1.2 μm; wild-type: 23.9 ± 1.2 μm; n = 6 ommatidia, six flies).

- $hdc^{JK910}$ rhabdomere contractions further implied reduced sensitivity (*Appendix 7—figure 11*), seen as a 10-fold right-shift in their displacement/logI-curve in respect to the wild-type data.

- Interestingly, this sensitivity difference resembles that seen between the wild-type and $hdc^{JK910}$ ERGs (*Appendix 7—figure 11H*). We have recently provided compelling evidence that the missing histaminergic (inhibitory) neurotransmission from $hdc^{JK910}$ photoreceptors to interneurons (LMCs and amacrine cells) causes a tonic excitatory synaptic feedback to R1-R6s, depolarizing them ~ 5 mV above the normal wild-type dark resting potential (*Dau et al., 2016*). Thus, $hdc^{JK910}$ R1-R6s should experience a tonic $Ca^{2+}$ influx and be permanently in a more 'light-adapted' state. Here, our results suggested that $Ca^{2+}$ overload may desensitize the biophysical machinery that moves the rhabdomeres, reducing its dynamic range.

## Light-adapted rhabdomere motion reflects rhodopsin/meta-rhodopsin balance

Given the slightly reduced back-to-front rhabdomere mobility of $hdc^{JK910}$ photoreceptors, we next asked whether prolonged light-adaptation itself would reduce wild-type photoreceptors' rhabdomere movement. To study this question, we examined how the rhabdomeres in seven individual flies responded to different light impulses at different adaptation states.

First, to obtain the baseline responses in each fly eye, we recorded their rhabdomere movements to a bright and a very bright full-field green-blue flash after 30 s of dark-adaption (*Appendix 7—figure 12A and B*, respectively). As before, we found that the brighter the light flash, the larger and the faster their rhabdomere movements were on average.

Interestingly, however, when the flies were adapted to a moderate or bright blue (470 nm) light field for 30 s, which converts most (if not all) rhodopsin Rh1 to its active meta-form (causing PDA, prolonged depolarizing afterpotential), the flashes now evoked only weak or no rhabdomere movement (*Appendix 7—figure 12C–D*). Such reduced mobility somewhat resembled that in some $hdc^{JK910}$ rhabdomeres (*Appendix 7—figure 11D*). These observations can be explained, at least in part, with the basic molecular model proposed for rhabdomere contraction (*Hardie and Franze, 2012*). Here, meta-rhodopsin would continuously activate G-protein and in turn phospholipase C (PLC). PLC would then cleave most $PIP_2$ off the microvillar membrane, causing a tonic photoreceptor contraction, which facilitates light-gated channel openings and thus increases $Ca^{2+}$ influx. But because a 10 ms bright green (545 nm) light pulse would convert only some fraction of meta-rhodopsin back to its non-activated form, this effect would be small proportionally, and could only partially rescue the rhabdomere contractibility.

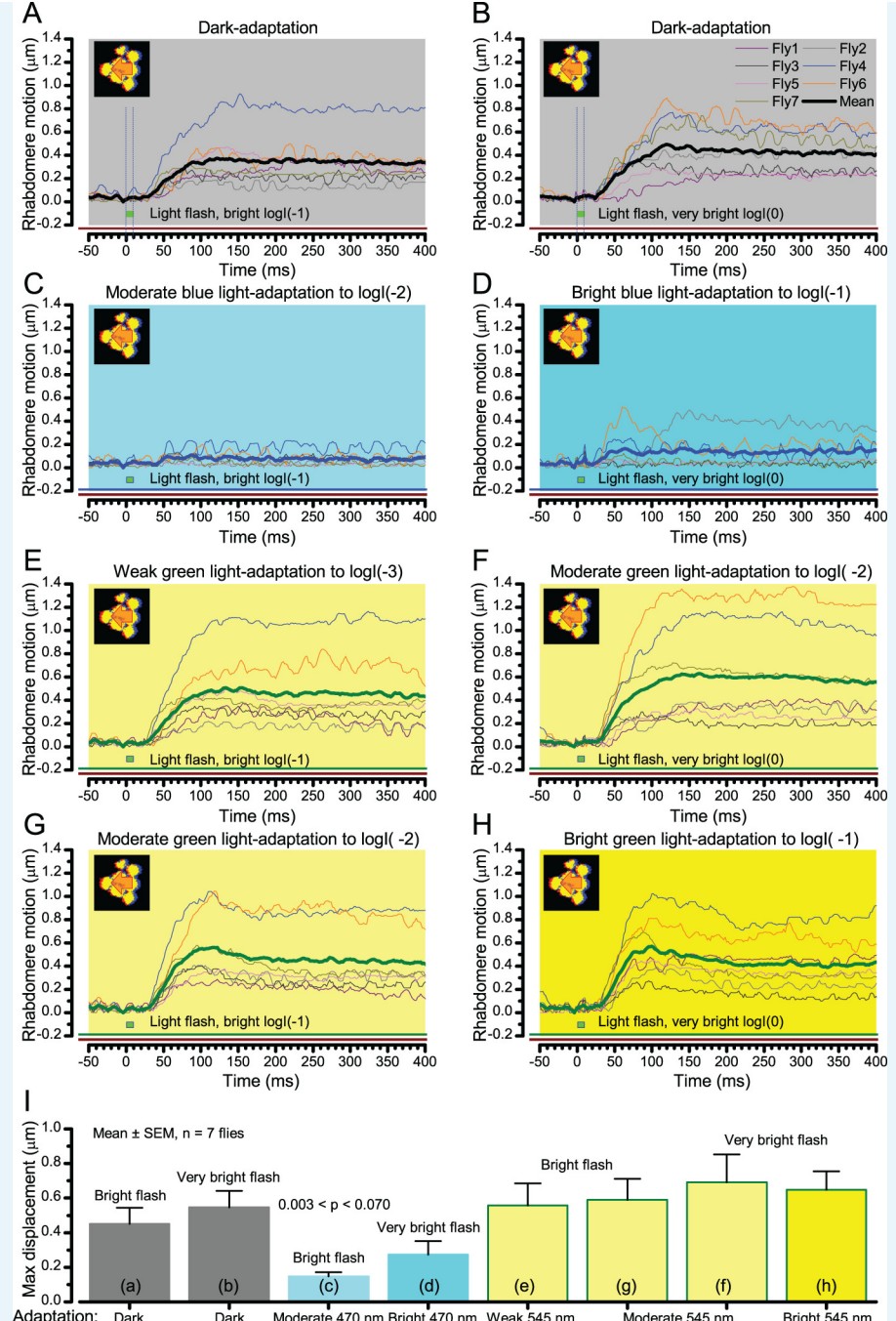

**Appendix 7—figure 12.** Wild-type photomechanical rhabdomere movement dynamics at different light-adaptation states. Characteristic non-averaged photomechanical rhabdomere movements of 7 wild-type flies and their means (thick traces) to bright (logI(−1); left) or to very bright (logI(0); right) light flashes. The recordings were performed in relative darkness (>720 nm) or when the eyes were adapted for 30 s to different green (545 nm) or blue (470 nm) light intensity levels; from weak (logI(−3)) to bright (logI(−1)). (**A–B**) In dark-adaptation, consistent with the results in *Appendix 7—figure 11*, the brighter the flash, the larger and faster the evoked mean rhabdomere movements. (**C–D**) Prolonged blue light exposure converted virtually all Rh1-rhodopsins to their active meta-form, causing a PDA (prolonged depolarizing afterpotential). PDA increases the photoreceptors' intracellular calcium load, cleaving PIP$_2$ from the plasma-membrane and so keeping them in a contracted state. A very bright green flash rapidly converts some of the meta-Rh1 back to Rh1, enabling small and brief rhabdomere

movements, which resemble those seen with *hdc*[*JK910*] flies (*Appendix 7—figure 11*). (**E–H**) Adaptation to different green-light intensity levels did not abolish rhabdomere movements to light increments. These movements can, in fact, be larger than in the same cell's dark-adapted state, as was seen in Fly4 and Fly6 recordings. Notice that a 1.2–1.4 μm rhabdomere displacements means ~4-5° shifts in a R1-R6 photoreceptor's receptive field (*Appendix 7—figure 4*), which can be more than the average *Drosophila* interommatidial angle.

DOI: https://doi.org/10.7554/eLife.26117.091

## Rhabdomere contractions resensitize refractory sampling units

On the other hand, under more natural green light-adaptation (*Appendix 7—figure 12E–H*), the eyes' normal back-to-front rhabdomere contractibility to light increments was retained, and sometimes even increased, in respect to their dark-adapted baseline responses (*Appendix 7—figure 12A–B*). Consequently in every fly eye (7/7), the rhabdomeres moved more when green-adapted than when moderately blue light-adapted (*Appendix 7—figure 12C*), as quantified by their flash-induced maximum displacements (*Appendix 7—figure 12I*).

These results indicate that in normal spectrally-broad natural environments, light increments (positive contrasts) will evoke fast evasive rhabdomere movements, steering them away from pointing directly to a bright light source. This novel photomechanical adaptation should together with the slower screening pigment migration (intracellular pupil mechanism, Appendix 2) help to recover (resensitize) a rhabdomere's refractory sampling units (30,000 microvilli). Thus, the rhabdomere movements likely participate in optimizing photon sampling for maximum information capture (**Appendixes 1–2**). This further means that a rhabdomere's state of contraction is constantly being reset to the ongoing light input, providing the capacity to respond to the next stimulus increment. Therefore, although being slower, in many sense, *Drosophila* photoreceptors' mechanical adaptation resembles the inner ear hair-cells' adaptive resensitization (*Howard et al., 1988*; *Corey et al., 2004*).

## Appendix 8

DOI: https://doi.org/10.7554/eLife.26117.092

# Microsaccadic sampling hypothesis for resolving fast-moving objects

## Overview

This appendix describes a new 'microsaccadic sampling'-hypothesis that predicts a *Drosophila* photoreceptor's voltage responses to moving dots, and provides important background information about the experimental and theoretical results presented in *Figures 7–9*.

In this appendix:

- We use our biophysical *Drosophila* R1-R6 model (Appendix 1), with different degrees of photomechanical rhabdomere movements modulating light input (Appendix 7), to simulate photoreceptor voltage output to moving bright dots that cross its receptive field in different directions, speeds and inter-distances.
- By comparing the model simulations to intracellular recordings, we reveal the likely biomechanics that allow R1-R6s to resolve adjacent dots at saccadic velocities (Appendix 6).
- We show that when a rhabdomere contracts away from the ommatidium lens's focal point, its receptive field must move and narrow dynamically. Together these processes actively reshape both the light input and photoreceptor output to separate and sharpen neighboring visual objects in time, improving their resolvability.
- Crucially, with such photomechanical light input modulation, the model photoreceptor output closely approximates that of the real R1-R6s, as recorded to two moving bright dots crossing their receptive fields at different speeds (*Figures 7–9*; Appendix 6).
- Hence, with refractory photon sampling and photomechanical rhabdomere movements, we can correctly predict and convincingly explain visual acuity of R1-R6s to moving objects.

## Modeling a R1-R6's receptive field dynamics to moving dots

Based on the combined results in Appendix 1-7, we develop a new 'microsaccadic sampling'-hypothesis, which predicts how photomechanical rhabdomere contractions (microsaccades) move and narrow *Drosophila* R1-R6 photoreceptors' receptive fields (RFs) to resolve fast-moving objects. We present extensive analytical and experimental evidence to show how these mechanisms operate with the photoreceptors' refractory information sampling to reduce light-adaptation and to increase the spatiotemporal resolution of their voltage responses, improving visual acuity.

Using the results in Appendix 1-7, we can now work out the biomechanics, which allow a R1-R6 to resolve two close bright dots crossing its receptive field at saccadic speeds. We do this systematically by comparing the output of our biophysical model (*Song et al., 2012*; *Song and Juusola, 2014*; *Juusola et al., 2015*) (Appendix 1), in which input is modulated by different degrees of rhabdomere photomechanics (*Appendix 8—figure 1A-C*), to the corresponding recorded real R1-R6 output (*Appendix 8—figure 1D*). Specifically, we consider three input modulation models:

A. *Stationary rhabdomere model (receptive field is fixed)*
B. *Photomechanical rhabdomere model (receptive field moves)*
C. *Photomechanical rhabdomere model (receptive field moves and narrows)*

In the following, to make these different models (**A-C**) directly comparable, we first present the findings for two bright dots, which cross the receptive field at 205 °/s (*Appendix 8—figure 1*), before generalizing the results for a vast range of stimuli and giving more examples. Note that these simulations and recordings were performed at 19–20°C. Later on in this appendix, we show how at the *Drosophila*'s preferred temperature range (24–25°C) (*Sayeed and Benzer, 1996*) these dynamics are naturally faster (*Juusola and Hardie, 2001b*) and improve visual acuity further. The scripts for light stimulus calibrations and for simulating

responses to two moving dots are in the repository: https://github.com/JuusolaLab/Microsaccadic_Sampling_Paper/tree/master/CalibrateLightInput-PhotoreceptorMovement.

## A. Stationary rhabdomere model (receptive field is fixed)

This approach is broadly analogous to the classic theory (Appendix 4 and Appendix 6). It was implemented in four steps (*Appendix 8—figure 1*):

i. Two bright dots, which were 6.8° apart, crossed a R1-R6's RF (Δρ = 8.1°) front-to-back at the saccadic speed of 205 °/s.
ii. Concurrently, the ommatidium lens focused their light onto a rhabdomere tip.
iii. The resulting dynamic light input at the rhabdomere tip was a convolution of the two dot intensities with the cell's receptive field over time (*Srinivasan and Bernard, 1975*; *Juusola and French, 1997*).
iv. The light input (photons/s) drove the photon sampling and refractory quantum bump (QB) production of 30,000 microvilli (*Song et al., 2012*; *Song and Juusola, 2014*; *Juusola et al., 2015*), which formed the rhabdomere. The resulting macroscopic photoreceptor output (*Hardie and Juusola, 2015*) dynamically integrated the QBs.

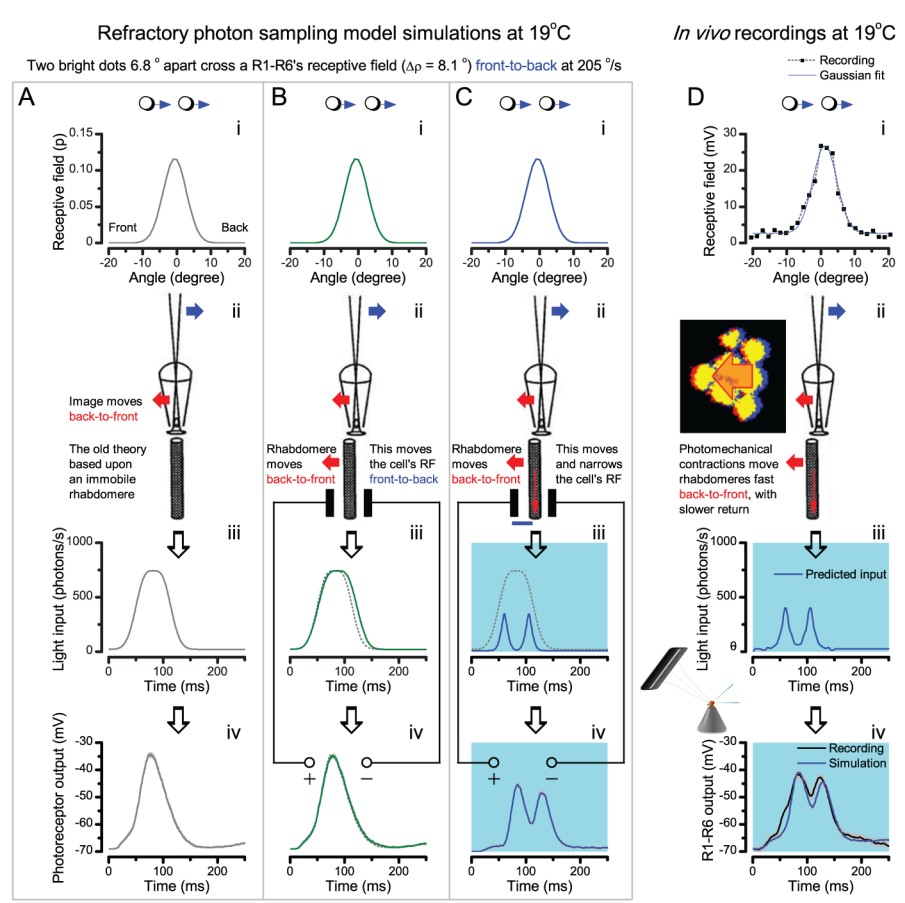

**Appendix 8—figure 1.** Microsaccadic sampling-hypothesis predicts R1-R6 output dynamics to fast moving bright dots. We compare stochastic sampling model predictions of increasingly realistic light input modulation (**A–C** to real recordings **D**). In each case, the light stimulus is two fast moving bright dots (i), which cross a R1-R6's receptive field (RF, half-width = 8.1°) front-to-back at 205 °/s. The immobile ommatidium lens focusses their light onto a rhabdomere tip (ii). But because of diffraction, their images blur (airy disk). Interestingly, however, in *Drosophila*, the airy disk half-width (~1.7°or ~ 0.6 µm) is smaller than the rhabdomere diameter (~1.7 µm; Appendix 5). The dynamic light input falling upon the

rhabdomere (iii) is then estimated by convolving the two dot intensities with the cell's receptive field over time (i). This light input is sampled stochastically by 30,000 microvilli, in which quantum bumps integrate the dynamic voltage response (iv). (**A**) In the simulations, where the rhabdomere is stationary during the dots flyby (corresponding to the classic theory (*Srinivasan and Bernard, 1975*; *Juusola and French, 1997*), ii), their light fuses together (iii). This is because the simulated photoreceptor's RF half-width (8.1°) is wider than the distance between the dots (6.8°). Accordingly, the photoreceptor output (iv) cannot distinguish the dots. (**B**) Next, we include the light-induced fast back-to-front rhabdomere movement in the model (ii; as measured in the high-speed video recordings in Appendix 7, *Appendix 7—figure 1*), but keep the cell's RF shape the same. The resulting RF movement makes the light input rise and decay slightly later than in the previous case (iii; *cf.* the green and dotted grey (**A**) traces), but still cannot separate the two dots. The resulting photoreceptor output (iv) shows a single peak, which is slightly broader than the output from the immobile cell in (**A**). (**C**) In the full model, as the contracting rhabdomere jumps away from the focal point (ii), the light input both moves with the cell's RF and dynamically narrows it from 8.1° to 4.0°. These processes differentiate the light input (from the dots) into two separate intensity spikes over time (iii), which the photoreceptor output can clearly separate in two distinctive peaks (iv). (**D**) *In vivo* intracellular photoreceptor recordings from a wild-type R1-R6 to the two moving dot stimulus show comparable dynamics to the full model's prediction. In the simulation, the same cell's RF (i) was moved and narrowed dynamically (ii-iii) according to our hypothesis in (**C**). This close correspondence, which occurred even without the additional response accentuation from the synaptic feedbacks (Appendix 6), implies that *Drosophila* photoreceptors' RFs must move and narrow with their photomechanical rhabdomere contractions. *Appendix 8—figure 3* proposes four mechanisms to narrow RF.
DOI: https://doi.org/10.7554/eLife.26117.093

Characteristically in this approach, input modulation reduced to a single peak as the RF convolution fused the light from the two dots together (iii). This made them irresolvable to the discrete photon sampling (iv). Specifically here, light input could not separate the two dots because the given receptive field half-width (8.1°) was wider than the distance between the dots (6.8°). The resulting model output, thus, failed to capture the resolvability of the real R1-R6 output (*Appendix 8—figure 1*, iv).

Markedly, the model prediction (*Appendix 8—figure 1A*, iv) much resembled those of the deterministic Volterra-models, in which similarly broad RFs were used in the calculations (see Appendix 6).

## B. Photomechanical rhabdomere model (receptive field moves).

In this case (*Appendix 8—figure 1B*), we kept the cell's receptive field shape the same (i; $\Delta\rho = 8.1°$) but included the light-induced fast horizontal (back-to-front) rhabdomere movement in the model (ii). The given model parameters (*Appendix 8—table 1*) were fixed to closely approximate the experimentally observed rhabdomere movement dynamics (see Appendix 7). Here, the following deductions were made

- Each photoreceptor contracted independently to light. Although this movement is linked to the synchronous contractions of its neighbors inside the same ommatidium, such cooperativity makes no difference to the model.
- Rhabdomere movement started 8 ms after the first dot reached the outer rim of a photoreceptor's receptive field, called the trigger zone. This delay matched both the delay in R1-R6 output (Appendix 7: *Appendix 7—figure 11*) and the apparent dead-time in AFM data (*Hardie and Franze, 2012*) to a bright light flash (Appendix 7: *Appendix 7—figure 9J*). The trigger zone was 14.6° from the photoreceptor receptive field center, matching the typical spatial threshold where dark-adapted wild-type R1-R6s responded faintly to subsaturating peripheral light flashes (Appendix 4, *Figure 7—figure supplement 1*).
- Maximum horizontal rhabdomere movement was set to be 0.58 μm, corresponding to a 1.6° shift in its receptive field. This value is close to the measured average of the maximum light-induced rhabdomere movements in wild-type fly ommatidia (Appendix 7, *Figure 8D*).

- Rhabdomere movements had two phases. In the first phase, a rhabdomere moved 1.6° in back-to-front direction for 100 ms, reaching its maximum displacement. This caused a receptive field to shift in the opposite, α (front-to-back) direction. Importantly, the first phase could not be disturbed. In the second phase: the rhabdomere slowly returned to the original position in 500 ms. The second phase could be disturbed. Both the phases followed linear motion.

**Appendix 8—table 1.** Parameters for modeling a R1-R6's receptive field (RF) dynamics caused by its rhabdomere contraction

| Trigger zone: trig | Starting RF half-width: $\Delta\rho_{start}$ | Delay before rhabdomere motion: lag | Rhabdomere motion: (time-to-peak) Phase 1 | Resulting parallel RF shift: $RF_{shift}$ | RF shift direction: α (front-to-back) | Rhabdomere motion: Phase 2 | Ending RF half-width: $\Delta\rho_{end}$ |
|---|---|---|---|---|---|---|---|
| 14.6 ° | 8.1° | 8 ms | 100 ms | 1.6° | 0 | 500 ms | 4.0° |

DOI: https://doi.org/10.7554/eLife.26117.094

As in the first case, the moving receptive field model (**Appendix 8—figure 1B**) was implemented for the same stimulus in four steps (i-iv):

i. Two bright dots, 6.8° apart, crossed a photoreceptor's RF (Δρ = 8.1°) front-to-back at the saccadic speed of 205 °/s.
ii. The ommatidium lens focused their light onto a rhabdomere tip. But here, after 8 ms delay, the rhabdomere started to move back-to-front as it contracted photomechanically.
iii. The resulting dynamic light input was, therefore, a convolution of the two dot intensities and the cell's receptive field, which moved at different speeds in the same direction (front-to-back).
iv. The light input drove the photon sampling and refractory QB production of 30,000 microvilli, while the resulting macroscopic photoreceptor output summed up the QBs.

We found that the resulting receptive field movement caused (iii) the light input rise and decay slightly later than when the rhabdomere was immobile (*cf.* green and dotted grey traces). However, because the two dots were close and crossed the cell's receptive field fast, the given co-directional receptive field motion failed to separate the light from them. Thus, (iv) the resulting photoreceptor output showed a single peak, which was slightly broader than the output from the immobile model.

## C. Photomechanical rhabdomere model (receptive field moves and narrows).

There is a large disparity between the measured (Appendix 4: Δρ = 7.00–11.65°) and the optical waveguide theory derived (**Stavenga, 2003b**) (Δρ = 3.5–5.3°) acceptance angles of dark-adapted R1-R6s. Even Snyner's simple formula (**Snyder, 1977**) (Appendix 4: **Equation A4.3**), which overestimates (**van Hateren, 1984**; **Stavenga, 2003b**) Δρ from the measured *Drosophila* ommatidium optical dimensions, gives a theoretical upper bound (Δρ <5°) that is smaller than the smallest intracellularly measured values.

$$\Delta\rho = \left( \sqrt{\left(\frac{0.545}{16}\right)^2 + \left(\frac{1.7}{21.36}\right)^2} \right) \times \frac{180}{\pi} = 4.9601^{\circ}$$

$$\Delta\rho = \left( \sqrt{\left(\frac{0.545}{17}\right)^2 + \left(\frac{1.7}{21.36}\right)^2} \right) \times \frac{180}{\pi} = 4.9161^{\circ}$$

These upper bounds for green light (545 nm) were obtained for the smallest and largest ommatidium lens (16–17 μm) with average rhabdomere diameter EM measurements (1.7 μm) (Appendix 5), using the experimentally estimated focal length (21.36 μm) (*Gonzalez-Bellido et al., 2011*).

Moreover, interestingly, Götz estimated from *Drosophila* optomotor behavior, using the early flight simulator system (*Götz, 1964*), that in bright illumination R1-R6 Δρ would be 3.5°.

To resolve the paradox between the conflicting experimental and theoretical Δρ-estimates, which in the past were based upon histological measurements of fixed/stained (dead/immobile) retinal structures, we hypothesized that the photomechanical rhabdomere contractions not only move a photoreceptor's RF but also dynamically narrow it (*Appendix 8—figure 1C*). What is more, we reasoned that the RF narrowing should depend upon stimulus history; the cell's ongoing light exposure. Therefore, our specific prediction was that when moving bright dot stimuli entered a R1-R6's RF, the resulting dynamic input modulation would transiently sharpen R1-R6 output, improving its temporal resolution.

Again, the feasibility of the hypothesis was assessed by analyzing and comparing the resulting biophysical model output to real R1-R6 recordings. The model was implemented in four steps (*Appendix 8—figure 1C*):

i.   Two bright dots, 6.8° apart, crossed a photoreceptor's RF (Δρ = 8.1°) front-to-back at the saccadic speed of 205 °/s.
ii.  The ommatidium lens focused their light onto a rhabdomere tip. After 8 ms delay, the rhabdomere started to move back-to-front as it contracted photomechanically. And now, with this movement, its acceptance angle, Δρ, also narrowed transiently, from 8.1° to 4.0° (*Appendix 8—table 1*). In the model, the further away the rhabdomere moved from its starting position at the focal plane, the more its receptive field narrowed (or skewed).
iii. The resulting dynamic light input was, therefore, a convolution of the two dot intensities and the cell's RF, which narrowed and moved at different speeds in the same direction (front-to-back).
iv.  The light input drove the microvillar photon sampling and refractory QB production, which were summed up over the whole rhabdomere to a macroscopic photoreceptor voltage output.

Importantly, the predicted photoreceptor output showed now two distinct peaks, indicating that the two dots (iv) were resolved neurally. Moreover, the simulations closely resembled the recordings to the similar stimulus (*cf. Appendix 8—figure 1C-D*).

In another test (*Appendix 8—figure 1D*), we estimated the same R1-R6's light input (iii) from its RF (i; through programmed look-up table operations, see *Appendix 6—figure 1*) and used this to predict its output (iv; blue dotted trace). We discovered that the simulated output was indeed similar to the cell's actual recorded output to the same stimulus (black trace), with the timing of their two peaks matching closely. This close dynamic correspondence between the simulations and recordings was robust and reproducible in different tested stimulus conditions (*Appendix 8—figure 2C-D*), meaning that the given model structure likely incorporated the basic biophysical mechanisms that R1-R6s use in encoding moving stimuli.

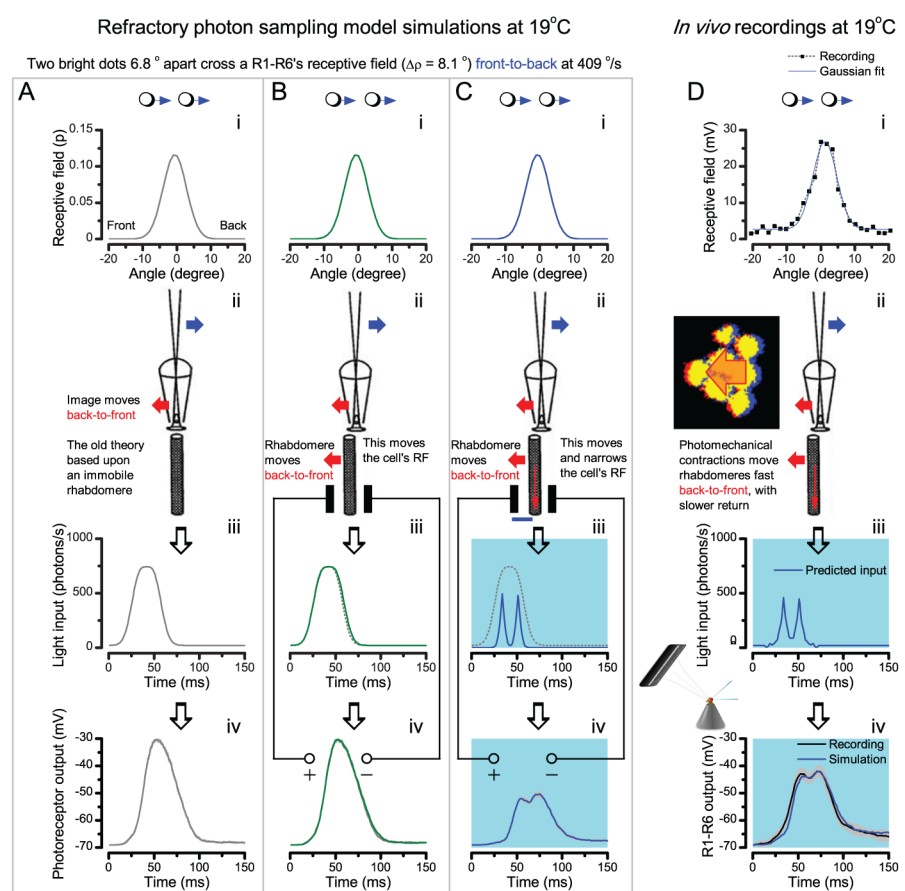

**Appendix 8—figure 2.** The hypothesis predicts R1-R6 output to double-fast moving point-objects. All the parameters of the simulations are fixed to the experimentally measured values, similar to *Appendix 8—figure 1*. The stimulus is the same two bright dots, but this time, they cross the cell's receptive field (RF) double-fast, 409°/s (i). (**A**) Again, if the rhabdomere remains stationary during their flyby (corresponding to the classic theory, ii), their light fuses (iii), and the photoreceptor output cannot distinguish the dots (iv). (**B**) By including the light-induced back-to-front rhabdomere movement (ii) but with the same RF shape, the light input broadens slightly (iii), but cannot separate the two dots. Consequently, the photoreceptor output (iv) shows a slightly narrower single peak than in the previous case in (**A**). (**C**) If, however, the rhabdomere contraction (away from the lens' focal point, ii) moves the RF and actively narrows it (from 8.8° to 4.9°), the light input from the dots is transformed into two intensity spikes (iii), which R1-R6 output separates into two peaks (iv). (**D**) Corresponding intracellular R1-R6 recordings show comparable dynamics to the full model (**C**). These and other simulations and recordings, which all show good correspondence, establish that photoreceptors' RFs must move and narrow dynamically during light stimulation.

DOI: https://doi.org/10.7554/eLife.26117.095

In summary, for the biophysical model to match up the real R1-R6 cells in resolving two bright moving dots, its receptive field must move and transiently narrow from its original size (Appendix 4: $\Delta\rho_{start}$ = 7.00–11.65 °) to the size predicted by the optical waveguide theory ($\Delta\rho_{end}$ = 3.5–5.3°). Moreover, within this dynamic $\Delta\rho$ change range, the resulting model outputs become realistic and robust. Using this modeling approach, we could appropriately predict the real photoreceptors' voltage responses to the different moving dot stimuli (Appendix 6), irrespective of the tested dot speed, direction (front-to-back or back-to-front) and inter-dot distance. For example, by replacing the mean measured rhabdomere displacements with some of the larger values (Appendix 7), the simulations resolved moving dots similarly well to the recordings even at very high stimulus speeds 400–800 °/s.

From the neural coding point of view, this broad agreement between our 'microsaccadic sampling'-hypothesis and the experiments makes it almost certain that photomechanical rhabdomere contractions (Appendix 7) move and narrow R1-R6 photoreceptors' RFs to enhance visual acuity. But from the viewpoint of reducing light-adaptation, these processes seem like by-products of a simple evasive action, which steers the rhabdomere away from pointing directly to the light source (to recover more refractory microvilli; see Appendix 2). Nevertheless, while elementary optics makes it clear why a horizontal rhabdomere motion must move a R1-R6's receptive field in the opposite way, it is harder to see what physical mechanisms could narrow it. We next consider four potential processes within the ommatidium lens system that could just do this.

## What could cause the receptive field narrowing during moving light stimuli?

Four hypothetical mechanisms, together or separately, could explain the required RF narrowing:

1. When a rhabdomere moves back-to-front (*Appendix 8—figure 3*), it moves away from the center axis, which remains fixed because the ommatidium lens system does not move (see Appendix 7, *Appendix 7—figure 7*). Therefore, as the rhabdomere tip moves horizontally (*Video 3*), the light input point-spread function (airy disc) should fall only partly upon it. This may clip or skew the rhabdomere's RF (acceptance angle, $\Delta\rho$), narrowing it.
2. Besides moving rhabdomeres horizontally (Appendix 7), photomechanical photoreceptor contractions also move them 0.5–1.7 µm inwards (*Video 2*; *Appendix 8—figure 3B*). In a dark-adapted state, the rhabdomeres are elongated towards the lens with their tips possibly not being at the focal point. Hence, in this position, the rhabdomeres should collect light from broader angles (Appendix 4, from the brief test pulses: *Appendix 4—figure 4*), and partially recover (re-elongate) before the next pulse comes. But during more continuous light stimulus, their contraction pulls their tips inwards, towards the possible focal point of the lens, which could narrow $\Delta\rho$ towards its theoretical values (3.5–5.3°).
3. Rhabdomere tips are linked by adherence junctions to the cone cells (above them) and pigment cells (at their upper corners) (*Tepass and Harris, 2007*) (*Appendix 8—figure 3C*). When the rhabdomeres contract these connections likely pull the pigment cells above, generating a dynamic aperture (*Video 4*), which moves and possibly tightens, to narrow $\Delta\rho$ (see Appendix 7, *Appendix 7—figure 7*).
4. In dark-adaptation, the waveguide crosstalk between neighboring rhabdomeres could broaden their receptive fields. But light-induced horizontal rhabdomere movement may eliminate the crosstalk between the neighbors, narrowing $\Delta\rho$ towards the theoretical values (3.5–5.3°).

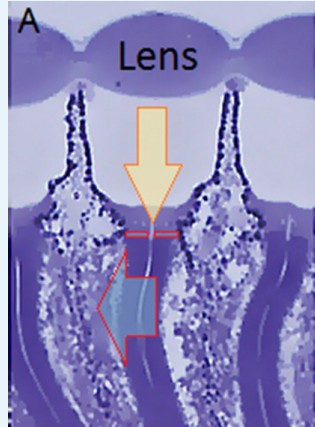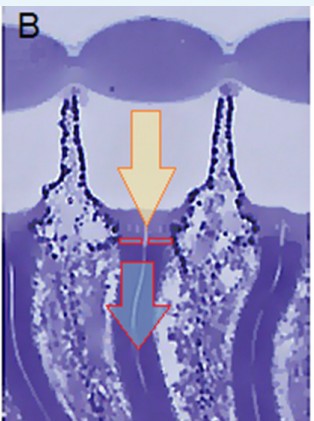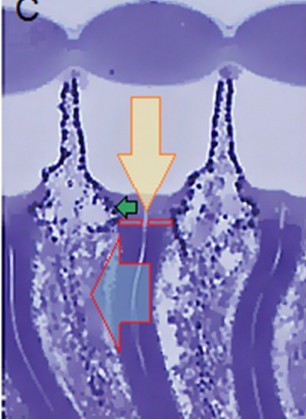

**Appendix 8—figure 3.** How ommatidium dimensions change during light-induced photoreceptor contractions. The three images show the same transverse section of three *Drosophila*

ommatidia, with their rhabdomeres appearing as six darker curvy strips. The rhabdomeres contract (cyan arrow) to light (yellow arrow, as focused by the lens), moving both (**A**) horizontally (back-to-front) and (**B**) vertically (down arrow), see Appendix 7. Because the rhabdomeres are connected to the surrounding structures by adherence junctions (*Tepass and Harris, 2007*) (red boxes highlight the transition areas), their contraction induces (**C**) moving and possibly narrowing of the aperture (green horizontal arrow) formed by the cone and pigment cells, which are directly above the rhabdomeres (lighter blue areas). Ultimately, the curvature of the lens and focal distance might also change slightly. Image modified from (*Gonzalez-Bellido et al., 2011*).

DOI: https://doi.org/10.7554/eLife.26117.096

## Specific predictions of our new hypothesis and their experimental validations

The microsaccadic sampling-hypothesis, as implemented by our biophysically realistic photoreceptor model (Appendix 1) with combined photomechanical rhabdomere dynamics, makes important predictions about the coding benefits of moving and narrowing R1-R6 receptive fields that can be tested experimentally.

The first prediction is that, for a given stimulus or saccadic velocity, R1-R6 output to a back-to-front moving bright dot should appear before the output to a front-to-back moving dot. This is because the back-to-front moving dot should enter and exit a contracting photoreceptor's front-back moving receptive field earlier; whereas the dot moving in the opposite direction should stay marginally longer inside its RF. *Appendix 8—figure 4* compares the theoretical predictions (output simulations) of the same three models as earlier (**A**, **B** and **C**; their details are above) and corresponding exemplary intracellular recordings (**D**) for these two stimuli.

Both the full model simulations (**C**) and many recordings (**D**) indicate that this prediction is indeed what happens for a R1-R6 with a symmetrical RF (i). The responses (red traces) to back-to-front moving dots rise and decay faster than the responses (blue traces) to front-to-back moving dots. Similar response dynamics of another intracellular recording series from another R1-R6 are highlighted in Appendix 6 (*Appendix 6—figure 4*). Notice, however, that these results are explicitly true for symmetrical receptive fields. If, on the other hand, a R1-R6's RF was asymmetrical - say, profoundly skewed towards the front of the eye, then its response to the front-to-back moving dot might, in fact, rise earlier, or there could be little difference between the responses. Thus, it is the mathematical relationship between the scale of RF asymmetricity and the scale of the rhabdomere back-to-front movement, which ultimately sets whether a front-to-back or back-to-front stimulus would win. Notice also that dynamic photomechanical rhabdomere movements and eye muscle activity make RF recordings difficult to perform, and consequently, experimental inaccuracies and limitations can influence the results. Thus, some of the natural variations in the recordings may result from imprecise stimulation control. For example, imperfect positioning of a 25 light-point stimulus array - either off-center of a cell's receptive field or if not aligned perfectly parallel in respect to the eye's back-to-front axis - could bias a recording (more about this variation and the cell numbers in Appendix 6).

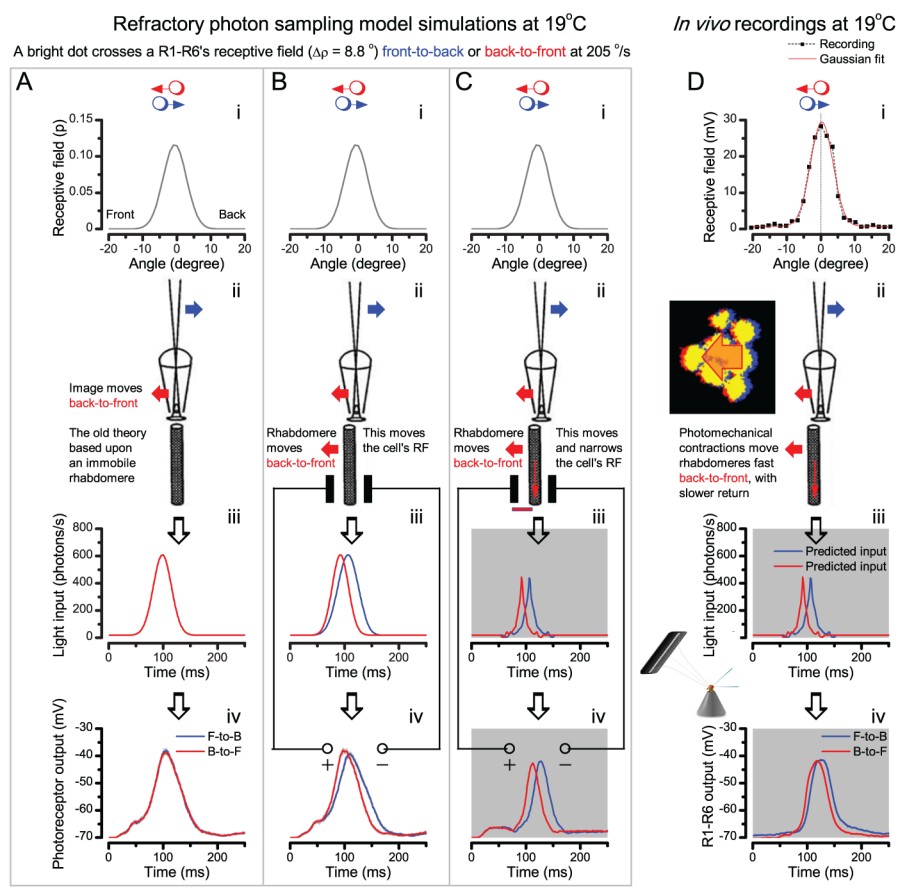

**Appendix 8—figure 4.** The new hypothesis predicts and the recordings show that R1-R6 output rises and decays earlier to a back-to-front moving bright dot than to a front-to-back moving dot of equal velocity. (**A**) A classic model with (i) a symmetrical receptive field and (ii) an immobile rhabdomere leads to identical (iii) light inputs and virtually identical (iv) photoreceptor outputs (minute differences result from stochastic photon sampling), respectively, for the dots moving in the opposite directions. (**B**) A model with (ii) a moving rhabdomere generates both dynamic (iii) light input and (iv) photoreceptor output, which rise and decay earlier for the back-to-front moving dot (red traces) than the front-to-back moving dot (blue traces). (**C**) Our full model with rhabdomere contraction dynamics that move and narrow its receptive field. This makes the light input and photoreceptor output rise and decay faster than in the other two models, with the back-to-front waveforms leading the front-to-back counterparts. (**D**) The intracellular responses of a R1-R6 photoreceptor to the given two dot stimuli, as recorded *in vivo*, show similar dynamics to the full model in (**C**) with its back-to-front signals (red) leading the front-to-back signals (blue).

DOI: https://doi.org/10.7554/eLife.26117.097

Interestingly, for both opposing object directions, the narrowing of a R1-R6's receptive field (*Appendix 8—figure 4C-D*) makes its voltage responses briefer than what would be the case without this process (A-B). Therefore, the resulting faster temporal photoreceptor output dynamics combat the effects of motion blur, supporting the theoretical and experimental results in Appendix 6 (*Appendix 6—figure 8*).

The second prediction is that, for high (saccadic) speeds, R1-R6s resolve two front-to-back moving bright dots better than when these move back-to-front (*Appendix 8—figure 5*). Thus, the normal back-to-front rhabdomere movement should improve the fly eye's spatiotemporal resolution during fast forward locomotion or object motion. This is indeed what we saw in the full model simulations (*Appendix 8—figure 5D*) and in some stable experimental recordings (*Appendix 8—figure 5E*). Because of the back-to-front rhabdomere

movement, which was inverted by the ommatidium lens, the light input (*Appendix 8—figure 5C*) for two front-to-back moving dots were separated further apart as intensity spikes (blue trace) than that for the opposite motion (red). Consequently, the resolvability in the resulting R1-R6 output (*Appendix 8—figure 5D*) also became greater for front-to-back moving stimuli. This of course further meant that in comparison to the case of the immobile rhabdomere with the same narrow acceptance angle ($\Delta\rho = 4°$), the neural resolvability of the back-to-front moving rhabdomere, which kept the same stimulus longer within its receptive field, would be still better.

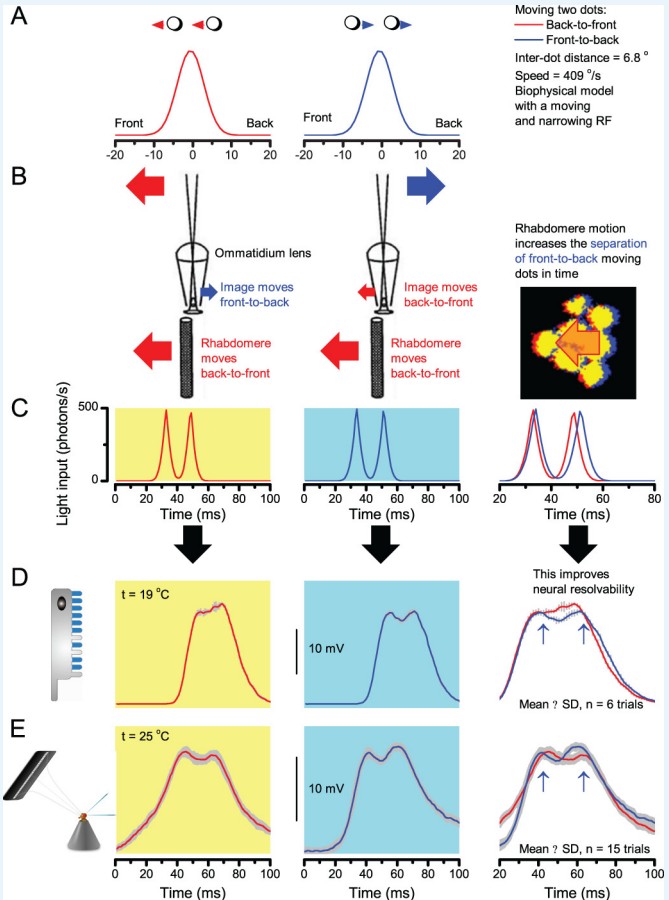

**Appendix 8—figure 5.** Fast rhabdomere movements are predicted to improve the resolvability of fast front-to-back moving objects the most. (**A**) Two bright dots, 6.8° apart, cross a photoreceptor's receptive field (RF; $\Delta\rho_{start} = 8.1°$) either in back-to-front (red, left) or front-to-back (blue, middle) at 409 °/s (i). (**B**) The new photoreceptor model translated the light-induced back-to-front rhabdomere motion into concurrent RF narrowing ($\Delta\rho_{end} = 4.0°$) and front-to-back movement (as reversed by the ommatidium lens). (**C**) Consequently, the light input from the dots was transformed into two intensity spikes. These spikes were further apart in time for the front-to-back moving dots and for the opposing stimuli. (**D**) The two peaks in in the corresponding model-predicted R1-R6 output (blue) for front-to-back moving dots (highlighted by arrows) indicated that the dots were neurally detectable. In contrast, the predicted R1-R6 output for back-to-front moving dots (red) failed to separate these two point-objects at 19°C. (**E**) An example of intracellular recordings from one R1-R6 photoreceptor to the same two stimuli at 25°C. This cell's voltage responses also resolved the front-to-back moving dots better than their back-to-front moving counterparts. These and other comparable simulations and recordings suggest that microsaccadic rhabdomere movements improve the neural resolution (and representation) of fast moving visual objects.

DOI: https://doi.org/10.7554/eLife.26117.098

## Photoreceptors resolve moving object up to high body-saccadic speeds

We next used the complete photoreceptor model (above) to estimate how well a typical R1-R6 photoreceptor can resolve two bright dots, which are less than the average acceptance angle (Δρ) apart, moving together at increasingly fast (saccadic) velocities. This time, however, the simulations were performed at the flies' preferred temperature (*Sayeed and Benzer, 1996*) of 25°C, rather than at 19°C (as in the previous data). We have shown earlier that warming accelerates R1-R6s' phototransduction dynamics and refractory microvilli recovery (*Juusola and Hardie, 2001b*; *Song et al., 2012*; *Song and Juusola, 2014*). Because the resulting increase in their sample (quantum bump) rate changes improves information transfer rate (*Juusola and Hardie, 2001b*; *Song et al., 2012*; *Song and Juusola, 2014*), we expect here that their output to moving dots should also show improved resolvability. We later compare these estimates to the measured head/body-saccade speeds of freely locomoting *Drosophila* (*Fry et al., 2003*; *Geurten et al., 2014*).

*Appendix 8—figure 6* shows the simulated light input (C) and photoreceptor output (D) to the dots (6.8° apart) crossing a R1-R6 photoreceptor's RF ($\Delta\rho_{start}$ = 8.1°, $\Delta\rho_{end}$ = 4.0°) in front-to-back direction at 205 (left), 409, 600, 700, 800 and 1000°/s (right). We found that the slower the dots moved (A), the better the predicted photoreceptor output distinguished them as two separate events (peaks) in time (D). Remarkably, the output resolved the dots at speeds until ~600 °/s (cyan background). As a neural threshold for representing sub-RF details, this image speed is indeed very high. It means that *Drosophila* should lose little neural image detail during its normal saccadic body rotations during walking; the measured rotation speed range is ~200–800 °/s (*Geurten et al., 2014*). At the higher speeds, the two response peaks fused into one. Notice that because the predicted light input (as modulated by the rhabdomere contraction) resolved the dots even at 1,000 °/s (*Appendix 8—figure 6C*), the resolution limit in the photoreceptor output (*Appendix 8—figure 6D*) resulted from its intrinsic signal integration time limit; for the given (experimentally measured) quantum bump size, latency and refractoriness distributions (*Juusola and Hardie, 2001a*; *Song et al., 2012*; *Song and Juusola, 2014*). Notice also how the response amplitude and half-width were reduced more the faster the dots crossed the receptive field. Thus, the rhabdomere then simply captured and integrated fewer photons in a given time unit.

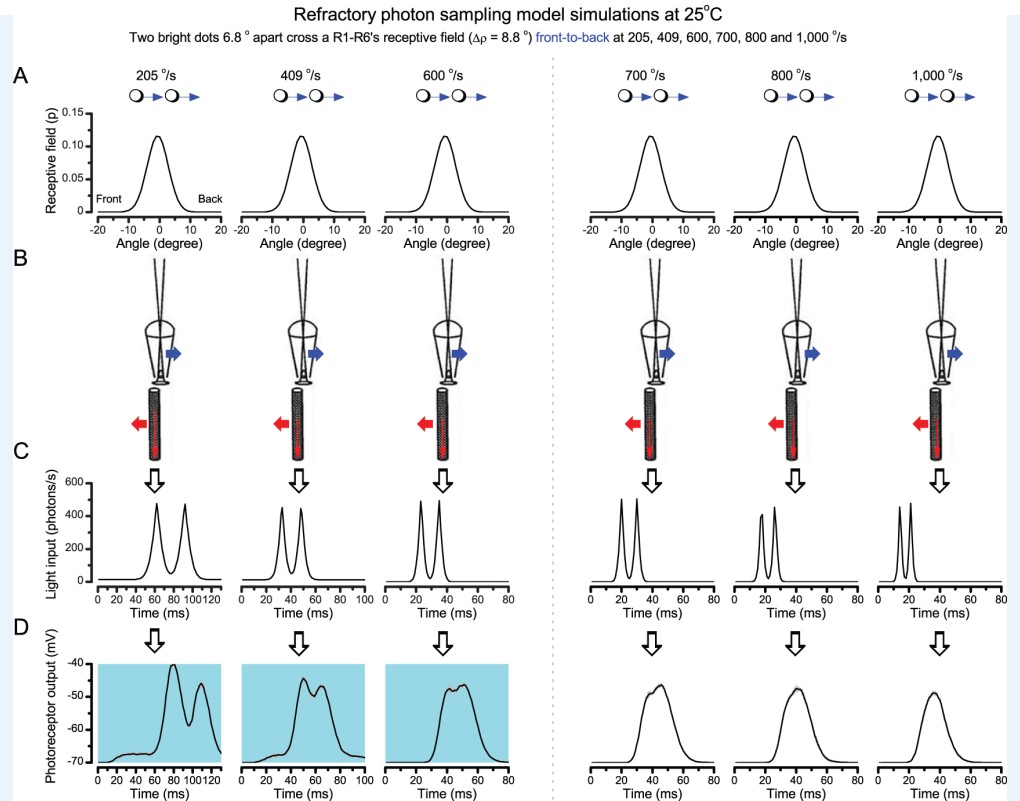

**Appendix 8—figure 6.** The hypothesis predicts that at the preferred temperature (25°C) *Drosophila* R1-R6 photoreceptors can distinguish two bright dots 6.8 ° apart travelling together at ~ 600 °/s. (**A**) The starting and ending receptive field size (RF; $\Delta\rho_{start}$ = 8.1°; $\Delta\rho_{end}$ = 4.0°) in all the simulation was the same. (**B**) in every case, the rhabdomere contracted, moving and narrowing the RF. (**C**) This translated light input into two clear sharp peaks. (**D**) Photoreceptor output separated the dots as two peaks at velocities until above 600 °/s (left, cyan background), indicating that they were neurally resolvable. At 700 °/s or higher speeds (right), the dots were not resolved as the outputs had only a single peak. Mean ± SD shown, n = 6 repeated stochastic simulations to each stimulus.

DOI: https://doi.org/10.7554/eLife.26117.099

## Prediction that R1-R6s encode hyperacute images in space-time

Given that R1-R6 output shows unexpectedly high acuity even at very fast saccadic velocities (**Appendix 8—figure 6**), we asked how well these cells could in fact resolve slower moving point-objects. Could a normal R1-R6 encode image details, which were less than the average interommatidial angle apart? That is, could *Drosophila* actually see the world in finer resolution than their compound eyes maximum sensor (or pixel) spacing, which is the limit predicted by the classic optical theory (**Land, 1997**)?

We tested this hypothesis theoretically by using the full 'microsaccadic sampling'-model (**Figure 9**). In these simulations, two bright dots were now either 1°, 2°, 3° or 4° apart. Thus the dot spacing was less than the *Drosophila* compound eye' average interommatidial angle ($\Delta\varphi$ ~4.5-5°). The dots were then moved across a R1-R6 photoreceptor's RF ($\Delta\rho_{start}$ = 8.1°; $\Delta\rho_{end}$ = 4.0°) at different speeds, ranging from 5 (slow gaze fixation) to 400 °/s (very fast body saccade).

The model predicted that a typical R1-R6 photoreceptor would resolve the dots in hyperacute details (**Figure 9**) over a broad range of velocities. These theoretical predictions were broadly confirmed by intracellular recordings (**Figure 9—figure supplement 1**), whilst flight simulator experiments verified that *Drosophila* indeed have hyperacute vision (**Figure 10**).

## Refractory sampling improves hyperacute motion vision

To quantify how refractory photon sampling contributes to the sharpening of the macroscopic responses during moving hyperacute 2-dot stimuli, we further compared the outputs of two different photoreceptor models for the same stimuli (*Appendix 8—figure 7*), with the brightness as in *Appendix 8—figure 1*. For both cases, the resulting dynamic light input – reflecting the narrowing and moving receptive field, as caused by photomechanical rhabdomere contraction - was the same, but the models' photon sampling differed. The test model had 30,000 stochastically operating refractory microvilli and the control was a comparable mock model, which converted every incoming photon into a quantum bump.

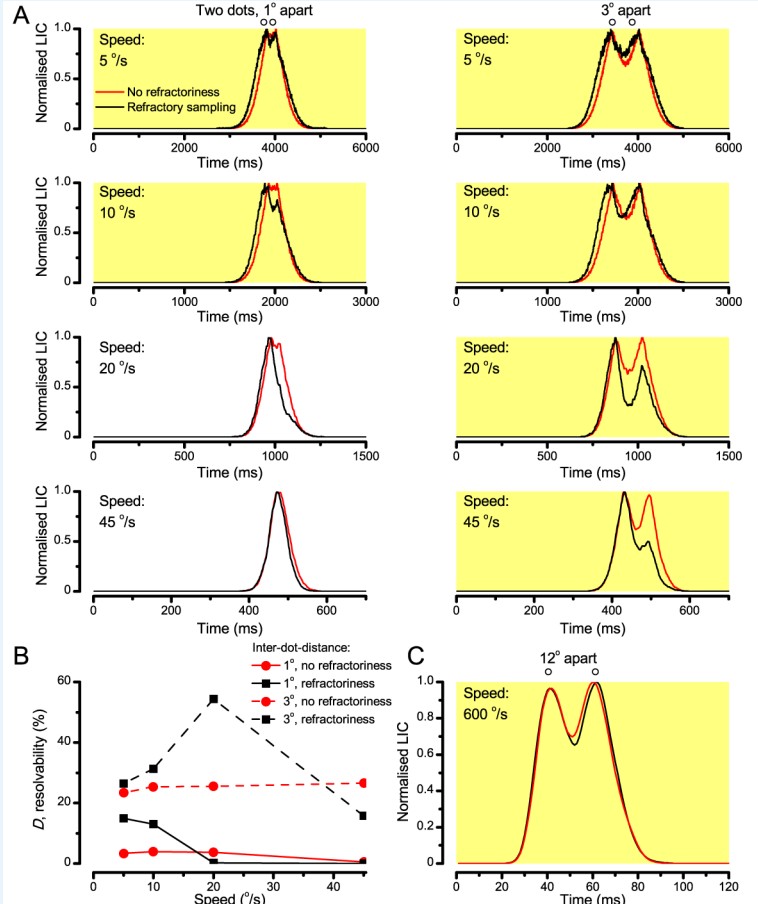

**Appendix 8—figure 7.** Refractory sampling enhances neural resolution for different aspects of hyperacute images. (**A**) Examples of simulated macroscopic light-induced current responses (normalized LICs) of two different photoreceptor models. Both models have 30,000 microvilli. In the first model (red), every photon causes a quantum bump; hence, its photon sampling has no refractoriness. In the second model (black), photon sampling is refractory. Both models are stimulated with the same moving two bright dots, with their actual light inputs being first modulated by photomechanical rhabdomere contractions (following the microsaccadic sampling hypothesis). Based on the tests with different velocities and inter-dot-distances, refractoriness consistently causes a phase lead in LIC responses. (**B**) Refractoriness improves response resolution for hyperacute stimuli (inter-dot-distance <4.5°) at slow velocities (≤20 °/s). The resolvability, *D*, of the recordings and simulations, was determined by Raleigh criterion (*cf. Figure 7C*). (**C**) An example of how refractoriness can enhance response resolution for larger stimulus separations at high saccadic velocities. The difference from the lower peak to the trough is larger in the black trace (refractory photon sampling) than in the red trace (complete photon sampling).

DOI: https://doi.org/10.7554/eLife.26117.100

Firstly, we found that the refractory sampling consistently improved the response resolution beyond that of the control for bright hyperacute dots (inter-dot-distance <4.5°) at slow velocities ($\leq$20 °/s) (*Appendix 8—figure 7A–B*). Thus, refractoriness enhances neural image resolution during slow self-motion or when high-resolution objects move slowly. Its effects were particularly well seen in the differing rising phases of the normalized light-induced current (LIC) responses of the two models. With the slow moving stimuli, the rising responses of the refractory sampling model (black traces) always led those of the non-refractory model (red).

Additionally, in other trials, we found that when the dots were more than the average acceptance angle apart ($\Delta\rho_{start} \geq 9.5°$) but moved across a R1-R6 photoreceptor's receptive field at very fast saccadic speeds ($\geq$400 °/s), the refractoriness often enhanced neural image resolution beyond that of the controls (*Appendix 8—figure 7C*). This observation is consistent with our previous finding that stochastic refractoriness of light-activated microvilli exerts a memory of past events in bump integration (*Song and Juusola, 2014*). This memory accentuates certain stimulus features relative to others so that a R1-R6 samples information from different inter-dot-distance/speed-combinations differently. Ultimately, it could well be that the real R1-R6s' refractory photon sampling statistics are adapted (through their visual lifestyle) to the statistics of moving high-resolution natural images. Of course, here, the used 2-dot stimuli and the models, which were isolated from the lamina network feedbacks, are too simple to fully explore such statics and the intricacies of hyperacute *Drosophila* vision.

## Horizontal vs. vertical motion hyperacuity

The two most important biophysical factors of *Drosophila* photoreceptors, which lead to motion hyperacuity - whereupon space is encoded in time - are their sufficiently narrow receptive fields ($\Delta\rho_{end} < 5°$) and refractory photon sampling (quantum bump dynamics). Therefore, theoretically, as R1-R6s' receptive fields should narrow when an object crosses them, irrespective of its motion direction; *Drosophila* is expected to have hyperacute vision for both horizontal and vertical motion.

However, as we considered in *Appendix 8—figure 4*, R1-R6 photoreceptors' neural resolvability should be the best for front-to-back moving objects. In this case, due to their back-to-front sweeping rhabdomeres, R1-R6s' receptive field can broaden slightly in horizontal direction. This dynamic may in part contribute to the curious observation that L2 monopolar cell terminals' receptive fields (in the medulla) are anisotropic, elongated in horizontal (yaw) direction and narrower in vertical (pitch) direction, as measured by calcium-imaging experiments (*Freifeld et al., 2013*).

## Mirror symmetric contractions may also provide navigational heading signal

As we showed in Appendix 7, light increments evoke mirror symmetric back-to-front rhabdomere movements in the left and right fly eye. Interestingly, during fast saccadic body rotations, this phenomenon could surprisingly help a fly's visual orientation (*Appendix 8—figure 8*). The microsaccadic sampling-hypothesis predicts that image rotation causes a phasic difference in photoreceptor outputs between the left and the right eye, with the signals always arriving slightly faster from the eye, towards which the fly rotates. Because this difference depends upon the rotation speed, it could be used for signaling changes in the fly's heading direction or to improve visual navigation. For example, when flying across more homogenous surroundings, such as an open field with few distinctive visual landmarks, the central brain could use saccadic turns to recalibrate the fly's head-direction in its internal world map near instantaneously; matching the intended direction to the new direction, as pointed by the global phase difference between the left and right eye signals.

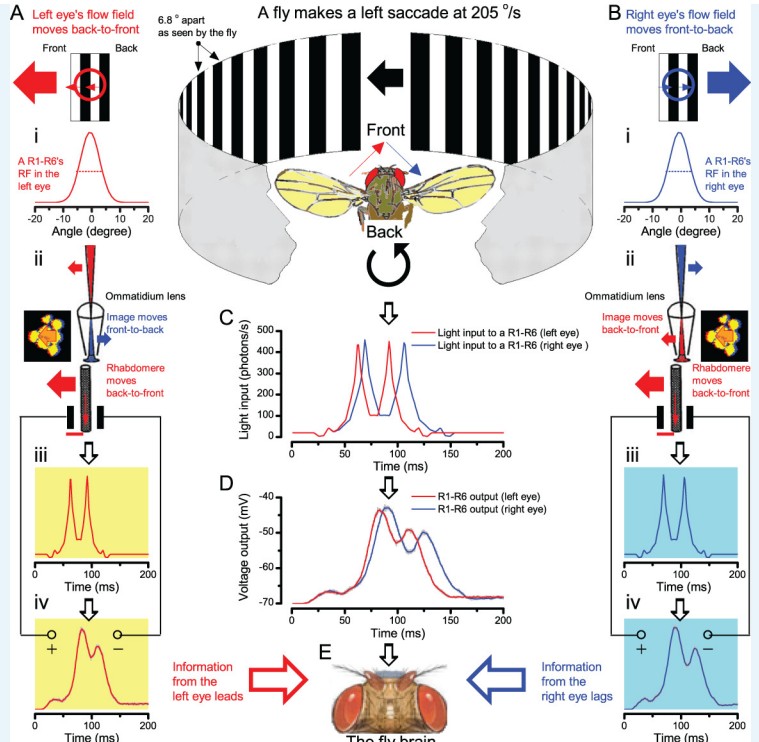

**Appendix 8—figure 8.** During a left saccade, signals from the left-eye lead the signals from the right-eye. Because incoming light, in both the left and right eyes, contracts their rhabdomeres in the back-to-front direction, photoreceptors can transmit information about the field rotation or the fly's orientation changes to the brain. Specifically, the output from the eye towards which the fly turns phase leads, and thus neurally signals rotation direction. Here we illustrate the underlying mechanism by using a brief 205 °/s left saccade (in a world of 6.8° black-and-white gratings), as an example. (**A**) During the left saccade, the flow field facing the left eye moves back-to-front (red arrow) across its photoreceptors' receptive fields (i). The image of the moving flow field is inverted by the ommatidium lens, and so moves front-to-back (blue arrow) while being sampled by the rhabdomere (ii). Light contracts the rhabdomeres in back-to-front direction (red arrow). With the projected image and the photon-sampler (rhabdomere) moving against each other, the light input from two bright bars (iii) becomes narrowed and accelerated. Accordingly, photoreceptor output shows two prominent peaks, in which distance to each other is compacted in time (iv). (**B**) During the left saccade, the flow field facing the right eye moves front-to-back (blue arrow) across its photoreceptors' receptive fields (i). Again, the ommatidium lens inverts the image, which now moves back-to-front (red arrow) over the rhabdomere (ii), which contracts in the same direction (red arrow). With the projected image and the rhabdomere moving together, the light input from two bright bars (iii) excites the photoreceptor longer, in which output shows two prominent peaks elongated in time (iv). (**C**) The light input to the left eye photoreceptors (red trace) rise and decay faster than the corresponding light input (blue trace) to the right eye photoreceptors. (**D**) R1-R6 output in the left eye (red trace) is faster and briefer than the photoreceptor output (blue trace) in the right eye. (**E**) Consequently, by correlating the outputs from the left and right eye photoreceptors in time, the fly brain can obtain information about the directional changes in the visual space, in respect to its head orientation.

DOI: https://doi.org/10.7554/eLife.26117.101

## Appendix 9

DOI: https://doi.org/10.7554/eLife.26117.102

# Microsaccadic rhabdomere movements and R1-R6s' information capture

## Overview

This appendix describes how photomechanical rhabdomere movements affect *Drosophila* R1-R6 photoreceptors' information capture, compares microsaccadic information sampling of dark and bright objects and provides useful background information about the experimental and theoretical results presented in *Figures 1–9*.

In this appendix:

- We explain why and how rhabdomere movement noise influences R1-R6 output mostly at low frequencies, causing relatively little information loss.
- We test and compare how microsaccadic sampling affects encoding of bursty bright or *dark* image contrasts, using intracellular recordings.
- We further examine how well R1-R6s encode two *dark* moving point-objects (dots), and compare these recordings with those to corresponding bright moving dots.
- The results confirm that *Drosophila* R1-R6 photoreceptors resolve both bright and dark moving hyperacute patterns (<interommatidial angle, Δφ ~4.5-5°), and can respond to bright or dark point-objects, which are less than their acceptance angles (Δρ ~ 9.5°) apart, even at high saccadic velocities. Thus, microsaccadic sampling hypothesis provides a robust functional explanation for *Drosophila*'s hyperacute vision (Appendix 10).
- The results support the idea that a fly's optimal viewing strategy would involve fixating on dark features, which recover refractory microvilli, and then shifting gaze to bright features, to maximize information capture. This of course would require that it can neurally shift attention (across the eyes) to visual objects of interest, as some results suggest (*Tang et al., 2004*; *van Swinderen, 2007*; *Tang and Juusola, 2010*; *Paulk et al., 2014*; *Seelig and Jayaraman, 2015*).

## Microsaccades accentuate high-frequency resolution but generate low-frequency noise

Photomechanical rhabdomere contractions (microsaccades; Appendix 7) can maximally shift the center of a R1-R6 photoreceptor's receptive field by ~5°, and through this self-induced light input modulation (*Figure 8*) cause variations (noise) in its voltage output. Such 'rhabdomere movement noise' is inevitable if the photoreceptor signal is classified and estimated as the average of the repeated responses, just as we did in the performance calculations (*e.g. Figure 2*).

The condition itself bears resemblance to taking snapshots of a stationary scene from different positions and averaging these. The mean image shows an obvious smear, even if the positions were only a fraction of a photoreceptor's receptive field ('pixel') apart. However, during repeated light stimulation, the rhabdomere movements adapt rapidly (*Figure 8E*, *Figure 2—figure supplement 2*), with this noise affecting less the subsequent performance estimates. Thus, when quantifying the photoreceptor performance to repeated light intensity time series stimulation (*e.g. Figure 2*), we removed the first 3–10 responses, in which these movements had the largest effect. In the recordings, this noise would then be rather constant across the collected responses.

Rhabdomere movement noise is missing from the simulated R1-R6 output (*Figures 8–9*). Therefore, given that the stochastic photoreceptor model's transduction noise is adapted to the mean light intensity (Appendix 2), similarly to that of the recordings (*Figure 2—figure supplement 2A–B*), we could isolate it as the difference between the recorded and simulated R1-R6 output (*Figure 2—figure supplement 2C–D*). The analysis suggests that rhabdomere

movement noise affects mostly low-frequency R1-R6 output, reducing its signal-to-noise ratio, and importantly, it effectively matches the rhabdomere jitter in high-speed video footage (*Figure 2—figure supplement 2E,F*).

The contractions deviate the rhabdomere from directly facing the light source, reducing photon influx especially during bright stimulation (*Figure 8C–D*). Such evasive action, however, has surprisingly little detrimental effect on the R1-R6s' information transfer. This is because bright stimulation ($>10^6$ photons/s) contains too much light to be transduced by 30,000 microvilli into quantum bumps, and R1-R6s actively screen off excess photons to maximize information in their voltage output. In Appendix 2, we showed that R1-R6s' photomechanical adaptations (the contractions and intracellular pupil mechanism) are jointly optimized with refractory sampling (to modulate quantum efficiency [*Song et al., 2012*; *Song and Juusola, 2014*]) for maximal information intake at different stimulus conditions. Moreover, owing to bump adaptation and microvilli refractoriness, which accentuate light fast changes in macroscopic voltage output (*Song et al., 2012*; *Song and Juusola, 2014*; *Juusola et al., 2015*) (e.g. (*Song and Juusola, 2014*): *Figures 9–10*, improving high-frequency resolution), slower signals in return become compressed. Importantly, this low-frequency response range (<~10 Hz), where also rhabdomere movement noise mostly resides (*Figure 2—figure supplement 2D*), carries relatively little information (*Song and Juusola, 2014*) (about the behaviorally more relevant faster changes in the world).

## Are saccades and fixations optimized to microsaccadic sampling?

Recordings and simulations (*Figures 1–2*) showed unequivocally that R1-R6s information capture is maximized for high-frequency saccade-like bursts with dark fixation intervals. This suggests that the optimal daytime viewing strategy would be to fixate on dark features in the visual scenes, as this recovers refractory microvilli, and then rapidly move gaze to over bright features, as this increases quantum bump (sample) rate changes and thus information capture in time. And indeed, in behavioral experiments, *Drosophila* readily fixates on and track dark objects, such as vertical bars (*Götz, 1980*; *Tang and Juusola, 2010*; *Bahl et al., 2013*). But because the fly eye photoreceptors sample a continuous panoramic view of the world, many of them - at any one time - would unavoidably face bright contrasts, which reduce their sensitivity even when their photomechanical adaptations (Appendix 2) operate maximally. We therefore also tested by intracellular R1-R6 recordings how encoding of dynamic bright or dark contrast changes may differ.

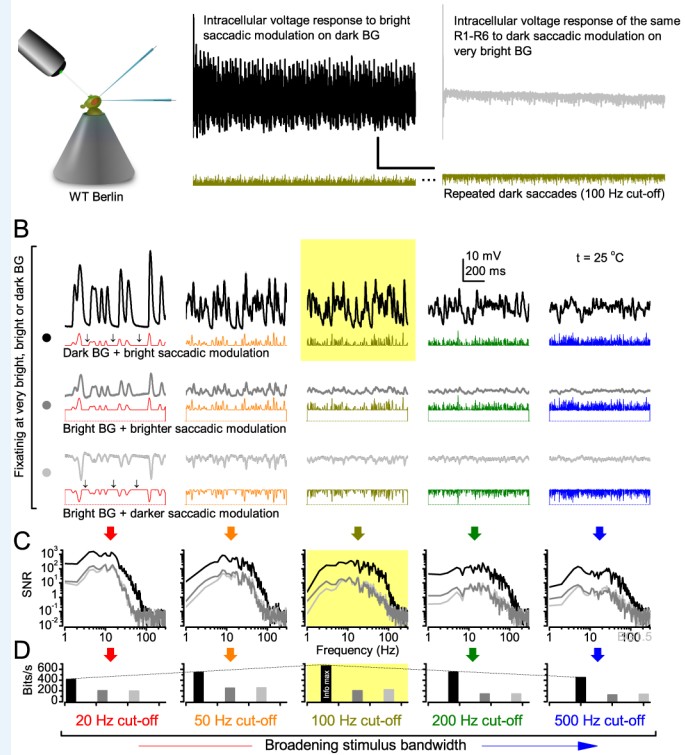

**Appendix 9—figure 1.** Optimal saccadic viewing strategy for maximizing R1-R6 photoreceptors' information capture requires fixating on darker image features. (**A**) Intracellular responses of a R1-R6 to repeated saccadic bright (left) and dark (right) bursts. (**B**) Output to bright saccadic bursts was always more vigorous when mixed with dark fixation periods (BG, above) than with bright periods (middle), irrespective of the stimulus frequency distribution. When the same R1-R6s were fixated (light-adapted) on a bright background (below), they still responded well to dark saccadic bursts (of the inverse waveforms), but the amplitude range of these responses was less than to bright bursts (above, dark BG). (**C**) Corresponding R1-R6 output signal-to-noise ratios were the highest for the bright-saccadic-burst/dark-fixation-period stimuli (black traces). Signal-to-noise ratios were lower but alike for the brighter saccades with bright fixation periods (grey) and the dark saccades with bright fixation periods (light grey). (**D**) Information transfer rates confirmed the global maximum for 100 Hz bright saccade-like bursts with dark fixations (black bars; *cf. Figure 2C*). Whilst bright fixation periods reduced information (with more microvilli becoming refractory), the bit rates for bright (grey) and dark (light grey) saccadic modulation, of equal but opposite contrasts, were similar.

DOI: https://doi.org/10.7554/eLife.26117.103

*Appendix 9—figure 1A* shows examples of consecutive responses recorded from a R1-R6 to repeated high-frequency saccadic bright or dark contrast bursts (with 100 Hz cut-off). Expectedly, following the stochastic adaptive visual information sampling theory (**Appendixes 1–2**), the responses to bright (positive) contrasts after dark 'fixation' periods (background, BG) were significantly larger than those to dark (negative) contrasts after bright 'fixation' periods. Notice, however, that although in terms of absolute light intensity (*I*) changes (or peak-to-peak amplitude modulation) the two stimuli were equal, the negative bursts had smaller absolute contrast values ($C = \Delta I/I$) than the positive ones. This is because the darkening bursts reached their absolute contrast maximum (darkness, −1) only occasionally, whereas the corresponding brightening bursts reached higher absolute contrasts (>1; owing to their lower mean light intensity).

To counter this bias, we used three sets of stimuli to examine individual R1-R6s' response dynamics to both positive and negative bursts of equal or different contrast distributions (*Appendix 9—figure 1B*). First (top row), the photoreceptors were stimulated with positive contrast bursts (peak-to-peak modulation = 1 intensity units), which contained high-frequency

saccade-like events on a dark background, having different cut-off frequencies (from 20 to 500 Hz, as in *Figure 1*). In the second set (middle), the same stimuli were superimposed on a bright background (one unit). Finally (bottom), the stimulus modulation was inverted (to negative contrast bursts) and superimposed on the same bright background (one unit). Thus, for the second and third sets, the stimulus contrasts were equal but opposite.

We found (again in agreement with the theory) that whilst the responses were always the largest to positive contrast bursts on a dark background, the corresponding responses to the positive and negative contrast bursts on bright background, although smaller, were about the same size (*Appendix 9—figure 1B*). Since we further know that the larger responses contain more quantum bumps (*Figure 2*), with the average bumps light-adapting to about the same size (*e.g. Figure 2—figure supplement 2A–B*), the responses' signal-to-noise ratios (*Appendix 9—figure 1A*) and information transfer rates (*Appendix 9—figure 1D*) were predictable. R1-R6s' signaling performance was the greatest to the larger positive contrast bursts on a dark background (black), and more than halved to the smaller corresponding positive (grey) and negative (light grey) contrast bursts over the test bandwidths. Notably, the responses to the opposite but equal positive and negative contrast bursts carried effectively equal information contents, underscoring the importance of contrast invariance at the primary visual encoding stage (*Juusola, 1993*; *Song et al., 2012*; *Juusola et al., 2015*).

Therefore, given the fast speed of adaptation (microvilli refractoriness and dynamic quantum bump size modulation) and its photomechanical counterbalancing (**Appendixes 2, 7**), the sensitivity of neighboring photoreceptors across the eyes can differ greatly at any one moment, depending upon whether they face dark or bright contrasts. This realization also implies that when a fly moves its gaze in saccades, the dark and bright spatial contrast differences in the world should be automatically translated into large temporal contrast changes between the neighboring retinotopic image pixels (neuro-ommatidia). Enhancement of local differences and similarities in neural images by spatiotemporal synaptic (*Zheng et al., 2006*; *Freifeld et al., 2013*) and gap-junctional (*Wardill et al., 2012*) co-processing (including network adaptation [*Nikolaev et al., 2009*; *Zheng et al., 2009*]) across the first optic ganglia, the lamina and medulla, should further improve object detection and fly vision.

## Microsaccadic sampling of bright or dark moving point-objects

Natural scenes are rich with dark features: shadows, object boundaries, surfaces of lesser reflectance etc., which have shaped visual circuit functions, perception and behaviors (*Barlow H, 1961*; *Yeh et al., 2009*; *Joesch et al., 2010*; *Ratliff et al., 2010*; *Kremkow et al., 2014*; *Song and Juusola, 2014*). Consequently, a fly's self-motion generates both dark and bright moving features travelling across its eyes. We have shown that R1-R6s can resolve fast-moving and hyperacute bright dots (*Figures 7–9* and *Figure 9—figure supplement 1*). But how well can these cells resolve dark moving dots?

We studied this question with the 25 light-point stimulus array (explained in Appendix 4 and Appendix 6). As before, the stimulus array was first carefully placed at the studied R1-R6's receptive field center, but this time, all the light-points were switched on, and we generated two travelling dark points of specified speeds and interdistances. As during these experiments the cells were light-adapted (depolarized) by the lit stimulus array, the two moving dark dots evoked hyperpolarizing responses.

*Appendix 9—figure 2A* shows R1-R6s' characteristic responses to two dark (black traces) and bright (red) dots of specific speeds (102, 205 and 409 °/s) and interdistances (3.4, 5.1 and 6.8° apart), recorded from the same cells. In all these cases, the hyperpolarizing responses resolved the two dots, generating two troughs separated by a peak, but these responses were considerably smaller than those to the corresponding bright dots. However, when normalized, the photoreceptors' relative neural resolvability of the dark dots matched that of the bright dots (*Appendix 9—figure 2B*).

Thus, in concordance with the behavioral experiments (*Figure 10*), these and other intracellular recordings established that *Drosophila* R1-R6 photoreceptors see both bright and dark moving hyperacute patterns (<interommatidial angle, Δφ ~4.5-5°), and can resolve point-

objects, which are less than their acceptance angles ($\Delta\rho \sim 9.5°$) apart, even at high saccadic velocities.

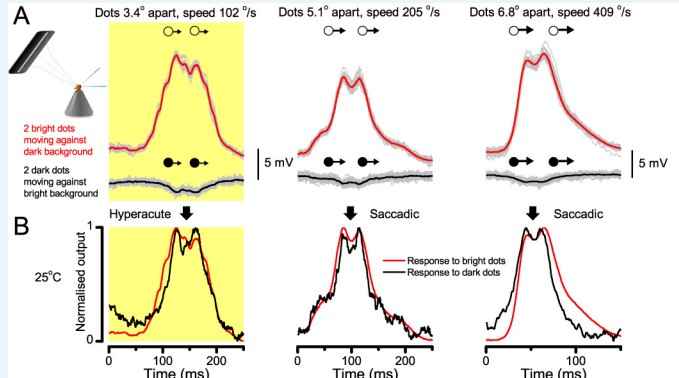

**Appendix 9—figure 2.** Comparing R1-R6s' intracellular responses to hyperacute or saccadic bright or dark moving dots. (**A**) Mean (thick traces) and individual responses (thin) of three individual R1-R6s to two bright (red, above) or dark (black, below) dots (of different inter-dot-distances and speeds), crossing the cells' receptive fields in front-to-back direction. Responses to the hyperacute dots (3.4° inter-dot-distance) are shown left; responses to the saccadic speed stimuli in the middle (205 °/s) and right (409 °/s). In all cases, these outputs resolved the two dots, but predictably the responses were always smaller to the dark dots at the bright background than to the bright dots at the dark background (as more microvilli should remain refractory when adapting to bright background). (**B**) The normalized and sign-inverted R1-R6 outputs to dark dots were similar to their normalized outputs to bright dots, with both showing equally good Raleigh-resolvability (*cf. Figure 7C*). Naturally, as the *Drosophila* eye samples light information from each point in space by eight photoreceptors (due to neural superposition) and balances this estimate with those of the neighboring lamina cartridges, its perception/resolvability of hyperacute and saccadic image motion is improved further.
DOI: https://doi.org/10.7554/eLife.26117.104

## Microsaccadic sampling hypothesis and efficient coding

Our results indicate that R1-R6 photoreceptors' information transfer adapts to the context of stimulus statistics; with refractory microvilli, fast quantum bump adaptation (*Henderson et al., 2000*; *Juusola and Hardie, 2001a*; *Song et al., 2012*) and photomechanical microsaccades maximizing encoding of phasic information from high-contrast bursts. Remarkably, the extraction of phasic stimulus features, which characterize object boundaries and line elements in visual scenes, already starts during sampling and integration of visual information in the microvilli, at the first visual processing stage. The darker periods in stimuli relieve the effects of microvilli refractoriness, enabling greater sensitivity: more and slightly larger samples (quantum bumps) can be generated transiently to the next light change (*Figure 1*). This increases the signal-to-noise ratio of the integrated macroscopic response, especially in its phasic (fast rising/decaying) components (Appendix 3, *Appendix 3—figure 1*). However, unlike later information processing in the network, during which presynaptic inputs are often translated to postsynaptic spike-bursts of high sparseness for specific features, the neural code of photoreceptors must consider all stimulus features together. It adapts to allocate information in high-contrast bursts into continuous Gaussian broadband voltage signals (*Figure 2A–B*), utilizing the output range optimally.

In the viewpoint of efficient coding, stochastic refractory sampling, fast quantum bump adaptation and photomechanical contractions benefit vision in three important ways:

- They exchange redundant information in mean voltage to more useful information in relative modulation, enabling photoreceptors to encode reliable estimates of the world within their limited output ranges, despite strongly and quickly changing intensities.

- They lower the metabolic cost of information with fewer bumps integrating a lower membrane potential, consuming less ATP (*Song and Juusola, 2014*).
- And when linked to bursty saccadic head/body movements, they increase high-frequency information capture from the world and reduce motion blur.

These results further imply that saccadic behaviors enable the fly eye to convey to the fly brain a far more efficient and accurate neural image of the variable world than what was believed before. Thus, saccades not only contribute to gaze-stabilization (*Land, 1973*; *Fox and Frye, 2014*), which historically is considered to be their major function (*Land, 1999*), but they also "burstify" light input for efficient and accurate sampling.

## Appendix 10

DOI: https://doi.org/10.7554/eLife.26117.105

# *Drosophila* behavior in a flight simulator system confirms hyperacute vision

## Overview

This appendix describes the used optomotor behavioral paradigm in a classic *Drosophila* flight simulator system to study visual resolution, and provides important background information about the experimental and theoretical results presented in *Figure 10* in the main paper.

In this appendix:

- We test in open-loop conditions whether wild-type *Drosophila* can generate yaw-torque (optomotor responses) to hyperacute vertical black-and-white stripe-scenes that rotate clockwise and counterclockwise at 45 °/s.
- Our results clearly indicate that wild-type *Drosophila* have hyperacute vision (seeing finer image details than the average interommatidial angle, Δφ ~4.5, of their compound eyes), with the measured behavioral responses closely following the prediction of our new 'micro-saccadic sampling'-hypothesis (see Appendix 8).

## Testing *Drosophila* visual behavior in flight simulator system

Wild-type 'Berlin' *Drosophila* were raised on standard food medium at 25°C and 60% relative humidity with a 12 hr light and 12 hr dark cycle, with light on at 8 a.m. In the experiments, we used 2- to 3-day-old female flies. Under cold-anesthesia (lasting <3 min), a small copper-wire hook was fixed with a droplet of UV-light sensitive glue (Loctite) between each fly's head and thorax. After preparation, flies were left to familiarize themselves with their hooks overnight in single vials, which provided them water and sucrose.

A custom-built, computer controlled flight simulator system (*Wardill et al., 2012*) was used to study *Drosophila*'s optomotor behavior. A tethered *Drosophila* was connected to the torque-meter (*Tang and Guo, 2001*) by a small clamp holding the copper-wire hook, which fixed the fly's head in a rigid position and orientation, but allowed stationary flight (*Götz, 1964*; *Heisenberg and Buchner, 1977*). The torque meter transduced yaw torque into electrical voltage.

A fly, tethered from the torque meter, was lowered by a mechanical micromanipulator in the center of a white featureless plastic hollow cylinder (a diffuser). Inside it, we placed high-resolution visual patterns (bars, stripe patterns, etc.), which were laser-printed on a transparent film, forming a 360° panorama around the fly's long axis. The panorama could be rotated around its vertical axis by a servomotor. Outside, the diffuse cylinder faced a surrounding ring-shaped light-tube (special full-band: 350–900 nm) that provided uniform illumination on the panorama. The light intensity during the panoramic motion stimulation, although bright, was always less (0.5–1.5 log-intensity units) than the direct stimuli used in the intracellular recordings (*cf. Figure 1*).

## Open-loop experiments

Inside the flight simulator, a flying fly saw a continuous (360°) stripe-scene (black-and-white bars) of predetermined spectral and spatial resolution, which was free of motion artefacts, flashing or aliasing. After one second of viewing the still scene, it was spun to right (clockwise) by a linear stepping motor (in which output was recorded simultaneously by a separate potentiometer coupled to the motor) for two seconds, stopped for two seconds, before rotating to left (counterclockwise) for two seconds, and stopped again for a second. This eight-second stimulus was repeated 10–25 times and each trial, together with the fly's coincident yaw torque responses, was sampled at 1 kHz and stored in a PC's hard-drive for

later analysis, using custom-written software (Biosyst) (*Juusola and Hardie, 2001a*). Presumably to stabilize gaze, flies tend to follow the scene rotations, generating yaw torque responses (optomotor responses to right or left), the strength of which is believed to reflect the strength of their motion perception (*Götz, 1964*; *Heisenberg and Buchner, 1977*; *Wardill et al., 2012*). The fixed stimulus parameters for moving stripe scenes, as shown in the figures, were: azimuth ±360°; elevation ±45°; velocity, 45, 50, 200 or 300 °/s; contrast, 1.0, as seen by the fly. *Figure 10A* show the averages (*n* = 9 flies) of the mean optomotor responses (*n* = 22–35 trials for each fly).

We first tested optomotor responses of wild-type flies to black-and-white stripe-scenes (spectral full-width: 380–900 nm) of three different spatial resolutions (wavelength: 1.16°, 2.88° and 14.4°), rotating at 45°/s, as shown in *Figure 10A–C*. To verify that air flow, or some hidden features in the stimulus panorama, was not affecting optomotor responses, we used the white diffuser cylinder alone, which showed no clear contrast to human eye, as the control stimulus. These control field rotation experiments were repeated using the same flies (*Figure 10—figure supplement 1*). We found by that the white control stimulus did not evoke torque responses.

We also tested optomotor responses of five flies to 3.9° (hyperacute) and 14.4° (control) wavelength panoramic stipe-scenes, rotating at 50, 200 and 300 °/s (*Figure 10D–F*). The results were consistent with the predictions of the full photoreceptor model (*cf. Figure 9A*, two dots 4° apart), which incorporated both the refractory photon sampling and photomechanical rhabdomere motion dynamics.

*Quantifying optomotor behavior*. The optomotor responses of individual flies to the same repeated field rotations vary in strength and repeatability (*Figure 10—figure supplement 1A*), but their visual performance to different spatial resolution stripe scenes is clearly different. These differences can be quantified by measuring the mean torque response of a single fly to stimulus repetitions and by averaging the mean responses of the many flies of the same stripe scene resolution (*Figure 10—figure supplement 1B*; here 9). This reduces noise and non-systematic (arbitrary) trends of single experiments, revealing the underlying response strength and optomotor behavior characteristics. These population responses are shown in *Figure 10* for a straightforward comparison.

In open-loop experiments, a fly's torque response returns gradually to baseline after the optomotor stimulus stops, but this can take seconds (varying with individual flies). Accordingly, in our experiments, which contain only brief 2-s-long inter-stimulus-intervals, the torque responses typically recover only fractionally (10–70%) during these still periods toward the baseline. Therefore, for comparing the optomotor behavior different stripe scene resolutions, we used the maximum range (or peak-to-peak) of the torque response, evoked by the combined leftward and rightward field rotation stimulus. The maximum range and variability in the torque responses to the same optomotor stimulus are shown with controls in *Figure 10—figure supplement 1C and D*, respectively.

Markedly, the optomotor responses to hyperacute stripe-scenes were not caused by aliasing. This is because perceptual aliasing (such as the wagon-wheel effect or Moiré patterns), if induced by the rotating hyperacute scenes, would have been perceived as slowed down image rotation, eventually reversing to the opposite direction (the reverse rotation effect). And thus, if the tested flies had seen such motion patterns, they would have consequently followed them slower and rotated against the real scene rotation direction. Such optomotor behavior was never observed in our experiments.

## Why did the previous behavioral studies not find hyperacute vision?

In 1976, Buhner probed *Drosophila*'s visual acuity by stimulating the upper frontal part in one of its eyes with small local moving grating patterns (covering about 50 ommatidia) while a fly walked on track-ball (*Buchner, 1976*). Notably, the aim of his study was not to find the finest resolution what a *Drosophila* can resolve but instead to deduce the likely columnar organization of its directionally sensitive elementary motion detectors from a fixed fly's

tendency to follow moving stimulus patterns. Thus, this was also an open-loop paradigm, but the used microscope-mediated local grating stimulation was very different from the global hyperacute panoramic visual scenes of our study. Specifically, we note that in Buchner's study:

- Visual acuity was not tested below the interommatidial angle ($\Delta\varphi$ ~4.5); with the overall results deduced by eliminating the presumed boundary elements and contrast attenuation from the data.
- The used mean stimulus light intensity (luminance; 16 cd/m$^2$) was low. Therefore, the resulting image grating at the level of individual photoreceptors would have been dim and spatio-temporal signal-to-noise ratio of light input and photoreceptor output low. Based on our intracellular data (*Juusola and Hardie, 2001a*; *Song et al., 2012*), this dim light intensity would have made it practically impossible for R1-R6 photoreceptors to resolve very fine (or hyperacute) visual patterns.
- The sensitivity and the time resolution of the used trackball system (*Buchner, 1976*) seem significantly less than in our bespoke torque meter (*Tang et al., 2004*), requiring extensive data averaging. This would have made it more difficult for the trackball system to resolve the weaker (small amplitude) behavioral responses to fine spatial contrast changes (*Figure 10A*).

More recently, because of the historical belief that interommatidial angle limits a fly's visual acuity, many experimentalists have started using coarse LED-matrixes, typically with 4.5-5° maximum resolution, to probe visual learning and optomotor responses. As our study here shows, these kinds of visual stimuli are very different from the panoramic high-resolution printed scenes with thin continuous lines and symbols and thus are expected evoke quite different neural responses.

