## [Decision Letter]

Thank you for submitting your article "Microsaccadic information sampling provides *Drosophila* hyperacute vision" for consideration by *eLife*. Your article has been reviewed by two peer reviewers, one of whom is a member of our Board of Reviewing Editors and the evaluation has been overseen David Van Essen as the Senior Editor. The reviewers have opted to remain anonymous.

The reviewers have discussed the reviews with one another and the Reviewing Editor has drafted this decision to help you prepare a revised submission.

This is an impressively comprehensive paper on visual acuity in flies. Both reviewers appreciated the depth with which the paper tackled a number of important issues. The work in the paper has an opportunity both to make an important point about the relation between eye movements and visual acuity and to become a reference on a number of issues about fly vision. To reach these dual goals, however, the work in the paper needs to be presented more clearly. Several options emerged from discussions among the editors and reviewers, including splitting the work in the paper into two related papers, with the intention that they would appear back-to-back, or revising the paper to into account suggestions in the individual reviews (below). Most important in considering the revisions is to target the paper or papers to a broad audience and to deliver the message in a way that is broadly accessible. These and some other points are elaborated below.

Reviewer #1:

This is a difficult paper to evaluate. I am not an expert on insect vision, but struggled more than I should have to follow the main arguments. I will leave detailed comments for the more expert reviewers, and focus primarily on accessibility. The paper contains several interesting ideas. Unfortunately, the data and analysis supporting these ideas is sufficiently spread out that it is very hard to follow and evaluate. I also felt that some of the essential aspects of the paper were not developed sufficiently.

Major points:

1) Stimulus manipulations in experiments of Figure 1.

The results illustrated in Figure 1 are interpreted in terms of changes in stimuluscontrast, "burstiness" and temporal frequency. But the mean light level is also changing substantially, which confounds interpretation of these experiments. Either a clear explanation for why the mean is not important needs to be given or the data needs to be interpreted with this in mind.

2) Hyperacuity introduced late and not clearly connected to sampling.

The focus of the paper, from the title, is on visual acuity. Yet the underlying data only starts in Figure 7. This is part of a more general issue of the number of ideas packed into the paper, and the difficulty as a reader in keeping track of the key points. This is exacerbated because the relation between the key ideas is not clear (e.g. how the sampling and signal to noise experiments from the first figures relate to the acuity experiments at the end of results). I appreciate the comprehensiveness of the paper, but worry that few readers will make it through.

3) Information and coding efficiency calculations.

The description of the coding efficiency measurements is quite spread out in the paper. Parts of it are in Results, Materials and methods and in the Appendices. Further, the measure is introduced (in Figure 2) before the relevant calculations are described. In addition to these issues about presentation, the calculations assume Gaussian signals and noise (i.e. for Equation 1 to hold). This is clearly not the case. Is the non-Gaussian nature of the signal (and possibly noise) an issue and if so how much is it likely to bias the information calculations and efficiency measures?

4) Figure 6 and sampling of inputs via natural movements.

The differences between the intensities sampled for saccades and a linear sampling in Figure 6 are quite small. It is hard to know how much importance to place on such small differences.

5) Speed tuning of behavior.

Figure 9 makes clear predictions about the speed dependence of spatial resolution. The behavioral experiments of Figure 10 would be enhanced greatly if these predictions were tested; if there are reasons that this is difficult, a discussion of why would be helpful.

6) Appendices and supplementary material.

There is a good deal of information central to the paper in the Appendices. It is unclear how much of this repeats previous papers and how much is new. I did not have time to evaluate the appendices. Is it possible to reduce them substantially in length by referencing out pieces?

Reviewer #2:

Understanding the inputs to a sensory system is fundamental to any quantitative of neural processing. The fruit fly visual system has fundamentally enlightened our understanding both of invertebrate phototransduction mechanisms and, more recently, has provided a number of important insights into neural computation. However, despite an extensive body of work spanning more than 50 years, Juusola and colleagues convincingly demonstrate that a property of the system as fundamental as acuity has been incompletely understood.

In this monumental work, Juusola and colleagues systematically measure and model photoreceptor responses to light across a wide range of viewing conditions and substantially extend our understanding. More specifically, by quantitatively modeling and measuring a combination of electrical signals and rhadomere movements, combined with behavior, the authors demonstrate that the fly retina offers a much high acuity view of the world than previous appreciated, and that the saccadic structure of fly movements may play a central role in enhancing acuity, a finding that runs contrary to accepted intuition.

This work is substantial, spanning 48 figures and more than 200 pages, making it both challenging to grasp in its entirety, and certain to emerge as the go-to reference standard in this field. I am therefore strongly supportive of its publication. I have only a small number of conceptual points where I believe the authors overstate the implications of their case, which I believe can be readily addressed by changes to the text. These issues are as follows:

1) The role of network connections to "add" information. In multiple places in the text, the authors refer to the ability of network connections to add information (for example, in the discussion of Figure 5 "extra network information…is mostly at high burst frequencies" and "the real R1-R6 cells receive extra information from their neighbors"). The stimuli the authors use to make this point are spatially homogenous; as a result, the only possible role that lateral synaptic connections can possibly play in enhancing the SNR/infomax of the system is through additional averaging of independent photoreceptor estimates of photon count. If this is correct, please make this point more transparent in the text. If this is incorrect, please explain how the network can provide additional information. Is it also correct to infer that if the lateral connections are really enhancing infomax/bitrate of photoreceptors, these lateral synaptic connections must contribute substantially little noise themselves? Can the authors comment on this from their data? In the same vein, the authors note that inputs from R7 and R8 would also contribute additional information. However, while this is true for R8, it is unlikely to be true for R7 under the conditions used by the authors (in which the white LED used is unlikely to have a significant UV component to drive R7).

2) As far as I can tell, the authors have no evidence to support the role for network connection asymmetries to produce differences in response in photoreceptors to motion in different directions. And despite the striking connectivity of L2 and L4, which has inspired a number of different functional models over the last 40 years, none of these models have received strong experimental support. Therefore please truncate this discussion in Appendix 6.

3) I am also unclear of the evidence in support of the phasic differences in inputs from the two eyes having information relevant to "measuring directional changes in the visual space, in respect to its head direction" (Figure 46). That is, while I agree that the photoreceptor movement should produce such a phase lag to rotational stimuli, I don't see a circumstance in which this lag would add any new information to the representation of the direction of movement that isn't already available from monocular inputs. Either articulate a circumstance under which detecting this phase delay would provide useful, unique information, or reduce the speculation.

4) Figure 6 – It appears that the authors examined responses to only six different natural scenes. How were these samples chosen? How representative are they of the image statistics in the van Hateren database? I could imagine how scenes with different spatial structure (particularly lower spatial frequency structure) would tend to bring contrast features into ideal sampling range at different turn velocities (notwithstanding the ability of saccades to make sampling shorter). Does this either match any existing data on saccade statistics, or make any predictions for future studies?

[Editors' note: further revisions were requested prior to acceptance, as described below.]

Thank you for submitting your article "Microsaccadic sampling of moving image information provides *Drosophila* hyperacute vision" for consideration by *eLife*. Your article has been reviewed by two peer reviewers, one of whom is a member of our Board of Reviewing Editors, and the evaluation has been overseen David Van Essen as the Senior Editor. The reviewers have opted to remain anonymous.

The reviewers have discussed the reviews with one another and the Reviewing Editor has drafted this decision to help you prepare a revised submission. Both reviewers felt the paper had improved. At the same time, several issues remain and we cannot reach a final decision until those are taken care of. In discussion, both reviewers felt these issues were important.

Reviewer #1:

This is a rereview of a paper about coding of naturalistic inputs by fly photoreceptors. The paper has improved in revision, but several substantial issues remain. These and some smaller points follow:

Figure 1 stimulus.

The new paragraph added on line 161 did not really answer the concern about whether changes in information rate could be attributed to contrast and burstiness, or whether the (changing) mean also should be considered. The argument in that paragraph is certainly reasonable, but is suggestive rather than a real test of the issue. It seems to me that the stimulus design confounds changes in mean and contrast – and acknowledging that and the resulting difficulty in knowing what controls the responses in Figure 1 would be quite helpful.

Information calculations.

I still have two concerns about the information calculations. First, testing for gaussian-distributed amplitudes for signal and noise is not a good test of whether the distributions are actually gaussian, since that also requires randomness of the phases of the different frequency components (e.g. a perfectly deterministic and periodic stimulus could have a gaussian amplitude distribution). Second, the argument that the Shannon formula is reasonable for highly non-gaussian responses cites previous work which, at least to my understanding, relies on extrapolation to obtain good estimates of information rates. It is not clear if that procedure was followed here, and if not whether the past work can be used to validate the current approach.

Histograms in Figure 6.

I asked previously about the small differences between the linear and saccadic walk distributions in this figure. The additional supplementary data helps make the case that these differences are systematic, but not that they are large enough to really matter. Given the small size of the differences, I think a better argument needs to be give as to why one should care about them.

Reviewer #2:

The authors have thoughtfully responded to all of my concerns, and I am now fully supportive of publication in *eLife*.

---

## [Author Response]

*Reviewer #1:*

*This is a difficult paper to evaluate. I am not an expert on insect vision, but struggled more than I should have to follow the main arguments. I will leave detailed comments for the more expert reviewers, and focus primarily on accessibility. The paper contains several interesting ideas. Unfortunately, the data and analysis supporting these ideas is sufficiently spread out that it is very hard to follow and evaluate. I also felt that some of the essential aspects of the paper were not developed sufficiently.*

To make it easier for a reader to follow the arguments and to obtain a new integrated viewpoint of saccadic information sampling, we now briefly introduce its key discoveries in the beginning of the Results section, and indicate where these are explained in detail. The reviewer’s concern that some essential aspects were not developed sufficiently is addressed in our following responses to each raised point.

*Major points:*

1) Stimulus manipulations in experiments of Figure 1.

*The results illustrated in Figure 1 are interpreted in terms of changes in stimuluscontrast, "burstiness" and temporal frequency. But the mean light level is also changing substantially, which confounds interpretation of these experiments. Either a clear explanation for why the mean is not important needs to be given or the data needs to be interpreted with this in mind.*

Our response: we have now added a new paragraph to make this point clearer:

“Notably, whilst all the stimuli were very bright, the largest responses (to bursts) were induced at the dimmest background (BG0, darkness) and the smallest responses (to GWN) at the brightest background (BG1.5) (Figure 1). […] More important for good vision are the stimulus contrast and bandwidth, which drive the dynamic quantum bump rate changes, summing up the photoreceptor output.”

2) Hyperacuity introduced late and not clearly connected to sampling.

*The focus of the paper, from the title, is on visual acuity. Yet the underlying data only starts in Figure 7. This is part of a more general issue of the number of ideas packed into the paper, and the difficulty as a reader in keeping track of the key points. This is exacerbated because the relation between the key ideas is not clear (e.g. how the sampling and signal to noise experiments from the first figures relate to the acuity experiments at the end of results). I appreciate the comprehensiveness of the paper, but worry that few readers will make it through.*

We have made many changes/additions/edits to link the key ideas better.

· The paper has new title: “Seeing through moving eyes – microsaccadic information sampling provides *Drosophila* hyperacute vision”.

· To make it easier to understand the paper’s various results and to obtain a new integrated viewpoint of saccadic information sampling, the Abstract ends now:” These discoveries elucidate how acuity depends upon photoreceptor function and eye movements.”

· We now also spell out the value of this study (Introduction, the last sentence), highlighting its dual contributions to the field:” By demonstrating how fly photoreceptors’ fast microsaccadic information sampling provides hyperacute vision, these results change our understanding of insect vision, whilst showing an important relationship between eye movements and visual acuity.”

· And perhaps most importantly, we now briefly introduce and link its main results in a new results summary.

3) Information and coding efficiency calculations.

*The description of the coding efficiency measurements is quite spread out in the paper. Parts of it are in Results, Materials and methods and in the Appendices. Further, the measure is introduced (in Figure 2) before the relevant calculations are described. In addition to these issues about presentation, the calculations assume Gaussian signals and noise (i.e. for Equation 1 to hold). This is clearly not the case. Is the non-Gaussian nature of the signal (and possibly noise) an issue and if so how much is it likely to bias the information calculations and efficiency measures?*

To address this concern, we have added a new paragraph:

“There are two reasons why these information rate estimates, which were calculated from equal-sized datasets by the Shannon formula (Eq. 1, Material and Methods), should be robust and largely bias-free. […] Thus here, the Shannon formula should provide a sufficiently accurate information estimate also for the 20 Hz high-contrast burst responses, making this evaluation fair (see Appendix 2).”

4) Figure 6 and sampling of inputs via natural movements.

*The differences between the intensities sampled for saccades and a linear sampling in Figure 6 are quite small. It is hard to know how much importance to place on such small differences.*

In Appendix 3, Figure 18—figure supplement 1, we compare six examples of such differences from six different naturalistic images. Each one shows similar behaviour, indicating that these differences are both consistent and predictable for saccadic viewing.

5) Speed tuning of behavior.

*Figure 9 makes clear predictions about the speed dependence of spatial resolution. The behavioral experiments of Figure 10 would be enhanced greatly if these predictions were tested; if there are reasons that this is difficult, a discussion of why would be helpful.*

We thank the reviewer for this useful suggestion, which made Figure 10 stronger. We performed new experiments and added new subfigures D-F to show this data. We now write: “Moreover, when a fine-grained (3.9o) panoramic image was rotated faster (Figure 10), the response declined as predicted (cf. two dots 4^o^ apart in Figure 9). This result is consistent with photoreceptor output setting the perceptual limit for vision and demonstrates that *Drosophila* see hyperacute details even at saccadic speeds (Figure 10).”

6) Appendices and supplementary material.

*There is a good deal of information central to the paper in the Appendices. It is unclear how much of this repeats previous papers and how much is new. I did not have time to evaluate the appendices. Is it possible to reduce them substantially in length by referencing out pieces?*

We believe this would make our paper less accessible, as each Appendix is very carefully constructed to clearly explain the new concepts and methods of this study. However, we have edited the Appendixes carefully to further increase their clarity.

*Reviewer #2:*

[…]

*1) The role of network connections to "add" information. In multiple places in the text, the authors refer to the ability of network connections to add information (for example, in the discussion of Figure 5 "extra network information…is mostly at high burst frequencies" and "the real R1-R6 cells receive extra information from their neighbors"). The stimuli the authors use to make this point are spatially homogenous; as a result, the only possible role that lateral synaptic connections can possibly play in enhancing the SNR/infomax of the system is through additional averaging of independent photoreceptor estimates of photon count. If this is correct, please make this point more transparent in the text. If this is incorrect, please explain how the network can provide additional information. Is it also correct to infer that if the lateral connections are really enhancing infomax/bitrate of photoreceptors, these lateral synaptic connections must contribute substantially little noise themselves? Can the authors comment on this from their data? In the same vein, the authors note that inputs from R7 and R8 would also contribute additional information. However, while this is true for R8, it is unlikely to be true for R7 under the conditions used by the authors (in which the white LED used is unlikely to have a significant UV component to drive R7).*

These clarifying points are now integrated in the corresponding paragraph.

“In other words, since our stimuli (from a white LED) were spatially homogenous, these synaptic feedbacks should be able to enhance the system’s signal-to-noise by averaging the photoreceptors’ independent photon count estimates from the same visual area, reducing noise (Zheng et al., 2006; Juusola and Song, 2017).”

”And yet whilst R7s also share gap-junctions with R6s (Shaw et al., 1989), our stimuli contained little UV component to drive them.”

*2) As far as I can tell, the authors have no evidence to support the role for network connection asymmetries to produce differences in response in photoreceptors to motion in different directions. And despite the striking connectivity of L2 and L4, which has inspired a number of different functional models over the last 40 years, none of these models have received strong experimental support. Therefore please truncate this discussion in Appendix 6.*

This is now truncated as suggested. See the new reduced discussion in Appendix 6.

*3) I am also unclear of the evidence in support of the phasic differences in inputs from the two eyes having information relevant to "measuring directional changes in the visual space, in respect to its head direction" (Figure 46). That is, while I agree that the photoreceptor movement should produce such a phase lag to rotational stimuli, I don't see a circumstance in which this lag would add any new information to the representation of the direction of movement that isn't already available from monocular inputs. Either articulate a circumstance under which detecting this phase delay would provide useful, unique information, or reduce the speculation.*

We have now extended this point. Appendix 8:

“The microsaccadic sampling-hypothesis predicts that image rotation causes a phasic difference in photoreceptor outputs between the left and the right eye, with the signals always arriving slightly faster from the eye, towards which the fly rotates. Because this difference depends upon the rotation speed, it could be used for signaling changes in the fly’s heading direction or to improve visual navigation. For example, when flying across more homogenous surroundings, such as an open field with few distinctive visual landmarks, the central brain could use saccadic turns to recalibrate the fly’s head-direction in its internal world map near instantaneously; matching the intended direction to the new direction, as pointed by the global phase difference between the left and right eye signals.”

4) Figure 6 – It appears that the authors examined responses to only six different natural scenes. How were these samples chosen? How representative are they of the image statistics in the van Hateren database? I could imagine how scenes with different spatial structure (particularly lower spatial frequency structure) would tend to bring contrast features into ideal sampling range at different turn velocities (notwithstanding the ability of saccades to make sampling shorter). Does this either match any existing data on saccade statistics, or make any predictions for future studies?

We have added small text sections in Appendix 3 to address these useful points.

“These natural scenes were arbitrarily chosen from Google image search results, and we do not know how representative their image statistics are, for example, in respect to the van Hateren database (van Hateren, 1997a).”

“Finally, we note that it is possible that in scenes with different spatial structure (particularly lower spatial frequency structure), flies would use different turn velocities to bring contrast features into ideal sampling range (irrespective of saccades making sampling shorter). Future studies need to explore whether such a match with saccade statistics exists.”

[Editors' note: further revisions were requested prior to acceptance, as described below.]

*Reviewer #1:*

*This is a rereview of a paper about coding of naturalistic inputs by fly photoreceptors. The paper has improved in revision, but several substantial issues remain. These and some smaller points follow:*

Figure 1 stimulus.

*The new paragraph added on line 161 did not really answer the concern about whether changes in information rate could be attributed to contrast and burstiness, or whether the (changing) mean also should be considered. The argument in that paragraph is certainly reasonable, but is suggestive rather than a real test of the issue. It seems to me that the stimulus design confounds changes in mean and contrast – and acknowledging that and the resulting difficulty in knowing what controls the responses in Figure 1 would be quite helpful.*

We concur and have edited the paragraph to increase its clarity, including the reviewer comment. It now reads: “Notably, whilst all the stimuli were very bright, the largest responses (to bursts) were induced at the dimmest background (BG0, darkness) and the smallest responses (to GWN) at the brightest background (BG1.5) (Figure 1). […] And, as such, this stimulus design, by containing four different BGs, makes it difficult to see the exact contributions of contrast, bandwidth and mean in controlling the responses.”

Information calculations.

*I still have two concerns about the information calculations. First, testing for gaussian-distributed amplitudes for signal and noise is not a good test of whether the distributions are actually gaussian, since that also requires randomness of the phases of the different frequency components (e.g. a perfectly deterministic and periodic stimulus could have a gaussian amplitude distribution). Second, the argument that the Shannon formula is reasonable for highly non-gaussian responses cites previous work which, at least to my understanding, relies on extrapolation to obtain good estimates of information rates. It is not clear if that procedure was followed here, and if not whether the past work can be used to validate the current approach.*

· We agree and – to test these points further – we have now (1) added in the Materials and methods that we generated the Gaussian white-noise stimuli by Matlab’s randn-function. Post-generation, we tested that these inputs were spectrally white and had Gaussian distributions. Furthermore, (2) we have reconfirmed that apart from the 20 Hz burst stimulus, the evoked photoreceptor signals and noises were Gaussian or very close to that. Since we had already provided the corresponding raw datasets to *eLife* (now in Dryad repository), we felt that there was no real need to produce extra graphs of these distributions. Finally, (3) we made some longer recordings to 20 Hz bursts (having 40 stimulus repetitions, with the last 30 used in the analyses). Analysing these responses both with the Shannon method and the triple extrapolation method, which does not have Gaussian signal and noise assumptions, gives similar information rates over a broad range of reliable parameter values (in agreement with our earlier studies), endorsing the current approach and supporting the Figure 2 results. This new data/analyses are now included in Figure 2—figure supplement 4 (see below). We have further added a new sentence in the given paragraph: “And, indeed, new tests using additional recordings to longer stimulus repetitions (Figure 2—figure supplement 4) indicated the same.”

· We added a brief description of the triple extrapolation in the Materials and methods section.

Histograms in Figure 6.

*I asked previously about the small differences between the linear and saccadic walk distributions in this figure. The additional supplementary data helps make the case that these differences are systematic, but not that they are large enough to really matter. Given the small size of the differences, I think a better argument needs to be give as to why one should care about them.*

We thank for this very useful suggestion, which made us to better quantify the differences between the saccadic and linear walking histograms. These differences are highly significant, as indicated by the peak-value and kurtosis differences of the corresponding histograms. Saccadic walks (all data: 6 six panoramic images x 15 horizontal line scans) Peak_sac_ = 4478.66 ± 1424.55 vs linear walks Peak_lin_ = 3379.98 ± 1753.44 counts (mean ± SD, p = 1.4195 x 10^-32^, pair-wise t-test). Kurtosis_sac_ = 48.22 ± 99.80 vs Kurtosis_lin_ = 30.25 ± 37.85 (mean ± SD, p = 0.01861, pair-wise t-test). This information is now added in Figure 6 legends.